# On Function Approximation in Reinforcement Learning: Optimism in the Face of Large State Spaces

**Zhuoran Yang**
Princeton University
zy6@princeton.edu

**Chi Jin**
Princeton University
chij@princeton.edu

**Zhaoran Wang**
Northwestern University
zhaoran.wang@northwestern.edu

**Mengdi Wang**
Princeton University
mengdiw@princeton.edu

**Michael I. Jordan**
University of California, Berkeley
jordan@cs.berkeley.edu

## Abstract

The classical theory of reinforcement learning (RL) has focused on tabular and linear representations of value functions. Further progress hinges on combining RL with modern function approximators such as kernel functions and deep neural networks, and indeed there have been many empirical successes that have exploited such combinations in large-scale applications. There are profound challenges, however, in developing a theory to support this enterprise, most notably the need to take into consideration the exploration-exploitation tradeoff at the core of RL in conjunction with the computational and statistical tradeoffs that arise in modern function-approximation-based learning systems. We approach these challenges by studying an optimistic modification of the least-squares value iteration algorithm, in the context of the action-value function represented by a kernel function or an overparameterized neural network. We establish both polynomial runtime complexity and polynomial sample complexity for this algorithm, without additional assumptions on the data-generating model. In particular, we prove that the algorithm incurs an $\widetilde{\mathcal{O}}(\delta_{\mathcal{F}} H^2 \sqrt{T})$ regret, where $\delta_{\mathcal{F}}$ characterizes the intrinsic complexity of the function class $\mathcal{F}$, $H$ is the length of each episode, and $T$ is the total number of episodes. Our regret bounds are independent of the number of states, a result which exhibits clearly the benefit of function approximation in RL.

## 1 Introduction

Reinforcement learning (RL) algorithms combined with modern function approximators such as kernel functions and deep neural networks have produced empirical successes in a variety of application problems [e.g., 27, 60, 61, 72, 70]. However, theory has lagged, and when these powerful function approximators are employed, there is little theoretical guidance regarding the design of RL algorithms that are efficient computationally or statistically, or regarding whether they even converge. In particular, function approximation blends statistical estimation issues with dynamical optimization issues, resulting in the need to balance the bias-variance tradeoffs that arise in statistical estimation with the exploration-exploitation tradeoffs that are inherent in RL. Accordingly, full theoretical treatments are mostly restricted to the tabular setting, where both the state and action spaces are

discrete and the value function can be represented as a table [see, e.g., 33, 52, 6, 35, 50, 56], and there is a disconnect between theory and the most compelling applications.

Provably efficient exploration in the function approximation setting has been addressed only recently, with most of the existing work considering (generalized) linear models [78, 77, 36, 12, 80, 73]. These algorithms and their analyses stem from classical upper confidence bound (UCB) or Thompson sampling methods for linear contextual bandits [11, 41] and it seems difficult to extend them beyond the linear setting. Unfortunately, the linear assumption is rather rigid and rarely satisfied in practice; moreover, when such a model is misspecified, sublinear regret guarantees can vanish. There has been some recent work that has presented sample-efficient algorithms with general function approximation. However, these methods are either computationally intractable [39, 34, 20, 22] or hinge on strong assumptions on the transition model [75, 24]. Thus, the following question remains open:

Can we design RL algorithms that incorporate powerful nonlinear function approximators such as neural networks or kernel functions and provably achieve both computational and statistical efficiency?

In this work, we provide an affirmative answer to this question. Focusing on the setting of an episodic Markov decision process (MDP) where the value function is represented by either a kernel function or an overparameterized neural network, we propose an RL algorithm with polynomial runtime complexity and sample complexity, without imposing any additional assumptions on the data-generating model. Our algorithm is relatively simple—it is an optimistic modification of the least-squares value iteration algorithm (LSVI) [10]—a classical batch RL algorithm—to which we add a UCB bonus term to each iterate. Specifically, when using a kernel function, each LSVI update becomes a kernel ridge regression, and the bonus term is derived from that proposed for kernelized contextual bandits [62, 67, 18]. For the neural network setting, motivated by the NeuralUCB algorithm for contextual bandits [84], we construct a UCB bonus from the tangent features of the neural network and we perform the LSVI updates via projected gradient descent. In both of these settings, the usage of the UCB bonus ensures that the value functions constructed by the algorithm are always optimistic in the sense that they serve as uniform upper bounds of the optimal value function. Furthermore, for both the kernel and neural settings, we prove that the proposed algorithm incurs an $\widetilde{\mathcal{O}}(\delta_{\mathcal{F}} H^2 \sqrt{T})$ regret, where $H$ is the length of each episode, $T$ is the total number of episodes, and $\delta_{\mathcal{F}}$ quantifies the intrinsic complexity of the function class $\mathcal{F}$. Specifically, as we will show in §4, $\delta_{\mathcal{F}}$ is determined by the interplay between the $\ell_{\infty}$-covering number of the function class used to represent the value function and the effective dimension of function class $\mathcal{F}$. (See Table 1 for a summary.)

A key feature of our regret bounds is that they depend on the complexity of the state space only through $\delta_{\mathcal{F}}$ and thus allow the number of states to be very large or even divergent. This clearly exhibits the benefit of function approximation by tying it directly to sample efficiency. To the best of our knowledge, this is the first provably efficient framework for reinforcement learning with kernel and neural network function approximations.

**Related Work.** There is a vast literature on establishing provably efficient RL methods in the absence of a generative model or an explorative behavioral policy. Much of this literature has focused on the tabular setting; see [33, 52, 6, 21, 65, 35, 56] and the references therein. In particular, [6, 35] prove that an RL algorithm necessarily incurs a $\Omega(\sqrt{SAT})$ regret under the tabular setting, where $S$ and $A$ are the cardinalities of the state and action spaces, respectively. Thus, algorithms designed for the tabular setting cannot be directly applied to the function approximation setting, where the number of effective states is large. A recent literature has accordingly focused on the function approximation setting, specifically the (generalized) linear setting where the value function (or the transition model) can be represented using a linear transform of a known feature mapping [77, 78, 36, 12, 80, 73, 5, 83, 37]. Among these papers, our work is most closely related to [36]. In particular, in our kernel setting when the kernel function has a finite rank, both our LSVI algorithm and the corresponding regret bound reduce to those established in [36]. However, the sample complexity and regret bounds in [36] diverge when the dimension of the feature mapping goes to infinity and thus cannot be directly applied to the kernel setting.

| function class $\mathcal{F}$ | regret bound |
| --- | --- |
| general RKHS $\mathcal{H}$ | $H^2 \cdot \sqrt{d_{\mathrm{eff}} \cdot [d_{\mathrm{eff}} + \log N_\infty(\epsilon^*)]} \cdot \sqrt{T}$ |
| $\gamma$-finite spectrum | $H^2 \cdot \sqrt{\gamma^3 T} \cdot \log(\gamma T H)$ |
| $\gamma$-exponential decay | $H^2 \cdot \sqrt{(\log T)^{3/\gamma} \cdot T} \cdot \log(TH)$ |
| overparameterized neural network | $H^2 \cdot \sqrt{d_{\mathrm{eff}} \cdot [d_{\mathrm{eff}} + \log N_\infty(\epsilon^*)]} \cdot \sqrt{T} + \mathrm{poly}(T, H) \cdot m^{-1/12}$ |

Table 1: Summary of the main results. Here $H$ is the length of each episode, $T$ is the number of episodes in total, and $2m$ is the number of neurons of the overparameterized networks in the neural setting. For an RKHS $\mathcal{H}$ in general, $d_{\mathrm{eff}}$ denotes the effective dimension of $\mathcal{H}$ and $N_\infty(\epsilon^*)$ is the $\ell_\infty$-covering number of the value function class, where $\epsilon^* = H/T$. Note that to obtain concrete bounds, we apply the general result to RKHS's with various eigenvalue decay conditions. Here $\gamma$ is a positive integer in the case of $\gamma$-finite spectrum and is a positive number in the case of $\gamma$-exponential decay. Finally, in the last case we present the regret bound for the neural setting in general, where $d_{\mathrm{eff}}$ is the effective dimension of the neural tangent kernel (NTK) induced by the overparameterized neural network with $2m$ neurons and $\mathrm{poly}(T, H)$ is a polynomial in $T$ and $H$. Such a general regret bound can be expressed concretely as a function of the spectrum of the NTK.

Also closely related to our work is [71], which studies a similar optimistic LSVI algorithm for general function approximation. This work focuses on value function classes with bounded eluder dimension [57, 51]. It is unclear whether whether this formulation can be extended to the kernel or neural network settings. [78] studies a kernelized MDP model where the transition model can be directly estimated. Under a slightly more general model, [5] recently propose an optimistic model-based algorithm via value-targeted regression, where the model class is the set of functions with bounded eluder dimension. In other recent work, [37] studies a nonlinear control formulation in which the transition dynamics belongs to a known RKHS and can be directly estimated from the data. Our work differs from this work in that we impose an explicit assumption on the transition model and our proposed algorithm is model-free.

Other authors who have presented regret bounds and sample complexities beyond the linear setting include [39, 34, 20, 22]. These algorithms generally involve either high computational costs or require possibly restrictive assumptions on the transition model [74, 75, 24].

Our work is also related to the literature on contextual bandits with either kernel function classes [62, 38, 63, 67, 18, 28] or neural network function classes [84]. Our construction of a bonus function for the RL setting has been adopted from this previous work. However, while contextual bandits can be viewed formally as special cases of our episodic MDP formulation with the episode length equal to one, the temporal dependence in the MDP setting raises significant challenges. In particular, the covering number $N_\infty(\epsilon^*)$ in Table 1 arises as a consequence of the fundamental challenge of performing temporally extended exploration in RL.

Finally, our analysis of the optimistic LSVI algorithm is related to recent work on optimization and generalization in overparameterized neural networks within the framework of the neural tangent kernel [32]. See also [19, 32, 76, 25, 26, 3, 2, 85, 17, 44, 4, 15, 16, 43]. This literature focuses principally on supervised learning, however; in the RL setting we need an additional bonus term in the least-squares problem and thus require a novel analysis.

## 2 Background

In this section, we provide essential background on reinforcement learning, reproducing kernel Hilbert space (RKHS), and overparameterized neural networks.

**Episodic Markov Decision Processes**

We focus on episodic MDPs, denoted $\mathrm{MDP}(\mathcal{S}, \mathcal{A}, H, \mathbb{P}, r)$, where $\mathcal{S}$ and $\mathcal{A}$ are the state and action spaces, respectively, the integer $H > 0$ is the length of each episode, $\mathbb{P} = \{\mathbb{P}_h\}_{h \in [H]}$ and $r = \{r_h\}_{h \in [H]}$ are the Markov transition kernel and the reward functions, respectively, where we let $[n]$

denote the set $\{1, \ldots, n\}$ for integers $n \geq 1$. We assume that $\mathcal{S}$ is a measurable space of possibly infinite cardinality while $\mathcal{A}$ is a finite set. Finally, for each $h \in [H]$, $\mathbb{P}_h(\cdot \,|\, x, a)$ denotes the probability transition kernel when action $a$ is taken at state $x \in \mathcal{S}$ in timestep $h \in [H]$, and $r_h \colon \mathcal{S} \times \mathcal{A} \to [0, 1]$ is the reward function at step $h$ which is assumed to be deterministic for simplicity.

A *policy* $\pi$ of an agent is a set of $H$ functions $\pi = \{\pi_h\}_{h \in [H]}$ such that each $\pi_h(\cdot \,|\, x)$ is a probability distribution over $\mathcal{A}$. Here $\pi_h(a \,|\, x)$ is the probability of the agent taking action $a$ at state $x$ at the $h$-th step in the episode.

The agent interacts with the environment as follows. For any $t \geq 1$, at the beginning of the $t$-th episode, the agent determines a policy $\pi^t = \{\pi_h^t\}_{h \in [H]}$ while an initial state $x_1^t$ is picked arbitrarily by the environment. Then, at each step $h \in [H]$, the agent observes the state $x_h^t \in \mathcal{S}$, picks an action $a_h^t \sim \pi_h^t(\cdot \,|\, x_h^t)$, and receives a reward $r_h(x_h^t, a_h^t)$. The environment then transitions into a new state $x_{h+1}^t$ that is drawn from the probability measure $\mathbb{P}_h(\cdot \,|\, x_h^t, a_h^t)$. The episode terminates when the $H$-th step is reached and $r_H(x_H^t, a_H^t)$ is thus the final reward that the agent receives.

The performance of the agent is captured by the *value function*. For any policy $\pi$, and $h \in [H]$, we define the value function $V_h^\pi \colon \mathcal{S} \to \mathbb{R}$ as

$$V_h^\pi(x) = \mathbb{E}_\pi\left[\sum_{h'=h}^{H} r_{h'}(x_{h'}, a_{h'}) \,\Big|\, x_h = x\right], \qquad \forall x \in \mathcal{S}, h \in [H],$$

where $\mathbb{E}_\pi[\cdot]$ denotes the expectation with respect to the randomness of the trajectory $\{(x_h, a_h)\}_{h=1}^{H}$ obtained by following the policy $\pi$. We also define the action-value function $Q_h^\pi \colon \mathcal{S} \times \mathcal{A} \to \mathbb{R}$ as follows:

$$Q_h^\pi(x, a) = \mathbb{E}_\pi\left[\sum_{h'=h}^{H} r_{h'}(x_{h'}, a_{h'}) \,\Big|\, x_h = x, \, a_h = a\right].$$

Moreover, let $\pi^\star$ denote the optimal policy which by definition yields the optimal value function, $V_h^\star(x) = \sup_\pi V_h^\pi(x)$, for all $x \in \mathcal{S}$ and $h \in [H]$. To simplify the notation, we write

$$(\mathbb{P}_h V)(x, a) := \mathbb{E}_{x' \sim \mathbb{P}_h(\cdot \,|\, x, a)}[V(x')],$$

for any measurable function $V \colon \mathcal{S} \to [0, H]$. Using this notation, the Bellman equation associated with a policy $\pi$ becomes

$$Q_h^\pi(x, a) = (r_h + \mathbb{P}_h V_{h+1}^\pi)(x, a), \qquad V_h^\pi(x) = \langle Q_h^\pi(x, \cdot), \pi_h(\cdot \,|\, x)\rangle_{\mathcal{A}}, \qquad V_{H+1}^\pi(x) = 0. \tag{2.1}$$

Here we let $\langle \cdot, \cdot \rangle_{\mathcal{A}}$ denote the inner product over $\mathcal{A}$. Similarly, the Bellman optimality equation is given by

$$Q_h^\star(x, a) = (r_h + \mathbb{P}_h V_{h+1}^\star)(x, a), \qquad V_h^\star(x) = \max_{a \in \mathcal{A}} Q_h^\star(x, a), \qquad V_{H+1}^\star(x) = 0. \tag{2.2}$$

Thus, the optimal policy $\pi^\star$ is the greedy policy with respect to $\{Q_h^\star\}_{h \in [H]}$. Moreover, we define the Bellman optimality operator $\mathbb{T}_h^\star$ by letting

$$(\mathbb{T}_h^\star Q)(x, a) = r(x, a) + (\mathbb{P}_h V)(x, a) \qquad \text{for all } Q \colon \mathcal{S} \times \mathcal{A} \to \mathbb{R},$$

where $V(x) = \max_{a \in \mathcal{A}} Q(x, a)$. By definition, the Bellman equation in (2.2) is equivalent to $Q_h^\star = \mathbb{T}_h^\star Q_{h+1}^\star, \forall h \in [H]$. The goal of the agent is to learn the optimal policy $\pi^\star$. For any policy $\pi$, the difference between $V_1^\pi$ and $V_1^\star$ quantifies the sub-optimality of $\pi$. Thus, for a fixed integer $T > 0$, after playing for $T$ episodes, the total (expected) regret [11] of the agent is defined as

$$\text{Regret}(T) = \sum_{t=1}^{T}\left[V_1^\star(x_1^t) - V_1^{\pi^t}(x_1^t)\right],$$

where $\pi^t$ is the policy executed in the $t$-th episode and $x_1^t$ is the initial state.

# 3  Optimistic Least-Squares Value Iteration Algorithms

In this section, we introduce the optimistic least-squares value iteration algorithm where the action-value functions are estimated using a class of functions defined on $\mathcal{Z} = \mathcal{S} \times \mathcal{A}$. The value iteration algorithm [53, 66] is one of the most classical method in reinforcement learning, which finds

$\{Q_h^\star\}_{h\in[H]}$ by applying the Bellman equation in (2.2) recursively. Specifically, value iteration constructs a sequence of action-value functions $\{Q_h\}_{h\in[H]}$ via

$$Q_h(x,a) \leftarrow (\mathbb{T}_h^\star Q_{h+1}) = \big[r_h + \mathbb{P}_h V_{h+1}\big](x,a), \qquad (3.1)$$
$$V_{h+1}(x) \leftarrow \max_{a'\in\mathcal{A}} Q_{h+1}(x,a'), \qquad \forall (x,a) \in \mathcal{S} \times \mathcal{A}, \forall h \in [H],$$

where $Q_{H+1}$ is set to be the zero function. However, this algorithm is impractical to implement in real-world RL problems due to the following two reasons: (i) the transition kernel $\mathbb{P}_h$ is unknown and (ii) we can neither iterate over all state-action pairs nor store a table of size $|\mathcal{S} \times \mathcal{A}|$ when the number of states is large. To tackle these challenges, the least-squares value iteration [10, 52] algorithm implements the update in (3.1) approximately by solving a least-squares regression problem based on historical data, which consists of the trajectories generated by the RL agent in previous episodes. Specifically, let $\mathcal{F}$ be a function class. Before the beginning of the $t$-th episode, we have observed $t-1$ transition tuples $\{(x_h^\tau, a_h^\tau, x_{h+1}^\tau)\}_{\tau\in[n]}$. Then, for estimating $Q_h^\star$, LSVI proposes to replace (3.1) with a least-squares regression problem

$$\widehat{Q}_h^t \leftarrow \underset{f\in\mathcal{F}}{\text{minimize}} \left\{ \sum_{\tau=1}^{t-1} \big[ r_h(x_h^\tau, a_h^\tau) + V_{h+1}^t(x_{h+1}^\tau) - f(x_h^\tau, a_h^\tau) \big]^2 + \text{pen}(f) \right\}, \qquad (3.2)$$

where $\text{pen}(f)$ is a regularization term. Moreover, to foster exploration, following the principle of optimism in the face of uncertainty [66], we further incorporate a bonus function $b_h^t \colon \mathcal{Z} \to \mathbb{R}$ and define

$$Q_h^t(\cdot,\cdot) = \min\big\{ \widehat{Q}_h^t(\cdot,\cdot) + \beta \cdot b_h^t(\cdot,\cdot), H - h + 1 \big\}^+, \qquad V_h^t(\cdot) = \max_{a\in\mathcal{A}} Q_h^t(\cdot,a), \qquad (3.3)$$

where $\beta > 0$ is a parameter and $\min\{\cdot, H-h+1\}^+$ denotes the truncation to the interval $[0, H-h-1]$. Here we truncate the value function to $[0, H - h + 1]$ as each reward function is bounded in $[0, 1]$. Then, in the $t$-the episode, we let $\pi^t$ be the greedy policy with respect to $\{Q_h^t\}_{h\in[H]}$ and execute $\pi^t$. Hence, combining (3.2) and (3.3) yields the optimistic least-squares value iteration algorithm, whose details are given in Algorithm 1.

---

**Algorithm 1** Optimistic Least-Squares Value Iteration with Function Approximation

1: **Input:** Function class $\mathcal{F}$, penalty function $\text{pen}(\cdot)$, and parameter $\beta$.
2: **for** episode $t = 1, \dots, T$ **do**
3:     Receive the initial state $x_1^t$.
4:     Set $V_{H+1}^t$ as the zero function.
5:     **for** step $h = H, \dots, 1$ **do**
6:         Obtain $Q_h^t$ and $V_h^t$ according to (3.2) and (3.3).
7:     **end for**
8:     **for** step $h = 1, \dots, H$ **do**
9:         Take action $a_h^t \leftarrow \text{argmax}_{a\in\mathcal{A}} Q_h^t(x_h^t, a)$.
10:       Observe the reward $r_h(x_h^t, a_h^t)$ and the next state $x_{h+1}^t$.
11:     **end for**
12: **end for**

---

We note that the both the bonus function $b_h^t$ in (3.3) and the penalty function in (3.2) relies on the choice of function class $\mathcal{F}$. The optimistic LSVI in Algorithm 1 is only implementable when $\mathcal{F}$ is specified. For instance, when $\mathcal{F}$ consists of functions of linear the form $\theta^\top \phi(z)$, where $\phi \colon \mathcal{Z} \to \mathbb{R}^d$ is a known feature mapping and $\theta \in \mathbb{R}^d$ is the parameter, we choose the ridge penalty $\|\theta\|_2^2$ in (3.2) and define $b_h^t(z)$ as $[\phi(z)^\top A_h^t \phi(z)]^{1/2}$ for some invertible matrix $A_h^t$. Then, Algorithm 1 recovers the LSVI-UCB algorithm studied in [36], which further reduces to the tabular UCBVI algorithm [6] when $\phi$ is the canonical basis.

In the rest of this section, we instantiate the optimistic LSVI framework by setting $\mathcal{F}$ as an RKHS and the class of overparameterized neural networks.

## 3.1 The Kernel Setting

In the following, we consider the case where function class $\mathcal{F}$ is an RKHS $\mathcal{H}$ with kernel $K$. In this case, by setting $\mathrm{pen}(f)$ as the ridge penalty, (3.2) reduces to a kernel ridge regression problem. Besides, we define $b_h^t$ in (3.3) as the UCB bonus function that also appears in kernelized contextual bandit [62, 67, 18, 28, 78, 58, 14]. With these two modifications, we obtain the Kernel Optimistic Least-Squares Value Iteration (KOVI) algorithm, which is summarized in Algorithm 2.

Specifically, for each $t \in [T]$, before the beginning of the $t$-th episode, we first obtain value functions $\{Q_h^t\}_{h \in [H]}$ by solving a sequence of kernel ridge regressions with the data obtained from the previous $t - 1$ episodes. In particular, we let $Q_{H+1}^t$ be a zero function. For any $h \in [H]$, we replace (3.2) by a kernel ridge regression given by

$$\widehat{Q}_h^t \leftarrow \underset{f \in \mathcal{H}}{\mathrm{minimize}} \sum_{\tau=1}^{t-1} \big[ r_h(x_h^\tau, a_h^\tau) + V_{h+1}^t(x_{h+1}^\tau) - f(x_h^\tau, a_h^\tau) \big]^2 + \lambda \cdot \|f\|_{\mathcal{H}}^2, \qquad (3.4)$$

where $\lambda > 0$ is the regularization parameter. Then, we obtain $Q_h^t$ and $V_h^t$ as in (3.3), where the bonus function $b_h^t$ will be specified later. That is,

$$Q_h^t(s,a) = \min \big\{ \widehat{Q}_h^t(s,a) + \beta \cdot b_h^t(s,a), H - h + 1 \big\}^+, \qquad V_h^t(s) = \max_a Q_h^t(s,a), \quad (3.5)$$

where $\beta > 0$ is a parameter.

The solution to (3.4) can be written in closed-form as follows. We define the response vector $y_h^t \in \mathbb{R}^{t-1}$ by letting its $\tau$-th entry be

$$[y_h^t]_\tau = r_h(x_h^\tau, a_h^\tau) + V_{h+1}^t(x_{h+1}^\tau), \qquad \forall \tau \in [t-1]. \qquad (3.6)$$

Recall that we denote $z = (x, a)$ and $\mathcal{Z} = \mathcal{S} \times \mathcal{A}$. Besides, based on the kernel function $K$ of the RKHS, we define the Gram matrix $K_h^t \in \mathbb{R}^{(t-1) \times (t-1)}$ and function $k_h^t \colon \mathcal{Z} \to \mathbb{R}^{t-1}$ respectively as

$$K_h^t = [K(z_h^\tau, z_h^{\tau'})]_{\tau, \tau' \in [t-1]} \in \mathbb{R}^{(t-1) \times (t-1)}, \qquad k_h^t(z) = \big[ K(z_h^1, z), \dots K(z_h^{t-1}, z) \big]^\top \in \mathbb{R}^{t-1}. \qquad (3.7)$$

Then $\widehat{Q}_h^t$ in (3.4) can be written as $\widehat{Q}_h^t(z) = k_h^t(z)^\top \alpha_h^t$, where we define $\alpha_h^t = (K_h^t + \lambda \cdot I)^{-1} y_h^t$.

Using $K_h^t$ and $k_h^t$ defined in (3.7), the bonus function is defined as

$$b_h^t(x,a) = \lambda^{-1/2} \cdot \big[ K(z,z) - k_h^t(z)^\top (K_h^t + \lambda I)^{-1} k_h^t(z) \big]^{1/2}, \qquad (3.8)$$

which can be interpreted as the posterior variance of Gaussian process regression and characterizes the uncertainty of $\widehat{Q}_h^t$ [55]. Such a bonus term also appears in the literature on kernelized contextual bandits [62, 67, 18, 28, 78, 58, 14] and is reduced to the UCB bonus proposed for linear bandits [11, 41] when the feature mapping $\phi$ of the RKHS is finite-dimensional. In this case, KOVI reduces to the LSVI-UCB algorithm proposed in [36] for linear value functions.

Furthermore, we remark that the bonus defined in (3.8) is called the UCB bonus because, when added by such a bonus function, $Q_h^t$ defined in (3.5) serves as an upper bound of $Q_h^\star$ for all state-action pair. Intuitively, the target function of the kernel ridge regression in (3.4) is $\mathbb{T}_h^\star Q_{h+1}^t$. However, due to having limited data, the solution $\widehat{Q}_h^t$ has some estimation error, which is quantified $b_h^t$. Thus, when $\beta$ is properly chosen, the bonus term triumphs the uncertainty of estimation, which yields that $Q_h^t \geq \mathbb{T}_h^\star Q_{h+1}^t$ elementwisely. Notice that $Q_{H+1}^t = Q_{H+1}^\star = 0$. The Bellman equation $Q_h^\star = \mathbb{T}_h^\star Q_{h+1}^\star$ directly implies that $Q_h^t$ is an elementwise upper bound of $Q_h^\star$ for all $h \in [H]$. Our algorithm is called "optimistic value iteration" as the policy $\pi^t$ is greedy with respect to $\{Q_h^t\}_{h \in [H]}$, which are upper bounds of the optimal value function. In other words, compared with the standard value iteration algorithm, we always over-estimate the value function. Such an optimistic approach is pivotal for the RL agent to perform efficient temporally extended exploration.

# 4 Theory of Kernel Optimistic Least-Squares Value Iteration

In this section, we prove that KOVI achieves $\mathcal{O}(\delta_{\mathcal{H}} H^2 \sqrt{T})$-regret bounds, where $\delta_{\mathcal{H}}$ characterizes the intrinsic complexity of the RKHS $\mathcal{H}$ that is used to approximate $\{Q_h^\star\}_{h \in [H]}$. Before presenting

the theory, we first lay out a structural assumption for the kernel setting, which postulates that the Bellman operator maps any bounded value function to a bounded RKHS-norm ball.

**Assumption 4.1.** Let $R_Q > 0$ be a fixed constant. We define $\mathcal{Q}^\star = \{f \in \mathcal{H} : \|f\|_\mathcal{H} \leq R_Q H\}$. We assume that for any $h \in [H]$ and any $Q : \mathcal{S} \times \mathcal{A} \to [0, H]$, we have $\mathbb{T}_h^\star Q \in \mathcal{Q}^\star$.

Since $Q_h^\star$ is bounded by in $[0, H]$ for each all $h \in [H]$, Assumption 4.1 ensures the optimal value functions are contained in the RKHS-norm ball $\mathcal{Q}^\star$. Thus, there is no approximation bias when using functions in $\mathcal{H}$ to approximate $\{Q_h^\star\}_{h \in [H]}$. Moreover, it is shown in [23] that only assuming $\{Q_h^\star\}_{h \in [H]} \subseteq \mathcal{Q}^\star$ is not sufficient for achieving a regret that is polynomial in $H$. Thus, we further assume that $\mathcal{Q}^\star$ contains the image of the Bellman operator. A sufficient condition for Assumption 4.1 to hold is that

$$r_h(\cdot, \cdot), \ \mathbb{P}_h(x' \,|\, \cdot, \cdot) \in \{f \in \mathcal{H} : \|f\|_\mathcal{H} \leq 1\}, \qquad \forall h \in [H], \ \forall x' \in \mathcal{S}. \tag{4.1}$$

That is, both the reward function and the Markov transition kernel can be represented by functions in the unit ball of $\mathcal{H}$. When (4.1) holds, for any $V : \mathcal{S} \to [0, H]$, it holds that $r_h + \mathbb{P}_h V \in \mathcal{H}$ with its RKHS norm bounded by $H + 1$. Hence, Assumption 4.1 holds with $R_Q = 2$. Moreover, similar assumptions are also made in [77, 78, 36, 80, 81, 73] for (generalized) linear functions. Also see [23, 68, 42] for related discussions on the necessity of such an assumption.

Moreover, as $\mathcal{Q}^\star$ contains the image of the Bellman operator, the complexity of $\mathcal{H}$ plays an important role in the performance of KOVI. To characterize the intrinsic complexity of $\mathcal{F}$, we consider a notion of effective dimension named the maximal information gain [62], which is defined as

$$\Gamma_K(T, \lambda) = \sup_{\mathcal{D} \subseteq \mathcal{Z}} \left\{ 1/2 \cdot \text{logdet}(I + K_\mathcal{D}/\lambda) \right\}, \tag{4.2}$$

where the supremum is taken over all $\mathcal{D} \subseteq \mathcal{Z}$ with $|\mathcal{D}| \leq T$. Here in (4.2) $K_\mathcal{D}$ is the Gram matrix defined in the same way as in (3.7) based on $\mathcal{D}$, $\lambda > 0$ is a parameter, and the subscript $K$ in $\Gamma_K$ indicates the kernel $K$. The magnitude of $\Gamma_K(T, \lambda)$ relies on how fast the the eigenvalues $\mathcal{H}$ decay to zero and can be viewed as a proxy of the dimension of $\mathcal{H}$ when $\mathcal{H}$ is infinite-dimensional. In the special case where $\mathcal{H}$ is finite-rank, it holds that $\Gamma_K(T, \lambda) = \mathcal{O}(\gamma \cdot \log T)$ where $\gamma$ is the rank of $\mathcal{H}$.

Furthermore, for any $h \in [H]$, note that each $Q_h^t$ constructed by KOVI takes the form of

$$Q(z) = \min \left\{ Q_0(z) + \beta \cdot \lambda^{-1/2} \big[ K(z, z) - k_\mathcal{D}(z)^\top (K_\mathcal{D} + \lambda I)^{-1} k_\mathcal{D}(z) \big]^{1/2}, H - h + 1 \right\}^+, \tag{4.3}$$

where $Q_0 \in \mathcal{H}$, similar to $\widehat{Q}_h^t$ in (3.4), is the solution to a kernel ridge regression problem and $\mathcal{D} \subseteq \mathcal{Z}$ is a discrete subset of $\mathcal{Z}$ with no more than $T$ state-action pairs. Moreover, $K_\mathcal{D}$ and $k_\mathcal{D}$ are defined similarly as in (3.7) based on data in $\mathcal{D}$. Then, for any $R, B > 0$, we define a function class $\mathcal{Q}_{\text{ucb}}(h, R, B)$ as

$$\mathcal{Q}_{\text{ucb}}(h, R, B) = \left\{ Q : Q \text{ takes the form of (4.3) with } \|Q_0\|_\mathcal{H} \leq R, \beta \in [0, B], |\mathcal{D}| \leq T \right\}. \tag{4.4}$$

As we will show in Lemma H.1, we have $\|\widehat{Q}_h^t\|_\mathcal{H} \leq R_T$ for all $(t, h) \in [T] \times [H]$, where $R_T = 2H\sqrt{\Gamma_K(T, \lambda)}$. Thus, when $B$ exceeds parameter $\beta$ in (3.5), each $Q_h^t$ is contained in $\mathcal{Q}_{\text{ucb}}(h, R_T, B)$.

Moreover, since $r_h + \mathbb{P}_h V_{h+1}^t = \mathbb{T}_h^\star Q_{h+1}^t$ is the population ground truth of the ridge regression in (3.4), the complexity of $\mathcal{Q}_{\text{ucb}}(h + 1, R_T, B)$ naturally appears when quantifying the uncertainty of $\widehat{Q}_h^t$. To this end, for any $\epsilon > 0$, let $N_\infty(\epsilon; h, B)$ be the $\epsilon$-covering number of $\mathcal{Q}_{\text{ucb}}(h, R_T, B)$ with respect to the $\ell_\infty$-norm on $\mathcal{Z}$, which is also determined by the spectral structure of $\mathcal{H}$ and characterizes the complexity of the value functions constructed by KOVI.

Now we are ready to present the regret bound of KOVI.

**Theorem 4.2.** Assume that there exists $B_T > 0$ satisfying

$$8 \cdot \Gamma_K(T, 1 + 1/T) + 8 \cdot \log N_\infty(\epsilon^*; h, B_T) + 16 \cdot \log(2TH) + 22 + 2R_Q^2 \leq (B_T/H)^2 \tag{4.5}$$

for all $h \in [H]$, where $\epsilon^* = H/T$. We set $\lambda = 1 + 1/T$ and $\beta = B_T$ in Algorithm 2. Then, under Assumption 4.1, with probability at least $1 - (T^2 H^2)^{-1}$, we have

$$\text{Regret}(T) \leq 5\beta H \cdot \sqrt{T \cdot \Gamma_K(T, \lambda)}. \tag{4.6}$$

As shown in (D.6), the regret can be written as $\mathcal{O}(H^2 \cdot \delta_{\mathcal{H}} \cdot \sqrt{T})$, where $\delta_{\mathcal{H}} = B_T/H \cdot \sqrt{\Gamma_K(T,\lambda)}$ reflects the complexity of $\mathcal{H}$ and $B_T$ satisfies (4.5). Specifically, $\delta_{\mathcal{H}}$ involves (i) the $\ell_\infty$-covering number $N_\infty(\epsilon^*, h, B_T)$ of $\mathcal{Q}_{\mathrm{ucb}}(h, R_T, B_T)$ and (ii) the effective dimension $\Gamma_K(T,\lambda)$, both characterize the intrinsic complexity of $\mathcal{H}$. Moreover, when neglecting the constant and logarithmic terms in (4.5), it suffices to choose $B_T$ satisfying

$$B_T/H \asymp \sqrt{\Gamma_K(T,\lambda)} + \max_{h \in [H]} \sqrt{\log N_\infty(\epsilon^*, h, B_T)},$$

which reduces the regret bound in (D.6) to

$$\mathrm{Regret}(T) = \widetilde{\mathcal{O}}\Big(H^2 \cdot \Big[\Gamma_K(T,\lambda) + \max_{h \in [H]} \sqrt{\Gamma_K(T,\lambda) \cdot \log N_\infty(\epsilon^*, h, B_T)}\Big] \cdot \sqrt{T}\Big). \quad (4.7)$$

To further obtain some intuition of (4.7), let us consider the tabular case where $\mathcal{Q}^*$ consists of all measurable functions defined on $\mathcal{S} \times \mathcal{A}$ with range $[0, H]$. In this case, the value function class $\mathcal{Q}_{\mathrm{ucb}}(h, R_T, B_T)$ can be set to $\mathcal{Q}^*$, whose $\ell_\infty$-covering number $N_\infty(\epsilon^*, h, B_T) \le |\mathcal{S} \times \mathcal{A}| \cdot \log T$. Moreover, it can be shown that the effective dimension is also $\mathcal{O}(|\mathcal{S} \times \mathcal{A}| \cdot \log T)$. Thus, ignoring the logarithmic terms, Theorem 4.2 implies that by choosing $\beta \asymp H \cdot |\mathcal{S} \times \mathcal{A}|$, optimistic least-squares value iteration achieves an $\widetilde{\mathcal{O}}(H^2 \cdot |\mathcal{S} \times \mathcal{A}| \cdot \sqrt{T})$ regret.

Furthermore, we remark that the regret bound in (D.6) holds for any RKHS in general. It hinges on (i) Assumption 4.1, which postulates that the RKHS-norm ball $\{f \in \mathcal{H}: \|f\|_{\mathcal{H}} \le R_Q H\}$ contains the image of the Bellman operator, and (ii) the inequality in (4.5) admits a solution $B_T$, which is set to be $\beta$ in Algorithm 2. Here we set $\beta$ to be sufficiently large as to dominate the uncertainty of $\widehat{Q}_h^t$, whereas to quantify such uncertainty, we utilize the uniform concentration over the value function class $\mathcal{Q}_{\mathrm{ucb}}(h+1, R_T, \beta)$ whose complexity metric, the $\ell_\infty$-covering number, in turn depends on $\beta$. Such an intricate desideratum leads to (4.5) which determines $\beta$ implicitly.

It is worth noting that the uniform concentration is unnecessary when $H = 1$. In this case, it suffices to choose $\beta = \widetilde{\mathcal{O}}(\sqrt{\Gamma_K(T,\lambda)})$ and KOVI incurs an $\widetilde{\mathcal{O}}(\Gamma_K(T,\lambda) \cdot \sqrt{T})$ regret, which matches the regret bounds of UCB algorithms for kernelized contextual bandits in [62, 18]. Here $\widetilde{O}(\cdot)$ omits logarithmic terms. Thus, the covering number in (4.7) is specific for MDPs and arises due to the temporal dependence within an episode.

Furthermore, to obtain a concrete regret bound from (D.6), it remains to further characterize $\Gamma_K(T,\lambda)$ and $\log N_\infty(\epsilon^*, h, B_T)$ using characteristics of $\mathcal{H}$. To this end, in the following, we specify the eigenvalue decay property of $\mathcal{H}$.

**Assumption 4.3** (Eigenvalue Decay of $\mathcal{H}$). Recall that the integral operator $T_K$ defined in (B.1) has eigenvalues $\{\sigma_j\}_{j \ge 1}$ and eigenfunctions $\{\psi_j\}_{j \ge 1}$. We assume that $\{\sigma_j\}_{j \ge 1}$ satisfies one of the following two eigenvalue decay conditions for some constant $\gamma > 0$:

  (i) $\gamma$-finite spectrum: we have $\sigma_j = 0$ for all $j > \gamma$, where $\gamma$ is a positive integer.

  (ii) $\gamma$-exponential decay: there exist absolute constants $C_1$ and $C_2$ such that $\sigma_j \le C_1 \cdot \exp(-C_2 \cdot j^\gamma)$ for all $j \ge 1$.

Moreover, for case (ii), we further assume that there exist constants $\tau \in [0, 1/2)$ $C_\psi > 0$ such that $\sup_{z \in \mathcal{Z}} \sigma_j^\tau \cdot |\psi_j(z)| \le C_\psi$ for all $j \ge 1$.

Case (i) implies that $\mathcal{H}$ is a $\gamma$-dimensional RKHS. When this is the case, under Assumption 4.1, there exists a feature mapping $\phi: \mathcal{Z} \to \mathbb{R}^\gamma$ such that, for any $V: \mathcal{S} \to [0, H]$, $r_h + \mathbb{P}_h V$ is a linear function of $\phi$. Such a property is satisfied by the linear MDP model studied in [77, 78, 36, 80]. Moreover, when $\mathcal{H}$ satisfies case (i), KOVI reduces to the LSVI-UCB algorithm studied in [36]. In addition, case (ii) postulates that the eigenvalues of $T_K$ decays exponentially fast, where $\gamma$ is a constant that might depend on the input dimension $d$, which is assumed fixed throughout this paper. For example, the squared exponential kernel belongs to case (ii) with $\gamma = 1/d$ [62]. Moreover, we assume that there exists $\tau \in [0, 1/2)$ such that $\sigma_j^\tau \cdot \|\psi_j\|_\infty$ is universally bounded. Since $K(z, z) \le 1$, this condition is naturally satisfied for $\tau = 1/2$. However, here we assume that $\tau \in (0, 1/2)$, which is satisfied when the magnitudes of the eigenvectors do grow not too fast compared with the decay of the eigenvalues.

Such a condition is significantly weaker than assuming $\|\psi_j\|_\infty$ is universally bounded, which is also commonly made in the literature of nonparametric statistics [40, 59, 82, 45, 79]. It can be shown that the squared exponential kernel on unit sphere in $\mathbb{R}^d$ satisfy this condition for any $\tau > 0$. See [46] for a more detailed discussion.

Now we present the regret bounds for the two eigenvalue decay conditions separately.

**Corollary 4.4.** Under Assumptions 4.1 and 4.3, we set $\lambda = 1 + 1/T$ and $\beta = B_T$ in Algorithm 2, where $B_T$ is defined as

$$B_T = \begin{cases} C_b \cdot \gamma H \cdot \sqrt{\log(\gamma \cdot TH)} & \gamma\text{-finite spectrum,} \\ C_b \cdot H\sqrt{\log(TH)} \cdot (\log T)^{1/\gamma} & \gamma\text{-exponential decay} \end{cases} \tag{4.8}$$

Here $C_b$ is an absolute constant that does not depend on $T$ or $H$. Then, there exists an absolute constant $C_r$ such that, with probability at least $1 - (T^2 H^2)^{-1}$, we have

$$\text{Regret}(T) \le \begin{cases} C_r \cdot H^2 \cdot \sqrt{\gamma^3 T} \cdot \log(\gamma TH) & \gamma\text{-finite spectrum,} \\ C_r \cdot H^2 \cdot \sqrt{(\log T)^{3/\gamma} \cdot T} \cdot \log(TH) & \gamma\text{-exponential decay.} \end{cases} \tag{4.9}$$

Corollary 4.4 asserts that when $\beta$ is chosen properly according to the eigenvalue decay property of $\mathcal{H}$, KOVI incurs a sublinear regret under both the two cases specified in Assumption 4.3. Note that the linear MDP [36] satisfies the $\gamma$-finite spectrum condition and KOVI recovers the LSVI-UCB algorithm studied in [36] when restricted to this setting. Moreover, our $\widetilde{\mathcal{O}}(H^2 \cdot \sqrt{\gamma^3 T})$ also matches the regret bound in [36]. In addition, under the $\gamma$-exponential eigenvalue decay condition, as we will show in §I, the log-covering number and the effective dimension are bounded by $(\log T)^{1+2/\gamma}$ and $(\log T)^{1+1/\gamma}$, respectively. Plugging these facts into (4.7), we obtain the sublinear regret in (D.6). As a concrete example, for the squared exponential kernel, we obtain an $\mathcal{O}(H^2 \cdot (\log T)^{1+1.5d} \cdot \sqrt{T})$ regret, where $d$ is the input dimension. This such a regret is $(\log T)^{d/2}$ worse than that in [62] for kernel contextual bandits, which is due to bounding the log-covering number. See §G.1 for details.

Furthermore, similarly to the discussion in Section 3.1 of [35], the regret bound in (D.6) directly translates to an upper bound on the sample complexity as follows. When the initial state is fixed for all episodes, for any fixed $\epsilon > 0$, with at least a constant probability, KOVI returns a policy $\pi$ satisfying $V_1^\star(x_1) - V_1^\pi(x_1) \le \epsilon$ using $\mathcal{O}(H^4 B_T^2 \cdot \Gamma_K(T, \lambda)/\epsilon^2)$ samples. Specifically, for the two cases considered in Assumption 4.3, such a sample complexity guarantee reduces to $\widetilde{\mathcal{O}}(H^4 \cdot \gamma^3/\epsilon^2)$ and $\widetilde{\mathcal{O}}(H^4 \cdot (\log T)^{2+3/\gamma}/\epsilon^2)$, respectively. Moreover, similar to [36], our analysis can also be extended to the misspecified setting where $\inf_{f \in \mathcal{Q}^\star} \|f - \mathcal{T}_h^\star Q\|_\infty \le \text{err}_{\text{mis}}$ for all $Q: \mathcal{Z} \to [0, H]$. Here $\text{err}_{\text{mis}}$ is the model misspecification error. Under this setting, KOVI will suffer from an extra $\text{err}_{\text{mis}} \cdot TH$ regret. The analysis for the misspecified setting is similar to that for the neural setting that will be presented in §D.

## 5 Conclusion

In this paper, we have presented an algorithmic framework for reinforcement learning with general function approximation. Such a framework is based on an optimistic least-squares value iteration algorithm that incorporates an additional bonus term in the solution to a least-squares value estimation problem. The bonus term promotes exploration. When deploying this framework in the settings of kernel function and overparameterized neural networks, respectively, we obtain two algorithms KOVI and NOVI. Both algorithms are provably efficient, both computationally and in terms of the number of samples. Specifically, under the kernel and neural network settings respectively, KOVI and NOVI both achieve sublinear regret, $\widetilde{O}(\delta_{\mathcal{F}} H^2 \sqrt{T})$, where $\delta_{\mathcal{F}}$ is a quantity that characterizes the intrinsic complexity of the function class $\mathcal{F}$. To the best of our knowledge, this is the first provably efficient reinforcement learning algorithm in the general settings of kernel and neural function approximations.

## Broader Impact

This is a theoretical paper. We do not foresee our work directly having any societal consequences. However, reinforcement learning is a tool that is increasingly used in practical machine learning applications, especially in the setting where nonlinear function approximation is involved. Theoretical explorations related to reinforcement learning with function approximation may help provide frameworks through which to reason about, and design safer and more reliable practical systems.

## Acknowledgements

We would like to thank the Simons Institute for the Theory of Computing in Berkeley, where this project was initiated. Zhuoran Yang would like to thank Jianqing Fan, Csaba Szepsvári, Tuo Zhao, Simon Shaolei Du, Ruosong Wang, and Yiping Lu for valuable discussions. Mengdi Wang gratefully acknowledges funding from the U.S. National Science Foundation (NSF) grant CMMI1653435, Air Force Office of Scientific Research (AFOSR) grant FA9550-19-1-020, and C3.ai DTI. Michael Jordan gratefully acknowledges funding from the Mathematical Data Science program of the Office of Naval Research under grant number N00014-18-1-2764.

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
