[Supplementary Material 1]

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

 without having access to a generative model or a explorative behavioral policy. The tabular setting is well studied the existing works. See, e.g., [33, 52, 6, 21, 65, 35, 56] and the references therein. It is shown in [6, 35] that any RL algorithm necessarily incurs a $\Omega(\sqrt{SAT})$ regret under the tabular setting, where $S$ and $A$ are the cardinalities of the state and action spaces, respectively. Thus, the algorithms designed for the tabular setting cannot be directly applied to the function approximation setting where the number of states is gigantic. When function approximation is employed, [77, 78, 36, 12, 80, 73, 5, 83, 37] focus on the (generalized) linear setting where the value function (or the transition model) can be represented using a linear transform of a known feature mapping. Among these works, our work is most related to [36]. In particular, in our kernel setting, when kernel function has a finite rank, both our LSVI algorithm and the corresponding regret bound are reduced to the those established in [36]. However, their sample complexity or regret bounds all diverge when the dimension of the feature mapping goes to infinity and thus cannot be directly extended to the kernel setting. Another closely related work is [71], which studies a similar optimistic LSVI algorithm for general function approximation. Their work focuses on value function classes with bounded eluder dimensions [57, 51] and it is unclear whether their construction of the bonus function can be extended to the kernel or neural settings. Besides, [78] also study a kernelized MDP model where the transition model can be directly estimated. Under a slightly more general model, [5] recently propose an optimistic model-based algorithm via value-targeted regression, where the model class is allowed to be general functions with bounded eluder dimension. In another recent work, [37] study a nonlinear control problem where the system dynamics belongs to a known RKHS and can be directly estimated from the data. As opposed to these works, we do not pose an explicit assumption on the transition model and our proposed algorithm is model-free. Furthermore, regret or sample complexity results have also been studied beyond linear function approximation. However, these algorithms are either computational challenging [39, 34, 20, 22] or require additional assumptions on the transition model that might be restrictive [74, 75, 24]

In addition, our work is also related to the literature on contextual bandits with kernel or [62, 38, 63, 67, 18, 28] neural network functions [84], which are special cases of our episodic MDP with the episode length equal to one. The construction of our bonus function are adopted from these works. However, our reinforcement learning problem has temporal dependence caused by state transitions according to the Markov transition kernel, which is absent in bandit models. Specifically, the covering number $N_\infty(\epsilon^*)$ in Table 1 arises due to such an additional structure captures the fundamental challenge of temporally extended exploration in RL. When applying our algorithm to kernel contextual bandits, the regret bound reduces to $d_{\mathrm{eff}} \cdot \sqrt{T}$ where $d_{\mathrm{eff}}$ is the effective dimension of the RKHS. Such a regret bound matches those in [62, 18].

Furthermore, our analysis of the optimistic LSVI algorithm is akin to the recent study of the optimization and generalization of over-parameterized neural networks via the framework of the neural tangent kernel [32]. Most of these works focus on the supervised learning [19, 32, 76, 25, 26, 3, 2, 85, 17, 44, 4, 15, 16, 43]. In contrast, our algorithm incorporates an additional bonus term in the least-squares problem and thus requires novel analysis.

# B  Additional Background

In this section, we present the background of reproducing kernel Hilbert space and overparameterized neural networks.

## B.1  Reproducing Kernel Hilbert Space

In the next section, we aim to estimate the optimal value function $Q_h^\star$ using functions in a reproducing kernel Hilbert space (RKHS) [31]. To this end, hereafter, to simplify the notation, we let $z = (x, a)$ denote a state-action pair and denote $\mathcal{Z} = \mathcal{S} \times \mathcal{A}$. Without loss of generality, we regard $\mathcal{Z}$ as

a compact subset of $\mathbb{R}^d$ where the dimension $d$ is assumed fixed. This can be achieved if there exists a known embedding mapping $\psi_{\text{embed}} \colon \mathcal{Z} \to \mathbb{R}^d$ that pre-processes the input $(x, a)$. Let $\mathcal{H}$ be an RKHS defined on $\mathcal{Z}$ with kernel function $K \colon \mathcal{Z} \times \mathcal{Z} \to \mathbb{R}$, which contains a family of functions defined on $\mathcal{Z}$. Let $\langle \cdot, \cdot \rangle_{\mathcal{H}} \colon \mathcal{H} \times \mathcal{H} \to \mathbb{R}$ and $\| \cdot \|_{\mathcal{H}} \colon \mathcal{H} \to \mathbb{R}$ denote the inner product and RKHS norm on $\mathcal{H}$, respectively. Since $\mathcal{H}$ is an RKHS, there exists a feature mapping $\phi \colon \mathcal{Z} \to \mathcal{H}$ such that $f(z) = \langle f(\cdot), \phi(z) \rangle_{\mathcal{H}}$ for all $f \in \mathcal{H}$ and all $z \in \mathcal{Z}$. Moreover, for any $x, y \in \mathcal{Z}$, we have $K(x, y) = \langle \phi(x), \phi(y) \rangle_{\mathcal{H}}$. In this work, we assume that the kernel function $K$ is uniformly bounded in the sense that $\sup_{z \in \mathcal{Z}} K(z, z) < \infty$. Without loss of generality, we assume that $\sup_{z \in \mathcal{Z}} K(z, z) \le 1$, which implies that $\|\phi(z)\|_{\mathcal{H}} \le 1$ for all $z \in \mathcal{Z}$.

Furthermore, let $\mathcal{L}^2(\mathcal{Z})$ be the space of square-integrable functions on $\mathcal{Z}$ with respect to the Lebesgue measure and let $\langle \cdot, \cdot \rangle_{\mathcal{L}^2}$ be the inner product on $\mathcal{L}^2(\mathcal{Z})$. The kernel function $K$ induces a integral operator $T_K \colon \mathcal{L}^2(\mathcal{Z}) \to \mathcal{L}^2(\mathcal{Z})$ defined as

$$T_K f(z) = \int_{\mathcal{Z}} K(z, z') \cdot f(z') \, \mathrm{d}z', \qquad \forall f \in \mathcal{L}^2(\mathcal{Z}). \tag{B.1}$$

By Mercer's Theorem [64], the integral operator $T_K$ has countable and positive eigenvalues $\{\sigma_i\}_{i \ge 1}$ and the corresponding eigenfunctions $\{\psi_i\}_{i \ge 1}$ form an orthonormal basis of $\mathcal{L}^2(\mathcal{Z})$. Moreover, the kernel function admits a spectral expansion

$$K(z, z') = \sum_{i=1}^{\infty} \sigma_i \cdot \psi_i(z) \cdot \psi_j(z'). \tag{B.2}$$

Then, the RKHS $\mathcal{H}$ can be written as a subset of $\mathcal{L}^2(\mathcal{Z})$ as

$$\mathcal{H} = \left\{ f \in \mathcal{L}^2(\mathcal{Z}) \colon \sum_{i=1}^{\infty} \frac{\langle f, \psi_i \rangle_{\mathcal{L}^2}^2}{\sigma_i} < \infty \right\},$$

and the inner product of $\mathcal{H}$ can be written as

$$\langle f, g \rangle_{\mathcal{H}} = \sum_{i=1}^{\infty} 1/\sigma_i \cdot \langle f, \psi_i \rangle_{\mathcal{L}^2} \cdot \langle g, \psi_i \rangle_{\mathcal{L}^2}, \qquad \text{for all} \quad f, g \in \mathcal{H}.$$

By such a construction, the scaled eigenfunctions $\{\sqrt{\sigma_i} \psi_i\}_{i \ge 1}$ form an orthogonal basis of RKHS $\mathcal{H}$ and the feature mapping $\phi(z) \in \mathcal{H}$ can be written as $\phi(z) = \sum_{i=1}^{\infty} \sigma_i \psi_i(z) \cdot \psi_i$ for any $z \in \mathcal{Z}$.

## B.2 Overparameterized Neural Networks

In addition to RKHS, we also study the setting where the value functions are approximated by overparameterized neural networks. In the sequel, we define the class of neural networks that will be used in the algorithm.

Recall that we denote $\mathcal{Z} = \mathcal{S} \times \mathcal{A}$ and view it as a subset of $\mathbb{R}^d$. For neural networks, we further regard $\mathcal{Z}$ as a subset of the unit sphere in $\mathbb{R}^d$. That is, $\|z\|_2 = 1$ for all $z = (x, a) \in \mathcal{Z}$. A two-layer neural network $f(\cdot; b, W) \colon \mathcal{Z} \to \mathbb{R}$ with $2m$ neurons and weights $(b, W)$ is defined as

$$f(z; b, W) = \frac{1}{\sqrt{2m}} \sum_{j=1}^{2m} b_j \cdot \mathrm{act}(W_j^{\top} z), \qquad \forall z \in \mathcal{Z}. \tag{B.3}$$

Here $\mathrm{act} \colon \mathbb{R} \to \mathbb{R}$ is the activation function, $b_j \in \mathbb{R}$ and $W_j \in \mathbb{R}^d$ for all $j \in [2m]$, and $b = (b_1, \ldots, b_{2m})^{\top} \in \mathbb{R}^{2m}$ and $W = (W_1, \ldots, W_{2m}) \in \mathbb{R}^{2dm}$. During training, we initialize $(b, W)$ via the symmetric initialization scheme [30, 9] as follows. For any $j \in [m]$, we set $b_j \stackrel{\text{i.i.d.}}{\sim} \mathrm{Unif}(\{-1, 1\})$ and $W_j \stackrel{\text{i.i.d.}}{\sim} N(0, I_d/d)$, where $I_d$ is the identity matrix in $\mathbb{R}^d$. For any $j \in \{m+1, \ldots, 2m\}$, we set $b_j = -b_{j-m}$ and $W_j = W_{j-m}$. We remark that such an initialization implies that the initial neural network is a zero function, which is used only to simply the theoretical analysis. Besides, for ease of presentation, during training we fix $b$ at its initial value and only optimize over $W$. Moreover, we denote $f(z; b, W)$ by $f(z; W)$ to simplify the notation.

Furthermore, we assume that the neural network in is overparameterized in the sense that the width $2m$ is much larger than the number of episodes $T$. Overparameterization is shown to be pivotal for

neural training in both theory and practice [49, 2, 4]. Under the such a regime, the dynamics of training neural networks are well captured by the framework of neural tangent kernel (NTK) [32]. Specifically, let $\varphi(\cdot; W)\colon \mathcal{Z} \to \mathbb{R}^{2md}$ be the gradient of $f(; W)$ with respect to $W$, which is given by

$$\varphi(z; W) = \nabla_W f(z; W) = \big(\nabla_{W_1} f(z; W), \dots, \nabla_{W_{2m}} f(z; W)\big), \qquad \forall z \in \mathcal{Z}. \tag{B.4}$$

Let $W^{(0)}$ be the initial value of $W$. Condition on the realization of $W^{(0)}$, we define a kernel matrix $K_m\colon \mathcal{Z} \to \mathcal{Z}$ as

$$K_m(z, z') = \big\langle \varphi(z; W^{(0)}), \varphi(z'; W^{(0)}) \big\rangle, \qquad \forall (z, z') \in \mathcal{Z} \times \mathcal{Z}. \tag{B.5}$$

When $m$ is sufficiently large, for all $W$ that is in a neighborhood of $W^{(0)}$, it can be shown that $f(\cdot, W)$ is close to its linearization at $W^{(0)}$,

$$f(\cdot; W) \approx \widehat{f}(\cdot; W) = f(\cdot, W^{(0)}) + \big\langle \phi(\cdot; W^{(0)}), W - W^{(0)} \big\rangle = \big\langle \phi(\cdot; W^{(0)}), W - W^{(0)} \big\rangle. \tag{B.6}$$

The linearized function $\widehat{f}(\cdot; W)$ belongs to the RKHS with kernel $K_m$. Moreover, as $m$ goes to infinity, due to random initialization, $K_m$ converges to a kernel $K_{\mathrm{ntk}}\colon \mathcal{Z} \times \mathcal{Z}$, dubbed as neural tangent kernel (NTK), which is given by

$$K_{\mathrm{ntk}}(z, z') = \mathbb{E}\big[\mathrm{act}'(w^\top z) \cdot \mathrm{act}'(w^\top z') \cdot z^\top z'\big], \qquad (z, z') \in \mathcal{Z} \times \mathcal{Z}, \tag{B.7}$$

where $\mathrm{act}'$ is the derivative of the activation function, and the expectation in (B.7) is taken with respect to $w \sim N(0, I_d/d)$.

## C   Kernel and Neural Optimistic Least-Squares Value Iteration

In this section, we lay out the details of KOVI and NOVI, which are omitted for brevity. We remark that the loss function $L_h^t$ in Line 7 of Algorithm 4 is given in (C.1) and its global minimizer $\widehat{W}_h^t$ can be efficiently obtained by first-order optimization methods.

---

**Algorithm 2** Kernelized Optimistic Least-Squares Value Iteration (KOVI)

---

1: **Input:** Parameters $\lambda$ and $\beta$.
2: **for** episode $t = 1, \dots, T$ **do**
3:     Receive the initial state $x_1^t$.
4:     Set $V_{H+1}^t$ as the zero function.
5:     **for** step $h = H, \dots, 1$ **do**
6:         Compute the response $y_h^t \in \mathbb{R}^{t-1}$, the Gram matrix $K_h^t \in \mathbb{R}^{(t-1)\times(t-1)}$, and function $k_h^t$
            as in (3.6) and (3.7), respectively.
7:         Compute
8:             $\alpha_h^t = (K_h^t + \lambda \cdot I)^{-1} y_h^t,$
9:             $b_h^t(\cdot, \cdot) = \lambda^{-1/2} \cdot \big[K(\cdot, \cdot; \cdot, \cdot) - k_h^t(\cdot, \cdot)^\top (K_h^t + \lambda I)^{-1} k_h^t(\cdot, \cdot)\big]^{1/2}.$
10:        Obtain value functions

$$Q_h^t(\cdot, \cdot) \leftarrow \min\{k_h^t(\cdot, \cdot)^\top \alpha_h^t + \beta \cdot b_h^t(\cdot, \cdot), H - h + 1\}^+, \qquad V_h^t(\cdot) = \max_a Q_h^t(\cdot, a).$$

11:    **end for**
12:    **for** step $h = 1, \dots, H$ **do**
13:        Take action $a_h^t \leftarrow \mathrm{argmax}_{a \in \mathcal{A}} Q_h^t(x_h^t, a).$
14:        Observe the reward $r_h(x_h^t, a_h^t)$ and the next state $x_{h+1}^t$.
15:    **end for**
16: **end for**

---

### C.1   Neural Optimistic Value Iteration

In this subsection, we estimate the value functions $\{Q_h^\star\}_{h \in [H]}$ using overparameterized neural networks. We aim to estimate each $Q_h^\star$ using a neural network given in (B.3), which is initialized via the symmetric initialization scheme [30, 9] introduced in §B.2. Moreover, for simplicity, we assume

---
**Algorithm 3** Neural Optimistic Least-Squares Value Iteration (NOVI)
---
1: **Input:** Parameters $\lambda$ and $\beta$.
2: Initialize the network weights $(b^{(0)}, W^{(0)})$ via the symmetric initialization scheme.
3: **for** episode $t = 1, \ldots, T$ **do**
4:      Receive the initial state $x_1^t$.
5:      Set $V_{H+1}^t$ as the zero function.
6:      **for** step $h = H, \ldots, 1$ **do**
7:          Solve the neural network optimization problem $\widehat{W}_h^t = \operatorname{argmin}_W L_h^t(W)$.
8:          Update $\Lambda_h^t = \Lambda_h^{t-1} + \varphi(x_h^{t-1}, a_h^{t-1}; \widehat{W}_h^t)\varphi(x_h^t, a_h^t; \widehat{W}_h^t)^\top$.
9:          Obtain the bonus function $b_h^t$ defined in (C.4).
10:          Obtain value functions

$$Q_h^t(\cdot, \cdot) \leftarrow \min\big\{ f(\cdot, \cdot; \widehat{W}_h^t) + \beta \cdot b_h^t(\cdot, \cdot), H - h + 1 \big\}^+, \qquad V_h^t(\cdot) = \max_a Q_h^t(\cdot, a).$$

11:      **end for**
12:      **for** step $h = 1, \ldots, H$ **do**
13:          Take action $a_h^t \leftarrow \operatorname{argmax}_{a \in \mathcal{A}} Q_h^t(x_h^t, a)$.
14:          Observe the reward $r_h(x_h^t, a_h^t)$ and the next state $x_{h+1}^t$.
15:      **end for**
16: **end for**
---

that all the neural networks share the same initial weights, denoted by $(b^{(0)}, W^{(0)})$. Besides, we fix $b = b^{(0)}$ in (B.3) and only update the value of $W \in \mathbb{R}^{2md}$.

Under such a neural setting, we replace the least-squares regression in (3.2) by a nonlinear ridge regression. In particular, for any $(t, h) \in [T] \times [H]$, we define the loss function $L_h^t : \mathbb{R}^{2md} \to \mathbb{R}$ as

$$L_h^t(W) = \sum_{\tau=1}^{t-1} \big[ r_h(x_h^\tau, a_h^\tau) + V_{h+1}^t(x_{h+1}^\tau) - f(x_h^\tau, a_h^\tau; W) \big]^2 + \lambda \cdot \big\| W - W^{(0)} \big\|_2^2, \qquad (\text{C.1})$$

where $\lambda > 0$ is the regularization parameter. Then we define $\widehat{Q}_h^t$ as

$$\widehat{Q}_h^t(\cdot, \cdot) = f(\cdot, \cdot; \widehat{W}_h^t), \qquad \text{where} \qquad \widehat{W}_h^t = \operatorname*{argmin}_{W \in \mathbb{R}^{2md}} L_h^t(W). \qquad (\text{C.2})$$

Here we assume that there is an optimization oracle that returns the global minimizer of the loss function $L_h^t$. It has been shown in a large body of literature that, when $m$ is sufficiently large, with random initialization, simple optimization methods such as gradient descent provably find the global minimizer of the empirical loss function at a linear rate of convergence [26, 25, 4]. Thus, such an optimization oracle can be realized by gradient descent with sufficiently large number of iterations and the computational cost of realizing such a oracle is polynomial in $m$, $T$, and $H$.

It remains to construct the bonus function $b_h^t$. Recall that we define $\varphi(\cdot; W) = \nabla_W f(\cdot; W)$ in (B.4). We define matrix $\Lambda_h^t \in \mathbb{R}^{2md \times 2md}$ as

$$\Lambda_h^t = \lambda \cdot I_{2md} + \sum_{\tau=1}^{t-1} \varphi(x_h^\tau, a_h^\tau; \widehat{W}_h^{\tau+1}) \varphi(x_h^\tau, a_h^\tau; \widehat{W}_h^{\tau+1})^\top, \qquad (\text{C.3})$$

which can be recursively computed by letting

$$\Lambda_h^1 = \lambda \cdot I_{2md}, \qquad \Lambda_h^t = \Lambda_h^{t-1} + \varphi(x_h^{t-1}, a_h^{t-1}; \widehat{W}_h^t) \varphi(x_h^{t-1}, a_h^{t-1}; \widehat{W}_h^t)^\top, \qquad \forall t \geq 2.$$

Then the bonus function $b_h^t$ is defined as

$$b_h^t(x, a) = \big[ \varphi(x, a; \widehat{W}_h^t)^\top (\Lambda_h^t)^{-1} \varphi(x, a; \widehat{W}_h^t) \big]^{1/2}, \qquad \forall (x, a) \in \mathcal{S} \times \mathcal{A}. \qquad (\text{C.4})$$

Finally, we obtain the value functions $Q_h^t$ and $V_h^t$ via (3.5), with $\widehat{Q}_h^t$ and $b_h^t$ defined in (C.2) and (C.4), respectively. By letting $\pi^t$ be the greedy policy with respect to $\{Q_h^t\}_{h \in [H]}$, we obtain the Neural Optimistic Least-Squares Value Iteration (NOVI) algorithm, whose details are stated in Algorithm 4 in §F.

The intuition of the bonus term in (C.4) can be understood via the connection between overparameterized neural networks and NTK. Specifically, when $m$ is sufficiently large, it can be shown that each $\widehat{W}_h^t$ is not far from the initial value $W^{(0)}$. When this is the case, suppose we replace the neural tangent features $\{\varphi(\cdot; \widehat{W}_h^\tau)\}_{\tau \in [T]}$ in (C.3) and (C.4) by $\varphi(\cdot; W^{(0)})$, then $b_h^t$ recovers the UCB bonus in linear contextual bandits and linear MDPs with feature mapping $\varphi(\cdot; W^{(0)})$ [1, 36, 73]. Moreover, when $m$ converges to infinity, it will become the UCB bonus defined in (3.8) for the RKHS setting with the kernel being $K_{\text{ntk}}$. Thus, when the neural networks are overparameterized, value functions $\{Q_h^t\}_{h \in [H]}$ are approximately elementwise upper bounds of the optimal value functions and thus we achieve optimism approximately.

## D    Theory of Neural Optimistic Least-Squares Value Iteration

In this section, we establish the regret of NOVI. Throughout this subsection, we let $\mathcal{H}$ be the RKHS whose kernel function is $K_{\text{ntk}}$ define in (B.7). Also recall that we regard $\mathcal{Z} = \mathcal{S} \times \mathcal{A}$ as a subset of the unit sphere $\mathbb{S}^{d-1} = \{z \in \mathbb{R}^d \colon \|z\|_2 = 1\}$. Moreover, let $(b^{(0)}, W^{(0)})$ be the initial value of the network weights obtained via the symmetric initialization scheme introduced in §B.2. Conditioning on the randomness of the initialization, we define a finite-rank kernel $K_m \colon \mathcal{Z} \times \mathcal{Z} \to \mathbb{R}$ by letting $K_m(z, z') = \langle \nabla_W f(z; b^{(0)}, W^{(0)}), \nabla_W f(z'; b^{(0)}, W^{(0)}) \rangle$. Notice that the rank of $K_m$ is $md$, where $m$ is much larger than $T$ and $H$ and is allowed to increase to infinity. Besides, with a slight abuse of notation, we define

$$\mathcal{Q}^\star = \left\{ f_\alpha(z) = \int_{\mathbb{R}^d} \text{act}'(w^\top z) \cdot z^\top \alpha(w) \, dp_0(w) \colon \alpha \colon \mathbb{R}^d \to \mathbb{R}^d, \|\alpha\|_{2,\infty} \leq R_Q H / \sqrt{d} \right\}, \quad \text{(D.1)}$$

where $R_Q$ is a positive number, $p_0$ is the density of $N(0, I_d/d)$, and we define $\|\alpha\|_{2,\infty} = \sup_w \|\alpha(w)\|_2$. That is, $\mathcal{Q}^\star$ consists of functions that can be expressed as infinite number of random features. As shown in Lemma C.1 of [30], $\mathcal{Q}^\star$ is a dense subset of the RKHS $\mathcal{H}$. Thus, when $R_Q$ is sufficiently large, $\mathcal{Q}^\star$ in (D.1) is an expressive function class. We impose the following condition on $\mathcal{Q}^\star$.

**Assumption D.1.** We assume that for any $h \in [H]$ and any $Q \colon \mathcal{S} \times \mathcal{A} \to [0, H]$, we have $\mathbb{T}_h^\star Q \in \mathcal{Q}^\star$.

Assumption D.1 is in the same vein as Assumption 4.1. Here we focus on $\mathcal{Q}^*$ instead of an RKHS norm ball of NTK only due to technical considerations. However, since functions of the form in (D.1) are dense in $\mathcal{H}$, Assumptions D.1 and 4.1 are indeed very similar.

To characterize the value function class associated with NOVI, for any discrete set $\mathcal{D} \subseteq \mathcal{Z}$, similar to (C.3), we define

$$\overline{\Lambda}_\mathcal{D} = \lambda \cdot I_{2md} + \sum_{z \in \mathcal{D}} \varphi(z; W^{(0)}) \varphi(z; W^{(0)})^\top,$$

where $\varphi(\cdot; W^{(0)})$ is the neural tangent feature defined in (B.4). With a slight abuse of notation, for any $R, B > 0$, we let $\mathcal{Q}_{\text{ucb}}(h, R, B)$ denote that class of functions that take the form of

$$Q(z) = \min \left\{ \langle \varphi(z; W^{(0)}), W \rangle + \beta \cdot \left[ \varphi(z; W^{(0)})^\top (\overline{\Lambda}_\mathcal{D})^{-1} \varphi(z; W^{(0)}) \right]^{1/2}, H - h + 1 \right\}^+,$$
$$\text{(D.2)}$$

where $W \in \mathbb{R}^{2md}$ satisfies $\|W\|_2 \leq R$, $\beta \in [0, B]$, and $\mathcal{D}$ has cardinality no more than $T$. Intuitively, when both $R$ and $B$ are sufficiently large, $\mathcal{Q}_{\text{ucb}}(h, R, B)$ contains the counterpart of neural-based value function $Q_h^t$ that is based on neural tangent features. When $m$ is sufficiently large, it is expected that $Q_h^t$ is well-approximately by functions in $\mathcal{Q}_{\text{ucb}}(h, R, B)$ where the approximation error decays with $m$. It is worth noting the class of linear functions of $\varphi(\cdot; W^{(0)})$ forms an RKHS with kernel $K_m$ in (B.5). Any function $f$ in this class can be written as $f(\cdot) = \langle \varphi(\cdot; W^{(0)}), W_f \rangle$ for some $W_f \in \mathbb{R}^{2md}$. Moreover, the RKHS norm of $f$ is given by $\|W_f\|_2$. Thus, $\mathcal{Q}_{\text{ucb}}(h, R, B)$ defined above coincides with the counterpart defined in (4.4) with the kernel function being $K_m$. We set $R_T = H\sqrt{2T/\lambda}$ and let $N_\infty(\epsilon; h, B)$ denote the $\epsilon$-covering number of $\mathcal{Q}_{\text{ucb}}(h, R_T, B)$ with respect to the $\ell_\infty$-norm on $\mathcal{Z}$.

In the following theorem, we present a general regret bound for NOVI.

**Theorem D.2.** Under Assumptions D.1, We also assume that $m$ is sufficiently large such that $m = \Omega(T^{13}H^{14} \cdot (\log m)^3)$. In Algorithm 4, we let $\lambda$ be a sufficiently large constant and let $\beta = B_T$ which satisfies inequality

$$16\Gamma_{K_m}(T, \lambda) + 16 \cdot \log N_\infty(\epsilon^*, h+1, B_T) + 32 \cdot \log(2TH) + 4R_Q^2 \cdot (1 + \lambda/d) \le (B_T/H)^2 \tag{D.3}$$

for all $h \in [H]$. Here $\epsilon^* = H/T$ and $\Gamma_{K_m}(T, \lambda)$ is the maximal information gain defined for kernel $K_m$. In addition, for the neural network in (B.3), we assume the activation function act is $C_{\text{act}}$-smooth, i.e., its derivative $\text{act}'$ is $C_{\text{act}}$-Lipschitz, and $m$ is sufficiently large such that

$$m = \Omega\big(\beta^{12} \cdot T^{13} \cdot H^{14} \cdot (\log m)^3\big). \tag{D.4}$$

Then with probability at least $1 - (T^2H^2)^{-1}$, we have

$$\text{Regret}(T) = 5\beta H \cdot \sqrt{T \cdot \Gamma_{K_m}(T, \lambda)} + 10\beta TH \cdot \iota, \tag{D.5}$$

where we define $\iota = T^{7/12} \cdot H^{1/6} \cdot m^{-1/12} \cdot (\log m)^{1/4}$.

This theorem shows that, when $m$ is sufficiently large, NOVI enjoys a similar regret bound as KOVI. Specifically, the choice of $\beta$ in (D.3) is similar to that in (4.5) for kernel $K_m$. Here we set $\lambda$ to be an absolute constant as $\sup_z K_m(z, z) \le 1$ no longer holds. In addition, here we assume that $\text{act}'$ is $C_{\text{act}}$-Lipschitz on $\mathbb{R}$, which can be relaxed to only assuming $\text{act}'$ is Lipschitz continous on a bounded interval of $\mathbb{R}$ that contains $w^\top z$ with high probability, where $w$ is drawn from the initial distribution of $W_j$, $j \in [m]$.

Moreover, comparing (D.6) and (D.5) we observe that, when $m$ is sufficiently large, NOVI can be viewed as a misspecified version of KOVI for the RKHS with kernel $K_m$, where the model misspecification error is $\text{err}_{\text{mis}} = 10\beta \cdot \iota$. Specifically, the first term in (D.5) is the same as that in (D.6), where the choice of $\beta$ and $\Gamma_{K_m}(T, \lambda)$ reflect the intrinsic complexity of $K_m$. Whereas the second term is equal to $\text{err}_{\text{mis}} \cdot TH$, which arises due to approximating neural network value functions by functions in $\mathcal{Q}_{\text{ucb}}(h, R_T, B_T)$, which are constructed using kernel functions with feature mapping $\varphi(\cdot; W^{(0)})$. Moreover, when $\beta$ is bounded by a polynomial of $TH$, to make $\text{err}_{\text{mis}} \cdot TH$ negligible, it suffices to let $m$ be a polynomial of $TH$. That is, when the network width is a polynomial of the total number of steps, NOVI achieves the same performance as KOVI.

Furthermore, when neglecting the constants and logarithmic terms in (D.3), we simplify the regret bound in (D.5) into

$$\text{Regret}(T) = \mathcal{O}\Big(H^2 \cdot \Big[\Gamma_{K_m}(T, \lambda) + \max_{h \in [H]} \sqrt{\Gamma_{K_m}(T, \lambda) \cdot \log N_\infty(\epsilon^*, h, B_T)}\Big] \cdot \sqrt{T} + \text{err}_{\text{mis}} \cdot T\Big).$$

which depends on the intrinsic complexity of $K_m$ through both the effective dimension $\Gamma_{K_m}(T, \lambda)$ and the log-covering number $\log N_\infty(\epsilon^*, h, B_T)$. To obtain a more concrete regret bounds, in the following, we pose an assumption on the spectral structure of $K_m$.

**Assumption D.3** (Eigenvalue Decay of the Empirical NTK). Conditioning on the randomness of $(b^{(0)}, W^{(0)})$, let $K_m$ be the kernel induced by the neural tangent features $\nabla f(\cdot; b^{(0)}, W^{(0)})$. Let $T_{K_m}$ be the integral operator induced by $K_m$ and the Lebesgue measure on $\mathcal{Z}$ and let $\{\sigma_j\}_{j \ge 1}$ and $\{\psi_j\}_{j \ge 1}$ be its eigenvalues and eigenvectors, respectively. We assume that $\{\sigma_j\}_{j \ge 1}$ and $\{\psi_j\}_{j \ge 1}$ satisfy either one of the two decay conditions specified in Assumption 4.3. Here we assume the constants $C_1, C_2, C_\psi, \gamma$, and $\tau$ do not depend on $m$.

Here we assume that $K_m$ satisfies Assumption 4.3. Since $K_m$ depends on the initial network weights, which are random, this assumption should be better understood in the limit sense. Specifically, as $m$ goes to infinity, $K_m$ converges to $K_{\text{ntk}}$, which is determined by both the activation function and the distribution of the initial network weights. Thus, if the RKHS with kernel $K_{\text{ntk}}$ satisfy Assumption 4.3, when $m$ is sufficiently large, it is reasonable to expect that such a condition also holds for $K_m$. Due to the space limit, we present concrete examples of $K_{\text{ntk}}$ satisfying Assumption 4.3 in §G.3 in the appendix.

Now we are ready to characterize the performances of NOVI for each case separately.

**Corollary D.4.** Under Assumptions D.1 and D.3, we assume the activation function is $C_{\mathrm{act}}$-smooth and the number of neurons of the neural network satisfies (D.4). Besides, in Algorithm 4 we let $\lambda$ be a sufficiently large constant and set $\beta = B_T$ as in (4.8). Then exists an absolute constant $C_r$ such that, with probability at least $1 - (T^2 H^2)^{-1}$, we have

$$\mathrm{Regret}(T) \leq \begin{cases} C_r \cdot H^2 \cdot \sqrt{\gamma^3 T} \cdot \log(\gamma T H) + 10\beta T H \cdot \iota & \gamma\text{-finite spectrum}, \\ C_r \cdot H^2 \cdot \sqrt{(\log T)^{3/\gamma} \cdot T} \cdot \log(T H) + 10\beta T H \cdot \iota & \gamma\text{-exponential decay}, \end{cases} \tag{D.6}$$

where we define $\iota = T^{7/12} \cdot H^{1/6} \cdot m^{-1/12} \cdot (\log m)^{1/4}$.

Corollary D.4 is parallel to Corollary 4.4, with an additional misspecification error $10\beta T H \cdot \iota$. It remains to see whether there exist concrete neural networks that induce NTKs satisfying each eigenvalue decay condition. As we will show in §G.3, neural network with quadratic and sine activation functions induce NTKs satisfying the finite-spectrum and exponential eigenvalue decay conditions, respectively. Corollary D.4 can be directly applied to these concrete examples to obtain sublinear regret bounds.

# E    Proofs of the Main Results

In this section, we provide the proofs of Theorems 4.2 and D.2. The proofs of the supporting lemmas and auxiliary results are deferred to the appendix.

## E.1    Proof of Theorem 4.2

*Proof.* For simplicity of presentation, we define the temporal-difference (TD) error as

$$\delta_h^t(x, a) = r_h(x, a) + (\mathbb{P}_h V_{h+1}^t)(x, a) - Q_h^t(x, a), \qquad \forall (x, a) \in \mathcal{S} \times \mathcal{A}. \tag{E.1}$$

Here $\delta_h^t$ is a function on $\mathcal{S} \times \mathcal{A}$ for all $h \in [H]$ and $t \in [T]$. Note that $V_h^t(\cdot) = \max_{a \in \mathcal{A}} Q_h^t(\cdot, a)$. Intuitively, $\{\delta_h^t\}_{h \in [H]}$ quantifies the how far the $\{Q_h^t\}_{h \in [H]}$ are from satisfying the Bellman optimality equation in (2.2). Next, recall that $\pi^t$ is the policy executed in the $t$-th episode, which generates a trajectory $\{(x_h^t, a_h^t)\}_{h \in [H]}$. For any $h \in [H]$ and $t \in [T]$, we further define $\zeta_{t,h}^1, \zeta_{t,h}^2 \in \mathbb{R}$ as

$$\zeta_{t,h}^1 = \left[ V_h^t(x_h^t) - V_h^{\pi^t}(x_h^t) \right] - \left[ Q_h^t(x_h^t, a_h^t) - Q_h^{\pi^t}(x_h^t, a_h^t) \right], \tag{E.2}$$

$$\zeta_{t,h}^2 = \left[ (\mathbb{P}_h V_{h+1}^t)(x_h^t, a_h^t) - (\mathbb{P}_h V_{h+1}^{\pi^t})(x_h^t, a_h^t) \right] - \left[ V_{h+1}^t(x_{h+1}^t) - V_{h+1}^{\pi^t}(x_{h+1}^t) \right]. \tag{E.3}$$

By definition, $\zeta_{t,h}^1$ and $\zeta_{t,h}^2$ capture two sources of randomness—the randomness of choosing an action $a_h^t \sim \pi_h^t(\cdot \,|\, x_h^t)$ and that of drawing the next state $x_{h+1}^t$ from $\mathbb{P}_h(\cdot \,|\, x_h^t, a_h^t)$, respectively. As we will see in Appendix §H.3, $\{\zeta_{t,h}^1, \zeta_{t,h}^2\}$ form a bounded martingale difference sequence with respect to a properly chosen filtration, which enables us to bound their total sum via the Azuma-Hoeffding inequality [7].

To establish an upper bound on the regret, the following lemma first decomposes the regret into three parts using the notation defined above. Similar regret decomposition results also appear in [12, 29].

**Lemma E.1** (Regret Decomposition). The temporal-difference error is the mapping $\delta_h^t : \mathcal{S} \times \mathcal{A} \to$ defined in (E.1) for all $(t, h) \in [T] \times [H]$. We can thus write the regret as

$$\mathrm{Regret}(T) = \underbrace{\sum_{t=1}^{T} \sum_{h=1}^{H} \left[ \mathbb{E}_{\pi^\star}[\delta_h^t(x_h, a_h) \,|\, x_1 = x_1^t] - \delta_h^t(x_h^t, a_h^t) \right]}_{\text{(i)}} + \underbrace{\sum_{t=1}^{T} \sum_{h=1}^{H} (\zeta_{t,h}^1 + \zeta_{t,h}^2)}_{\text{(ii)}}$$

$$\underbrace{\sum_{t=1}^{T} \sum_{h=1}^{H} \mathbb{E}_{\pi^\star} \left[ \left\langle Q_h^t(x_h, \cdot), \pi_h^\star(\cdot \,|\, x_h) - \pi_h^t(\cdot \,|\, x_h) \right\rangle_{\mathcal{A}} \,\middle|\, x_1 = x_1^t \right]}_{\text{(iii)}}, \tag{E.4}$$

where $\zeta_{t,h}^1$ and $\zeta_{t,h}^2$ are defined in (E.2) and (E.3), respectively.

*Proof.* See Appendix §H.1 for a detailed proof. $\qquad\square$

Returning to the main proof, notice that $\pi_h^t$ is the greedy policy with respect to $Q_h^t$ for all $(t, h) \in [T] \times [H]$. We have
$$\left\langle Q_h^t(x_h, \cdot), \pi_h^{\star}(\cdot \,|\, x_h) - \pi_h^t(\cdot \,|\, x_h) \right\rangle_{\mathcal{A}} = \left\langle Q_h^t(x_h, \cdot), \pi_h^{\star}(\cdot \,|\, x_h) \right\rangle_{\mathcal{A}} - \max_{a \in \mathcal{A}} Q_h^t(x_h, a) \leq 0,$$
for all $x_h \in \mathcal{S}$. Thus, Term (iii) in (E.4) is non-positive. Then, by Lemma E.1, we can upper bound the regret by
$$\mathrm{Regret}(T) \leq \underbrace{\left\{ \sum_{t=1}^{T} \sum_{h=1}^{H} \left[ \mathbb{E}_{\pi^*}[\delta_h^t(x_h, a_h) \,|\, x_1 = x_1^t] - \delta_h^t(x_h^t, a_h^t) \right] \right\}}_{\text{(i)}} + \underbrace{\left[ \sum_{t=1}^{T} \sum_{h=1}^{H} (\zeta_{t,h}^1 + \zeta_{t,h}^2) \right]}_{\text{(ii)}}.$$
$$\text{(E.5)}$$

For Term (i), since we do not observe trajectories from $\pi^*$, which is unknown, it appears that $\mathbb{E}_{\pi^*}[\delta_h^t(x_h, a_h) \,|\, x_1 = x_1^t]$ cannot be estimated. Fortunately, however, by adding the bonus term in Algorithm 2, we ensure that the temporal-difference error $\delta_h^t$ is a non-positive function, as shown in the following lemma.

**Lemma E.2** (Optimism). Let $\lambda = 1 + 1/T$ and $\beta = B_T$ in Algorithm 2, where $B_T$ satisfies (4.5). Under Assumptions 4.1, with probability at least $1 - (2T^2H^2)^{-1}$, we have that the following holds for all $(t, h) \in [T] \times [H]$ and $(x, a) \in \mathcal{S} \times \mathcal{A}$:
$$-2\beta \cdot b_h^t(x, a) \leq \delta_h^t(x, a) \leq 0.$$

*Proof.* See Appendix §H.2 for a detailed proof. $\qquad\square$

Applying Lemma E.2 to Term (i) in (E.5), we obtain that
$$\text{Term (i)} \leq \left[ \sum_{t=1}^{T} \sum_{h=1}^{H} -\delta_h^t(x_h^t, a_h^t) \right] \leq 2\beta \cdot \left[ \sum_{t=1}^{T} \sum_{h=1}^{H} b_h^t(x_h^t, a_h^t) \right] \qquad \text{(E.6)}$$
holds with probability at least $1 - (2T^2H^2)^{-1}$, where $\beta$ is equal to $B_T$ as specified in (4.5).

Finally, it remains to bound the sum of bonus terms in (E.6). As we show in (H.17), using the feature representation of $\mathcal{H}$, we can write each $b_h^t(x_h^t, a_h^t)$ as
$$b_h^t(x_h^t, a_h^t) = \left[ \phi(x_h^t, a_h^t)^{\top} (\Lambda_h^t)^{-1} \phi(x_h^t, a_h^t) \right]^{1/2},$$
where $\Lambda_h^t = \lambda \cdot I_{\mathcal{H}} + \sum_{\tau=1}^{t-1} \phi(x_h^t, a_h^t) \phi(x_h^t, a_h^t)^{\top}$ is a self-adjoint and positive-definite operator on $\mathcal{H}$ and $\mathcal{I}_{\mathcal{H}}$ is the identity mapping on $\mathcal{H}$. Thus, combining the Cauchy-Schwarz inequality and Lemma J.3, we have, for any $h \in [H]$, with probability at least $1 - (2T^2H^2)^{-1}$ the following:
$$\text{Term (i)} \leq 2\beta \cdot \sqrt{T} \cdot \sum_{h=1}^{H} \left[ \sum_{t=1}^{T} \phi(x_h^t, a_h^t)^{\top} (\Lambda_h^t)^{-1} \phi(x_h^t, a_h^t) \right]^{1/2}$$
$$\leq 2\beta \cdot \sum_{h=1}^{H} \left[ 2T \cdot \mathrm{logdet}(I + K_h^T/\lambda) \right]^{1/2} = 4\beta H \cdot \sqrt{T \cdot \Gamma_K(T, \lambda)}, \qquad \text{(E.7)}$$
where $\Gamma_K(T, \lambda)$ is the maximal information gain defined in (4.2) with parameter $\lambda$.

It remains to bound Term (ii) in (E.5), which is the purpose of the following lemma.

**Lemma E.3.** For $\zeta_{t,h}^1$ and $\zeta_{t,h}^2$ defined respectively in (E.2) and (E.3) and for any $\zeta \in (0, 1)$, with probability at least $1 - \zeta$, we have
$$\sum_{t=1}^{T} \sum_{h=1}^{H} (\zeta_{t,h}^1 + \zeta_{t,h}^2) \leq \sqrt{16TH^3 \cdot \log(2/\zeta)}.$$

*Proof.* See Appendix §H.3 for a detailed proof. $\qquad\square$

Setting $\zeta = (2T^2H^2)^{-1}$ in Lemma E.3 we obtain that

$$\text{Term (ii)} = \sum_{t=1}^{T} \sum_{h=1}^{H} (\zeta_{t,h}^1 + \zeta_{t,h}^2) \le \sqrt{16TH^3 \cdot \log(4T^2H^2)} = \sqrt{32TH^3 \cdot \log(2TH)} \quad \text{(E.8)}$$

holds with probability at least $1 - (2TH)^{-1}$.

Therefore, combining (4.5), (E.5), and (E.8), we conclude that, with probability at least $1 - (T^2H^2)^{-1}$, the regret is bounded by

$$\text{Regret}(T) \le 4\beta H \cdot \sqrt{T \cdot \Gamma_K(T, \lambda)} + \sqrt{32TH^3 \cdot \log(2TH)} \le 5\beta H \cdot \sqrt{T \cdot \Gamma_K(T, \lambda)},$$

where the last inequality follows from the choice of $\beta = B_T$, which implies that

$$\beta \ge H \cdot \sqrt{16 \log(TH)} \ge \sqrt{32H \cdot \log(2TH)}.$$

This concludes the proof of Theorem 4.2. $\qquad\qquad\qquad\qquad\qquad\qquad\qquad\qquad\qquad\qquad\qquad\square$

## E.2 Proof of Theorem D.2

*Proof.* The proof of Theorem D.2 is similar to that of Theorem 4.2. Recall that we let $\mathcal{Z}$ denote $\mathcal{S} \times \mathcal{A}$ for simplicity. Recall also that for all $(t, h) \in [T] \times [H]$, we define the temporal-difference (TD) error $\delta_h^t \colon \mathcal{Z} \to \mathbb{R}$ in (E.1) and define random variables $\zeta_{t,h}^1$ and $\zeta_{t,h}^2$ in (E.2) and (E.3), respectively.

Then, combining Lemma E.1 and the fact that $\pi^t$ is the greedy policy with respect to $\{Q_h^t\}_{h \in [H]}$, we bound the regret by

$$\text{Regret}(T) \le \underbrace{\left\{ \sum_{t=1}^{T} \sum_{h=1}^{H} \left[ \mathbb{E}_{\pi^\star} [\delta_h^t(x_h, a_h) \,|\, x_1 = x_1^t] - \delta_h^t(x_h^t, a_h^t) \right] \right\}}_{\text{(i)}} + \underbrace{\left[ \sum_{t=1}^{T} \sum_{h=1}^{H} (\zeta_{t,h}^1 + \zeta_{t,h}^2) \right]}_{\text{(ii)}}.$$

$$\text{(E.9)}$$

Here, Term (ii) is a sum of a martingale difference sequence. By setting $\zeta = (4T^2H^2)^{-1}$ in Lemma E.3, with probability at least $1 - (4T^2H^2)^{-1}$, we have

$$\text{Term (ii)} = \sum_{t=1}^{T} \sum_{h=1}^{H} (\zeta_{t,h}^1 + \zeta_{t,h}^2) \le \sqrt{16TH^3 \cdot \log(8T^2H^2)} \le H \cdot \sqrt{32TH \log(2TH)}. \quad \text{(E.10)}$$

It remains to bound Term (i) in (E.9). To this end, we aim to establish a counterpart of Lemma E.2 for neural value functions, which shows that, by adding a bonus term $\beta \cdot b_h^t$, the TD error $\delta_h^t$ is always a non-positive function approximately. This implies that bounding Term (i) in (E.9) reduces to controlling $\sum_{t=1}^{T} \sum_{h=1}^{H} b_h^t(x_h^t, a_h^t)$.

Note that the bonus functions $b_h^t$ are constructed based on the neural tangent features $\varphi(\cdot; \widehat{W}_h^t)$ and the matrix $\Lambda_h^t$. In order to relate $\sum_{t=1}^{T} \sum_{h=1}^{H} b_h^t(x_h^t, a_h^t)$ to the maximal information gain of the empirical NTK $K_m$, we define $\overline{\Lambda}_h^t$ and $\overline{b}_h^t$, by analogy with $\Lambda_h^t$ and $b_h^t$, as follows:

$$\overline{\Lambda}_h^t = \lambda \cdot I_{2md} + \sum_{\tau=1}^{t-1} \varphi(x_h^\tau, a_h^\tau; W^{(0)}) \varphi(x_h^\tau, a_h^\tau; W^{(0)})^\top, \qquad \overline{b}_h^t(z) = \left[ \varphi(z; W^{(0)})^\top (\overline{\Lambda}_h^t)^{-1} \varphi(z; W^{(0)}) \right]^{1/2}.$$

In the following lemma, we bound the TD error $\delta_h^t$ using $\overline{b}_h^t$ and show that $b_h^t$ and $\overline{b}_h^t$ are close in the $\ell_\infty$-norm on $\mathcal{Z}$ when $m$ is sufficiently large.

**Lemma E.4** (Optimism). Let $\lambda$ be an absolute constant and let $\beta = B_T$ in Algorithm 4, where $B_T$ satisfies (D.3). Under the assumptions made in Theorem D.2, with probability at least $1 - (2T^2H^2)^{-1} - m^2$, it holds for all $(t, h) \in [T] \times [H]$ and $(x, a) \in \mathcal{S} \times \mathcal{A}$ that

$$-5\beta \cdot \iota - 2\beta \cdot \overline{b}_h^t(x, a) \le \delta_h^t(x, a) \le 5\beta \cdot \iota, \qquad \sup_{(x,a) \in \mathcal{Z}} \left| b_h^t(x, a) - \overline{b}_h^t(x, a) \right| \le 2\iota, \quad \text{(E.11)}$$

where we define $\iota = T^{7/12} \cdot H^{1/12} \cdot m^{-1/12} \cdot (\log m)^{1/4}$.

*Proof.* See Appendix §H.4 for a detailed proof. $\qquad\qquad\qquad\qquad\qquad\qquad\qquad\qquad\qquad\qquad\square$

Applying Lemma E.2 to Term (i) in (E.5), we obtain that

$$\text{Term (i)} \leq \left[ \sum_{t=1}^{T} \sum_{h=1}^{H} -\delta_h^t(x_h^t, a_h^t) \right] + 5TH \cdot \iota \leq 2\beta \cdot \left[ \sum_{t=1}^{T} \sum_{h=1}^{H} \bar{b}_h^t(x_h^t, a_h^t) \right] + 10\beta TH \cdot \iota \quad \text{(E.12)}$$

holds with probability at least $1 - (2T^2H^2)^{-1} - m^{-2}$, where $\beta = B_T$. Moreover, combining the Cauchy-Schwarz inequality and Lemma J.3, we have

$$\sum_{t=1}^{T} \sum_{h=1}^{H} \bar{b}_h^t(x_h^t, a_h^t) \leq \sqrt{T} \cdot \sum_{h=1}^{H} \left[ \sum_{t=1}^{T} \varphi(x_h^t, a_h^t; W^{(0)})^\top (\overline{\Lambda}_h^t)^{-1} \varphi(x_h^t, a_h^t; W^{(0)}) \right]^{1/2}$$

$$\leq 2H \cdot \sqrt{T \cdot \Gamma_{K_m}(T, \lambda)}, \quad \text{(E.13)}$$

where $\Gamma_K(T, \lambda)$ is the maximal information gain defined in (4.2) for kernel $K_m$.

Notice that $(2T^2H^2)^{-1} + m^{-2} + (4T^2H^2)^{-1} \leq (T^2H^2)^{-1}$. Thus, combining (E.9), (E.10), (E.12), and (E.13), we obtain that

$$\text{Regret}(T) \leq 4\beta H \cdot \sqrt{T \cdot \Gamma_{K_m}(T, \lambda)} + 10\beta TH \cdot \iota + H \cdot \sqrt{32TH \log(2TH)}$$

$$\leq 5\beta H \cdot \sqrt{T \cdot \Gamma_{K_m}(T, \lambda)} + 10\beta TH \cdot \iota$$

holds with probability at least $1 - (2T^2H^2)^{-1}$. Here the last inequality follows from the fact that

$$\beta \geq H \cdot \sqrt{32 \log(TH)} \geq \sqrt{32H \log(2TH)}.$$

This concludes the proof of Theorem D.2. □

# F  Neural Optimistic Least-Squares Value Iteration

In this section, we provide the pseudocode for NOVI, which was omitted in the main text for brevity. We remark that the loss function $L_h^t$ in Line 7 is given in (C.1) and its global minimizer $\widehat{W}_h^t$ can be efficiently obtained by first-order optimization methods.

---

**Algorithm 4** Neural Optimistic Least-Squares Value Iteration (NOVI)

---

1: **Input:** Parameters $\lambda$ and $\beta$.
2: Initialize the network weights $(b^{(0)}, W^{(0)})$ via the symmetric initialization scheme.
3: **for** episode $t = 1, \ldots, T$ **do**
4:     Receive the initial state $x_1^t$.
5:     Set $V_{H+1}^t$ as the zero function.
6:     **for** step $h = H, \ldots, 1$ **do**
7:         Solve the neural network optimization problem $\widehat{W}_h^t = \text{argmin}_W L_h^t(W)$.
8:         Update $\Lambda_h^t = \Lambda_h^{t-1} + \varphi(x_h^{t-1}, a_h^{t-1}; \widehat{W}_h^t) \varphi(x_h^{t-1}, a_h^{t-1}; \widehat{W}_h^t)^\top$.
9:         Obtain the bonus function $b_h^t$ defined in (C.4).
10:        Obtain value functions

$$Q_h^t(\cdot, \cdot) \leftarrow \min\{f(\cdot, \cdot; \widehat{W}_h^t) + \beta \cdot b_h^t(\cdot, \cdot), H - h + 1\}^+, \qquad V_h^t(\cdot) = \max_a Q_h^t(\cdot, a).$$

11:     **end for**
12:     **for** step $h = 1, \ldots, H$ **do**
13:         Take action $a_h^t \leftarrow \text{argmax}_{a \in \mathcal{A}} Q_h^t(x_h^t, a)$.
14:         Observe the reward $r_h(x_h^t, a_h^t)$ and the next state $x_{h+1}^t$.
15:     **end for**
16: **end for**

---

# G  Proofs of the Corollaries

In this section, we prove Corollaries 4.4 and D.4, which establish the regret for KOVI and NOVI under each specific eigenvalue decay condition. in Appendix §G.3 we provide concrete examples of neural

tangent kernels that satisfy Assumption 4.3 and show how to apply Corollaries 4.4 and D.4 to these examples.

## G.1 Proof of Corollary 4.4

*Proof.* To prove this corollary, it suffices to verify that for each eigenvalue decay condition specified in Assumption 4.3, $B_T$ defined in (4.8) satisfies the condition in (4.5). Recall that we set $\lambda = 1 + 1/T$ in Algorithm 2 and denote $R_T = 2H\sqrt{\Gamma_K(T,\lambda)}$, $\epsilon^* = H/T$. Also recall that we let $N_\infty(\epsilon, h, B)$ denote the $\epsilon$-covering number of $\mathcal{Q}_{\mathrm{ucb}}(h, R_T, B)$ with respect to the $\ell_\infty$-norm. In the sequel, we consider the two cases separately.

**Case (i): $\gamma$-Finite Spectrum.** When $\mathcal{H}$ has at most $\gamma$ nonzero eigenvalues, by Lemma I.5, we have $\Gamma_K(T,\lambda) \le C_K \cdot \gamma \log T$, where $C_K$ is an absolute constant. Moreover, by Lemma I.1, for any $h \in [H]$, we have

$$\log N_\infty(\epsilon^*, h, B_T) \le C_N \cdot \gamma \cdot \big\{ 1 + \log\big[ 2\sqrt{\Gamma(T,\lambda)} \cdot T \big] \big\} + C_N \cdot \gamma^2 \cdot \big[ 1 + \log(B_T \cdot T/H) \big]$$
$$\le 2C_N \cdot \gamma^2 + C' \cdot \gamma \cdot \log(\gamma T) + C_N \cdot \gamma^2 \cdot \log(B_T \cdot T/H), \tag{G.1}$$

where $C_N > 0$ is the absolute constant given in Lemma I.1 and $C'$ is an absolute constant that depends on $C_N$ and $C_K$. Thus, setting $B_T = C_b \cdot \gamma H \cdot \sqrt{\log(dTH)}$ in (G.1), the left-hand side (LHS) of (4.5) is bounded by

$$\text{LHS of (4.5)} \le 8C_K \cdot \gamma \log T + 16 C_N \cdot \gamma^2 + 8C' \cdot \gamma \cdot \log(\gamma T) +$$
$$8C_N \cdot \gamma^2 \cdot \log(C_b \cdot \gamma T \cdot \sqrt{\log(dTH)}) + 16 \cdot \log(TH) + 22 + 2R_Q^2$$
$$\le \gamma^2 \cdot \big[ \overline{C}_1 \cdot \log(\gamma TH) + 8C_N \cdot \log(C_b) \big], \tag{G.2}$$

where $\overline{C}_1$ is an absolute constant that depends on $C'$, $C_N$, $C_K$, and $R_Q$. Thus, setting $C_b$ as a sufficiently large constant, by (G.2), we have

$$\text{LHS of (4.5)} \le C_b^2 \cdot \gamma^2 \cdot \log(dTH) = (B_T/H)^2,$$

which establishes (4.5) for the first case. Thus, applying Theorem 4.2 we obtain that

$$\text{Regret}(T) \le 8B_T \cdot H \cdot \sqrt{T \cdot \Gamma_K(T,\lambda)} \le C_{r,1} \cdot H^2 \cdot \sqrt{\gamma^3 T} \cdot \log(\gamma TH) = \widetilde{\mathcal{O}}(H^2\sqrt{\gamma^3 T})$$

holds with probability at least $1 - (T^2 H^2)^{-1}$, where $C_{r,1}$ is an absolute constant and $\widetilde{\mathcal{O}}(\cdot)$ omits the logarithmic factor. Therefore, we conclude the first case.

**Case (ii): $\gamma$-Exponential Decay.** For the second case, by Lemma I.5 we have

$$\Gamma_K(T,\lambda) \le C_K \cdot (\log T)^{1+1/\gamma}, \tag{G.3}$$

where $C_K$ is an absolute constant. Thus, by the choice of $B_T$ in (4.8), when $C_b$ is sufficiently large, it holds that $R_T = 2H\sqrt{\Gamma_K(T,\lambda)} \le B_T$. Then by Lemma I.1 we have

$$\log N_\infty(h, \epsilon^*, B_T) \le C_N \cdot \big[ 1 + \log(R_T/\epsilon^*) \big]^{1+1/\gamma} + C_N \cdot \big[ 1 + \log(B_T/\epsilon^*) \big]^{1+2/\gamma}$$
$$\le 2C_N \cdot \big[ 1 + \log(B_T/\epsilon^*) \big]^{1+2/\gamma} = 2C_N \cdot \big\{ 1 + \log\big[ C_b T \cdot \sqrt{\log(TH)} \cdot (\log T)^{1/\gamma} \big] \big\}^{1+2/\gamma},$$

where the absolute constant $C_N$ is given by Lemma I.1. By direct computation, there exists an absolute constant $\overline{C}_2$ such that

$$\log N_\infty(h, \epsilon^*, B_T) \le 2C_N \cdot \big[ 1 + \log(C_b) + \overline{C}_2 \cdot \log T + 1/2 \cdot \log \log H \big]^{1+2/\gamma}. \tag{G.4}$$

Thus, combining (G.3) and (G.4), the left-hand side of (4.5) is bounded by

$$\text{LHS of (4.5)} \le 8C_K \cdot (\log T)^{1+1/\gamma} + 16C \cdot \big[ 1 + \log(C_b) + \overline{C}_2 \cdot \log T + 1/2 \cdot \log \log H \big]^{1+2/\gamma}$$
$$+ 16 \cdot \log(TH) + 22 + 2R_Q^2$$
$$\le \overline{C}_3 \cdot \big[ (\log T)^{1+2/\gamma} + (\log \log H)^{1+2/\gamma} + \log(C_b) \big], \tag{G.5}$$

where $\overline{C}_3$ is an absolute constant that does not depend on $C_b$. Thus, when $C_b$ is sufficiently large, (G.5) implies that

$$\text{LHS of (4.5)} \le \overline{C}_3 \cdot \big[ (\log T)^{1+2/\gamma} + (\log \log H)^{1+2/\gamma} + \log(C_b) \big] \le C_b^2 \cdot (\log T)^{2/\gamma} \cdot \log(TH) = (B_T/H)^2.$$

Thus, for the case of $\gamma$-exponential eigenvalue decay, (4.5) holds true for $B_T$ defined in (4.8).

Finally, applying Theorem 4.2 and combining (4.8) and (G.3), we obtain that

$$\text{Regret}(T) \leq C_{r,2} \cdot H^2 \cdot \log(TH) \cdot \sqrt{(\log T)^{3/\gamma} \cdot T},$$

where $C_{r,2}$ is an absolute constant. Thus we conclude the second case. Therefore, we conclude the proof of Corollary 4.4. $\qquad\square$

## G.2    Proof of Corollary D.4

*Proof.* By Theorem D.2, we have

$$\text{Regret}(T) = 5\beta H \cdot \sqrt{T \cdot \Gamma_{K_m}(T, \lambda)} + 10\beta T H \cdot \iota, \tag{G.6}$$

where $\beta = B_T$ satisfies (D.3) and $\iota = T^{7/12} \cdot H^{1/6} \cdot m^{-1/12} \cdot (\log m)^{1/4}$. When Assumption D.3 holds, thanks to the similarity between (4.5) and (D.3), it can be similarly shown that $B_T$ defined in (4.8) satisfies the inequality in (D.3) when $C_b$ is sufficiently large. Moreover, Lemma I.5 provides upper bounds on $\Gamma_{K_m}(T, \lambda)$ for the two eigenvalue decay conditions. Finally, combining (4.8), (G.6), and Lemma I.5, we conclude the proof of Corollary D.4. $\qquad\square$

## G.3    Examples of Kernels Satisfying Assumption 4.3

In the following, we introduce concrete kernels and neural tangent kernels that satisfy Assumption 4.3. We consider each eigenvalue decay condition separately.

**Case (i): $\gamma$-Finite Spectrum.** Consider the polynomial kernel $K(z, z') = (1 + \langle z, z' \rangle)^n$ defined on the unit ball $\{z \in \mathbb{R}^d \colon \|z\|_2 \leq 1\}$, where $n$ is a fixed number. By direct computation, the kernel function can be written as

$$K(z, z') = \sum_{\alpha \colon \|\alpha\|_1 \leq n} z^\alpha \cdot z'^\alpha,$$

where $\alpha = (\alpha_1, \ldots, \alpha_d) \in \mathbb{N}^d$ is a multi-index and $z^\alpha$ is a monomial with degree $\alpha$. It can be shown that all monomials in $\mathbb{R}^d$ with degree no more than $n$ are linearly independent. Thus, the dimension of such an RKHS is $\binom{n+d}{d}$; i.e., it satisfies the $\gamma$-finite spectrum condition with $\gamma = \binom{n+d}{d}$.

Furthermore, for a finite-dimensional NTK, we consider the quadratic activation function $\text{act}(u) = u^2$. Note that we assume $\mathcal{Z} = \mathbb{S}^{d-1}$ for the neural network setting. Moreover, in (B.3), instead of sampling $W_j \sim N(0, I_d/d)$ for all $j \in [d]$, we draw $W_j$ uniformly over the unit sphere $\mathbb{S}^{d-1}$. Then it holds that $|W_j^\top z| \leq 1$ for all $j \in [2m]$ and $z \in \mathbb{S}^{d-1}$. Here we let the distribution be $\text{Unif}(\mathbb{S}^{d-1})$ in order to ensure that the $\text{act}'$ is Lipschitz continuous on $\{W_j^\top z \colon z \in \mathbb{S}^{d-1}\} \subseteq [-1, 1]$ for any $W_j$ sampled from the initial distribution, which is required when utilizing Proposition C.1 in [30] in the proof of Lemma E.4. Note that the covariance of $W_j$ is still $I_d/d$. Then by (B.7), the NTK is given by

$$K_{\text{ntk}}(z, z') = \mathbb{E}_{w \sim \text{Unif}(\mathbb{S}^{d-1})}[2(w^\top z) \cdot 2(w^\top z') \cdot (z^\top z')] = 4/d \cdot (z^\top z')^2, \qquad \forall z, z' \in \mathbb{S}^{d-1}. \tag{G.7}$$

Thus, $K_{\text{ntk}}(z, z')$ can be written as a univariate function of the inner product $\langle z, z' \rangle$. To characterize the spectral property $K_{\text{ntk}}$, we first introduce some background on spherical harmonic functions on $\mathbb{S}^{d-1}$, which are closely related to inner product kernels on $\mathbb{S}^{d-1} \times \mathbb{S}^{d-1}$.

Let $\mu$ be the uniform measure on $\mathbb{S}^{d-1}$. For any $j \geq 0$, let $\mathcal{Y}_j(d)$ be the set of all homogeneous harmonics of degree $j$ on $\mathbb{S}^{d-1}$, which is a finite-dimensional subspace of $\mathcal{L}^2_\mu(\mathbb{S}^{d-1})$, the space of square-integrable functions on $\mathbb{S}^{d-1}$ with respect to $\mu$. It can be shown that the dimensionality of $\mathcal{Y}_j(d)$ is given by $N(d, j)$, which is defined as

$$N(d, j) = \frac{(2j + d - 2)(d + j - 3)!}{j!(d - 2)!}. \tag{G.8}$$

In addition, let $\{Y_{j,\ell}\}_{\ell \in [N(d,j)]}$ be an orthonormal basis of $\mathcal{Y}_j(d)$, then $\{Y_{j,\ell}\}_{\ell \in [N(d,j)], j \in \mathbb{N}}$ form an orthonormal basis of $\mathcal{L}^2_\mu(\mathbb{S}^{d-1})$. In the next lemma, we present the Funk-Hecke formula [48, page 30], which relates spherical harmonics to inner product kernels.

**Lemma G.1** (Funk-Hecke formula). Let $k\colon [-1,1] \to \mathbb{R}$ be a continuous function, which gives rise to an inner product kernel $K(z,z') = k(\langle z, z'\rangle)$ on $\mathbb{S}^{d-1} \times \mathbb{S}^{d-1}$. For any $\ell \geq 2$, let $|\mathbb{S}^{\ell-1}|$ be the Lebesgue measure of $\mathbb{S}^{\ell-1}$, which is given by $|\mathbb{S}^{\ell-1}| = 2\pi^{\ell/2}/\Gamma(\ell/2)$, where $\Gamma(\cdot)$ is the Gamma function. Moreover, for any $j \geq 0$, let $Y_j\colon \mathbb{S}^{d-1} \to \mathbb{R}$ be any function in $\mathcal{Y}_j(d)$. Then for any $z \in \mathbb{S}^{d-1}$, we have

$$\int_{\mathbb{S}^{d-1}} K(z,z')Y_j(z')\,\mathrm{d}\mu(z') = \left[\frac{|\mathbb{S}^{d-2}|}{|\mathbb{S}^{d-1}|} \cdot \int_{-1}^{1} k(u) \cdot P_j(u;d) \cdot (1-u^2)^{(d-3)/2}\,\mathrm{d}u\right] \cdot Y_j(z), \tag{G.9}$$

where $P_j(\cdot;d)$ is the $j$-th Legendre polynomial in dimension $d$, which is given by

$$P_j(u;d) = \frac{(-1/2)^j \cdot \Gamma(\frac{d-1}{2})}{\Gamma(\frac{2j+d-1}{2})} \cdot (1-u^2)^{(3-d)/2} \cdot \left(\frac{\mathrm{d}}{\mathrm{d}u}\right)^j \left[(1-u^2)^{j+(d-3)/2}\right].$$

Thus, by the Funk-Hecke formula, for any inner product kernel $K$, its integral operator $T_K\colon \mathcal{L}^2_\mu(\mathbb{S}^{d-1}) \to \mathcal{L}^2_\mu(\mathbb{S}^{d-1})$ has eigenvalues

$$\varrho_j = \frac{|\mathbb{S}^{d-2}|}{|\mathbb{S}^{d-1}|} \cdot \int_{-1}^{1} k(u) \cdot P_j(u;d) \cdot (1-u^2)^{(d-3)/2}\,\mathrm{d}u, \qquad \forall j \geq 1, \tag{G.10}$$

each with multiplicity $N(d,j)$. Moreover, for each eigenvalue $\varrho_j$, the corresponding eigenfunctions are spherical harmonics $\{Y_{j,\ell}\}_{\ell \in [N(d,j)]}$. Furthermore, to compute the eigenvalues in (G.10), we can use Rodrigues' rule [48, page 23], as follows.

**Lemma G.2** (Rodrigues' Rule). For any $j \geq 0$, let $f\colon [-1,1] \to \mathbb{R}$ be any $j$-th continuously differentiable function. Then we have

$$\int_{-1}^{1} f(t) \cdot P_j(u;d) \cdot (1-u^2)^{(d-3)/2}\,\mathrm{d}u = R_j(d) \cdot \int_{-1}^{1} f^{(j)}(u) \cdot (1-u^2)^{(2j+d-3)/2}\,\mathrm{d}t,$$

where $f^{(j)}$ is the $j$-th order derivative of $f$ and $R_j(d) = 2^{-j} \cdot \Gamma((d-1)/2) \cdot [\Gamma((2j+d-1)/2)]^{-1}$ is the $j$-th Rodrigues constant.

Now we consider the NTK given in (G.7), which is the inner product kernel induced by the univariate function $k_1(u) = 4/d \cdot u^2$. Note that $k_1^{(3)}$ is a zero function. Combining Lemma G.2 and (G.10), we observe that $\varrho_j = 0$ for all $j \geq 3$. In addition, by direct computation, we have that

$$\varrho_1 = R_1(d) \cdot (8/d) \cdot \int_{-1}^{1} u \cdot (1-u^2)^{(d-1)/2}\,\mathrm{d}u = 0,$$

and $\varrho_0, \varrho_2 > 0$. Thus, $K_{\mathrm{ntk}}$ given in (G.7) has $N(d,0) + N(d,2) = d(d+1)/2$ nonzero eigenvalues, each with value $\varrho_2$. This implies that the NTK induced by neural networks with quadratic activation satisfies the $\gamma$-finite spectrum condition with $\gamma = d(d+1)/2$. For such a class of neural networks, Corollary D.4 asserts that the regret of NOVI is $\widetilde{\mathcal{O}}(H^2 d^3 \cdot \sqrt{T} + \beta T H \cdot \iota)$.

**Case (ii): $\gamma$-exponential Decay.** Now we consider the squared exponential kernel

$$K(z,z') = \exp(-\|z-z'\|_2^2 \cdot \sigma^{-2}) = k_2(\langle z, z'\rangle), \qquad \forall z, z' \in \mathbb{S}^{d-1}, \tag{G.11}$$

where $\sigma > 0$ is an absolute constant and we define $k_2(u) = \exp[-2\sigma^{-2} \cdot (1-u)]$. Note that $d$ is regarded as a fixed number. Applying Lemmas G.1 and G.2, we obtain the following lemma that bounds the eigenvalues of $T_K$.

**Lemma G.3** (Theorem 2 in [47]). For the squared quadratic kernel in (G.11), the corresponding integral operator has eigenvalues $\{\rho_j\}_{j\geq 0}$, where each $\rho_j$ is defined in (G.10) with $k$ replaced by $k_2$. Moreover, each $\varrho_j$ has multiplicity $N(d,j)$ and the corresponding eigenfunctions are $\{Y_{j,\ell}\}_{\ell \in [N(d,j)]}$. Finally, when $\sigma$ in (G.11) satisfy $\sigma^2 \geq 2/d$, $\{\varrho_j\}_{j\geq 0}$ form a decreasing sequence that satisfy

$$A_1 \cdot (2e/\sigma^2)^j \cdot (2j+d-2)^{-(2j+d-1)/2} < \varrho_j < A_2 \cdot (2e/\sigma^2)^j \cdot (2j+d-2)^{-(2j+d-1)/2} \tag{G.12}$$

for all $j \geq 0$, where $A_1, A_2$ are absolute constants that only depend on $d$ and $\sigma$.

The $\ell_\infty$-norm of each eigenfunction $Y_{j,\ell}$ is given by the following lemma.

**Lemma G.4** (Lemma 3 in [47]). For any $d \geq 2$, $j \geq 0$, and any $\ell \in [N(d,j)]$, we have

$$\|Y_{j,\ell}\|_\infty = \sup_{z \in \mathbb{S}^{d-1}} |Y_{j,\ell}(z)| \leq \sqrt{N(d,j)/|\mathbb{S}^{d-1}|}.$$

Now, let $\tau > 0$ be a sufficiently small constant. Combining Lemmas G.3 and G.4, we have

$$\varrho_j^\tau \cdot \|Y_{j,\ell}\|_\infty \leq C \cdot \left( \frac{2e}{\sigma^2 \cdot (2j + d - 2)} \right)^{-j \cdot \tau} \cdot \sqrt{N(d,j) \cdot (2j + d - 2)^{-(d-1) \cdot \tau}}, \quad \text{(G.13)}$$

where $C$ is a constant depending on $d$ and $\sigma$. By the definition of $N(d,j)$ in (G.8), when $j$ is sufficiently large, it holds that

$$N(d,j) \asymp \frac{(2j + d - 2) \cdot \sqrt{d + j - 3} \cdot [(d + j - 3)/e]^{d+j-3}}{\sqrt{j} \cdot (j/e)^j} \asymp j^{d-2}, \quad \text{(G.14)}$$

where we utilize the Stirling's formula and neglect constants involving $d$. Then, combining (G.13) and (G.14), we have

$$\sup_{j \geq 0} \sup_{\ell \in [N(d,j)]} \varrho_j^\tau \cdot \|Y_{j,\ell}\|_\infty \leq C_\varrho, \quad \text{(G.15)}$$

for some absolute constant $C_\varrho > 0$. Renaming the eigenvalues and eigenvectors as $\{\sigma_j, \psi_j\}_{j \geq 1}$ in the descending order of the eigenvalues, (G.15) equivalently states that $\sup_{j \geq 1} \sigma_j^\tau \cdot \|\psi_j\|_\infty \leq C_\varrho$.

Furthermore, to show that the squared exponential kernel satisfy the $\gamma$-exponential decay condition, we notice that

$$\sigma_j = \varrho_t \qquad \text{for } \sum_{i=1}^{t-1} N(d,i) \leq j < \sum_{i=1}^t N(d,i). \quad \text{(G.16)}$$

Then by (G.14), this implies that $\sigma_j \asymp \rho_t$ for $(t-1)^{d-1} \leq j \leq t^{d-1}$ when $j$ is sufficiently large. Thus, by Lemma G.3 we further obtain that

$$\sigma_j \asymp (2e/\sigma^2)^{j^{\frac{1}{d-1}}} \cdot (2j^{\frac{1}{d-1}} + d - 2)^{-j^{\frac{1}{d-1}} - (d-1)/2}$$

$$\asymp \exp\left(c_1 \cdot j^{\frac{1}{d-1}}\right) \cdot \exp\left(c_2 - j^{\frac{1}{d-1}} \cdot \log j\right) \leq \exp(-c \cdot j^{1/d}),$$

where $c$, $c_1$, and $c_2$ are constants depending on $d$. Therefore, we have shown that the squared exponential kernel satisfies the $\gamma$-exponential decay condition with $\gamma = 1/d$. Combining this with (G.15), we conclude that it satisfies Assumption 4.3.

In the sequel, we construct an NTK that satisfies Assumption 4.3. Specifically, we adopt the sine activation function and slightly modify the neural network in (B.3) by employing an intercept for each neuron. That is,

$$f(z; b, W, \theta) = \frac{1}{\sqrt{m}} \sum_{j=1}^m b_j \cdot \sin(W_j^\top z + \theta_j).$$

To initialize the network weights $(b, W, \theta)$, we set $b_j = -b_{j-m}$, $W_j = W_{j-m}$, and $\theta_j = \theta_{j-m}$ for any $j \in \{m + 1, \ldots, 2m\}$. For any $j \in [m]$, we independently sample $b_j \sim \text{Unif}(\{-1, 1\})$, $W_j \sim N(0, I_d)$, and $\theta_j \sim \text{Unif}([0, 2\pi])$. Only $W$ is updated during training.

For such a neural network, the corresponding NTK is given by

$$K_{\text{ntk}}(z, z') = 2\mathbb{E}\left[(z^\top z') \cdot \cos(w^\top z + \theta) \cdot \cos(w^\top z' + \theta)\right]$$

$$= (z^\top z') \cdot \exp(-\|z - z'\|_2^2/2) = (z^\top z') \cdot \exp[(z^\top z') - 1] = k_3(\langle z, z' \rangle), \quad \text{(G.17)}$$

where we define $k_3(u) = u \cdot \exp(u - 1)$. Here the second equality follows from [54]. By construction, such an NTK is closely related to the squared quadratic kernel in (G.11). To see that it satisfy the $\gamma$-exponential decay condition, let $\{\varrho_j\}_{j \geq 0}$ and $\{\widetilde{\varrho}_j\}_{j \geq 0}$ denote the eigenvalues of the NTK in (G.17) and the inner product kernel induced by $\widetilde{k}_2(u) = \exp(u - 1)$, respectively. By Lemma G.1, we have

$$\rho_j = C_1 \cdot \int_{-1}^1 k_3(u) \cdot P_j(u; d) \cdot (1 - u^2)^{(d-3)/2} \, du = C_1 \cdot \int_{-1}^1 \widetilde{k}_2(u) \cdot u \cdot P_j(u; d) \cdot (1 - u^2)^{(d-3)/2} \, du$$

$$= C_2 \cdot j/(2j + d - 2) \cdot \widetilde{\varrho}_{j-1} + C_2 \cdot (j + d - 2)/(2j + d - 2) \cdot \widetilde{\varrho}_{j+1} \leq C_2(\widetilde{\rho}_{j-1} + \widetilde{\rho}_{j+1}), \quad \text{(G.18)}$$

where $C_1$ and $C_2$ are constants and in the second equality, we utilize the following recurrence relation of Legendre polynomials:

$$u \cdot P_j(u; d) = j/(2j + d - 2) \cdot P_{j-1}(u; d) + (j + d - 2)/(2j + d - 2) \cdot P_{j+1}(u; d).$$

Notice that $\{\widetilde{\varrho}_j\}_{j \geq 0}$ satisfy (G.12). Thus, combining (G.12) and (G.18), we obtain (G.15). Moreover, when ordering all the eigenvalues of $K_{\mathrm{ntk}}$ in the descending order and renaming them as $\{\sigma_j\}_{j \geq 1}$, similar to (G.16), we have

$$\sigma_j \leq C_2 \cdot (\widetilde{\rho}_{t-1} + \widetilde{\rho}_{t+1}) \qquad \text{for } \sum_{i=1}^{t-1} N(d, i) \leq j < \sum_{i=1}^{t} N(d, i). \tag{G.19}$$

Using a similar analysis, we can show that $\{\sigma_j\}_{j \geq 1}$ satisfy the $\gamma$-exponential eigenvalue decay condition with $\gamma = 1/d$. Therefore, we have shown that the NTK given in (G.17) satisfy Assumption 4.3.

# H  Proofs of the Supporting Lemmas

## H.1  Proof of Lemma E.1

*Proof.* For ease of presentation, before presenting the proof, we first define two operators $\mathbb{J}_h^\star$ and $\mathbb{J}_{t,h}$ respectively by letting

$$(\mathbb{J}_h^\star f)(x) = \langle f(x, \cdot), \pi_h^\star(\cdot \,|\, x) \rangle_{\mathcal{A}}, \quad (\mathbb{J}_{t,h} f)(x) = \langle f(x, \cdot), \pi_h^t(\cdot \,|\, x) \rangle_{\mathcal{A}}, \tag{H.1}$$

for any $(t, h) \in [T] \times [H]$ and any function $f : \mathcal{S} \times \mathcal{A} \to \mathbb{R}$. Moreover, for any $(t, h) \in [T] \times [H]$ and any state $x \in \mathcal{S}$, we define

$$\xi_h^t(x) = (\mathbb{J}_h Q_h^t)(x) - (\mathbb{J}_{t,h} Q_h^t)(x) = \langle Q_h^t(x, \cdot), \pi_h^\star(\cdot \,|\, x) - \pi_h^t(\cdot \,|\, x) \rangle_{\mathcal{A}}. \tag{H.2}$$

After introducing this notation, to prove (E.4) we decompose the instantaneous regret at the $t$-th episode into the following two terms,

$$V_1^\star(x_1^t) - V_1^{\pi^t}(x_1^t) = \underbrace{V_1^\star(x_1^t) - V_1^t(x_1^t)}_{(i)} + \underbrace{V_1^t(x_1^t) - V_1^{\pi^t}(x_1^t)}_{(ii)}. \tag{H.3}$$

In the sequel, we consider the two terms in (H.3) separately.

**Term (i).** By the definitions of the value function $V_h^\star$ in (2.2) and the operator $\mathbb{J}_h^\star$ in (H.1), we have $V_h^\star = \mathbb{J}_h^\star Q_h^\star$. Similarly, for all the algorithms, we have $V_h^t(x) = \langle Q_h^t(x, \cdot), \pi_h^t(\cdot \,|\, x) \rangle$ for all $x \in \mathcal{S}$. Thus, by the definition of $\mathbb{J}_{t,h}$ in (H.1), we have $V_h^t = \mathbb{J}_{t,h} Q_h^t$. Thus, using $\xi_h^t$ defined in (H.2), for any $(t, h) \in [T] \times [H]$, we have

$$\begin{aligned} V_h^\star - V_h^t &= \mathbb{J}_h^\star Q_h^\star - \mathbb{J}_{t,h} Q_h^t = \left( \mathbb{J}_h^\star Q_h^\star - \mathbb{J}_h^\star Q_h^t \right) + \left( \mathbb{J}_h^\star Q_h^t - \mathbb{J}_{t,h} Q_h^t \right) \\ &= \mathbb{J}_h^\star (Q_h^\star - Q_h^t) + \xi_h^t, \end{aligned} \tag{H.4}$$

where the last equality follows from the definition of $\xi_h^t$ in (H.2) and the fact that $\mathbb{J}_h^\star$ is a linear operator. Moreover, by the definition of the temporal-difference error $\delta_h^t$ in (E.1) and the Bellman optimality condition, we have

$$Q_h^\star - Q_h^t = \left( r_h + \mathbb{P}_h V_{h+1}^\star \right) - \left( r_h + \mathbb{P}_h V_{h+1}^t - \delta_h^t \right) = \mathbb{P}_h(V_{h+1}^\star - V_{h+1}^t) + \delta_h^t. \tag{H.5}$$

Thus, combining (H.4) and (H.5), we obtain that

$$V_h^\star - V_h^t = \mathbb{J}_h^\star \mathbb{P}_h(V_{h+1}^\star - V_{h+1}^t) + \mathbb{J}_h^\star \delta_h^t + \xi_h^t, \qquad \forall (t, h) \in [T] \times [H]. \tag{H.6}$$

Equivalently, for all $x \in \mathcal{S}$, and all $(t, h) \in [T] \times [H]$, we have

$$\begin{aligned} V_h^\star(x) - V_h^t(x) =\ & \mathbb{E}_{a \sim \pi_h^\star(\cdot \,|\, x)} \left\{ \mathbb{E} \left[ V_{h+1}^\star(x_{h+1}) - V_{h+1}^t(x_{h+1}) \,\middle|\, x_h = x, a_h = a \right] \right\} \\ & + \mathbb{E}_{a \sim \pi_h^\star(\cdot \,|\, x)} \left[ \delta_h^t(x, a) \right] + \xi_h^t(x). \end{aligned}$$

Then, by recursively applying (H.6) for all $h \in [H]$, we have

$$V_1^\star - V_1^t = \left( \prod_{h=1}^{H} \mathbb{J}_h^\star \mathbb{P}_h \right)(V_{H+1}^\star - V_{H+1}^k) + \sum_{h=1}^{H} \left( \prod_{i=1}^{h-1} \mathbb{J}_i^\star \mathbb{P}_i \right) \mathbb{J}_h^\star \delta_h^t + \sum_{h=1}^{H} \left( \prod_{i=1}^{h-1} \mathbb{J}_i^\star \mathbb{P}_i \right) \xi_h^t. \tag{H.7}$$

Furthermore, notice that we have $V_{H+1}^\star = V_{H+1}^k = \mathbf{0}$. Thus, (H.7) can be equivalently written as

$$V_1^\star(x) - V_1^t(x) = \mathbb{E}_{\pi^\star}\left[\sum_{h=1}^{H}\langle Q_h^t(x_h,\cdot), \pi_h^\star(\cdot\,|\,x_h) - \pi_h^t(\cdot\,|\,x_h)\rangle_{\mathcal{A}} + \delta_h^t(x_h,a_h)\,\Big|\,x_1 = x\right],$$

where we utilize the definition of $\xi_h^t$ given in (H.2). Thus, we can write Term (i) on the right-hand side of (H.3) as

$$V_1^\star(x_t^t) - V_1^t(x_t^t) = \sum_{h=1}^{H}\mathbb{E}_{\pi^\star}\left[\langle Q_h^t(x_h,\cdot), \pi_h^\star(\cdot\,|\,x_h) - \pi_h^t(\cdot\,|\,x_h)\rangle_{\mathcal{A}}\,\big|\,x_1 = x_t^t\right]$$

$$+ \sum_{h=1}^{H}\mathbb{E}_{\pi^\star}[\delta_h^t(x_h,a_h)\,|\,x_1 = x_t^t], \qquad \forall t \in [T]. \tag{H.8}$$

**Term (ii).** It remains to bound the second term on the right-hand side of (H.3). By the definition of the temporal-difference error $\delta_h^t$ in (E.1), for any $(t,h) \in [T] \times [H]$, we have

$$\delta_h^t(x_h^t, a_h^t) = r_h(x_h^t, a_h^t) + (\mathbb{P}_h V_{h+1}^t)(x_h^t, a_h^t) - Q_h^t(x_h^t, a_h^t)$$

$$= \left[r_h(x_h^t, a_h^t) + (\mathbb{P}_h V_{h+1}^t)(x_h^t, a_h^t) - Q_h^{\pi^t}(x_h^t, a_h^t)\right] + \left[Q_h^{\pi^t}(x_h^t, a_h^t) - Q_h^t(x_h^t, a_h^t)\right]$$

$$= \left(\mathbb{P}_h V_{h+1}^t - \mathbb{P}_h V_{h+1}^{\pi^t}\right)(x_h^t, a_h^t) + (Q_h^{\pi^t} - Q_h^t)(x_h^t, a_h^t), \tag{H.9}$$

where the last equality follows from the Bellman equation (2.1). Moreover, recall that we define $\zeta_{t,h}^1$ and $\zeta_{t,h}^2$ in (E.2) and (E.3), respectively. Thus, from (H.9) we obtain that

$$V_h^t(x_h^t) - V_h^{\pi^t}(x_h^t) \tag{H.10}$$

$$= V_h^t(x_h^t) - V_h^{\pi^t}(x_h^t) + (Q_h^{\pi^t} - Q_h^t)(x_h^t, a_h^t) + \left(\mathbb{P}_h(V_{h+1}^t - V_{h+1}^{\pi^t})\right)(x_h^t, a_h^t) - \delta_h^t(x_h^t, a_h^t),$$

$$= \left(V_h^t - V_h^{\pi^t}\right)(x_h^t) - (Q_h^t - Q_h^{\pi^t})(x_h^t, a_h^t)$$

$$\quad + \left(\mathbb{P}_h(V_{h+1}^t - V_{h+1}^{\pi^t})\right)(x_h^t, a_h^t) - (V_{h+1}^t - V_{h+1}^{\pi^t})(x_{h+1}^t) + (V_{h+1}^t - V_{h+1}^{\pi^t})(x_{h+1}^t) - \delta_h^t(x_h^t, a_h^t)$$

$$= \left[V_{h+1}^t(x_{h+1}^t) - V_{h+1}^{\pi^t}(x_{h+1}^t)\right] + \zeta_{t,h}^1 + \zeta_{t,h}^2 - \delta_h^t(x_h^t, a_h^t).$$

Thus, recursively applying (H.10) for all $h \in [H]$, we obtain that

$$V_1^t(x_1^t) - V_1^{\pi^t}(x_1^t) = V_{H+1}^t(x_{H+1}^k) - V_{H+1}^{\pi^k, k}(x_{H+1}^t) + \sum_{h=1}^{H}(\zeta_{t,h}^1 + \zeta_{t,h}^2) - \sum_{h=1}^{H}\delta_h^t(x_h^t, a_h^t)$$

$$= \sum_{h=1}^{H}(\zeta_{t,h}^1 + \zeta_{t,h}^2) - \sum_{h=1}^{H}\delta_h^t(x_h^t, a_h^t), \qquad \forall t \in [T], \tag{H.11}$$

where the last equality follows from the fact that $V_{H+1}^t(x_{H+1}^t) = V_{H+1}^{\pi^t}(x_{H+1}^t) = 0$. Thus, we have simplified Term (ii) defined in (H.3).

Thus, combining (H.3), (H.8), and (H.11), we obtain that

$$\text{Regret}(T) = \sum_{t=1}^{T}\left[V_1^\star(x_1^t) - V_1^{\pi^t}(x_1^t)\right]$$

$$= \sum_{t=1}^{T}\sum_{h=1}^{H}\mathbb{E}_{\pi^\star}[\delta_h^t(x_h,a_h)\,|\,x_1 = x_t^t] + \sum_{t=1}^{T}\sum_{h=1}^{H}(\zeta_{t,h}^1 + \zeta_{t,h}^2) - \sum_{t=1}^{T}\sum_{h=1}^{H}\delta_h^t(x_h^t, a_h^t)$$

$$+ \sum_{t=1}^{T}\sum_{h=1}^{H}\mathbb{E}_{\pi^\star}\left[\langle Q_h^t(x_h,\cdot), \pi_h^\star(\cdot\,|\,x_h) - \pi_h^t(\cdot\,|\,x_h)\rangle_{\mathcal{A}}\,\big|\,x_1 = x_t^t\right].$$

Therefore, we conclude the proof of this lemma. □

## H.2 Proof of Lemma E.2

*Proof.* For ease of presentation, we utilize the feature representation induced by the kernel $K$. Let $\phi\colon \mathcal{Z} \to \mathcal{H}$ be the feature mapping such that $K(z, z') = \langle \phi(z), \phi(z')\rangle_{\mathcal{H}}$. For simplicity, we formally

view $\phi(z)$ as a vector and write $\langle\phi(z),\phi(z')\rangle_{\mathcal{H}}=\phi(z)^{\top}\phi(z')$. Then, any function $f\colon\mathcal{Z}\to\mathbb{R}$ in the RKHS satisfies $f(z)=\langle\phi(z),f\rangle_{\mathcal{H}}=f^{\top}\phi(z)$. Using the feature representation, we can rewrite the kernel ridge regression in (3.4) as

$$\underset{\theta\in\mathcal{H}}{\text{minimize}}\ L(\theta)=\sum_{\tau=1}^{t-1}\big[r_h(x_h^{\tau},a_h^{\tau})+V_{h+1}^{t}(x_{h+1}^{\tau})-\langle\phi(x_h^{\tau},a_h^{\tau}),\theta\rangle_{\mathcal{H}}\big]^2+\lambda\cdot\|\theta\|_{\mathcal{H}}^2. \qquad \text{(H.12)}$$

We define the feature matrix $\Phi_h^t\colon\mathcal{H}\to\mathbb{R}^{t-1}$ and "covariance matrix" $\Lambda_h^t\colon\mathcal{H}\to\mathcal{H}$ respectively as

$$\Phi_h^t=\big[\phi(z_h^1)^{\top},\ldots,\phi(z_h^{t-1})^{\top}\big]^{\top},\qquad \Lambda_h^t=\sum_{\tau=1}^{t-1}\phi(z_h^{\tau})\phi(z_h^{\tau})^{\top}+\lambda\cdot I_{\mathcal{H}}=\lambda\cdot I_{\mathcal{H}}+(\Phi_h^t)^{\top}\Phi_h^t,$$
$$\text{(H.13)}$$

where $I_{\mathcal{H}}$ is the identity mapping on $\mathcal{H}$. Thus, the Gram matrix $K_h^t$ in (3.7) is equal to $\Phi_h^t(\Phi_h^t)^{\top}$. More specifically, here $\Lambda_h^t$ is a self-adjoint and positive-definite operator. For any $f_1,f_2\in\mathcal{H}$, we denote

$$\Lambda_h^t f_1=\lambda\cdot f_1+\sum_{\tau=1}^{t-1}\phi(z_h^{\tau})\cdot f_1(x_h^{\tau})\in\mathcal{H},\qquad f_1^{\top}\Lambda_h^t f_2=\langle f_1,\Lambda_h^t f\rangle_{\mathcal{H}}.$$

It is not hard to see that all the eigenvalues of $\Lambda_h^t$ are positive and at least $\lambda$. Thus, the inverse operator of $\Lambda_h^t$, denoted by $(\Lambda_h^t)^{-1}$, is well-defined, which is also a self-adjoint and positive-definite operator on $\mathcal{H}$. Similarly, for any $f_1,f_2\in\mathcal{H}$, we let $f_1^{\top}(\Lambda_h^t)^{-1}f_2$ denote $\langle f_1,(\Lambda_h^t)^{-1}f_2\rangle_{\mathcal{H}}$. The eigenvalues of $(\Lambda_h^t)^{-1}$ are all bounded in interval $[0,1/\lambda]$.

In addition, using the feature matrix $\Phi_h^t$ defined in (H.13) and $y_h^t$ defined in (3.6), we can write (H.12) as

$$\underset{\theta\in\mathcal{H}}{\text{minimize}}\ L(\theta)=\|y_h^t-\Phi_h^t\theta\|_2^2+\lambda\cdot\theta^{\top}\theta,$$

whose solution is given by $\widehat{\theta}_h^t=(\Lambda_h^t)^{-1}(\Phi_h^t)^{\top}y_h^t$. and $\widehat{Q}_h^t$ in (3.4) satisfies $\widehat{Q}_h^t(z)=\phi(z)^{\top}\widehat{\theta}_h^t$.

In the sequel, to further simplify the notation, we let $\Phi$ denote $\Phi_h^t$ when its meaning is clear from the context. Since both $(\Phi\Phi^{\top}+\lambda\cdot I)$ and $(\Phi^{\top}\Phi+\lambda\cdot I_{\mathcal{H}})$ are strictly positive definite and

$$(\Phi^{\top}\Phi+\lambda\cdot I_{\mathcal{H}})\Phi^{\top}=\Phi^{\top}(\Phi\Phi^{\top}+\lambda\cdot I),$$

which implies that

$$(\Lambda_h^t)^{-1}\Phi^{\top}=(\Phi\Phi^{\top}+\lambda\cdot I_{\mathcal{H}})^{-1}\Phi^{\top}=\Phi^{\top}(\Phi\Phi^{\top}+\lambda\cdot I)^{-1}=\Phi^{\top}(K_h^t+\lambda\cdot I)^{-1}. \qquad \text{(H.14)}$$

Here $I$ is the identity matrix in $\mathbb{R}^{(t-1)\times(t-1)}$. Thus, by (H.14) we have

$$\widehat{\theta}_h^t=(\Lambda_h^t)^{-1}\Phi^{\top}y_h^t=\Phi^{\top}(K_h^t+\lambda\cdot I)^{-1}y_h^t=\Phi^{\top}\alpha_h^t. \qquad \text{(H.15)}$$

Moreover, $k_h^t$ defined in (3.7) can be written as $k_h^t(z)=\Phi\phi(z)$, which, combined with (H.14), implies

$$\begin{aligned}
\phi(z)&=(\Lambda_h^t)^{-1}\Lambda_h^t\phi(z)=(\Lambda_h^t)^{-1}(\Phi^{\top}\Phi+\lambda\cdot I_{\mathcal{H}})\phi(z)\\
&=(\Lambda_h^t)^{-1}(\Phi^{\top}\Phi)\phi(z)+\lambda\cdot(\Lambda_h^t)^{-1}\phi(z)\\
&=\Phi^{\top}(K_h^t+\lambda\cdot I)^{-1}k_h^t(z)+\lambda\cdot(\Lambda_h^t)^{-1}\phi(z). \qquad \text{(H.16)}
\end{aligned}$$

Thus, we can write $\|\phi(z)\|_{\mathcal{H}}^2=\phi(z)^{\top}\phi(z)$ as

$$\begin{aligned}
\|\phi(z)\|_{\mathcal{H}}^2&=\phi(z)^{\top}\cdot\big[\Phi^{\top}(K_h^t+\lambda\cdot I)^{-1}k_h^t(z)+\lambda\cdot(\Lambda_h^t)^{-1}\phi(z)\big]\\
&=k_h^t(z)^{\top}(K_h^t+\lambda\cdot I)^{-1}k_h^t(z)+\lambda\cdot\phi(z)(\Lambda_h^t)^{-1}\phi(z),
\end{aligned}$$

which implies that we can equivalently write the bonus $b_h^t$ defined in (3.8) as

$$b_h^t(x,a)=\big[\phi(x,a)^{\top}(\Lambda_h^t)^{-1}\phi(x,a)\big]^{1/2}=\|\phi(x,a)\|_{(\Lambda_h^t)^{-1}}. \qquad \text{(H.17)}$$

Combining (H.15) and (H.17), we equivalently write $Q_h^t$ in (3.5) as

$$\begin{aligned}
Q_h^t(x,a)&=\min\big\{\widehat{Q}_h^t(x,a)+\beta\cdot b_h^t(x,a),H-h+1\big\}^{+}\\
&=\min\big\{\phi(x,a)^{\top}\widehat{\theta}_h^t+\beta\cdot\|\phi(x,a)\|_{(\Lambda_h^t)^{-1}},H-h+1\big\}^{+}. \qquad \text{(H.18)}
\end{aligned}$$

Now we are ready to bound the temporal-difference error $\xi_h^t$ defined in (E.1). Noticing that $V_h^t(x) = \max_a Q_h^t(x,a)$ for all $(t,h) \in [T] \times [H]$, we have
$$\delta_h^t = r_h + \mathbb{P}_h V_{h+1}^t - Q_h^t = \mathbb{T}_h^\star Q_{h+1}^t - Q_h^t,$$
where $\mathbb{T}_h^\star$ is the Bellman optimality operator. Under the Assumption 4.1, for all $(t,h) \in [T] \times [H]$, since $Q_{h+1}^t \in [0, H]$, we have $\mathbb{T}_h^\star Q_{h+1}^t \in \mathcal{Q}^\star$. Using the feature representation of RKHS, there exists $\overline{\theta}_h^t \in \mathcal{Q}^\star$ such that $(\mathbb{T}_h^\star Q_{h+1}^t)(z) = \phi(z)^\top \overline{\theta}_h^t$ for all $z \in \mathcal{Z}$.

In the sequel, we consider the difference between $\phi(z)^\top \widehat{\theta}_h^t$ and $\phi(z)^\top \overline{\theta}_h^t$. To begin with, using (H.16), we can write $\phi(z)^\top \overline{\theta}_h^t$ as
$$\phi(z)^\top \overline{\theta}_h^t = k_h^t(z)^\top (K_h^t + \lambda \cdot I)^{-1} \Phi \overline{\theta}_h^t + \lambda \cdot \phi(z)^\top (\Lambda_h^t)^{-1} \overline{\theta}_h^t. \tag{H.19}$$
Hence, combining (H.15) and (H.19), we have
$$\phi(z)^\top \widehat{\theta}_h^t - \phi(z)^\top \overline{\theta}_h^t = \underbrace{k_h^t(z)^\top (K_h^t + \lambda \cdot I)^{-1} (y_h^t - \Phi \overline{\theta}_h^t)}_{\text{(i)}} - \underbrace{\lambda \cdot \phi(z)^\top (\Lambda_h^t)^{-1} \overline{\theta}_h^t}_{\text{(ii)}}. \tag{H.20}$$

We bound Term (i) and Term (ii) on the right-hand side of (H.20) separately. For Term (ii), by the Cauchy-Schwarz inequality, we have
$$\left| \lambda \cdot \phi(z)^\top (\Lambda_h^t)^{-1} \overline{\theta}_h^t \right| \le \left\| \lambda \cdot (\Lambda_h^t)^{-1} \phi(x) \right\|_{\mathcal{H}} \cdot \|\overline{\theta}_h^t\|_{\mathcal{H}} \le R_Q H \cdot \left\| \lambda \cdot (\Lambda_h^t)^{-1} \phi(x) \right\|_{\mathcal{H}} \tag{H.21}$$
$$= R_Q H \cdot \sqrt{\lambda \cdot \phi(z)^\top (\Lambda_h^t)^{-1} \cdot \lambda \cdot I_{\mathcal{H}} \cdot (\Lambda_h^t)^{-1} \phi(x)}$$
$$\le R_Q H \cdot \sqrt{\lambda \cdot \phi(z)(\Lambda_h^t)^{-1} \cdot \Lambda_h^t \cdot (\Lambda_h^t)^{-1} \phi(z)} = \sqrt{\lambda} R_Q H \cdot b_h^t(z).$$
Here the first inequality follows from the Cauchy-Schwarz inequality and the second inequality follows from the fact that $\overline{\theta}_h^t \in \mathcal{Q}^\star$, which implies that $\|\overline{\theta}_h^t\|_{\mathcal{H}} \le R_Q H$. Moreover, the last inequality follows from the fact that $\Lambda_h^t - \lambda \cdot I_{\mathcal{H}}$ is a self-adjoint and positive-semidefinite operator, which means that $f^\top (\Lambda_h^t - \lambda \cdot I_{\mathcal{H}}) f \ge 0$ for all $f \in \mathcal{H}$, and the last equality follows from (H.17).

Furthermore, for Term (i), by the Bellman equation in (2.2) and the definition of $y_h^t$ in (3.6), for any $\tau \in [t-1]$, the $\tau$-th entry of $(y_h^t - \Phi \overline{\theta}_h^t)$ can be written as
$$[y_h^t]_\tau - [\Phi \overline{\theta}_h^t]_\tau = r_h(x_h^\tau, a_h^\tau) + V_{h+1}^t(x_{h+1}^\tau) - \phi(x_h^\tau, a_h^\tau)^\top \overline{\theta}_h^t$$
$$= r_h(x_h^\tau, a_h^\tau) + V_{h+1}^t(x_{h+1}^\tau) - (\mathbb{T}_h^\star Q_{h+1}^t)(x_h^\tau, a_h^\tau)$$
$$= V_{h+1}^t(x_{h+1}^\tau) - (\mathbb{P}_h V_{h+1}^t)(x_h^\tau, a_h^\tau). \tag{H.22}$$
Thus, combining (H.14), (H.20), and (H.22) we have
$$\left| k_h^t(z)^\top (K_h^t + \lambda \cdot I)^{-1} (y_h^t - \Phi \overline{\theta}_h^t) \right|$$
$$= \left| \phi(z)^\top (\Lambda_h^t)^{-1} \left\{ \sum_{\tau=1}^{t-1} \phi(x_h^\tau, a_h^\tau) \cdot \left[ V_{h+1}^k(x_{h+1}^\tau) - (\mathbb{P}_h V_{h+1}^k)(x_h^\tau, a_h^\tau) \right] \right\} \right|$$
$$\le \|\phi(z)\|_{(\Lambda_h^t)^{-1}} \cdot \left\| \sum_{\tau=1}^{t-1} \phi(x_h^\tau, a_h^\tau) \cdot \left[ V_{h+1}^t(x_{h+1}^\tau) - (\mathbb{P}_h V_{h+1}^t)(x_h^\tau, a_h^\tau) \right] \right\|_{(\Lambda_h^t)^{-1}}, \tag{H.23}$$
where the last inequality follows from the Cauchy-Schwarz inequality. In the following, we aim to bound (H.23) by the concentration of self-normalized stochastic processes in the RKHS. However, here $V_{h+1}^t$ depends on the historical data in the first $(t-1)$ episodes and is thus not independent of $\{(x_h^\tau, a_h^\tau, x_{h+1}^\tau)\}_{\tau \in [t-1]}$. To bypass this challenge, in the sequel, we combine the concentration of self-normalized processes and uniform convergence over the function classes that contain each $V_{h+1}^t$.

Specifically, recall that we define function classes $\mathcal{Q}_{\text{ucb}}(h, R, B)$ in (4.4) for any $h \in [H]$, and any $R, B > 0$. We define $\mathcal{V}_{\text{ucb}}(h, R, B)$ as
$$\mathcal{V}_{\text{ucb}}(h, R, B) = \left\{ V : V(\cdot) = \max_{a \in \mathcal{A}} Q(\cdot, a) \text{ for some } Q \in \mathcal{Q}_{\text{ucb}}(h, R, B) \right\}. \tag{H.24}$$

In the following, we find a parameter $R_T$ such that $V_h^t \in \mathcal{V}_{\text{ucb}}(h, R_T, B_T)$ holds for all $h \in [H]$ and $t \in [T]$, where $B_T$ is specified in (4.5). Here both $R_T$ and $B_T$ depend on $T$. By (4.4) and (H.18), it

suffices to set $R_T$ as an upper bound of $\|\widehat{\theta}_h^t\|_{\mathcal{H}}$ for all $(t, h) \in [T] \times [H]$. In the following lemma, we bound the RKHS norm of each $\widehat{\theta}_h^t$.

**Lemma H.1** (RKHS Norm of $\widehat{\theta}_h^t$). When $\lambda \geq 1$, for any $(t, h) \in [T] \times [H]$, $\widehat{\theta}_h^t$ defined in (H.15) satisfies

$$\left\|\widehat{\theta}_h^t\right\|_{\mathcal{H}} \leq H\sqrt{2/\lambda \cdot \mathrm{logdet}(I + K_h^t/\lambda)} \leq 2H\sqrt{\Gamma_K(T, \lambda)},$$

where $K_h^t$ is defined in (3.7) and $\Gamma_K(T, \lambda)$ is defined in (I.16).

*Proof.* See §J.1 for a detailed proof. $\qquad\square$

By this lemma, in the sequel, we set $R_T = 2H\sqrt{\Gamma_K(T, \lambda)}$. To conclude the proof, we show that the sum of the two terms in (H.20) is bounded by $\beta \cdot \|\phi(z)\|_{(\Lambda_h^t)^{-1}}$, where we set $\beta = B_T$. To this end, for any two value functions $V, V' \colon \mathcal{S} \to \mathbb{R}$, we define their distance as $\mathrm{dist}(V, V') = \sup_{x \in \mathcal{S}} |V(x) - V'(x)|$. For any $\epsilon \in (0, 1/e)$, any $B > 0$, and any $h \in [H]$, we let $N_{\mathrm{dist}}(\epsilon; h, B)$ be the $\epsilon$-covering number of $\mathcal{V}_{\mathrm{ucb}}(h, R_T, B)$ with respect to distance $\mathrm{dist}(\cdot, \cdot)$. Recall that we define $N_\infty(\epsilon; h, B)$ as the $\epsilon$-covering number of $\mathcal{Q}_{\mathrm{ucb}}(h, R_T, B)$ with respect to the $\ell_\infty$-norm on $\mathcal{Z}$. Note that for any $Q, Q' \colon \mathcal{Z} \to \mathbb{R}$, we have

$$\sup_{x \in \mathcal{S}} \left| \max_{a \in \mathcal{A}} Q(x, a) - \max_{a \in \mathcal{A}} Q'(x, a) \right| \leq \sup_{(x, a) \in \mathcal{S} \times \mathcal{A}} |Q(x, a) - Q'(x, a)| = \|Q - Q'\|_\infty.$$

By (H.24) we have $N_{\mathrm{dist}}(\epsilon; h, B) \leq N_\infty(\epsilon; h, B)$. Then, by applying Lemma J.2 with $\delta = (2T^2H^3)^{-1}$ and taking a union bound over $h \in [H]$, we obtain that

$$\left\| \sum_{\tau=1}^{t-1} \phi(x_h^\tau, a_h^\tau) \cdot \left[ V_{h+1}^t(x_{h+1}^\tau) - (\mathbb{P}_h V_{h+1}^t)(x_h^\tau, a_h^\tau) \right] \right\|_{(\Lambda_h^t)^{-1}}^2$$

$$\leq \sup_{V \in \mathcal{V}_{\mathrm{ucb}}(h+1, R_T, B_T)} \left\| \sum_{\tau=1}^{t-1} \phi(x_h^\tau, a_h^\tau) \cdot \left[ V(x_{h+1}^\tau) - (\mathbb{P}_h V)(x_h^\tau, a_h^\tau) \right] \right\|_{(\Lambda_h^t)^{-1}}^2$$

$$\leq 2H^2 \cdot \mathrm{logdet}(I + K_h^t/\lambda) + 2H^2 t \cdot (\lambda - 1) + 8t^2\epsilon^2/\lambda$$

$$\qquad + 4H^2 \cdot \left[ \log N_\infty(\epsilon; h+1, B_T) + \log(2T^2H^3) \right] \tag{H.25}$$

holds uniformly for all $(t, h) \in [T] \times [H]$ with probability at least $1 - (2T^2H^2)^{-2}$, where we utilize the fact that $V_{h+1}^t \in \mathcal{V}_{\mathrm{ucb}}(h+1, R_T, B_T)$. Note that we set $\lambda = 1 + 1/T$. Then, setting $\epsilon$ as $\epsilon^* = H/T$, (H.25) is further reduced to

$$\left\| \sum_{\tau=1}^{t-1} \phi(x_h^\tau, a_h^\tau) \cdot \left[ V_{h+1}^t(x_{h+1}^\tau) - (\mathbb{P}_h V_{h+1}^t)(x_h^\tau, a_h^\tau) \right] \right\|_{(\Lambda_h^t)^{-1}}^2$$

$$\leq 4H^2 \cdot \Gamma_K(T, \lambda) + 11H^2 + 4H^2 \cdot \log N_\infty(\epsilon^*; h+1, B_T) + 8H^2 \cdot \log(TH). \tag{H.26}$$

Thus, combining (H.17), (H.20), (H.21), (H.23), and (H.26), we obtain that

$$\left| \phi(z)^\top (\widehat{\theta}_h^t - \overline{\theta}_h^t) \right|$$

$$\leq H \cdot \left\{ \left[ 4 \cdot \Gamma_K(T, \lambda) + 4 \cdot \log N_\infty(\epsilon^*; h+1, B_T) + 8 \cdot \log(TH) + 11 \right]^{1/2} + \sqrt{\lambda} R_Q \right\} \cdot b_h^t(z)$$

$$\leq H \cdot \left[ 8 \cdot \Gamma_K(T, \lambda) + 8 \cdot \log N_\infty(\epsilon^*; h+1, B_T) + 16 \cdot \log(TH) + 22 + 2R_Q^2\lambda \right]^{1/2} \cdot b_h^t(z)$$

$$\leq B_T \cdot b_h^t(z) = \beta \cdot b_h^t(z) \tag{H.27}$$

holds uniformly for all $(t, h) \in [T] \times [H]$ with probability at least $1 - (2T^2H^2)^{-1}$, where the second inequality follows from the elementary inequality $\sqrt{a} + \sqrt{b} \leq \sqrt{2(a^2 + b^2)}$, and the last inequality follows from the assumption on $B_T$ given in (4.5).

Finally, by (H.27) and the definition of the temporal-difference error $\delta_h^t$ in (E.1), we have

$$-\delta_h^t(z) = Q_h^t(z) - \phi(z)^\top \overline{\theta}_h^t \leq \phi(z)^\top (\widehat{\theta}_h^t - \overline{\theta}_h^t) + \beta \cdot b_h^t(z) \leq 2\beta \cdot b_h^t(z). \tag{H.28}$$

In addition, since $Q_{h+1}^t(z) \leq H - h$ for all $z \in \mathcal{Z}$, we have $(\mathbb{T}_h^\star Q_{h+1}^t) \leq H - h + 1$. Hence, we have

$$\delta_h^t(z) = \phi(z)^\top \overline{\theta}_h^t - \min\{\phi(z)^\top \widehat{\theta}_h^t + \beta \cdot b_h^t(z), H - h + 1\}^+$$

$$\leq \max\{\phi(z)^\top \overline{\theta}_h^t - \phi(z)^\top \widehat{\theta}_h^t - \beta \cdot b_h^t(z), \phi(z)^\top \overline{\theta}_h^t - (H - h + 1)\} \leq 0. \qquad \text{(H.29)}$$

Therefore, combining (H.28) and (H.29), we conclude the proof of Lemma E.2. $\qquad \square$

## H.3 Proof of Lemma E.3

*Proof.* Following [12], we prove this lemma by showing that $\{\zeta_{t,h}^1, \zeta_{t,h}^2\}_{(t,h)\in[T]\times[H]}$ can be written as a bounded martingale difference sequence with respect to a filtration. In particular, we construct the filtration explicitly as follows. For any $(t, h) \in [T] \times [H]$, we define $\sigma$-algebras $\mathcal{F}_{t,h,1}$ and $\mathcal{F}_{t,h,2}$ as follows:

$$\mathcal{F}_{t,h,1} = \sigma\big(\{(x_i^\tau, a_i^\tau)\}_{(\tau,i)\in[t-1]\times[H]} \cup \{(x_i^t, a_i^t)\}_{i\in[h]}\big),$$
$$\mathcal{F}_{t,h,2} = \sigma\big(\{(x_i^\tau, a_i^\tau)\}_{(\tau,i)\in[t-1]\times[H]} \cup \{(x_i^t, a_i^t)\}_{i\in[h]} \cup \{x_{h+1}^t\}\big), \qquad \text{(H.30)}$$

where $\sigma(\cdot)$ denotes the $\sigma$-algebra generated by a finite set. Moreover, for any $t \in [T]$, $h \in [H]$ and $m \in [2]$, we define the timestep index $\tau(t, h, m)$ as

$$\tau(t, h, m) = (t - 1) \cdot 2H + (h - 1) \cdot 2 + m, \qquad \text{(H.31)}$$

which offers an partial ordering over the triplets $(t, h, m) \in [T] \times [H] \times [2]$. Moreover, by the definitions in (H.30), for any $(t, h, m)$ and $(t', h', m')$ satisfying $\tau(k, h, m) \leq \tau(k', h', m')$, it holds that $\mathcal{F}_{k,h,m} \subseteq \mathcal{F}_{k',h',m'}$. Thus, the sequence of $\sigma$-algebras $\{\mathcal{F}_{t,h,m}\}_{(t,h,m)\in[T]\times[H]\times[2]}$ forms a filtration.

Furthermore, for any $(t, h) \in [T] \times [H]$, since both $Q_h^t$ and $V_h^t$ are obtained based on the trajectories of the first $(t - 1)$ episodes, they are both measurable with respect to $\mathcal{F}_{t,1,1}$, which is a subset of $\mathcal{F}_{t,h,m}$ for all $h \in [H]$ and $m \in [2]$. Thus, by (H.30), $\zeta_{t,h}^1$ defined in (E.2) and $\zeta_{t,h}^2$ defined in (E.3) are measurable with respect to $\mathcal{F}_{t,h,1}$ and $\mathcal{F}_{t,h,2}$, respectively. In addition, note that $a_h^t \sim \pi_h^t(\cdot \mid x_h^t)$ and that $x_{h+1}^t \sim \mathbb{P}_h(\cdot \mid x_h^t, a_h^t)$. Thus, we have

$$\mathbb{E}\big[\zeta_{t,h}^1 \mid \mathcal{F}_{t,h-1,2}\big] = 0, \qquad \mathbb{E}\big[\zeta_{t,h}^2 \mid \mathcal{F}_{t,h,1}\big] = 0, \qquad \text{(H.32)}$$

where we identify $\mathcal{F}_{t,0,2}$ with $\mathcal{F}_{t-1,H,2}$ for all $t \geq 2$ and let $\mathcal{F}_{1,0,2}$ be the empty set. Combining (H.31) and (H.32), we can define a martingale $\{M_{t,h,m}\}_{(t,h,m)\in[T]\times[H]\times[2]}$ indexed by $\tau(t, k, m)$, defined in (H.31), as follows. For any $(t, h, m) \in [T] \times [H] \times [2]$, we define

$$M_{t,h,m} = \Big\{\sum_{(s,g,\ell)} \zeta_{s,g}^\ell : \tau(s, g, \ell) \leq \tau(t, h, m)\Big\}; \qquad \text{(H.33)}$$

that is, $M_{t,h,m}$ is the sum of all terms of the form $\zeta_{s,g}^\ell$ defined in (E.2) or (E.3) such that its timestep index $\tau(s, g, \ell)$ is no greater than $\tau(t, h, m)$. By definition, we have

$$M_{K,H,2} = \sum_{t=1}^T \sum_{h=1}^H (\zeta_{t,h}^1 + \zeta_{t,h}^2). \qquad \text{(H.34)}$$

Moreover, since $V_h^t$, $Q_h^t$, $V_h^{\pi^t}$, and $Q_h^{\pi^t}$ all takes values in $[0, H]$, we have $|\zeta_{t,h}^1| \leq 2H$ and $|\zeta_{t,h}^2| \leq 2H$ for all $(t, h) \in [T] \times [H]$. This means that the martingale $M_{t,h,m}$ defined in (H.33) has uniformly bounded differences. Thus, applying the Azuma-Hoeffding inequality [7] to $M_{T,H,2}$ in (H.34), we obtain that

$$\mathbb{P}\Big(\Big|\sum_{t=1}^T \sum_{h=1}^H (\zeta_{t,h}^1 + \zeta_{t,h}^2)\Big| > t\Big) \leq 2\exp\Big(\frac{-t^2}{16TH^3}\Big) \qquad \text{(H.35)}$$

holds for all $t > 0$. Finally, we set the right-hand side of (H.35) to $\zeta$ for some $\zeta \in (0, 1)$, which yields $t = \sqrt{16TH^3 \cdot \log(2/\zeta)}$. Thus, we obtain that

$$\Big|\sum_{t=1}^T \sum_{h=1}^H (\zeta_{t,h}^1 + \zeta_{t,h}^2)\Big| \leq \sqrt{16TH^3 \cdot \log(2/\zeta)},$$

with probability at least $1 - \zeta$, which concludes the proof. $\qquad \square$

## H.4  Proof of Lemma E.4

*Proof.* The proof of this lemma utilizes the connection between overparameterized neural networks and NTKs. Recall that we denote $z = (x, a)$ and $\mathcal{Z} = \mathcal{S} \times \mathcal{A}$. Also recall that $(b^{(0)}, W^{(0)})$ is the initial value of the network parameters obtained by the symmetric initialization scheme introduced in §B.2. Thus, $f(\cdot; W^{(0)})$ is a zero function. For any $(t, h) \in [T] \times [H]$, since $\widehat{W}_h^t$ is the global minimizer of loss function $L_h^t$ defined in (C.1), we have

$$
L_h^t(\widehat{W}_h^t) = \sum_{\tau=1}^{t-1} \big[ r_h(x_h^\tau, a_h^\tau) + V_{h+1}^t(x_{h+1}^\tau) - f(x_h^\tau, a_h^\tau; \widehat{W}_h^t) \big]^2 + \lambda \cdot \big\| \widehat{W}_h^t - W^{(0)} \big\|_2^2
$$

$$
\leq L_h^t(W^{(0)}) = \sum_{\tau=1}^{t-1} \big[ r_h(x_h^\tau, a_h^\tau) + V_{h+1}^t(x_{h+1}^\tau) \big]^2 \leq (H - h + 1)^2 \cdot (t - 1) \leq TH^2,
$$
(H.36)

where the second-to-last inequality follows from the facts that $V_{h+1}^t$ is bounded by $H - h$ and that $r_h \in [0, 1]$. Thus, (H.36) implies that

$$
\big\| \widehat{W}_h^t - W^{(0)} \big\|_2^2 \leq TH^2/\lambda, \qquad \forall (t, h) \in [T] \times [H]. \tag{H.37}
$$

That is, each $\widehat{W}_h^t$ belongs to the Euclidean ball $\mathcal{B} = \{ W \in \mathbb{R}^{2md} \colon \|W - W^{(0)}\|_2 \leq H\sqrt{T/\lambda} \}$. Here the regularization parameter $\lambda$ is does not depend on $m$ and will be determined later. Notice that the radius of $\mathcal{B}$ does not depend on $m$. When $m$ is sufficiently large, it can be shown that $f(\cdot, W)$ is close to its linearization, $\widehat{f}(\cdot; W) = \langle \varphi(\cdot; W^{(0)}), W - W^{(0)} \rangle$, for all $W \in \mathcal{B}$, where $\varphi(\cdot; W) = \nabla_W f(\cdot; W)$.

Furthermore, recall that the temporal-difference error $\delta_h^t$ is defined as

$$
\delta_h^t = r_h + \mathbb{P}_h V_{h+1}^t - Q_h^t = \mathbb{T}_h^\star Q_{h+1}^t - Q_h^t.
$$

Under Assumption D.1, we have $\mathbb{T}_h^\star Q_{h+1}^t \in \mathcal{Q}^\star$ for all $(t, h) \in [T] \times [H]$, where $\mathcal{Q}^\star$ is defined in (D.1). That is, for all $(t, h) \in [T] \times [H]$, there exists a function $\alpha_h^t \colon \mathbb{R}^d \to \mathbb{R}^d$ such that

$$
(\mathbb{T}_h^\star Q_{h+1}^t)(z) = \int_{\mathbb{R}^d} \mathrm{act}'(w^\top z) \cdot z^\top \alpha_h^t(w) \, \mathrm{d}p_0(w), \qquad \forall (t, h) \in [T] \times [H], \forall z \in \mathcal{Z}. \tag{H.38}
$$

Moreover, it holds that $\|\alpha_h^t\|_{2,\infty} = \sup_w \|\alpha_h^t(w)\|_2 \leq R_Q H/\sqrt{d}$.

Now we are ready to bound the temporal-difference error $\delta_h^t$ defined in (E.1). Our proof is decomposed into three steps.

**Step I.** In the first step, we show that, with high probability, $\mathbb{T}_h^\star Q_{h+1}^t$ can be well-approximated by the class of linear functions of $\varphi(\cdot; W^{(0)})$ with respect to the $\ell_\infty$-norm.

Specifically, by Proposition C.1 in [30], with probability at least $1 - m^{-2}$ over the randomness of initialization, for any $(t, h) \in [T] \times [H]$, there exists a function $\widetilde{Q}_h^t \colon \mathcal{Z} \to \mathbb{R}$ that can be written as

$$
\widetilde{Q}_h^t(z) = \frac{1}{\sqrt{m}} \sum_{j=1}^m \mathrm{act}'(\langle W_j^{(0)}, z \rangle) \cdot z^\top \alpha_j, \tag{H.39}
$$

where $\|\alpha_j\|_2 \leq R_Q/\sqrt{dm}$ for all $j \in [m]$ and $\{W_j^{(0)}\}_{j \in [2m]}$ are the random weights generated in the symmetric initialization scheme. Moreover, $\widetilde{Q}_h^t$ satisfies that

$$
\|\widetilde{Q}_h^t - \mathbb{T}_h^\star Q_{h+1}^t\|_\infty \leq 10 C_{\mathrm{act}} R_Q H \cdot \sqrt{\log(mTH)/m}. \tag{H.40}
$$

Also, for any $j \in [2m]$, let $W_j^{(0)}$ and $b_j^{(0)}$ be the $j$-th component of $b^{(0)}$ and $W^{(0)}$, respectively.

Now we show that $\widetilde{Q}_h^t$ in (H.39) can be written as $\varphi(\cdot; W^{(0)})^\top (\widetilde{W}_h^t - W^{(0)})$ for some $\widetilde{W}_h^t \in \mathbb{R}^{2md}$. To this end, we define $\widetilde{W}_h^t = (\widetilde{W}_1, \ldots \widetilde{W}_{2m}) \in \mathbb{R}^{2md}$ as follows. For any $j \in [m]$, we let $\widetilde{W}_j = W_j^{(0)} + b_j^{(0)} \cdot \alpha_j/\sqrt{2}$, and for any $j \in \{m+1, \ldots, 2m\}$, we let $\widetilde{W}_j = W_j^{(0)} + b_j^{(0)} \cdot \alpha_{j-m}/\sqrt{2}$.

Then, by the symmetric initialization scheme, we have

$$
\begin{aligned}
\widetilde{Q}_h^t(z) &= \frac{1}{\sqrt{2m}} \sum_{j=1}^m \sqrt{2} \cdot (b_j^{(0)})^2 \cdot \mathrm{act}'\big(\langle W_j^{(0)}, z \rangle\big) \cdot z^\top \alpha_j \\
&= \frac{1}{\sqrt{2m}} \sum_{j=1}^m 1/\sqrt{2} \cdot (b_j^{(0)})^2 \cdot \mathrm{act}'\big(\langle W_j^{(0)}, z \rangle\big) \cdot z^\top \alpha_j \\
&\qquad + \frac{1}{\sqrt{2m}} \sum_{j=1}^m 1/\sqrt{2} \cdot (b_j^{(0)})^2 \cdot \mathrm{act}'\big(\langle W_j^{(0)}, z \rangle\big) \cdot z^\top \alpha_{j-m} \\
&= \frac{1}{\sqrt{2m}} \sum_{j=1}^{2m} b_j^{(0)} \cdot \mathrm{act}'\big(\langle W_j^{(0)}, z \rangle\big) \cdot z^\top \big(\widetilde{W}_j - W_j^{(0)}\big) = \varphi(z; W^{(0)})^\top \big(\widetilde{W}_h^t - W^{(0)}\big).
\end{aligned}
$$

$$(\mathrm{H}.41)$$

Moreover, since $\|\alpha_j\|_2 \leq R_Q H / \sqrt{dm}$, we have $\|\widetilde{W}_h^t - W^{(0)}\|_2 \leq R_Q H / \sqrt{d}$.

Therefore, for all $(t, h) \in [T] \times [H]$, we have constructed $\widetilde{Q}_h^t$ to be linear in $\varphi(\cdot; W^{(0)})$. Moreover, with probability at least $1 - m^{-2}$ over the randomness of initialization, $\widetilde{Q}_h^t$ is close to $\mathbb{T}_h^\star Q_{h+1}^t$ in the sense that (H.40) holds uniformly for all $(t, h) \in [T] \times [H]$. Thus, we conclude the first step.

**Step II.** In the second step, we show that $Q_h^t$ used in Algorithm 4 can be well approximated by functions based on the feature mapping $\varphi(\cdot; W^{(0)})$.

Recall that the bonus in $Q_h^t$ utilizes matrix $\Lambda_h^t$ defined in (C.3), which involves the neural tangent features $\{\varphi(\cdot; \widehat{W}_h^\tau)\}_{\tau \in [T]}$. Similar to $\Lambda_h^t$, we define $\overline{\Lambda}_h^t$ as

$$
\overline{\Lambda}_h^t = \lambda \cdot I_{2md} + \sum_{\tau=1}^{t-1} \varphi(x_h^\tau, a_h^\tau; W^{(0)}) \varphi(x_h^\tau, a_h^\tau; W^{(0)})^\top, \tag{H.42}
$$

which adopts the same feature mapping $\varphi(\cdot; W^{(0)})$. To simplify the notation, hereafter, we use $\varphi(\cdot)$ to denote $\varphi(\cdot; W^{(0)})$ when its meaning is clear from the text. Moreover, for any $(t, h) \in [T] \times [H]$, we define the response vector $y_h^t \in \mathbb{R}^{t-1}$ by letting its entries be

$$
[y_h^t]_\tau = r_h(x_h^\tau, a_h^\tau) + V_{h+1}^t(x_{h+1}^\tau), \qquad \forall \tau \in [t-1]. \tag{H.43}
$$

We define the feature matrix $\Phi_h^t \in \mathbb{R}^{(t-1) \times 2md}$ by

$$
\Phi_h^t = \big[\varphi(x_h^1, a_h^1)^\top, \ldots, \varphi(x_h^{t-1}, a_h^{t-1})^\top\big]^\top. \tag{H.44}
$$

Hence, by (H.42) and (H.44), we have $\overline{\Lambda}_h^t = \lambda \cdot I_{2md} + (\Phi_h^t)^\top \Phi_h^t$. Similar to the bonus function $b_h^t$ defined in (C.4), we define

$$
\overline{b}_h^t = \big[\varphi(x, a)^\top (\overline{\Lambda}_h^t)^{-1} \varphi(x, a)\big]^{1/2} = \|\varphi(x, a)\|_{(\overline{\Lambda}_h^t)^{-1}}. \tag{H.45}
$$

Similar to $L_h^t$ defined in (C.1), we define another least-squares loss function $\overline{L}_h^t : \mathbb{R}^{2md} \to \mathbb{R}$ as

$$
\overline{L}_h^t(W) = \sum_{\tau=1}^{t-1} \big[r_h(x_h^\tau, a_h^\tau) + V_{h+1}^t(x_{h+1}^\tau) - \langle \varphi(x_h^\tau, a_h^\tau), W - W^{(0)} \rangle\big]^2 + \lambda \cdot \|W - W^{(0)}\|_2^2
$$

$$(\mathrm{H}.46)$$

and let $\overline{W}_h^t$ be its global minimizer. By direct computation, $\overline{W}_h^t$ can be written in closed form as

$$
\overline{W}_h^t = W^{(0)} + (\overline{\Lambda}_h^t)^{-1} (\Phi_h^t)^\top y_h^t, \tag{H.47}
$$

where $\overline{\Lambda}_h^t$, $\Phi_h^t$, and $y_h^t$ are defined respectively in (H.42), (H.44), and (H.43). Similar to (H.36), utilizing the fact that $\overline{L}_h^t(\overline{W}_h^t) \leq \overline{L}_h^t(W^{(0)})$, we also have $\|\overline{W}_h^t - W^{(0)}\|_2 \leq H\sqrt{T/\lambda}$. Then, in a manner similar to the construction of $Q_h^t$ in Algorithm 4, we combine $\overline{b}_h^t$ in (H.45) and $\overline{W}_h^t$ in (H.47) to define $\overline{Q}_h^t : \mathcal{Z} \to \mathbb{R}$ as

$$
\overline{Q}_h^t(x, a) = \min\big\{\varphi(x, a)^\top (\overline{W}_h^t - W^{(0)}) + \beta \cdot \overline{b}_h^t(x, a), H - h + 1\big\}^+. \tag{H.48}
$$

Note that $\overline{Q}_h^t$ share the same form as $Q$ in (D.2). Thus, we have $\overline{Q}_h^t \in \mathcal{Q}_{\mathrm{ucb}}(h, H\sqrt{T/\lambda}, B)$ for any $B \geq \beta$. Moreover, we define $\overline{V}_h^t(\cdot) = \max_{a \in \mathcal{A}} \overline{Q}_h^t(\cdot, a)$.

In the following, we aim to show that $\overline{Q}_h^t$ is close to $Q_h^t$ when $m$ is sufficiently large. When this is true, $\overline{V}_h^t$ is also close to $V_h^t$. To bound $Q_h^t - \overline{Q}_h^t$, since the truncation operator is non-expansive, by the triangle inequality we have

$$\|Q_h^t - \overline{Q}_h^t\|_\infty \leq \underbrace{\|f(\cdot; \widehat{W}_h^t) - \varphi(\cdot)^\top (\overline{W}_h^t - W^{(0)})\|_\infty}_{\text{(i)}} + \underbrace{\beta \cdot \|b_h^t - \overline{b}_h^t\|_\infty}_{\text{(ii)}}. \tag{H.49}$$

Recall that we define $\mathcal{B} = \{W \in \mathbb{R}^{2md} : \|W - W^{(0)}\|_2 \leq H\sqrt{T/\lambda}\}$. To bound the two terms on the right-hand side of (H.49), we utilize the following lemma that quantifies the perturbation of $f(\cdot; W)$ and $\varphi(\cdot; W)$ within $W \in \mathcal{B}$.

**Lemma H.2.** When $TH^2 = \mathcal{O}(m \cdot \log^{-6} m)$, with probability at least $1 - m^{-2}$ with respect to the randomness of initialization, for any $W \in \mathcal{B}$ and any $z \in \mathcal{Z}$, we have

$$\left|f(z, W) - \varphi(z, W^{(0)})^\top (W - W^{(0)})\right| \leq \overline{C} \cdot T^{2/3} \cdot H^{4/3} \cdot m^{-1/6} \cdot \sqrt{\log m},$$

$$\left\|\varphi(z, W) - \varphi(z, W^{(0)})\right\|_2 \leq \overline{C} \cdot (TH^2/m)^{1/6} \cdot \sqrt{\log m}, \qquad \|\varphi(z, W)\|_2 \leq \overline{C}.$$

*Proof.* See [3, 30, 13] for a detailed proof. More specifically, this lemma is obtained from Lemmas F.1 and F.2 in [13], which are further based on results in [3, 30]. $\qquad\square$

By Lemma H.2 and triangle inequality, Term (i) on the right-hand side of (H.49) is bounded by

$$\text{Term (i)} \leq \left\|f(\cdot; \widehat{W}_h^t) - \varphi(\cdot)^\top (\widehat{W}_h^t - W^{(0)})\right\|_\infty + \left\|\varphi(\cdot)^\top (\widehat{W}_h^t - \overline{W}_h^t)\right\|_\infty$$

$$\leq \overline{C} \cdot T^{2/3} \cdot H^{4/3} \cdot m^{-1/6} \cdot \sqrt{\log m} + \overline{C} \cdot \left\|\widehat{W}_h^t - \overline{W}_h^t\right\|_2. \tag{H.50}$$

To bound $\left\|\widehat{W}_h^t - \overline{W}_h^t\right\|_2$, notice that $\widehat{W}_h^t$ and $\overline{W}_h^t$ are the global minimizers of $L_h^t$ in (C.1) and $\overline{L}_h^t$ in (H.46), respectively. Thus, by the first-order optimality condition, we have

$$\lambda \cdot \left(\widehat{W}_h^t - W^{(0)}\right) = \sum_{\tau=1}^{t-1} \{[y_h^t]_\tau - f(z_h^\tau; \widehat{W}_h^t)\} \cdot \varphi(z_h^\tau; \widehat{W}_h^t), \tag{H.51}$$

$$\lambda \cdot (\overline{W}_h^t - W^{(0)}) = \sum_{\tau=1}^{t-1} \{[y_h^t]_\tau - \langle \varphi(z_h^\tau; W^{(0)}), \overline{W}_h^t - W^{(0)}\rangle\} \cdot \varphi(z_h^\tau; W^{(0)}), \tag{H.52}$$

where $[y_h^t]_\tau$ is defined in (H.43) and $z_h^\tau = (x_h^\tau, a_h^\tau)$. In addition, by the definition of $\overline{\Lambda}_h^t$ in (H.42), (H.52) can be equivalently written as

$$\overline{\Lambda}_h^t (\overline{W}_h^t - W^{(0)}) = \sum_{\tau=1}^{t-1} [y_h^t]_\tau \cdot \varphi(z_h^\tau; W^{(0)}). \tag{H.53}$$

Similarly, for (H.51), by direct computation we have

$$\overline{\Lambda}_h^t (\widehat{W}_h^t - W^{(0)}) = \sum_{\tau=1}^{t-1} [y_h^t]_\tau \cdot \varphi(z_h^\tau; \widehat{W}_h^t) \tag{H.54}$$

$$+ \sum_{\tau=1}^{t-1} [\langle \varphi(z_h^\tau; W^{(0)}), \widehat{W}_h^t - W^{(0)}\rangle \cdot \varphi(z_h^\tau; W^{(0)}) - f(z_h^\tau; \widehat{W}_h^t) \cdot \varphi(z_h^\tau; \widehat{W}_h^t)].$$

For any $\tau \in [t-1]$, we have

$$\langle \varphi(z_h^\tau; W^{(0)}), \widehat{W}_h^t - W^{(0)}\rangle \cdot \varphi(z_h^\tau; W^{(0)}) - f(z_h^\tau; \widehat{W}_h^t) \cdot \varphi(z_h^\tau; \widehat{W}_h^t)$$

$$= \langle \varphi(z_h^\tau; W^{(0)}), \widehat{W}_h^t - W^{(0)}\rangle \cdot [\varphi(z_h^\tau; W^{(0)}) - \varphi(z_h^\tau; \widehat{W}_h^t)]$$

$$+ [\langle \varphi(z_h^\tau; W^{(0)}), \widehat{W}_h^t - W^{(0)}\rangle - f(z_h^\tau; \widehat{W}_h^t)] \cdot \varphi(z_h^\tau; \widehat{W}_h^t). \tag{H.55}$$

Thus, applying Lemma H.2 to (H.55), we have

$$\left\| \left\langle \varphi(z_h^\tau; W^{(0)}), \widehat{W}_h^t - W^{(0)} \right\rangle \cdot \varphi(z_h^\tau; W^{(0)}) - f(z_h^\tau; \widehat{W}_h^t) \cdot \varphi(z_h^\tau; \widehat{W}_h^t) \right\|_2$$

$$\leq \left\| \varphi(z_h^\tau; W^{(0)}) \right\|_2 \cdot \left\| \widehat{W}_h^t - W^{(0)} \right\|_2 \cdot \left\| \varphi(z_h^\tau; W^{(0)}) - \varphi(z_h^\tau; \widehat{W}_h^t) \right\|$$

$$+ \left| \left\langle \varphi(z_h^\tau; W^{(0)}), \widehat{W}_h^t - W^{(0)} \right\rangle - f(z_h^\tau; \widehat{W}_h^t) \right| \cdot \left\| \varphi(z_h^\tau; \widehat{W}_h^t) \right\|_2$$

$$\leq 2\overline{C}^2 \cdot T^{2/3} \cdot H^{4/3} \cdot m^{-1/6} \cdot \sqrt{\log m} \cdot \lambda^{-1/2}, \tag{H.56}$$

where we utilize the fact that $\|\widehat{W}_h^t - W^{(0)}\|_2 \leq H\sqrt{T/\lambda} \leq H\sqrt{T}$. Then, combining (H.53), (H.54), and (H.56), we have

$$\left\| \overline{\Lambda}_h^t (\widehat{W}_h^t - \overline{W}_h^t) \right\|_2$$

$$\leq \left\| \sum_{\tau=1}^{t-1} [y_h^t]_\tau \cdot \left[ \varphi(z_h^\tau; \widehat{W}_h^t) - \varphi(z_h^\tau; W^{(0)}) \right] \right\|_2 + 2\overline{C}^2 \cdot T^{5/3} \cdot H^{4/3} \cdot m^{-1/6} \cdot \sqrt{\log m}$$

$$\leq \overline{C} \cdot T^{7/6} \cdot H^{4/3} \cdot m^{-1/6} \cdot \sqrt{\log m} + 2\overline{C}^2 \cdot T^{5/3} \cdot H^{4/3} \cdot m^{-1/6} \cdot \sqrt{\log m}, \tag{H.57}$$

where in the last inequality we utilize the fact that $[y_h^t]_\tau \in [0, H]$. When $T$ is sufficiently large, the second term in (H.57) dominates. Since the eigenvalues of $(\overline{\Lambda}_h^t)^{-1}$ are all bounded by $1/\lambda$, we have

$$\left\| \widehat{W}_h^t - \overline{W}_h^t \right\|_2 \leq \left\| (\overline{\Lambda}_h^t)^{-1} \right\|_{\mathrm{op}} \cdot \left\| \overline{\Lambda}_h^t (\widehat{W}_h^t - \overline{W}_h^t) \right\|_2 \leq 1/\lambda \cdot \left\| \overline{\Lambda}_h^t (\widehat{W}_h^t - \overline{W}_h^t) \right\|_2. \tag{H.58}$$

In the sequel, we set $\lambda$ as

$$\lambda = \overline{C}^2 \cdot (1 + 1/T) \in \left[ \overline{C}^2, 2\overline{C}^2 \right]. \tag{H.59}$$

Thus, combining (H.50), (H.57), (H.58), and (H.59), we have

$$\text{Term (i)} \leq 4 \cdot T^{5/3} \cdot H^{4/3} \cdot m^{-1/6} \cdot \sqrt{\log m} \tag{H.60}$$

where we use the fact that $\overline{C}^2/\lambda \leq 1$.

Furthermore, to bound Term (ii), by the definitions of $b_h^t$ and $\overline{b}_h^t$, for any $z \in \mathcal{Z}$, we have

$$\left| b_h^t(z) - \overline{b}_h^t(z) \right| = \left| \sqrt{\varphi(z; \widehat{W}_h^t)^\top (\Lambda_h^t)^{-1} \varphi(z; \widehat{W}_h^t)} - \sqrt{\varphi(z; W^{(0)})^\top (\overline{\Lambda}_h^t)^{-1} \varphi(z; W^{(0)})} \right|$$

$$\leq \sqrt{\left| \varphi(z; \widehat{W}_h^t)^\top (\Lambda_h^t)^{-1} \varphi(z; \widehat{W}_h^t) - \varphi(z; W^{(0)})^\top (\overline{\Lambda}_h^t)^{-1} \varphi(z; W^{(0)}) \right|}, \quad \text{(H.61)}$$

where the inequality follows from the elementary inequality $|\sqrt{x} - \sqrt{y}| \leq \sqrt{|x - y|}$. By the triangle inequality

$$\left| \varphi(z; \widehat{W}_h^t)^\top (\Lambda_h^t)^{-1} \varphi(z; \widehat{W}_h^t) - \varphi(z; W^{(0)})^\top (\overline{\Lambda}_h^t)^{-1} \varphi(z; W^{(0)}) \right|$$

$$\leq \left| \left[ \varphi(z; \widehat{W}_h^t) - \varphi(z; W^{(0)}) \right]^\top (\Lambda_h^t)^{-1} \varphi(z; \widehat{W}_h^t) \right| + \left| \varphi(z; W^{(0)})^\top \left[ (\Lambda_h^t)^{-1} - (\overline{\Lambda}_h^t)^{-1} \right] \varphi(z; \widehat{W}_h^t) \right|$$

$$+ \left| \varphi(z; W^{(0)})^\top (\overline{\Lambda}_h^t)^{-1} \left[ \varphi(z; \widehat{W}_h^t) - \varphi(z; W^{(0)}) \right] \right|. \tag{H.62}$$

Combining Hölder's inequality and Lemma H.2, we bound the first term on the right-hand side of (H.62) by

$$\left| \left[ \varphi(z; \widehat{W}_h^t) - \varphi(z; W^{(0)}) \right]^\top (\Lambda_h^t)^{-1} \varphi(z; \widehat{W}_h^t) \right| \tag{H.63}$$

$$\leq \left\| \varphi(z; \widehat{W}_h^t) - \varphi(z; W^{(0)}) \right\|_2 \cdot \left\| (\Lambda_h^t)^{-1} \right\|_{\mathrm{op}} \cdot \left\| \varphi(z; \widehat{W}_h^t) \right\|_2 \leq \overline{C}^2 \cdot T^{1/6} \cdot H^{1/3} \cdot m^{-1/6} \cdot \lambda^{-1} \cdot \sqrt{\log m},$$

where $\|(\Lambda_h^t)^{-1}\|_{\mathrm{op}}$ is the matrix operator norm of $(\Lambda_h^t)^{-1}$, which is bounded by $1/\lambda$. Similarly, for the third term, we also have

$$\left| \varphi(z; W^{(0)})^\top (\overline{\Lambda}_h^t)^{-1} \left[ \varphi(z; \widehat{W}_h^t) - \varphi(z; W^{(0)}) \right] \right| \leq \overline{C}^2 \cdot T^{1/6} \cdot H^{1/3} \cdot m^{-1/6} \cdot \lambda^{-1} \cdot \sqrt{\log m}. \tag{H.64}$$

For the second term, since both $\Lambda_h^t$ and $\overline{\Lambda}_h^t$ are invertible, we have

$$\left\| (\Lambda_h^t)^{-1} - (\overline{\Lambda}_h^t)^{-1} \right\|_{\mathrm{op}} = \left\| (\Lambda_h^t)^{-1} (\Lambda_h^t - \overline{\Lambda}_h^t)(\overline{\Lambda}_h^t)^{-1} \right\|_{\mathrm{op}}$$

$$\leq \|(\Lambda_h^t)^{-1}\|_{\mathrm{op}} \cdot \|(\overline{\Lambda}_h^t)^{-1}\|_{\mathrm{op}} \cdot \|\Lambda_h^t - \overline{\Lambda}_h^t\|_{\mathrm{op}} \leq \lambda^{-2} \cdot \|\Lambda_h^t - \overline{\Lambda}_h^t\|_{\mathrm{fro}}. \tag{H.65}$$

By direct computation, we have

$$\|\Lambda_h^t - \overline{\Lambda}_h^t\|_{\mathrm{fro}}$$

$$= \left\|\sum_{\tau=1}^{t-1} \left[\varphi(z_h^\tau; \widehat{W}_h^{\tau+1})\varphi(z_h^\tau; \widehat{W}_h^{\tau+1})^\top - \varphi(z_h^\tau; W^{(0)})\varphi(z_h^\tau; W^{(0)})^\top\right]\right\|_{\mathrm{fro}}$$

$$\leq \sum_{\tau=1}^{t-1}\left\|\varphi(z_h^\tau; \widehat{W}_h^{\tau+1})\varphi(z_h^\tau; \widehat{W}_h^{\tau+1})^\top - \varphi(z_h^\tau; W^{(0)})\varphi(z_h^\tau; W^{(0)})^\top\right\|_{\mathrm{fro}}$$

$$\leq \sum_{\tau=1}^{t-1}\left\|\left[\varphi(z_h^\tau; \widehat{W}_h^{\tau+1}) - \varphi(z_h^\tau; W^{(0)})\right]\varphi(z_h^\tau; \widehat{W}_h^{\tau+1})^\top\right.$$

$$\left. + \varphi(z_h^\tau; W^{(0)})\left[\varphi(z_h^\tau; \widehat{W}_h^{\tau+1}) - \varphi(z_h^\tau; W^{(0)})\right]^\top\right\|_{\mathrm{fro}}.$$

Hence, by Lemma H.2 we can bound $\|\Lambda_h^t - \overline{\Lambda}_h^t\|_{\mathrm{fro}}$ by

$$\|\Lambda_h^t - \overline{\Lambda}_h^t\|_{\mathrm{fro}} \leq 2(t-1)\cdot\overline{C}^2\cdot T^{1/6}\cdot H^{1/3}\cdot m^{-1/6}\cdot\sqrt{\log m}$$

$$\leq 2\overline{C}^2\cdot T^{7/6}\cdot H^{1/3}\cdot m^{-1/6}\cdot\sqrt{\log m}. \tag{H.66}$$

Hence, combining (H.65) and (H.66), the second term on the right-hand side of (H.62) can be bounded by

$$\left|\varphi(z; W^{(0)})^\top\left[(\Lambda_h^t)^{-1} - (\overline{\Lambda}_h^t)^{-1}\right]\varphi(z; \widehat{W}_h^t)\right|$$

$$\leq \left\|\varphi(z; W^{(0)})\right\|_2\cdot\left\|\varphi(z; \widehat{W}_h^t)\right\|_2\cdot\left\|(\Lambda_h^t)^{-1} - (\overline{\Lambda}_h^t)^{-1}\right\|_{\mathrm{op}}$$

$$\leq 2\overline{C}^4\cdot T^{7/6}\cdot H^{1/3}\cdot m^{-1/6}\cdot\lambda^{-2}\cdot\sqrt{\log m}. \tag{H.67}$$

Notice that $\lambda$ defined in (H.59) satisfies that $\lambda \geq \overline{C}^2$. Thus, combining (H.61)-(H.64), and (H.67), we have

$$\left|b_h^t(z) - \overline{b}_h^t(z)\right| \leq 2\cdot T^{7/12}\cdot H^{1/6}\cdot m^{-1/12}\cdot(\log m)^{1/4}, \qquad \forall(t,h)\in[T]\times[H], \tag{H.68}$$

which establishes the second inequality in (E.11). Finally, combining (H.49), (H.60), and (H.68), we conclude that

$$\|Q_h^t - \overline{Q}_h^t\|_\infty \leq 4\cdot T^{5/3}\cdot H^{4/3}\cdot m^{-1/6}\cdot\sqrt{\log m} + 2\beta\cdot T^{7/12}\cdot H^{1/6}\cdot m^{-1/12}\cdot(\log m)^{1/4}.$$

Note that $\beta > 1$. When $m = \Omega(\beta^{12}\cdot T^{13}\cdot H^{14}\cdot(\log m)^3)$, the second term in the above inequality is the dominating term. Thus, we have

$$\sup_{x\in\mathcal{S}}\left|V_h^t(x) - \overline{V}_h^t(x)\right| \leq \|Q_h^t - \overline{Q}_h^t\|_\infty \leq 4\beta\cdot T^{7/12}\cdot H^{1/6}\cdot m^{-1/12}\cdot(\log m)^{1/4}. \tag{H.69}$$

This concludes the second step.

**Step III.** In the last step, we establish optimism by comparing $\varphi(\cdot)^\top(\overline{W}_h^t - W^{(0)})$ and the function $\widetilde{Q}_h^t$ defined in (H.39), where $\varphi(\cdot)$ denotes $\varphi(\cdot; W^{(0)})$. By the definition of $\overline{\Lambda}_h^t$ in (H.42), we have

$$\widetilde{W}_h^t - W^{(0)} = (\overline{\Lambda}_h^t)^{-1}\cdot\left[\lambda\cdot\left(\widetilde{W}_h^t - W^{(0)}\right) + (\Phi_h^t)^\top\Phi_h^t\left(\widetilde{W}_h^t - W^{(0)}\right)\right],$$

where $\widetilde{W}_h^t$ is given in (H.41). Hence, combining (H.47), we have

$$\overline{W}_h^t - \widetilde{W}_h^t = -\lambda\cdot(\overline{\Lambda}_h^t)^{-1}\left(\widetilde{W}_h^t - W^{(0)}\right) + (\overline{\Lambda}_h^t)^{-1}(\Phi_h^t)^\top\left[y_h^t - \Phi_h^t\left(\widetilde{W}_h^t - W^{(0)}\right)\right]. \tag{H.70}$$

Thus, for any $z\in\mathcal{Z}$, by (H.70) we have

$$\varphi(z)^\top\left(\overline{W}_h^t - \widetilde{W}_h^t\right)$$

$$= \underbrace{-\lambda\cdot\varphi(z)^\top(\overline{\Lambda}_h^t)^{-1}\cdot\left(\widetilde{W}_h^t - W^{(0)}\right)}_{\text{(iii)}} + \underbrace{\varphi(z)^\top(\overline{\Lambda}_h^t)^{-1}(\Phi_h^t)^\top\left[y_h^t - \Phi_h^t\left(\widetilde{W}_h^t - W^{(0)}\right)\right]}_{\text{(iv)}}.$$

$$\tag{H.71}$$

For Term (iii) on the right-hand side of (H.71), by the Cauchy-Schwarz inequality, we have

$$\big|\lambda \cdot \varphi(z)^\top (\overline{\Lambda}_h^t)^{-1} \cdot \big(\widetilde{W}_h^t - W^{(0)}\big)\big| \le \lambda \cdot \big\|\widetilde{W}_h^t - W^{(0)}\big\|_2 \cdot \big\|(\overline{\Lambda}_h^t)^{-1}\varphi(z)\big\|_2$$

$$\le \lambda \cdot R_Q H/\sqrt{d} \cdot \sqrt{\varphi(z)^\top (\overline{\Lambda}_h^t)^{-1}(\overline{\Lambda}_h^t)^{-1}\varphi(z)} \le R_Q H \cdot \sqrt{\lambda/d} \cdot \overline{b}_h^t(z). \tag{H.72}$$

For Term (iv) in (H.71), recall that $\widetilde{Q}_h^t(z) = \varphi(z)^\top(\widetilde{W}_h^t - W^{(0)})$. To simplify the notation, let $q^\star \in \mathbb{R}^{t-1}$ denote the vector whose $\tau$-th entry is $(\mathbb{T}_h^\star Q_{h+1}^t)(x_h^\tau, a_h^\tau)$ for any $\tau \in [t-1]$. Then, by (H.40), for any $\tau \in [t-1]$, the $\tau$-th entry of $\Phi_h^t(\widetilde{W}_h^t - W^{(0)})$ satisfies

$$\big|[\Phi_h^t\big(\widetilde{W}_h^t - W^{(0)}\big)]_\tau - [q^\star]_\tau\big| = \big|[\Phi_h^t\big(\widetilde{W}_h^t - W^{(0)}\big)]_\tau - (\mathbb{T}_h^\star Q_{h+1}^t)(x_h^\tau, a_h^\tau)\big|$$

$$\le 10 C_{\text{act}} \cdot R_Q H \cdot \sqrt{\log(mTH)/m}.$$

Moreover, for any $\tau \in [t-1]$, the $\tau$-th entry of $(y_h^t - q^\star)$ can be written as

$$[y_h^t]_\tau - [q^\star]_\tau = r_h(x_h^\tau, a_h^\tau) + V_{h+1}^t(x_{h+1}^\tau) - \varphi(x_h^\tau, a_h^\tau)^\top \overline{\theta}_h^t$$

$$= r_h(x_h^\tau, a_h^\tau) + V_{h+1}^t(x_{h+1}^\tau) - (\mathbb{T}_h^\star Q_{h+1}^t)(x_h^\tau, a_h^\tau)$$

$$= V_{h+1}^t(x_{h+1}^\tau) - (\mathbb{P}_h V_{h+1}^t)(x_h^\tau, a_h^\tau). \tag{H.73}$$

Then, by the triangle inequality and (H.73), we have

$$\big|\varphi(z)^\top (\overline{\Lambda}_h^t)^{-1}(\Phi_h^t)^\top \big[y_h^t - \Phi_h^t\big(\widetilde{W}_h^t - W^{(0)}\big)\big]\big|$$

$$\le \big|\varphi(z)^\top (\overline{\Lambda}_h^t)^{-1}(\Phi_h^t)^\top \big[y_h^t - q^\star\big]\big| + \big|\varphi(z)^\top (\overline{\Lambda}_h^t)^{-1}(\Phi_h^t)^\top \big[q^\star - \Phi_h^t\big(\widetilde{W}_h^t - W^{(0)}\big)\big]\big|$$

$$\le \|\varphi(z)\|_{(\overline{\Lambda}_h^t)^{-1}} \cdot \Big\|\sum_{\tau=1}^{t-1} \varphi(x_h^\tau, a_h^\tau) \cdot \big[V_{h+1}^t(x_{h+1}^\tau) - (\mathbb{P}_h V_{h+1}^t)(x_h^\tau, a_h^\tau)\big]\Big\|_{(\overline{\Lambda}_h^t)^{-1}}$$

$$+ 10 C_{\text{act}} \cdot R_Q H \cdot \sqrt{\log(mTH)/m} \cdot \|\varphi(z)\|_{(\overline{\Lambda}_h^t)^{-1}}. \tag{H.74}$$

Recall that we have shown in **Step II** that, with probability at least $1 - m^2$ with respect to the randomness of initialization, (H.69) holds for all $(t,h) \in [T] \times [H]$. To simplify the notation, we denote

$$\texttt{Err} = 4\beta \cdot T^{7/12} \cdot H^{1/6} \cdot m^{-1/12} \cdot (\log m)^{1/4}. \tag{H.75}$$

Moreover, we define functions $\Delta V_1 = V_{h+1}^t - \overline{V}_{h+1}^t$ and $\Delta V_2 = \mathbb{P}_h(V_{h+1}^t - \overline{V}_{h+1}^t)$. Then (H.69) implies that $\sup_{x \in S} |\Delta V_1(x)| \le \texttt{Err}$ and $\sup_{z \in \mathcal{Z}} |\Delta V_2(z)| \le \texttt{Err}$. By the elementary inequality $(a+b)^2 \le 2a^2 + 2b^2$, we have

$$\Big\|\sum_{\tau=1}^{t-1} \varphi(x_h^\tau, a_h^\tau) \cdot \big[V_{h+1}^t(x_{h+1}^\tau) - (\mathbb{P}_h V_{h+1}^t)(x_h^\tau, a_h^\tau)\big]\Big\|_{(\overline{\Lambda}_h^t)^{-1}}^2$$

$$\le 2\underbrace{\Big\|\sum_{\tau=1}^{t-1} \varphi(x_h^\tau, a_h^\tau) \cdot \big[\overline{V}_{h+1}^t(x_{h+1}^\tau) - (\mathbb{P}_h \overline{V}_{h+1}^t)(x_h^\tau, a_h^\tau)\big]\Big\|_{(\overline{\Lambda}_h^t)^{-1}}^2}_{(\text{v})}$$

$$+ 2\Big\|\sum_{\tau=1}^{t-1} \varphi(x_h^\tau, a_h^\tau) \cdot \big[\Delta V_1(x_{h+1}^\tau) - \Delta V_2(x_h^\tau, a_h^\tau)\big]\Big\|_{(\overline{\Lambda}_h^t)^{-1}}^2$$

$$\le 2 \cdot \text{Term (v)} + 8 \cdot \texttt{Err}^2 \cdot T^2, \tag{H.76}$$

where the last inequality follows from the fact that

$$\Big\|\sum_{\tau=1}^{t-1} \varphi(x_h^\tau, a_h^\tau) \cdot \big[\Delta V_1(x_{h+1}^\tau) - \Delta V_2(x_h^\tau, a_h^\tau)\big]\Big\|_{(\overline{\Lambda}_h^t)^{-1}}^2 \le 4\texttt{Err}^2 \cdot \Big\|\sum_{\tau=1}^{t-1} \varphi(x_h^\tau, a_h^\tau)\Big\|_{(\overline{\Lambda}_h^t)^{-1}}^2$$

$$\le 4 \cdot \texttt{Err}^2 \cdot (t-1) \cdot \lambda^{-1} \cdot \sum_{\tau=1}^{t-1} \|\varphi(x_h^\tau, a_h^\tau)\|_2^2 \le 4 \cdot \texttt{Err}^2 \cdot (t-1)^2 \cdot \overline{C}^2 \cdot \lambda^{-1} \le 4 \cdot \texttt{Err}^2 \cdot T^2.$$

Here the second-to-last inequality follows from Lemma H.2 and the definition of $\lambda$.

Recall that we define $\overline{b}_h^t(z) = \|\varphi(z)\|_{(\overline{\Lambda}_h^t)^{-1}}$. Combining (H.73), (H.74), and (H.77), we have

$$\left|\varphi(z)^\top (\overline{\Lambda}_h^t)^{-1} (\Phi_h^t)^\top \left[y_h^t - \Phi_h^t \big(\widetilde{W}_h^t - W^{(0)}\big)\right]\right|$$
$$\leq \overline{b}_h^t(z) \cdot \left[10 C_{\mathrm{act}} \cdot R_Q H \cdot \sqrt{\log(mTH)/m} + \sqrt{2 \cdot \mathrm{Term\ (v)}} + 2\sqrt{2} \cdot \mathtt{Err} \cdot T\right]$$
$$\leq \overline{b}_h^t(z) \cdot \left[R_Q H + \sqrt{2 \cdot \mathrm{Term\ (v)}}\right], \tag{H.77}$$

where we apply the elementary inequality $\sqrt{a+b} \leq \sqrt{a} + \sqrt{b}$. Here in the last inequality we let $m$ be sufficiently large such that

$$10 C_{\mathrm{act}} \cdot R_Q H \cdot \sqrt{\log(mTH)/m} + 2\sqrt{2} \cdot \mathtt{Err} \cdot T \leq R_Q H.$$

In the following, we aim to bound Term (v) in (H.77) by combining the concentration of the self-normalized stochastic process and uniform concentration. To characterize the function class that contains each $\overline{V}_h^t$, we define $\widetilde{\varphi} \colon \mathcal{Z} \to \mathbb{R}$ by $\widetilde{\varphi}(z) = \varphi(z)/\overline{C}$. Then, conditioning on the event where the statements in Lemma H.2 are true, we have $\|\widetilde{\varphi}(z)\|_2 \leq 1$ for all $z \in \mathcal{Z}$. Furthermore, we define a kernel function $\widetilde{K} \colon \mathcal{Z} \times \mathcal{Z} \to \mathbb{R}$ by letting $\widetilde{K}(z, z') = \widetilde{\varphi}(z)^\top \widetilde{\varphi}(z')$ for all $z, z' \in \mathcal{Z}$. That is, $\widetilde{K}$ is the normalized version of the empirical NTK $K_m$. By construction, $\widetilde{K}$ is a bounded kernel such that $\sup_{z \in \mathcal{Z}} \widetilde{K}(z, z) \leq 1$. We can also consider the maximal information gain in (4.2) for $\widetilde{K}$ and $K_m$. These two quantities are linked via

$$\Gamma_{\widetilde{K}}(T, \sigma) = \Gamma_{K_m}\big(T, \overline{C}^2 \sigma\big), \qquad \forall \sigma > 0. \tag{H.78}$$

Furthermore, we define $\widetilde{\lambda} = \lambda/\overline{C}^2$ and $\widetilde{\Lambda}_h^t = \overline{\Lambda}_h^t // \overline{C}^2$ for all $(t, h) \in [T] \times [H]$. By the definition of $\lambda$ in (H.59), we have $\widetilde{\lambda} = 1 + 1/T \in [1, 2]$. Moreover, by (H.42) we have

$$\widetilde{\Lambda}_h^t = \widetilde{\lambda} \cdot I_{2md} + \sum_{\tau=1}^{t-1} \widetilde{\varphi}(x_h^\tau, a_h^\tau) \widetilde{\varphi}(x_h^\tau, a_h^\tau)^\top.$$

Since $\widetilde{\lambda} > 1$, $\widetilde{\Lambda}_h^t$ is an invertible matrix and the eigenvalues of $(\widetilde{\Lambda}_h^t)^{-1}$ are all bounded above by one. Using $\widetilde{\varphi}$ and $\widetilde{\Lambda}_h^t$, we rewrite each $\overline{Q}_h^t$ as follows. For $\overline{W}_h^t$ defined in (H.47), we have

$$\varphi(x, a)^\top \big(\overline{W}_h^t - W^{(0)}\big) = \overline{C} \cdot \widetilde{\varphi}(x, a)^\top \big(\overline{W}_h^t - W^{(0)}\big), \tag{H.79}$$

where $\overline{C} \cdot \|\overline{W}_h^t - W^{(0)}\|_2 \leq \overline{C} \cdot H\sqrt{T/\lambda} \leq H\sqrt{T}$ since $\lambda \geq (\overline{C})^2$. Meanwhile, we also have

$$\overline{b}_h^t(z) = \|\varphi(z)\|_{(\overline{\Lambda}_h^t)^{-1}} = \left[\widetilde{\varphi}(z)^\top \big(\widetilde{\Lambda}_h^t\big)^{-1} \widetilde{\varphi}(z)\right]^{1/2}. \tag{H.80}$$

Thus, combining (H.79) and (H.80), $\overline{Q}_h^t$ defined in (H.48) can be written equivalently as

$$\overline{Q}_h^t(z) = \min\left\{\widetilde{\varphi}(z)^\top \overline{\vartheta}_h^t + \beta \cdot \|\widetilde{\varphi}(z)\|_{(\widetilde{\Lambda}_h^t)^{-1}}, H - h + 1\right\}^+,$$

where $\overline{\theta}_h^t = \overline{C} \cdot (\overline{W}_h^t - W^{(0)})$, which satisfies $\|\overline{\vartheta}_h^t\|_2 \leq H\sqrt{T}$.

Let $\mathcal{D}$ be a finite subset of $\mathcal{Z}$ with no more than $T$ elements. For any fixed $\mathcal{D}$, we define

$$\widetilde{\Lambda}_{\mathcal{D}} = \widetilde{\lambda} \cdot I_{2dm} + \sum_{z \in \mathcal{D}} \widetilde{\varphi}(z) \widetilde{\varphi}(z)^\top \in \mathbb{R}^{2md \times 2md}. \tag{H.81}$$

For any $h \in [H]$, $R, B > 0$, we let $\widetilde{Q}_{\mathrm{ucb}}(h, R, B)$ consists of functions that take the form of

$$Q(\cdot) = \min\left\{\widetilde{\varphi}(\cdot)^\top \vartheta + \beta \cdot \|\widetilde{\varphi}(\cdot)\|_{(\widetilde{\Lambda}_{\mathcal{D}})^{-1}}; H - h + 1\right\}^+,$$

for some $\vartheta \in \mathbb{R}^{2md}$ with $\|\vartheta\|_2 \leq R$ and some $\mathcal{D} \subseteq \mathcal{Z}$. Then $\widetilde{Q}_{\mathrm{ucb}}(h, R, B)$ corresponds to the function class in (4.4) with the kernel being $\widetilde{K}$. Moreover, we define $\widetilde{\mathcal{V}}_{\mathrm{ucb}}(h, R, B)$ as

$$\widetilde{\mathcal{V}}_{\mathrm{ucb}}(h, R, B) = \left\{V : V(\cdot) = \max_a Q(\cdot, a) \text{ for some } Q \in \widetilde{Q}_{\mathrm{ucb}}(h, R, B)\right\}.$$

By definition, for all $h \in [H]$ and any $R, B > 0$, we have that $\mathcal{Q}_{\mathrm{ucb}}(h, R, B) = \widetilde{Q}_{\mathrm{ucb}}(h, \overline{C}R, B)$. Meanwhile, since $(\overline{C})^2 \leq \lambda \leq 2(\overline{C})^2$, for all $R > 0$, we have

$$\mathcal{Q}_{\mathrm{ucb}}(h, R, B) \subseteq \widetilde{Q}_{\mathrm{ucb}}(h, R\sqrt{\lambda}, B) \subseteq \mathcal{Q}_{\mathrm{ucb}}(h, \sqrt{2}R, B). \tag{H.82}$$

Recall that we define $R_T = H\sqrt{2T/\lambda}$ and let $N_\infty(\epsilon; h, B)$ denote the $\epsilon$-covering number of $\mathcal{Q}_{\mathrm{ucb}}(h, R_T, B)$ with respect to the $\ell_\infty$-norm on $\mathcal{Z}$. Moreover, hereafter, we denote $\epsilon^* = H/T$

and set $B = B_T$ which satisfy (D.3). Since we set $\beta = B_T$ in Algorithm 4, it holds for all $(t, h) \in [T] \times [H]$ that

$$\overline{Q}_h^t \in \widetilde{\mathcal{Q}}_{\mathrm{ucb}}(h, H\sqrt{T}, B) \subseteq \mathcal{Q}_{\mathrm{ucb}}(h, R_T, B), \qquad \overline{V}_h^t \in \widetilde{\mathcal{V}}_{\mathrm{ucb}}(h, H\sqrt{T}, B).$$

Now, to bound Term (v) in (H.77), similar to the analysis the proof of Lemma E.2, we apply the concentration of self-normalized stochastic process and uniform concentration over $\widetilde{\mathcal{V}}_{\mathrm{ucb}}(h, H\sqrt{T}, B_T)$. Specifically, similar to (H.25) and (H.26), with probability at least $1 - (2T^2 H^2)^{-1}$, we have

$$
\begin{aligned}
\text{Term (v)} &= \left\| \sum_{\tau=1}^{t-1} \varphi(x_h^\tau, a_h^\tau) \cdot \left[ \overline{V}_{h+1}^t(x_{h+1}^\tau) - (\mathbb{P}_h \overline{V}_{h+1}^t)(x_h^\tau, a_h^\tau) \right] \right\|_{(\overline{\Lambda}_h^t)^{-1}}^2 \\
&= \left\| \sum_{\tau=1}^{t-1} \widetilde{\varphi}(x_h^\tau, a_h^\tau) \cdot \left[ \overline{V}_{h+1}^t(x_{h+1}^\tau) - (\mathbb{P}_h \overline{V}_{h+1}^t)(x_h^\tau, a_h^\tau) \right] \right\|_{(\widetilde{\Lambda}_h^t)^{-1}}^2 \\
&\le 4H^2 \cdot \Gamma_{\widetilde{K}}(T, \widetilde{\lambda}) + 11H^2 + 4H^2 \cdot \log N_\infty(\epsilon^*, h+1, B_T) + 8H^2 \cdot \log(TH).
\end{aligned}
$$
(H.83)

Thus, combining (H.71), (H.72), (H.77), and (H.83), we obtain that

$$\left| \varphi(z)^\top (\overline{W}_h^t - \widetilde{W}_h^t) \right|$$
$$\le \left| \text{Term (iii)} \right| + \left| \text{Term (iv)} \right| \le \left[ R_Q H + \sqrt{2 \cdot \text{Term (v)}} + R_Q H \cdot \sqrt{\lambda/d} \right] \cdot \overline{b}_h^t(z)$$
$$\le H \cdot \left\{ \left[ 8\Gamma_{K_m}(T, \lambda) + 22 + 8 \cdot \log N_\infty(\epsilon^*, h+1, B_T) + 16 \cdot \log(TH) \right]^{1/2} + R_Q \cdot (1 + \sqrt{\lambda/d}) \right\} \cdot \overline{b}_h^t(z).$$

Using the elementary inequality $a + b \le \sqrt{2(a^2 + b^2)}$, we have

$$\left| \varphi(z)^\top (\overline{W}_h^t - \widetilde{W}_h^t) \right|$$
$$\le H \cdot \left[ 16\Gamma_{K_m}(T, \lambda) + 16 \cdot \log N_\infty(\epsilon^*, h+1, B_T) + 32 \cdot \log(TH) + 2R_Q^2 \cdot (1 + \sqrt{\lambda/d})^2 \right]^{1/2} \cdot \overline{b}_h^t(z)$$
$$\le H \cdot \left[ 16\Gamma_{K_m}(T, \lambda) + 16 \cdot \log N_\infty(\epsilon^*, h+1, B_T) + 32 \cdot \log(TH) + 4R_Q^2 \cdot (1 + \lambda/d) \right]^{1/2} \cdot \overline{b}_h^t(z).$$

By the choice of $B_T$ in (D.3), we have that

$$\left| \varphi(z)^\top (\overline{W}_h^t - \widetilde{W}_h^t) \right| = \left| \varphi(z)^\top (\overline{W}_h^t - W^{(0)}) - \widetilde{Q}_h^t(z) \right| \le \beta \cdot \overline{b}_h^t(z)$$

holds simultaneously for all $(t, h) \in [T] \times [H]$ and $z \in \mathcal{Z}$ with probability at least $1 - (2T^2 H^2)^{-1}$.

Thus, combining this with (H.39) and (H.40), we have

$$\left| \varphi(z)^\top (\overline{W}_h^t - W^{(0)}) - \mathbb{T}_h^\star Q_{h+1}^t(z) \right| \le \beta \cdot \overline{b}_h^t(z) + 10C_{\mathrm{act}} \cdot R_Q H \cdot \sqrt{\log(mTH)/m}. \quad \text{(H.84)}$$

By the definition of $\overline{Q}_h^t$ in (H.48), we have

$$
\begin{aligned}
\overline{Q}_h^t(z) - \mathbb{T}_h^\star Q_{h+1}^t(z) &\le \varphi(z)^\top (\overline{W}_h^t - W^{(0)}) - \mathbb{T}_h^\star Q_{h+1}^t(z) + \beta \cdot \overline{b}_h^t(z) \\
&\le 2\beta \cdot \overline{b}_h^t(z) + 10C_{\mathrm{act}} \cdot R_Q \cdot \sqrt{\log(mTH)/m}.
\end{aligned}
$$
(H.85)

Moreover, since $\mathbb{T}_h^\star Q_{h+1}^t \le H - h + 1$, by (H.84) we have

$$
\begin{aligned}
\mathbb{T}_h^\star Q_{h+1}^t(z) - \overline{Q}_h^t(z) &= \mathbb{T}_h^\star Q_{h+1}^t(z) - \min \left\{ \varphi(x, a)^\top (\overline{W}_h^t - W^{(0)}) + \beta \cdot \overline{b}_h^t(x, a), H - h + 1 \right\}^+ \\
&= \max \left\{ \mathbb{T}_h^\star Q_{h+1}^t(z) - \varphi(z)^\top (\overline{W}_h^t - W^{(0)}) - \beta \cdot \overline{b}_h^t(z), 0 \right\}^+ \\
&\le 10C_{\mathrm{act}} \cdot R_Q \cdot \sqrt{\log(mTH)/m}.
\end{aligned}
$$
(H.86)

Let $\iota$ denote $T^{7/12} \cdot H^{1/12} \cdot m^{-1/12} \cdot (\log m)^{1/4}$. When $m$ is sufficiently large, it holds that

$$10C_{\mathrm{act}} \cdot R_Q \cdot \sqrt{\log(mTH)/m} \le \iota \le \beta \cdot \iota.$$

Meanwhile, combining the definition of the TD error $\delta_h^t$ in (E.1) and (H.69), we have

$$
\begin{aligned}
\left| \delta_h^t(z) - \left[ \mathbb{T}_h^\star Q_{h+1}^t(z) - \overline{Q}_h^t(z) \right] \right| &= \left| Q_h^t(z) - \overline{Q}_h^t(z) \right| \\
&\le 4\beta \cdot T^{7/12} \cdot H^{1/12} \cdot m^{-1/12} \cdot (\log m)^{1/4}.
\end{aligned}
$$
(H.87)

Finally, combining (H.85), (H.86), and (H.87), we establish that, with probability at least $1 - (2T^2H^2)^{-1}$,

$$\delta_h^t(z) \le [\mathbb{T}_h^\star Q_{h+1}^t(z) - \overline{Q}_h^t(z)] + 4\beta \cdot \iota \le 5\beta \cdot \iota$$

$$\delta_h^t(z) \ge [\mathbb{T}_h^\star Q_{h+1}^t(z) - \overline{Q}_h^t(z)] - 4\beta \cdot \iota \ge -2\beta \cdot \overline{b}_h^t(z) - 5\beta \cdot \iota$$

hold for all $(t, h) \in [T] \times [H]$ simultaneously. Finally, combining this with (H.68), we have

$$-2\beta \cdot b_h^t - 9\beta \cdot \iota \le -2\beta \cdot \overline{b}_h^t - 5\beta \cdot \iota \le \delta_h^t(z) \le 5\beta \cdot \iota,$$

which, together with (H.68), concludes the proof of Lemma E.4. $\qquad\square$

# I  Covering Number and Effective Dimension

In this section, we present results on the covering number of the class of value functions that we study and the effective dimension of the corresponding RKHS. Both of these results play a key role in establishing our regret bounds.

## I.1  Covering Number of the Classes of Value Functions

For any $R, B > 0$, any $h \in [H]$, and fixed $\mathcal{D}$, we define $\mathcal{Q}_{\mathrm{ucb}}(h, R, B)$ as the function class that contains functions on $\mathcal{Z}$ that take the following form:

$$Q(z) = \min\big\{\theta(z) + \beta \cdot \lambda^{-1/2}\big[K(z, z) - k_t(z)^\top (K_t + \lambda I)^{-1} k_t(z)\big]^{1/2}, H - h + 1\big\}^+, \quad \text{(I.1)}$$

where $\theta \in \mathcal{H}$ satisfies $\|\theta\|_{\mathcal{H}} \le R$, $\beta \in [0, B]$, $h \in [H]$, and $\mathcal{D} = \{z^\tau = (x^\tau, a^\tau),\}_{\tau \in [t]}$ is a finite subset of $\mathcal{Z}$ with $t$ elements, where $t \le T$. Here $T$ is the total number of the episodes. Moreover, $K_t \in \mathbb{R}^{t \times t}$ and $k_t \colon \mathcal{Z} \to \mathbb{R}^t$ are defined similarly as in (3.7) based on state-action pairs in $\mathcal{D}$, that is,

$$K_t = [K(z^\tau, z^{\tau'})]_{\tau, \tau' \in [t]} \in \mathbb{R}^{t \times t}, \qquad k_t(z) = \big[K(z^1, z), \dots K(z^t, z)\big]^\top \in \mathbb{R}^t.$$

By definition, $Q$ in (I.1) is determined by $Q_0 \in \mathcal{H}$ and a bonus term constructed using $\mathcal{D}$. Thus, the function $Q_h^t$ constructed in Algorithm 2 belongs to $\mathcal{Q}_{\mathrm{ucb}}(h, R, B)$ when $\beta \le B$ and $\|\alpha_h^t\|_{\mathcal{H}} \le R$. In the following, for any $\epsilon \in (0, 1)$, we let $\mathcal{C}(\mathcal{Q}_{\mathrm{ucb}}(h, R, B), \epsilon)$ be the minimal $\epsilon$-cover of $\mathcal{Q}_{\mathrm{ucb}}(h, R, B)$ with respect to the $\ell_\infty$-norm on $\mathcal{Z}$. That is, for any $Q \in \mathcal{Q}_{\mathrm{ucb}}(h, R, B)$, there exists $Q' \in \mathcal{C}(\mathcal{Q}_{\mathrm{ucb}}(h, R, B), \epsilon)$ satisfying $\|Q - Q'\|_\infty \le \epsilon$. Moreover, among all function classes that possess such a property, $\mathcal{C}(\mathcal{Q}_{\mathrm{ucb}}(h, R, B), \epsilon)$ has the smallest cardinality. Thus, by definition, $|\mathcal{C}(\mathcal{Q}_{\mathrm{ucb}}(h, R, B), \epsilon)|$ is the $\epsilon$-covering number of $\mathcal{Q}_{\mathrm{ucb}}(h, R, B)$ with respect to the $\ell_\infty$-norm on $\mathcal{Z}$.

In addition, based on $\mathcal{Q}_{\mathrm{ucb}}(h, R, B)$, we define the function class $\mathcal{V}_{\mathrm{ucb}}(h, R, B)$ as

$$\mathcal{V}_{\mathrm{ucb}}(h, R, B) = \big\{V \colon V(\cdot) = \max_a Q(\cdot, a) \text{ for some } Q \in \mathcal{Q}_{\mathrm{ucb}}(h, R, B)\big\}. \quad \text{(I.2)}$$

For any two value functions $V_1, V_2 \colon \mathcal{S} \to \mathbb{R}$, we denote their supremum norm distance as

$$\mathrm{dist}(V_1, V_2) = \sup_{x \in \mathcal{S}}\big|V_1(x) - V_2(x)\big|. \quad \text{(I.3)}$$

For any $\epsilon \in (0, 1)$, we let $\mathcal{C}(\mathcal{V}_{\mathrm{ucb}}(h, R, B), \epsilon)$ denote the minimal $\epsilon$-cover of $\mathcal{V}_{\mathrm{ucb}}(h, R, B)$ with respect to $\mathrm{dist}(\cdot, \cdot)$ defined in (I.3).

The main result of this section is a set of upper bounds on the size of $\mathcal{C}(\mathcal{V}_{\mathrm{ucb}}(h, R, B), \epsilon)$ under the two eigenvalue decay conditions specified in Assumption 4.3.

**Lemma I.1.** Let Assumption 4.3 hold and $\lambda$ be bounded in $[c_1, c_2]$, where both $c_1$ and $c_2$ are absolute constants. Then, for any $h \in [H]$, any $R, B > 0$, and any $\epsilon \in (0, 1/e)$, there exists a positive constant $C_N$ such that

$$\log\big|\mathcal{C}\big(\mathcal{V}_{\mathrm{ucb}}(h, R, B), \epsilon\big)\big| \le \log\big|\mathcal{C}\big(\mathcal{Q}_{\mathrm{ucb}}(h, R, B), \epsilon\big)\big| \quad \text{(I.4)}$$

$$\le \begin{cases} C_N \cdot \gamma \cdot \big[1 + \log(R/\epsilon)\big] + C_N \cdot \gamma^2 \cdot \big[1 + \log(B/\epsilon)\big] & \text{case (i),} \\ C_N \cdot \big[1 + \log(R/\epsilon)\big]^{1+1/\gamma} + C_N \cdot \big[1 + \log(B/\epsilon)\big]^{1+2/\gamma} & \text{case (ii),} \end{cases} \quad \text{(I.5)}$$

where cases (i) and (ii) above correspond to the two eigenvalue decay conditions specified in Assumption 4.3, respectively. Moreover, here $C_N$ in (I.4) is independent of $T$, $H$, $R$, and $B$, and only depends on $C_\psi$, $C_1$, $C_2$, $c_1$, $c_2$, $\gamma$, and $\tau$.

*Proof.* For any fixed subset $\mathcal{D} = \{z^\tau\}_{\tau \in [t]}$ of $\mathcal{Z}$ with size $t \in [T]$, we define $\Phi_{\mathcal{D}} \colon \mathcal{H} \to \mathbb{R}^t$ and $\Lambda_{\mathcal{D}} \colon \mathcal{H} \to \mathcal{H}$ respectively as

$$\Phi_{\mathcal{D}} = \left[ \phi(z^1)^\top, \phi(z^2)^\top \ldots, \phi(z^t)^\top \right]^\top,$$

$$\Lambda_{\mathcal{D}} = \sum_{\tau=1}^t \phi(z^\tau)\phi(z^\tau)^\top + \lambda \cdot I_{\mathcal{H}} = \lambda \cdot I_{\mathcal{H}} + (\Phi_{\mathcal{D}})^\top \Phi_{\mathcal{D}}, \tag{I.6}$$

where $\phi \colon \mathcal{Z} \to \mathcal{H}$ is the feature mapping of $\mathcal{H}$ and $\mathcal{I}_{\mathcal{H}}$ is the identity mapping on $\mathcal{H}$. Then, we can equivalently write $Q_1 \in \mathcal{Q}_{\mathrm{ucb}}(h, R, B)$ as

$$Q_1(z) = \phi(z)^\top \theta_1 + \beta \cdot \sqrt{\phi(z)^\top \Lambda_{\mathcal{D}_1}^{-1} \phi(z)}, \tag{I.7}$$

where $\theta_1 \in \mathcal{H}$ has an RKHS norm bounded by $R$, $\beta_1 \in [0, B]$, and $\mathcal{D}_1$ is a finite subset of $\mathcal{Z}$ with size $t_1 \leq T$. Let $V_1(\cdot) = \max_{a \in \mathcal{A}} Q_1(\cdot, a)$. Combining (I.2) and (I.7), we can write $V_1 \in \mathcal{V}_{\mathrm{ucb}}(h, R, B)$ as

$$V_1(\cdot) = \min \left\{ \max_a \left\{ \phi(\cdot, a)^\top \theta_1 + \beta_1 \cdot \sqrt{\phi(\cdot, a)^\top \Lambda_{\mathcal{D}_1}^{-1} \phi(\cdot, a)} \right\}, H - h + 1 \right\}^+, \tag{I.8}$$

Similar to $V_1$ in (I.8), consider any $V_2 \colon \mathcal{S} \to \mathbb{R}$ that can be written as

$$V_2(\cdot) = \min \left\{ \max_a \left\{ f_1(\cdot, a) + \beta_2 \cdot f_2(\cdot, a) \right\}, H - h + 1 \right\}^+, \tag{I.9}$$

where $Q_2 = f_1 + \beta_2 \cdot f_2$ for some $f_1, f_2 \colon \mathcal{Z} \to \mathbb{R}$ and $\beta_2 \in [0, B]$. Since both $\min\{\cdot, H - h + 1\}^+$ and $\max_a$ are contractive mappings, by (I.8) and (I.9) we have

$$\mathrm{dist}(V_1, V_2) \leq \sup_{(x,a) \in \mathcal{Z}} \left| \left[ \phi(x, a)^\top \theta_1 + \beta_1 \cdot \sqrt{\phi(x, a)^\top \Lambda_{\mathcal{D}_1}^{-1} \phi(x, a)} \right] \right.$$

$$\left. - \left[ f(x, a) + \beta_2 \cdot f_2(x, a) \right] \right| = \|Q_1 - Q_2\|_\infty,$$

which implies that

$$\log \left| \mathcal{C}\big(\mathcal{V}_{\mathrm{ucb}}(h, R, B), \epsilon\big) \right| \leq \log \left| \mathcal{C}\big(\mathcal{Q}_{\mathrm{ucb}}(h, R, B), \epsilon\big) \right|.$$

Moreover, by the triangle inequality, we have

$$\|Q_1 - Q_2\|_\infty \leq \sup_{(x,a) \in \mathcal{Z}} \left| \phi(x, a)^\top \theta_1 - f_2(x, a) \right| + |\beta_1 - \beta_2| \cdot \sup_{(x,a) \in \mathcal{Z}} \left\| \phi(x, a) \right\|_{\Lambda_{\mathcal{D}_1}^{-1}}$$

$$+ B \cdot \sup_{(x,a) \in \mathcal{Z}} \left| \left\| \phi(x, a) \right\|_{\Lambda_{\mathcal{D}_1}^{-1}} - f_2(x, a) \right|, \tag{I.10}$$

where we denote $\|\phi(x, a)\|_{\Lambda_{\mathcal{D}_1}^{-1}}^2 = \phi(x, a)^\top \Lambda_{\mathcal{D}_1}^{-1} \phi(x, a)$. Notice that by the reproducing property we have $\phi(x, a)^\top \theta = \langle \theta, \phi(x, a) \rangle_{\mathcal{H}} = \theta(x, a)$ for all $\theta \in \mathcal{H}$ and $(x, a) \in \mathcal{Z}$. Also note that

$$\|\phi(x, a)\|_{\Lambda_{\mathcal{D}_1}^{-1}}^2 \leq 1/\lambda \cdot \|\phi(x, a)\|^2 \leq 1/\lambda \cdot K(z, z) \leq 1/\lambda.$$

Thus, by (I.10) we have

$$\|Q_1 - Q_2\|_\infty \leq \sup_{(x,a) \in \mathcal{Z}} \left| \theta_1(x, a) - f_1(x, a) \right| + |\beta_1 - \beta_2|/\lambda$$

$$+ B \cdot \sup_{(x,a) \in \mathcal{Z}} \left| \left\| \phi(x, a) \right\|_{\Lambda_{\mathcal{D}_1}^{-1}} - f_2(x, a) \right|. \tag{I.11}$$

Thus, by (I.11), to get the covering number of $\mathcal{Q}_{\mathrm{ucb}}(h, R, B)$ with respect to $\mathrm{dist}(\cdot, \cdot)$, it suffices to bound the covering numbers of the RKHS norm ball $\{f \in \mathcal{H} \colon \|f\|_{\mathcal{H}} \leq R\}$, the interval $[0, B]$, and the set of functions that are of the form of $\|\phi(\cdot)\|_{\Lambda_{\mathcal{D}}^{-1}}$, respectively.

Notice that, by the definition in (I.6), $\Lambda_{\mathcal{D}} \colon \mathcal{H} \to \mathcal{H}$ is a self-adjoint operator on $\mathcal{H}$ with eigenvalues bounded in $[0, 1/\lambda]$. To simplify the notation, we define the function class $\mathcal{F}(\lambda)$ as

$$\mathcal{F}(\lambda) = \left\{ \|\phi(\cdot)\|_{\Upsilon} = \left[ \phi(\cdot)^\top \Upsilon \phi(\cdot) \right]^{1/2} \colon \|\Upsilon\|_{\mathrm{op}} \leq 1/\lambda \right\}, \tag{I.12}$$

where $\Upsilon \colon \mathcal{H} \to \mathcal{H}$ in (I.12) is a self-adjoint operator on $\mathcal{H}$ whose eigenvalues are all bounded by $1/\lambda$ in magnitude. Here, the operator norm of $\Upsilon$ is defined as

$$\|\Upsilon\|_{\mathrm{op}} = \sup \{ f^\top \Upsilon f \colon f \in \mathcal{H}, \|f\|_{\mathcal{H}} = 1 \} = \sup \{ \langle f, \Upsilon f \rangle_{\mathcal{H}} \colon f \in \mathcal{H}, \|f\|_{\mathcal{H}} = 1 \}.$$

Thus, by definition, for any finite subset $\mathcal{D}$ of $\mathcal{Z}$, $\|\phi(\cdot)\|_{\Lambda_{\mathcal{D}}^{-1}}$ belongs to $\mathcal{F}(\lambda)$, where $\Lambda_{\mathcal{D}}$ is defined in (I.6). For any $\epsilon \in (0, 1)$, we let $N_\infty(\epsilon, \mathcal{F}, \lambda)$ denote the $\epsilon$-covering number of $\mathcal{F}(\lambda)$ in (I.12)

with respect to the $\ell_\infty$-norm. Moreover, let $N_\infty(\epsilon, \mathcal{H}, R)$ denote the $\epsilon$-covering number of $\{f \in \mathcal{H} \colon \|f\|_\mathcal{H} \leq R\}$ with respect to the $\ell_\infty$-norm and let $N(\epsilon, B)$ denote the $\epsilon$-covering number of the interval $[0, B]$ with respect the Euclidean distance. Then, by (I.11) we obtain that

$$\big|\mathcal{C}\big(\mathcal{Q}_{\mathrm{ucb}}(h, R, B), \epsilon\big)\big| \leq N_\infty(\epsilon/3, \mathcal{H}, R) \cdot N(\epsilon \cdot \lambda/3, B) \cdot N_\infty\big(\epsilon/(3B), \mathcal{F}, \lambda\big). \tag{I.13}$$

As shown in [69, Corollary 4.2.13], it holds that

$$N(\epsilon \cdot \lambda/3, B) \leq 1 + 6B/(\epsilon \cdot \lambda) \leq 1 + 6B/\epsilon, \tag{I.14}$$

where the last inequality follows from the fact that $\lambda \in [1, 2]$.

It remains to bound the first and the third terms on the right-hand side of (I.13) separately. We establish the $\ell_\infty$-covering of the RKHS norm ball and $F(\lambda)$ in the following two lemmas, respectively.

**Lemma I.2** ($\ell_\infty$-norm covering number of RKHS ball). For any $\epsilon \in (0, 1)$, we let $N_\infty(\epsilon, \mathcal{H}, R)$ denote the $\epsilon$-covering number of the RKHS norm ball $\{f \in \mathcal{H} \colon \|f\|_\mathcal{H} \leq R\}$ with respect to the $\ell_\infty$-norm. Consider the two eigenvalue decay conditions given in Assumption 4.3. Then, under Assumption 4.3, there exist absolute constants $C_3$ and $C_4$ such that

$$\log N_\infty(\epsilon, \mathcal{H}, R) \leq \begin{cases} C_3 \cdot \gamma \cdot \big[\log(R/\epsilon) + C_4\big] & \gamma\text{-finite spectrum,} \\ C_3 \cdot \big[\log(R/\epsilon) + C_4\big]^{1+1/\gamma} & \gamma\text{-exponential decay,} \end{cases}$$

where $C_3$ and $C_4$ are independent of $T$, $H$, $R$, and $\epsilon$, and only depend on absolute constants $C_\psi$, $C_1$, $C_2$, $\gamma$, and $\tau$ specified in Assumption 4.3.

*Proof.* See §J.2 for a detailed proof. $\qquad\square$

**Lemma I.3.** For any $\epsilon \in (0, 1/e)$, let $N_\infty(\epsilon, \mathcal{F}, \lambda)$ be the $\epsilon$-covering number of function class $\mathcal{F}(\lambda)$ with respect to the $\ell_\infty$-norm, where $\mathcal{F}(\lambda)$ is defined in (I.12). Here we assume that $\lambda$ is bounded in $[c_1, c_2]$, where both $c_1$ and $c_2$ are absolute constants. Then, under Assumption 4.3, there exist absolute constants $C_5$ and $C_6$ such that

$$\log N_\infty(\epsilon, \mathcal{F}, \lambda) \leq \begin{cases} C_5 \cdot \gamma^2 \cdot \big[\log(1/\epsilon) + C_6\big] & \gamma\text{-finite spectrum,} \\ C_5 \cdot \big[\log(1/\epsilon) + C_6\big]^{1+2/\gamma} & \gamma\text{-exponential decay} \end{cases}$$

where $C_5$ and $C_6$ only depend on $C_\psi$, $C_1$, $C_2$, $\gamma$, $\tau$, $c_1$, and $c_2$, and do not rely on $T$, $H$, or $\epsilon$.

*Proof.* See §J.3 for a detailed proof. $\qquad\square$

Finally, we conclude the proof by combining Lemmas I.2 and I.3. Specifically, by (I.13) and (I.14), we have

$$\log\big|\mathcal{C}\big(\mathcal{Q}_{\mathrm{ucb}}(h, R, B), \epsilon\big)\big| \leq \log N_\infty(\epsilon/3, \mathcal{H}, R) + \log N(\epsilon \cdot \lambda/3, B) + \log N_\infty\big(\epsilon/(3B), \mathcal{F}, \lambda\big) \tag{I.15}$$

$$\leq \log\big[1 + 6B/(\epsilon \cdot \lambda)\big] + \log N_\infty(\epsilon/3, \mathcal{H}, R) + \log N_\infty\big(\epsilon/(3B), \mathcal{F}, \lambda\big).$$

We consider the two eigenvalue decay conditions separately. For the $\gamma$-finite spectrum case, by Lemmas I.2 and I.3 and (I.15) we have

$$\log\big|\mathcal{C}\big(\mathcal{Q}_{\mathrm{ucb}}(h, R, B), \epsilon\big)\big|$$

$$\leq \log\big[1 + 6B/(\epsilon \cdot \lambda)\big] + C_3 \cdot \gamma \cdot \big[\log(3R/\epsilon) + C_4\big] + C_5 \cdot \gamma^2 \cdot \big[\log(3B/\epsilon) + C_6\big]$$

$$\leq C_N \cdot \gamma \cdot \big[1 + \log(R/\epsilon)\big] + C_N \cdot \gamma^2 \cdot \big[1 + \log(B/\epsilon)\big],$$

where $C_N$ is an absolute constant. Similarly, for the case where the eigenvalues satisfy the $\gamma$-exponential decay condition, by Lemmas I.2 and I.3 we have

$$\log\big|\mathcal{C}\big(\mathcal{Q}_{\mathrm{ucb}}(h, R, B), \epsilon\big)\big|$$

$$\leq \log\big[1 + 6B/(\epsilon \cdot \lambda)\big] + C_3 \cdot \big[\log(3R/\epsilon) + C_4\big]^{1+1/\gamma} + C_5 \cdot \big[\log(3B/\epsilon) + C_6\big]^{1+2/\gamma}$$

$$\leq C_N \cdot \big[1 + \log(R/\epsilon)\big]^{1+1/\gamma} + C_N \cdot \big[1 + \log(B/\epsilon)\big]^{1+2/\gamma}$$

for some absolute constant $C_N > 0$. Therefore, we conclude the proof. $\qquad\square$

## I.2 Effective Dimension of RKHS

**Definition I.4** (Maximal information gain). For any fixed integer $T$ and any $\sigma > 0$, we define the maximal information gain associated with the RKHS $\mathcal{H}$ as

$$\Gamma_K(T, \sigma^2) = \sup_{\mathcal{D} \subseteq \mathcal{Z}} \left\{ 1/2 \cdot \text{logdet}(I + \sigma^{-2} \cdot K_\mathcal{D}) \right\}, \qquad (I.16)$$

where the supremum is taken over all discrete subsets of $\mathcal{Z}$ with cardinality no more than $T$, and $K_D$ is the Gram matrix induced by $\mathcal{D} \subseteq \mathcal{Z}$, which is defined similarly as in (3.7). Here the subscript $K$ in $\Gamma_K(T, \sigma^2)$ denotes the kernel function of $\mathcal{H}$.

The maximal information gain naturally arises in Gaussian process regression. Specifically, let $f \sim \text{GP}(0, K)$ be draw from the Gaussian process with covariance kernel $K$. Let $\mathcal{D} = \{z_1, \ldots, z_{|\mathcal{D}|}\}$ be a subset of $\mathcal{Z}$ with $|\mathcal{D}| \leq T$ elements. Suppose that we observe noisy observations of $f$ at points in $\mathcal{D}$. That is, for any $z_i \in \mathcal{D}$, we have $y_i = f(z_i) + \epsilon_i$, where $\epsilon_i \sim N(0, \sigma^2)$ is a random Gaussian noise. We let $y_\mathcal{D}$ denote the vector whose entries are $y_i$. Then, the information gain of $y_\mathcal{D}$ is defined as the mutual information between $f$ and the observations $y_\mathcal{D}$, denoted by $I(f, y_\mathcal{D})$. By direct computation, we have

$$I(f, y_\mathcal{D}) = 1/2 \cdot \text{logdet}(I + \sigma^{-2} \cdot K_\mathcal{D}).$$

The mutual information $I(f, y_\mathcal{D})$ quantifies the reduction of the uncertainty about $f$ when we observe $y_\mathcal{D}$. Thus, the maximal mutual information $\Gamma_K(T, \sigma^2)$ characterizes the maximal possible reduction of the uncertainty of $f$ when having no more than $T$ observations.

Moreover, we note that, when $\sigma^2$ is a constant, $\Gamma_K(T, \sigma^2)$ depends on the eigenvalue decay of the RKHS and thus can be viewed as an effective dimension of the RKHS. Specifically, as shown in [62], when the kernel is the $d$-dimensional linear kernel, $\Gamma_K(T, \sigma^2) = \mathcal{O}(d \log T)$. Moreover, for the squared exponential kernel that satisfies the exponential eigenvalue decay condition, the maximal information gain is $\mathcal{O}((\log T)^{d+1})$. In the following lemma, similar to Theorem 5 in [62], we establish upper bounds on the maximal information gain of the RKHS under the eigenvalue decay conditions specified in Assumption 4.3.

**Lemma I.5** (Theorem 5 in [62]). Let $\mathcal{Z}$ be a compact subset of $\mathbb{R}^d$ and $K: \mathcal{Z} \times \mathcal{Z} \to \mathbb{R}$ be the RKHS kernel of $\mathcal{H}$. We assume that $K$ is a bounded kernel in the sense that $\sup_{z \in \mathcal{Z}} K(z, z) \leq 1$, and $K$ is continuously differentiable on $\mathcal{Z} \times \mathcal{Z}$. Moreover, let $T_K$ be the integral operator induced by $K$ and the Lebesgue measure on $\mathcal{Z}$, whose definition is given in (B.1). Let $\{\sigma_j\}_{j \geq 1}$ be the eigenvalues of $T_K$ in the descending order. We assume that $\{\sigma_j\}_{j \geq 1}$ satisfy either one of the following three eigenvalue decay conditions:

(i) $\gamma$-finite spectrum: We have $\sigma_j = 0$ for all $j \geq \gamma + 1$, where $\gamma$ is a positive integer.

(ii) $\gamma$-exponential eigenvalue decay: There exist constants $C_1, C_2 > 0$ such that $\sigma_j \leq C_1 \exp(-C_2 \cdot j^\gamma)$ for all $j \geq 1$, where $\gamma > 0$ is positive constant.

Let $\sigma$ be bounded in interval $[c_1, c_2]$ with $c_1$ and $c_2$ being absolute constants. Then, for conditions (i)–(iii) respectively, we have

$$\Gamma_K(T, \sigma^2) \leq \begin{cases} C_K \cdot \gamma \cdot \log T & \gamma\text{-finite spectrum,} \\ C_K \cdot (\log T)^{1+1/\gamma} & \gamma\text{-exponential decay,} \end{cases}$$

where $C_K$ is an absolute constant that depends on $d$, $\gamma$, $C_1$, $C_2$, $C$, $c_1$, and $c_2$.

We note that Lemma I.5 is a generalization of Theorem 5 in [62], which establishes the maximal information gain for the linear, squared exponential, and Matérn kernels, respectively. Specifically, the squared exponential kernel satisfies the $\gamma$-exponential eigenvalue decay condition with $\gamma = 1/d$. Lemma I.5 implies that the $\Gamma_K(T, \sigma^2) = \mathcal{O}((\log T)^{d+1})$, which matches Theorem 5 in [62].

*Proof.* The proof of this lemma is based on a modification of that of Theorem 5 in [62]. To begin with, for any $j \in \mathbb{N}$, we define $B_K(j) = \sum_{s > j} \sigma_s$, i.e., the sum of eigenvalues with indices larger

than $j$. Then, we use the following lemma obtained from [62] to bound $\Gamma_K(T, \sigma^2)$ using function $B_K$.

**Lemma I.6** (Theorem 8 in [62])**.** Under the same condition as in Lemma I.5, for any fixed $\tau > 0$, we denote $C_\tau = 2\mu(\mathcal{Z}) \cdot (2\tau + 1)$ where $\mu(\mathcal{Z})$ is the Lebesgue measure of $\mathcal{Z}$. Let $n_T$ denote $C_\tau \cdot T^\tau \cdot \log T$. Then, for any $T_\star \in \{1, \ldots, n_T\}$, we have

$$\Gamma_K(T, \sigma^2) \leq T_\star \cdot \log(T \cdot n_T/\sigma^2) + C_\tau \cdot \sigma^{-2} \cdot \log T \cdot \left[T^{\tau+1} \cdot B_K(T_\star) + 1\right] + \mathcal{O}(T^{1-\tau/d}).$$

*Proof.* See [62] for a detailed proof. $\qquad\square$

In the following, we choose proper $\tau$ and $T_\star$ in Lemma I.6 for the two eigenvalue decay conditions separately.

**Case (i): $\gamma$-Finite Spectrum.** When $\sigma_j = 0$ for all $j \geq \gamma + 1$, we set $\tau = d$ and $T_\star = \gamma$ in Lemma I.6. Then we have $B_K(T_\star) = 0$ and $n_T = C_d \cdot T^d \cdot \log T$. When $T$ is sufficiently large, it holds that $T_\star < n_T$. Then Lemma I.6 implies that

$$\Gamma_K(T, \sigma^2) \leq \gamma \cdot \log\left(C_d \cdot T^{d+1} \cdot \log T/\sigma^2\right) + C_d \cdot \sigma^{-2} \cdot \log T + \mathcal{O}(1) \leq C_K \cdot \gamma \cdot \log T,$$

for some absolute constant $C_K > 0$. Thus, we conclude the proof for the first case.

**Case (ii): $\gamma$-Exponential Decay.** When $\{\sigma_j\}_{j\geq 1}$ satisfies the $\gamma$-exponential eigenvalue decay condition, for any $T_\star \in \mathbb{N}$, we have

$$B_K(T_\star) = \sum_{j > T_\star} \sigma_j \leq C_1 \cdot \sum_{j > T_\star} \exp(-C_2 \cdot j^\gamma) \leq C_1 \cdot \int_{T_\star}^\infty \exp(-C_2 \cdot u^\gamma)\, \mathrm{d}u. \tag{I.17}$$

In a manner similar to the derivation of (J.16), by direct computation we have

$$\int_{T_\star}^\infty \exp(-C_2 \cdot u^\gamma)\, \mathrm{d}u \leq \begin{cases} C_2^{-1} \cdot \exp(-C_2 \cdot T_\star^\gamma), & \text{if } \gamma \geq 1, \\ 2 \cdot (\gamma \cdot C_2)^{-1} \cdot \exp(-C_2 \cdot T_\star^\gamma) \cdot T_\star^{1-\gamma}, & \text{if } \gamma \in (0,1). \end{cases} \tag{I.18}$$

In the following, we set $\tau = d$. Then we have $n_T = C_d \cdot T^d \cdot \log T$ where $C_d = 2\mu(\mathcal{Z}) \cdot (2d+1)$. Then we have

$$\log(T \cdot n_T) = \log(C_d) + \log \cdot (T^{d+1} \cdot \log T) \leq \log(C_d) + 2(d+1) \cdot \log T, \tag{I.19}$$

when $T$ is sufficiently large. Moreover, combining Lemma I.6 and (I.19), when $\sigma$ is sandwiched by absolute constants $c_1$ and $c_2$, we have

$$\Gamma_K(T, \sigma^2) \leq \widetilde{C}_1 \cdot T_\star \cdot \log T + \widetilde{C}_2 \cdot \log T \cdot \left[T^{d+1} \cdot B_K(T_\star) + 1\right] + \widetilde{C}_3, \tag{I.20}$$

where $\widetilde{C}_1$, $\widetilde{C}_2$, and $\widetilde{C}_3$ are absolute constants that depend on $d$, $\gamma$, $c_1$, $c_2$, $C_1$, and $C_2$. Now we choose $T_\star$ such that

$$\exp(C_2 \cdot T_\star^\gamma) \asymp T \cdot n_T = C_d \cdot T^{d+1} \cdot \log T, \tag{I.21}$$

that is, $T_\star = \widetilde{C}_4 \cdot (\log T)^{1/\gamma}$ where $\widetilde{C}_4$ is an absolute constant. Notice that $T_\star < n_T$ when $T$ is sufficiently large.

Thus, combining (I.17), (I.18), and (I.21), for $\gamma \geq 1$, we have

$$\log T \cdot \left[T^{d+1} \cdot B_K(T_\star) + 1\right]$$
$$\leq C_1 \cdot C_2^{-1} \log T \cdot T^{d+1} \cdot \exp(-C_2 \cdot T_\star^\gamma) + \log T \leq 2\log T, \tag{I.22}$$

where the last inequality follows from (I.21). Similarly, for $\gamma \in (0,1)$, by (I.17), (I.18), and (I.21), we have

$$\log T \cdot \left[T^{d+1} \cdot B_K(T_\star) + 1\right]$$
$$\leq 2C_1 \cdot (\gamma \cdot C_2)^{-1} \cdot \exp(-C_2 \cdot T_\star^\gamma) \cdot \log T \cdot T^{d+1} \cdot T_\star^{1-\gamma} + \log T \asymp (\log T)^{1/\gamma-1} + \log T. \tag{I.23}$$

Thus, combining (I.20), (I.22), (I.23), we conclude that

$$\Gamma_K(T, \sigma^2) \leq C_K \cdot \log(T)^{1+1/\gamma}$$

for any $\gamma \geq 0$, where $C_K$ is an absolute constant that depends on $d$, $\gamma$, $c_1$, $c_2$, $C_1$, and $C_2$. Thus, we conclude the proof for the second case. Therefore, we conclude the proof of Lemma I.5. $\qquad\square$

# J Proofs of Auxiliary Results

In this section, we provide the proofs of the auxiliary results.

## J.1 Proof of Lemma H.1

*Proof.* For any function $f \in \mathcal{H}$, using the feature representation induced by the kernel $K$, we have

$$\left| \langle f, \widehat{\theta}_h^t \rangle_{\mathcal{H}} \right| = \left| f^\top \widehat{\theta}_h^t \right| \le \left| f^\top (\Lambda_h^t)^{-1} \Phi^\top y_h^t \right|$$

$$= \left| f^\top (\Lambda_h^t)^{-1} \sum_{\tau=1}^{t-1} \phi(x_h^\tau, a_h^\tau) \cdot [r_h(x_h^\tau, a_h^\tau) + V_{h+1}^t(x_{h+1}^\tau)] \right|, \tag{J.1}$$

where we let $\Phi$ denote $\Phi_h^t$ defined in (H.13) for simplicity. Since $|r_h(x_h^\tau, a_h^\tau)| \le 1$ and $|V_{h+1}^t(x_{h+1}^\tau)| \le H - h$, we have $|[r_h(x_h^\tau, a_h^\tau) + V_{h+1}^t(x_{h+1}^\tau)]| \le H$ for all $h \in [H]$ and $\tau \in [t-1]$. Then, by (J.1) and the Cauchy-Schwarz inequality, we have

$$\left| \langle f, \widehat{\theta}_h^t \rangle_{\mathcal{H}} \right| \le H \cdot \sum_{\tau=1}^{t-1} \left| f^\top (\Lambda_h^t)^{-1} \phi(x_h^\tau, a_h^\tau) \right|$$

$$\le H \cdot \left[ \sum_{\tau=1}^{t-1} f^\top (\Lambda_h^t)^{-1} f \right]^{1/2} \cdot \left[ \sum_{\tau=1}^{t-1} \phi(x_h^\tau, a_h^\tau)^\top (\Lambda_h^t)^{-1} \phi(x_h^\tau, a_h^\tau) \right]^{1/2}$$

$$\le H/\sqrt{\lambda} \cdot \|f\|_{\mathcal{H}} \cdot \left[ \sum_{\tau=1}^{t-1} \phi(x_h^\tau, a_h^\tau)^\top (\Lambda_h^t)^{-1} \phi(x_h^\tau, a_h^\tau) \right]^{1/2}, \tag{J.2}$$

where the last inequality follows from the fact that $(\Lambda_h^t)^{-1} \colon \mathcal{H} \to \mathcal{H}$ is a self-adjoint and positive-definite operator whose eigenvalues are bounded by $1/\lambda$. Furthermore, by Lemma J.3, we have

$$\left[ \sum_{\tau=1}^{t-1} \phi(x_h^\tau, a_h^\tau)^\top (\Lambda_h^t)^{-1} \phi(x_h^\tau, a_h^\tau) \right] \le 2\mathrm{logdet}(I + K_h^t/\lambda). \tag{J.3}$$

Thus, combining (J.2), (J.3), and the fact that $\lambda \ge 1$, we obtain that

$$\left| \langle f, \widehat{\theta}_h^t \rangle_{\mathcal{H}} \right| \le H \cdot \|f\|_{\mathcal{H}} \cdot \sqrt{2/\lambda \cdot \mathrm{logdet}(I + K_h^t/\lambda)} \le H \cdot \|f\|_{\mathcal{H}} \cdot \sqrt{2 \cdot \mathrm{logdet}(I + K_h^t/\lambda)}.$$

Finally, utilizing the definition of $\Gamma_K(T, \lambda)$ in (I.16), we conclude the proof of this lemma. $\qquad\square$

## J.2 Proof of Lemma I.2

*Proof.* Recall that we have defined the integral operator $T_K \colon \mathcal{L}^2(\mathcal{Z}) \to \mathcal{L}^2(\mathcal{Z})$ defined in (B.1), which has eigenvalues $\{\sigma_j\}_{j \ge 0}$ and eigenvectors $\{\psi_j\}_{j \ge 0}$. Moreover, $\{\psi_j\}$ and $\{\sqrt{\sigma_j} \cdot \psi_j\}_{j \ge 0}$ are orthonormal bases of $\mathcal{L}_2(\mathcal{Z})$ and $\mathcal{H}$, respectively. Then, any $\in \mathcal{H}$ with $\|f\|_{\mathcal{H}} \le R$ can be written as

$$f = \sum_{j=1}^{\infty} w_j \cdot \sqrt{\sigma_j} \cdot \psi_j, \tag{J.4}$$

where $\{w_j\}_{j \ge 0}$ satisfy $\sum_{j=1}^{\infty} w_j^2 = \|f\|_{\mathcal{H}}^2 \le R^2$. Let $m$ be any positive integer and let $\Pi_m \colon \mathcal{H} \to \mathcal{H}$ denote the projection onto the subspace spanned by $\{\psi_j\}_{j \in [m]}$, i.e., $\Pi_m(f) = \sum_{j=1}^{m} w_j \cdot \sqrt{\sigma_j} \cdot \psi_j$ for any $f \in \mathcal{H}$ written as in (J.4). Then we have

$$\|f - \Pi_m(f)\|_\infty = \sum_{j=m+1}^{\infty} |w_j| \cdot \sqrt{\sigma_j} \cdot \sup_{z \in \mathcal{Z}} |\psi_j(z)|. \tag{J.5}$$

In the following, we consider the two eigenvalue decay conditions specified in Assumption 4.3 separately.

**Case (i): $\gamma$-Finite Spectrum.** Consider the case where $\sigma_j = 0$ for all $j > \gamma$. Then, by the definition of $\Pi_m$, we have $f = \Pi_\gamma(f)$ for all $f \in \mathcal{H}$. That is, (J.4) is reduced to

$$f = \sum_{j=1}^{\gamma} w_j \cdot \sqrt{\sigma_j} \cdot \psi_j,$$

where $\{w_j\}_{j \in [\gamma]}$ satisfies $\sum_{j=1}^{\gamma} w_j^2 \leq R^2$. Let $\mathcal{C}_\gamma(\epsilon, R)$ be the minimal $\epsilon$-cover of the $\gamma$-dimensional Euclidean ball $\{w \in \mathbb{R}^\gamma \colon \|w\|_2 \leq R\}$ with respect to the Euclidean norm. Then, by construction, there exists $\widetilde{w} \in \mathbb{R}^\gamma$ such that $\sum_{j=1}^{\gamma} (w_j - \widetilde{w}_j)^2 \leq \epsilon^2$. Then, by the Cauchy-Schwarz inequality, we have

$$\left\| f - \sum_{j=1}^{\gamma} \widetilde{w}_j \cdot \sqrt{\sigma_j} \cdot \psi_j \right\|_\infty = \sup_{z \in \mathcal{Z}} \left| \sum_{j=1}^{\gamma} (w_j - \widetilde{w}_j) \cdot \sqrt{\sigma_j} \cdot \psi_j(z) \right| \tag{J.6}$$

$$= \left[ \sum_{j=1}^{\gamma} (w_j - \widetilde{w}_j)^2 \right]^{1/2} \cdot \sup_{z \in \mathcal{Z}} \left\{ \left[ \sum_{j=1}^{\gamma} \sigma_j \cdot |\psi_j(z)|^2 \right]^{1/2} \right\} \leq \epsilon \cdot \sup_z \sqrt{K(z,z)} \leq \epsilon,$$

where the last equality follows from the fact that $K(z,z) = \sum_{j=1}^{\gamma} \sigma_j \cdot |\psi_j(z)|^2$. Thus, the $\epsilon$-covering of $\{f \in \mathcal{H} \colon \|f\|_\mathcal{H} \leq R\}$ is bounded by the cardinality of $\mathcal{C}_\gamma(\epsilon, R)$. As shown in [69, Corollary 4.2.13], we have

$$\big| \mathcal{C}_\gamma(\epsilon, R) \big| \leq (1 + 2R/\epsilon)^\gamma. \tag{J.7}$$

Thus, combining (J.6) and (J.7), we have

$$\log N_\infty(\epsilon, \mathcal{H}, R) \leq \gamma \cdot \log(1 + 2R/\epsilon) \leq C_3 \cdot \gamma \cdot \big[ \log(R/\epsilon) + C_4 \big],$$

where both $C_3$ and $C_4$ are absolute constants. Thus, we conclude the proof for the first case.

**Case (ii): $\gamma$-Exponential Decay.** In the following, we assume the eigenvalues $\{\sigma_j\}_{j \geq 1}$ satisfy the $\gamma$-exponential decay condition and $\|\psi_j\|_\infty \leq C_\psi \cdot \sigma_j^{-\tau}$ for all $j \geq 1$. Thus, by (J.5) we have

$$\|f - \Pi_m(f)\|_\infty \leq \sum_{j=m+1}^{\infty} C_\psi \cdot |w_j| \cdot \sigma_j^{1/2 - \tau}$$

$$\leq \sum_{j=m+1}^{\infty} C_\psi \cdot C_1^{1/2 - \tau} \cdot |w_j| \cdot \exp\big[ -C_2 \cdot (1/2 - \tau) \cdot j^\gamma \big]. \tag{J.8}$$

To simplify the notation, we define $C_{1,\tau} = C_\psi \cdot C_1^{1/2 - \tau}$ and $C_{2,\tau} = C_2 \cdot (1 - 2\tau)$. Then, applying the Cauchy-Schwarz inequality to (J.8), we have

$$\|f - \Pi_m(f)\|_\infty \leq C_{1,\tau} \cdot \left( \sum_{j=m+1}^{\infty} |w_j|^2 \right)^{1/2} \cdot \left[ \sum_{j=m+1}^{\infty} \exp(-C_{2,\tau} \cdot j^\gamma) \right]^{1/2}$$

$$\leq C_{1,\tau} \cdot R \cdot \left[ \sum_{j=m+1}^{\infty} \exp(-C_{2,\tau} \cdot j^\gamma) \right]^{1/2}, \tag{J.9}$$

where the second inequality follows from the fact that $\sum_{j \geq 1} w_j^2 \leq R^2$. Since $\gamma > 0$, $\exp(-u^\gamma)$ is monotonically decreasing in $u$. Thus, we have

$$\sum_{j=m+1}^{\infty} \exp(-C_{2,\tau} \cdot j^\gamma) \leq \int_m^\infty \exp(-C_{2,\tau} \cdot u^\gamma) \, \mathrm{d}u. \tag{J.10}$$

In the following, we bound the integral in (J.10) by considering the cases where $\gamma \geq 1$ and $\gamma \in (0, 1)$ separately. First, when $\gamma \geq 1$, since $d \geq 1$, we have $u^{\gamma-1} \geq 1$ for all $u \geq d$. Hence, we have

$$\int_m^\infty \exp(-C_{2,\tau} \cdot u^\gamma) \, \mathrm{d}u \leq \int_m^\infty u^{\gamma-1} \cdot \exp(-C_{2,\tau} \cdot u^\gamma) \, \mathrm{d}u$$

$$\leq \int_{m^\gamma}^\infty \exp(-C_{2,\tau} \cdot v) \, \mathrm{d}v = C_{2,\tau}^{-1} \cdot \exp(-C_{2,\tau} \cdot m^\gamma), \tag{J.11}$$

where the second inequality follows from the change of variable $v = u^\gamma$ and the fact that $\gamma \geq 1$. Second, when $\gamma < 1$, by letting $v = u^\gamma$, we have

$$\int_m^\infty \exp(-C_{2,\tau} \cdot u^\gamma) \, \mathrm{d}u = \frac{1}{\gamma} \cdot \int_{m^\gamma}^\infty \exp(-C_{2,\tau} \cdot v) \cdot v^{1/\gamma - 1} \, \mathrm{d}v = \frac{1}{\gamma \cdot C_{2,\tau}} \int_{m^\gamma}^\infty v^{1/\gamma - 1} \, \mathrm{d}[-\exp(-C_{2,\tau} \cdot v)]$$

$$= \frac{1}{\gamma \cdot C_{2,\tau}} \cdot \exp(-C_{2,\tau} \cdot m^\gamma) \cdot m^{1-\gamma} + \frac{(1 - \gamma)}{\gamma^2 \cdot C_{2,\tau}} \int_{m^\gamma}^\infty \exp(-C_{2,\tau} \cdot v) \cdot v^{1/\gamma - 2} \, \mathrm{d}v, \tag{J.12}$$

where the last equality follows from integration by parts. Moreover, by direct calculation, we have

$$\frac{1}{\gamma} \int_{m^\gamma}^\infty \exp(-C_{2,\tau} \cdot v) \cdot v^{1/\gamma-2} \, \mathrm{d}v \le \frac{1}{m^\gamma} \cdot \frac{1}{\gamma} \int_{m^\gamma}^\infty \exp(-C_{2,\tau} \cdot v) \cdot v^{1/\gamma-1} \, \mathrm{d}v$$

$$= \frac{1}{m^\gamma} \int_m^\infty \exp(-C_{2,\tau} \cdot u^\gamma) \, \mathrm{d}u, \qquad (J.13)$$

where the first inequality follows from the fact that $v \ge m^\gamma$ in the integral and the second equality follows from letting $u = v^{1/\gamma}$. Then, combining (J.12) and (J.13), we have

$$\int_m^\infty \exp(-C_{2,\tau} \cdot u^\gamma) \, \mathrm{d}u$$

$$\le \frac{1}{\gamma \cdot C_{2,\tau}} \cdot \exp(-C_{2,\tau} \cdot m^\gamma) \cdot m^{1-\gamma} + \frac{1/\gamma - 1}{C_{2,\tau} \cdot m^\gamma} \cdot \int_m^\infty \exp(-C_{2,\tau} \cdot u^\gamma) \, \mathrm{d}u. \quad (J.14)$$

Thus, when $m$ is sufficiently large such that $m^\gamma \cdot C_{2,\tau} > 2/\gamma - 2$, by (J.14) we have

$$\int_m^\infty \exp(-C_{2,\tau} \cdot u^\gamma) \, \mathrm{d}u \le \left( 1 - \frac{1/\gamma - 1}{C_{2,\tau} m^\gamma} \right)^{-1} \cdot \frac{1}{\gamma \cdot C_{2,\tau}} \exp(-C_{2,\tau} \cdot m^\gamma) \cdot m^{1-\gamma}$$

$$\le \frac{2}{\gamma \cdot C_{2,\tau}} \exp(-C_{2,\tau} \cdot m^\gamma) \cdot m^{1-\gamma}. \qquad (J.15)$$

Therefore, combining (J.10), (J.11), and (J.15), we obtain that

$$\int_m^\infty \exp(-C_{2,\tau} \cdot u^\gamma) \, \mathrm{d}u \le \begin{cases} C_{2,\tau}^{-1} \cdot \exp(-C_{2,\tau} \cdot m^\gamma), & \text{if } \gamma \ge 1, \\ 2 \cdot (\gamma \cdot C_{2,\tau})^{-1} \cdot \exp(-C_{2,\tau} \cdot m^\gamma) \cdot m^{1-\gamma}, & \text{if } \gamma \in (0,1). \end{cases}$$

$$(J.16)$$

In the sequel, we let $m^*$ be the smallest integer such that

$$\int_m^\infty \exp(-C_{2,\tau} \cdot u^\gamma) \, \mathrm{d}u \le \left( \frac{\epsilon}{2C_{1,\tau} \cdot R} \right)^2, \qquad \forall m \ge m^*. \qquad (J.17)$$

Hence, combining (J.9), (J.10), and (J.17), we have $\|f - \Pi_{m^*}(f)\|_\infty \le \epsilon/2$ for any $f \in \mathcal{H}$ with $\|f\|_{\mathcal{H}} \le R$. Note, moreover, that $C_{1,\tau}, C_{2,\tau}$, and $\gamma$ are all absolute constants. By (J.16) and (J.17), there exist absolute constants $C_{1,m}$ and $C_{2,m}$ such that

$$m^* \le C_{1,m} \cdot \left[ \log(R/\epsilon) + C_{2,m} \right]^{1/\gamma}. \qquad (J.18)$$

Finally, it remains to approximate $\Pi_{m^*}(f)$ up to error $\epsilon/2$ for $m^*$ specified in (J.17). By the expansion of $f$ in (J.4), we have $\Pi_{m^*}(f) = \sum_{j=1}^{m^*} w_j \cdot \sqrt{\sigma_j} \cdot \psi_j$. For any $m^*$ real numbers $\{\widetilde{w}_j\}_{j \in [m^*]}$, by the Cauchy-Schwarz inequality, we have

$$\left| [\Pi_{m^*}(f)](z) - \sum_{j=1}^{m^*} \widetilde{w}_j \cdot \sqrt{\sigma_j} \cdot \psi_j(z) \right| = \left| \sum_{j=1}^{m^*} (w_j - \widetilde{w}_j) \cdot \sqrt{\sigma_j} \cdot \psi_j(z) \right|$$

$$\le \left[ \sum_{j=1}^{m^*} (w_j - \widetilde{w}_j)^2 \right]^{1/2} \cdot \left\{ \sum_{j=1}^{m^*} \sigma_j \cdot [\psi_j(z)]^2 \right\}^{1/2} \le \sqrt{K(z,z)} \cdot \left[ \sum_{j=1}^{m^*} (w_j - \widetilde{w}_j)^2 \right]^{1/2},$$

$$(J.19)$$

where the last inequality follows from the fact that $K(z,z) = \sum_{j=1}^\infty \sigma_j \cdot [\psi_j(z)]^2$. Under Assumption 4.3, we have $\sup_{z \in \mathcal{Z}} K(z,z) \le 1$. Notice that $\sum_{j=1}^{m^*} \omega_j^2 \le \|f\|_{\mathcal{H}}^2 \le R^2$. Let $\mathcal{C}_{m^*}(\epsilon/2, R)$ be the minimal $\epsilon/2$-cover of $\{w \in \mathbb{R}^{m^*} : \|w\|_2 \le R\}$ with respect to the Euclidean norm. By definition, for any $f \in \mathcal{H}$ with $\|f\|_{\mathcal{H}} \le R$, there exist $\widetilde{w} \in \mathcal{C}_{m^*}(\epsilon/2, R)$ such that $\sum_{j=1}^{m^*} (w_j - \widetilde{w}_j)^2 \le \epsilon^2/4$. Therefore, by (J.19) we have

$$\left\| f - \sum_{j=1}^{m^*} \widetilde{w}_j \cdot \sqrt{\sigma_j} \cdot \psi_j \right\|_\infty \le \|f - \Pi_{m^*}(f)\|_\infty + \left\| \Pi_{m^*}(f) - \sum_{j=1}^{m^*} \widetilde{w}_j \cdot \sqrt{\sigma_j} \cdot \psi_j \right\|_\infty \le \epsilon, \quad (J.20)$$

which implies that the $\epsilon$-covering number of the RKHS norm ball $\{f \in \mathcal{H} : \|f\|_{\mathcal{H}} \le R\}$ is bounded by the cardinality of $\mathcal{C}_{m^*}(\epsilon/2, R)$, i.e., $N_\infty(\epsilon, \mathcal{H}, R) \le |\mathcal{C}_{m^*}(\epsilon/2, R)|$. As shown in [69, Corollary

4.2.13], we have

$$\left|\mathcal{C}_{m^*}(\epsilon/2, R)\right| \leq (1 + 4R/\epsilon)^{m^*}. \tag{J.21}$$

Therefore, combining (J.18) and (J.21), we have

$$\log N_\infty(\epsilon, \mathcal{H}, R) \leq m^* \cdot \log(1 + 4R/\epsilon) \leq C_{1,m} \cdot \left[\log(R/\epsilon) + C_{2,m}\right]^{1/\gamma} \cdot \left[\log(1 + 4R/\epsilon)\right]$$

$$\leq C_3 \cdot \left[\log(R/\epsilon) + C_4\right]^{1+1/\gamma},$$

where $C_3$ and $C_4$ are absolute constants that only depend on $C_\Psi$, $C_1$, $C_2$, $\gamma$, and $\tau$, which are specified in Assumption 4.3. Thus we conclude the proof of this lemma. $\qquad \square$

## J.3 Proof of Lemma I.3

*Proof.* As shown in §B.1, the feature mapping $\phi \colon \mathcal{Z} \to \mathcal{H}$ satisfies

$$\phi(z) = \sum_{j=1}^{\infty} \sigma_j \cdot \psi_j(z) \cdot \psi_j = \sum_{j=1}^{\infty} \sqrt{\sigma_j} \cdot \psi_j(z) \cdot (\sqrt{\sigma_j} \cdot \psi_j). \tag{J.22}$$

That is, when expanding $\phi(z) \in \mathcal{H}$ in the basis $\{\sqrt{\sigma_j} \cdot \psi_j\}_{j \geq 0}$ as in (J.4), the $j$-th coefficient is equal to $\sqrt{\sigma_j} \cdot \psi_j(z)$ for all $j \geq 1$. Similar to the proof of Lemma I.2, in the following, we consider the two eigenvalue decay conditions separately.

**Case (i): $\gamma$-Finite Spectrum.** When $\mathcal{H}$ has only $\gamma$ nonzero eigenvalues, for any $z \in \mathcal{Z}$, we define a vector $w_z \in \mathbb{R}^\gamma$ by letting its $j$-th entry be $\sqrt{\sigma_j} \cdot \psi_j(z)$ for all $j \in [\gamma]$. Moreover, for any self-adjoint operator $\Upsilon \colon \mathcal{H} \to \mathcal{H}$ satisfying $\|\Upsilon\|_{\mathrm{op}} \leq 1/\lambda$, we define a matrix $A_\Upsilon \in \mathbb{R}^{\gamma \times \gamma}$ as follows. For any $j, k \in [\gamma]$, we define the $(j, k)$-th entry of $A_\Upsilon$ as

$$[A_\Upsilon]_{j,k} = \left\langle \sqrt{\sigma_j} \cdot \psi_j, \sqrt{\sigma_k} \cdot \Upsilon\psi_k \right\rangle_{\mathcal{H}}.$$

By (J.22) and the definition of $A_\Upsilon$, we have

$$\|\phi(z)\|_\Upsilon^2 = \sum_{j,k=1}^{\gamma} \sqrt{\sigma_j} \cdot \psi_j(z) \cdot \sqrt{\sigma_k} \cdot \psi_k(z) \cdot [A_\Upsilon]_{j,k} = w_z^\top A_\Upsilon w_z. \tag{J.23}$$

With a slight abuse of notation, we define $\mathcal{C}_\gamma(\epsilon, \lambda)$ denote the minimal $\epsilon^2$-cover of

$$\left\{ A \in \mathbb{R}^{\gamma \times \gamma} \colon \|A\|_{\mathrm{fro}} \leq \sqrt{\gamma}/\lambda \right\}$$

with respect to the Frobenius norm. Then by definition, there exists $\widetilde{A}_\Upsilon \in \mathcal{C}_\gamma(\epsilon, \lambda)$ such that $\|A_\Upsilon - \widetilde{A}_\Upsilon\|_{\mathrm{fro}} \leq \epsilon^2$, which implies that

$$\left| w_z^\top A_\Upsilon w_z - w_z^\top \widetilde{A}_\Upsilon w_z \right| \leq \|w_z\|_2^2 \cdot \|A_\Upsilon - \widetilde{A}_\Upsilon\|_{\mathrm{op}} \leq \|A_\Upsilon - \widetilde{A}_\Upsilon\|_{\mathrm{fro}} \leq \epsilon^2, \tag{J.24}$$

where we use the fact that

$$\|w_z\|_2^2 = \sum_{j=1}^{\gamma} |w_j|^2 = \sum_{j=1}^{\gamma} \sigma_j \cdot |\psi_j(z)|^2 = K(z, z) \leq 1.$$

Thus, combining (J.23) and (J.24), and utilizing Corollary 4.2.13 in [69], we have

$$\log N_\infty(\epsilon, \mathcal{F}, \lambda) \leq \log\left|\mathcal{C}_\gamma(\epsilon, \lambda)\right| \leq \gamma^2 \cdot \log\left[1 + 8\sqrt{\gamma}/(\lambda \cdot \epsilon^2)\right] \leq C_5 \cdot \gamma^2 \cdot \left[\log(1/\epsilon) + C_6\right],$$

where $C_5$ and $C_6$ are absolute constants that depend solely on $\lambda$ and $\gamma$. Thus, we conclude the proof for the first case.

**Case (ii): $\gamma$-Exponential Decay.** In the following, we focus on the second case where the eigenvalues satisfy the $\gamma$-exponential decay condition. For any $m \in \mathbb{N}$, we define $\Pi_m \colon \mathcal{H} \to \mathcal{H}$ as the projection operator onto the subspace spanned by $\{\psi_j\}_{j \in [m]}$. Then, by the Cauchy-Schwarz inequality and Assumption 4.3, for any $z \in \mathcal{Z}$, by (J.22) we have

$$\left\|\phi(z) - \Pi_m\big[\phi(z)\big]\right\|_{\mathcal{H}} = \left\| \sum_{j=m+1}^{\infty} \sqrt{\sigma_j} \cdot \psi_j(z) \cdot \sqrt{\sigma_j} \cdot \psi_j \right\|_{\mathcal{H}} = \left\{ \sum_{j=m+1}^{\infty} \sigma_j \cdot [\psi_j(z)]^2 \right\}^{1/2}$$

$$\leq \left( \sum_{j=m+1}^{\infty} \sigma_j \cdot \|\psi_j\|_\infty^2 \right)^{1/2} \leq C_\psi \cdot \left( \sum_{j=m+1}^{\infty} \sigma_j^{1-2\tau} \right)^{1/2}, \tag{J.25}$$

where the second equality follows from the fact that $\{\sqrt{\sigma_j} \cdot \psi_j\}_{j \geq 0}$ form an orthonormal basis of $\mathcal{H}$, the first inequality follows from taking a supremum over $z \in \mathcal{Z}$, and the last inequality follows from the assumption that $\|\psi_j\|_\infty \leq C_\psi \cdot \sigma_j^{-\tau}$. Then, for any self-adjoint operator $\Upsilon \colon \mathcal{H} \to \mathcal{H}$ satisfying $\|\Upsilon\|_{\mathrm{op}} \leq 1/\lambda$ and any $z \in \mathcal{Z}$, by (J.25) and the triangle inequality we have

$$\left| \|\phi(z)\|_\Upsilon - \big\|\Pi_m\big[\phi(z)\big]\big\|_\Upsilon \right| \leq \big\|\phi(z) - \Pi_m\big[\phi(z)\big]\big\|_\Upsilon \leq C_\psi/\sqrt{\lambda} \cdot \left( \sum_{j=m+1}^\infty \sigma_j^{1-2\tau} \right)^{1/2}. \quad \text{(J.26)}$$

Note that the eigenvalues $\{\sigma_j\}_{j \geq 0}$ admit $\gamma$-exponential decay under Assumption 4.3. We now upper bound the right-hand side of (J.26) by

$$\sup_{z \in \mathcal{Z}} \left| \|\phi(z)\|_\Upsilon - \big\|\Pi_m\big[\phi(z)\big]\big\|_\Upsilon \right| \leq C_\psi/\sqrt{\lambda} \cdot \left\{ \sum_{j=m+1}^\infty C_1^{1-2\tau} \cdot \exp\big[-C_2 \cdot (1-2\tau) \cdot j^\gamma\big] \right\}^{1/2}. \quad \text{(J.27)}$$

To simplify the notation, we define $C_{3,\tau} = C_\psi \cdot C_1^{1/2-\tau}/\sqrt{\lambda}$ and $C_{4,\tau} = C_2 \cdot (1-2\tau)$, which are both absolute constants. Then, by (J.27) and the monotonicity of $\exp(-u^\gamma)$, we further obtain

$$\sup_{z \in \mathcal{Z}} \left| \|\phi(z)\|_\Upsilon - \big\|\Pi_m\big[\phi(z)\big]\big\|_\Upsilon \right| \leq C_{3,\tau} \cdot \left[ \int_m^\infty \exp(-C_{4,\tau} \cdot u^\gamma) \, \mathrm{d}u \right]^{1/2}. \quad \text{(J.28)}$$

Here we can take the supremum over $\mathcal{Z}$ because the right-hand side of (J.27) does not depend on $z$. Note that we have shown in (J.16) that

$$\int_m^\infty \exp(-C_{4,\tau} \cdot u^\gamma) \, \mathrm{d}u \leq \begin{cases} C_{4,\tau}^{-1} \cdot \exp(-C_{4,\tau} \cdot m^\gamma), & \text{if } \gamma \geq 1, \\ 2 \cdot (\gamma \cdot C_{4,\tau})^{-1} \cdot \exp(-C_{4,\tau} \cdot m^\gamma) \cdot m^{1/\gamma-1}, & \text{if } \gamma \in (0,1), \end{cases} \quad \text{(J.29)}$$

where for the case of $\gamma \in (0,1)$, (J.29) holds for sufficient large $m$ such that $m^\gamma \cdot C_{4,\tau} > 2/\gamma - 2$.

We now define $m^*$ as the smallest integer such that

$$\int_{m^*}^\infty \exp(-C_{4,\tau} \cdot u^\gamma) \, \mathrm{d}u \leq \big[\epsilon/(2C_{3,\tau})\big]^2. \quad \text{(J.30)}$$

By (J.29), since both $C_{3,\tau}$, $C_{4,\tau}$ and $\gamma$ are absolute constants, there exist absolute constants $C_{3,m}$ and $C_{4,m}$ such that

$$m^* \leq C_{3,m} \cdot \big[\log(1/\epsilon) + C_{4,m}\big]^{1/\gamma}. \quad \text{(J.31)}$$

It is worth noting that the choice of $m^*$ in (J.31) is uniform over all $z \in \mathcal{Z}$. Moreover, by (J.28), for such an $m^*$, it holds that

$$\sup_{z \in \mathcal{Z}} \left| \|\phi(z)\|_\Upsilon - \big\|\Pi_{m^*}\big[\phi(z)\big]\big\|_\Upsilon \right| \leq \epsilon/2. \quad \text{(J.32)}$$

Thus, it remains to approximate $\|\Pi_{m^*}[\phi(z)]\|_\Upsilon$ up to accuracy $\epsilon/2$. Note that the subspace spanned by $\{\psi_j\}_{j \in [m^*]}$ is $m^*$-dimensional. When restricted to such a subspace, $\Upsilon$ can be expressed using a matrix $A_\Upsilon \in \mathbb{R}^{m^* \times m^*}$. Specifically, for any $j, k \in [m^*]$, we define the $(j,k)$-th entry of $A_\Upsilon$ as

$$[A_\Upsilon]_{j,k} = \big\langle \sqrt{\sigma_j} \cdot \psi_j, \sqrt{\sigma_k} \cdot \Upsilon\psi_k \big\rangle_{\mathcal{H}}. \quad \text{(J.33)}$$

Moreover, let $w_z \in \mathbb{R}^{m^*}$ be a vector whose $j$-th entry is given by $\sqrt{\sigma_j} \cdot \psi_j(z), \forall j \in [m^*]$. Then, by (J.33) it holds that

$$\big\|\Pi_{m^*}\big[\phi(z)\big]\big\|_\Upsilon^2 = \big\langle \Pi_{m^*}\big[\phi(z)\big], \Upsilon\Pi_{m^*}\big[\phi(z)\big] \big\rangle_{\mathcal{H}} = w_z^\top A_\Upsilon w_z. \quad \text{(J.34)}$$

Also, since $\|\Upsilon\|_{\mathrm{op}} \leq 1/\lambda$, the matrix operator norm of $A_\Upsilon$ is bounded by $1/\lambda$; i.e., $\|A_\Upsilon\|_{\mathrm{op}} \leq 1/\lambda$. This means that the Frobenius norm of $A_\Upsilon$ is bounded by $\sqrt{m^*}/\lambda$. Let $\mathcal{C}_{m^*}(\epsilon/2, \lambda)$ denote the minimal $\epsilon^2/4$-cover of $\{A \in \mathbb{R}^{m^* \times m^*} \colon \|A\|_{\mathrm{fro}} \leq \sqrt{m^*}/\lambda\}$ with respect to the Frobenius norm. By definition, there exists $\widetilde{A}_\Upsilon \in \mathcal{C}_{m^*}(\epsilon/2, \lambda)$ such that $\|A_\Upsilon - \widetilde{A}_\Upsilon\|_{\mathrm{fro}} \leq \epsilon^2/4$. Hence, we have

$$\big| w_z^\top A_\Upsilon w_z - w_z^\top \widetilde{A}_\Upsilon w_z \big| \leq \|w_z\|_2^2 \cdot \|A_\Upsilon - \widetilde{A}_\Upsilon\|_{\mathrm{op}} \leq \|A_\Upsilon - \widetilde{A}_\Upsilon\|_{\mathrm{fro}} \leq \epsilon^2/4. \quad \text{(J.35)}$$

Finally, for any $z \in \mathcal{Z}$, we define

$$f_\Upsilon(z) = w_z^\top \widetilde{A}_\Upsilon w_z = \sum_{j,k=1}^{m^*} \sqrt{\sigma_j \cdot \sigma_k} \cdot \psi_j(z) \cdot \psi_k(z) \cdot \big[\widetilde{A}_\Upsilon\big]_{jk}, \quad \text{(J.36)}$$

where $[\widetilde{A}_\Upsilon]_{jk}$ is the $(j,k)$-th entry of $\widetilde{A}_\Upsilon$ and $m^*$ is specified in (J.30). We remark that $f_\Upsilon\colon \mathcal{Z} \to \mathbb{R}$ is well defined since $m^*$ does not depend on $z$.

Finally, combining (J.32), (J.34), (J.35), and (J.36), we obtain
$$\big\| \|\phi(z)\|_\Upsilon - f_\Upsilon \big\|_\infty = \sup_{z \in \mathcal{Z}} \big| \|\phi(z)\|_\Upsilon - f_\Upsilon(z) \big|$$
$$\leq \sup_{z \in \mathcal{Z}} \Big| \|\phi(z)\|_\Upsilon - \big\| \Pi_{m^*}[\phi(z)] \big\|_\Upsilon \Big| + \sup_{z \in \mathcal{Z}} \Big| \big\| \Pi_{m^*}[\phi(z)] \big\|_\Upsilon - f_\Upsilon(z) \Big|$$
$$\leq \epsilon/2 + \sup_{z \in \mathcal{Z}} \Big| \sqrt{w_z^\top A_\Upsilon w_z} - \sqrt{w_z^\top \widetilde{A}_\Upsilon w_z} \Big| \leq \epsilon/2 + \sup_{z \in \mathcal{Z}} \sqrt{\big| w_z^\top A_\Upsilon w_z - w_z^\top \widetilde{A}_\Upsilon w_z \big|} \leq \epsilon.$$

This implies that $\{f_\Upsilon \colon \Upsilon \in \mathcal{C}_{m^*}(\epsilon, \lambda)\}$ forms an $\epsilon$-cover of $\mathcal{F}(\lambda)$ in (I.12). Hence, we have that
$$N_\infty(\epsilon, \mathcal{F}, \lambda) \leq \big| \mathcal{C}_{m^*}(\epsilon/2, \lambda) \big|. \tag{J.37}$$

Furthermore, using Corollary 4.2.13 in [69], we have
$$\big| \mathcal{C}_{m^*}(\epsilon/2, \lambda) \big| \leq \big[ 1 + 8\sqrt{m^*}/(\lambda \cdot \epsilon^2) \big]^{m^{*2}}. \tag{J.38}$$

Combining (J.31), (J.37), and (J.38), we finally have
$$\log N_\infty(\epsilon, \mathcal{F}, \lambda) \leq m^{*2} \cdot \log\big[ 1 + 8\sqrt{m^*}/(\lambda \cdot \epsilon^2) \big]$$
$$\leq C_{3,m}^2 \cdot \big[ \log(1/\epsilon) + C_{4,m} \big]^{2/\gamma} \cdot \log \Big\{ 1 + 8 C_{3,m}^{1/2} \cdot \big[ \log(1/\epsilon) + C_{4,m} \big]^{1/(2\gamma)}/(\lambda \cdot \epsilon^2) \Big\}$$
$$\leq C_5 \cdot [\log(1/\epsilon) + C_6]^{1+2/\gamma},$$

where $C_5$ and $C_6$ are absolute constants that depend on $C_\psi$, $C_1$, $C_2$, $\tau$, $\gamma$, and $\lambda$, but are independent of $T$, $H$, and $\epsilon$. Here in the last inequality we use the fact that $\log(1/\epsilon) \leq 1/\epsilon$, which holds when $\epsilon \leq 1/e$. Therefore, we conclude the proof for the second case and thus conclude the proof of the lemma. $\qquad\square$

## J.4 Technical Lemmas

Next, we present a few concentration inequalities. The first one provides concentration for standard self-normalized processes.

**Lemma J.1** (Concentration of Self-Normalized Processes in RKHS [18])**.** Let $\mathcal{H}$ be an RKHS defined over $\mathcal{X} \subseteq \mathbb{R}^d$ with kernel function $K(\cdot, \cdot)\colon \mathcal{X} \times \mathcal{X} \to \mathbb{R}$. Let $\{x_\tau\}_{\tau=1}^\infty \subseteq \mathcal{X}$ be a discrete time stochastic process that is adapted to the filtration $\{\mathcal{F}_t\}_{t=0}^\infty$. That is, $x_\tau$ is $\mathcal{F}_{\tau-1}$ measurable for all $\tau \geq 1$. Let $\{\epsilon_t\}_{\tau=1}^\infty$ be a real-valued stochastic process such that (i) $\epsilon_\tau \in \mathcal{F}_\tau$ and (ii) $\epsilon_\tau$ is zero-mean and $\sigma$-sub-Gaussian conditioning on $\mathcal{F}_{\tau-1}$:
$$\mathbb{E}[\epsilon_\tau | \mathcal{F}_{\tau-1}] = 0, \qquad \mathbb{E}[e^{\lambda \epsilon_\tau} | \mathcal{F}_{\tau-1}] \leq e^{\lambda^2 \sigma^2/2}, \qquad \forall \lambda \in \mathbb{R}.$$
Moreover, for any $t \geq 2$, let $E_t = (\epsilon_1, \ldots, \epsilon_{t-1})^\top \in \mathbb{R}^{t-1}$ and $K_t \in \mathbb{R}^{(t-1)\times(t-1)}$ be the Gram matrix of $\{x_\tau\}_{\tau \in [t-1]}$. Then, for any $\eta > 0$ and any $\delta \in (0,1)$, with probability at least $1 - \delta$, simultaneously for all $t \geq 1$, we have
$$E_t^\top \big[ (K_t + \eta \cdot I)^{-1} + I \big]^{-1} E_t \leq \sigma^2 \cdot \mathrm{logdet}\big[ (1+\eta) \cdot I + K_t \big] + 2\sigma^2 \cdot \log(1/\delta). \tag{J.39}$$
Moreover, if $K_t$ is positive definite for all $t \geq 2$ with probability one, then the inequality in (J.39) also holds with $\eta = 0$.

*Proof.* See Theorem 1 in [18] for a detailed proof. $\qquad\square$

**Lemma J.2** (Lemma D.4 of [35])**.** Let $\{x_\tau\}_{\tau=1}^\infty$ and $\{\phi_\tau\}_{\tau=1}^\infty$ be $\mathcal{S}$-valued and $\mathcal{H}$-valued stochastic processes adapted to filtration $\{\mathcal{F}_\tau\}_{\tau=0}^\infty$, respectively, where we assume that $\|\phi_\tau\|_\mathcal{H} \leq 1$ for all $\tau \geq 1$. Moreover, for any $t \geq 1$, we let $K_t \in \mathbb{R}^{t \times t}$ be the Gram matrix of $\{\phi_\tau\}_{\tau \in [t]}$ and define an operator $\Lambda_t\colon \mathcal{H} \to \mathcal{H}$ as $\Lambda_t = \lambda \cdot I_\mathcal{H} + \sum_{\tau=1}^t \phi_\tau \phi_\tau^\top$ with $\lambda > 1$. Let $\mathcal{V} \subseteq \{V\colon \mathcal{S} \to [0, H]\}$ be a class of bounded functions on $\mathcal{S}$. Then for any $\delta \in (0,1)$, with probability at least $1 - \delta$, we have

simultaneously for all $t \geq 1$ that

$$\sup_{V \in \mathcal{V}} \left\| \sum_{\tau=1}^{t} \phi_\tau \{ V(x_\tau) - \mathbb{E}[V(x_\tau)|\mathcal{F}_{\tau-1}] \} \right\|_{\Lambda_t^{-1}}^2 \tag{J.40}$$

$$\leq 2H^2 \cdot \mathrm{logdet}(I + K_t/\lambda) + 2H^2 t(\lambda - 1) + 4H^2 \log(\mathcal{N}_\epsilon/\delta) + 8t^2 \epsilon^2/\lambda,$$

where $\mathcal{N}_\epsilon$ is the $\epsilon$-covering number of $\mathcal{V}$ with respect to the distance $\mathrm{dist}(\cdot, \cdot)$.

*Proof.* Let $\mathcal{V}_\epsilon \subseteq \{ V \colon \mathcal{S} \to [0, H] \}$ be the minimal $\epsilon$-cover of $\mathcal{V}$ such that $N_\epsilon = |\mathcal{V}_\epsilon|$. Then for any $V \in \mathcal{V}$, there exists a value function $V' \colon \mathcal{S} \to \mathbb{R}$ in $\mathcal{N}_\epsilon$ such that $\mathrm{dist}(V, V') \leq \epsilon$. Let $\Delta_V = V - V'$. By the inequality $(a+b)^2 \leq 2a^2 + 2b^2$, we have

$$\left\| \sum_{\tau=1}^{t} \phi_\tau \{ V(x_\tau) - \mathbb{E}[V(x_\tau)|\mathcal{F}_{\tau-1}] \} \right\|_{\Lambda_t^{-1}}^2 \tag{J.41}$$

$$\leq 2 \cdot \left\| \sum_{\tau=1}^{t} \phi_\tau \{ V'(x_\tau) - \mathbb{E}[V'(x_\tau)|\mathcal{F}_{\tau-1}] \} \right\|_{\Lambda_t^{-1}}^2 + 2 \cdot \left\| \sum_{\tau=1}^{t} \phi_\tau \{ \Delta_V(x_\tau) - \mathbb{E}[\Delta_V(x_\tau)|\mathcal{F}_{\tau-1}] \} \right\|_{\Lambda_t^{-1}}^2.$$

To bound the first term on the right-hand side of (J.41), we apply Lemma J.1 to $V'$ and take a union bound over $V' \in \mathcal{V}_\epsilon$. While for the second term, since $\sup_{x \in \mathcal{S}} |\Delta_V(x)| \leq \epsilon$, we have

$$\left\| \sum_{\tau=1}^{t} \phi_\tau \{ \Delta_V(x_\tau) - \mathbb{E}[\Delta_V(x_\tau)|\mathcal{F}_{\tau-1}] \} \right\|_{\Lambda_t^{-1}}^2 \leq t^2 \cdot (2\epsilon)^2/\lambda = 4t^2 \epsilon^2/\lambda. \tag{J.42}$$

Thus, combining (J.41) and (J.42), we have

$$\sup_{V \in \mathcal{V}} \left\| \sum_{\tau=1}^{t} \phi_\tau \{ V(x_\tau) - \mathbb{E}[V(x_\tau)|\mathcal{F}_{\tau-1}] \} \right\|_{\Lambda_t^{-1}}^2$$

$$\leq \sup_{V' \in \mathcal{V}_\epsilon} 2 \cdot \left\| \sum_{\tau=1}^{t} \phi_\tau \{ V'(x_\tau) - \mathbb{E}[V'(x_\tau)|\mathcal{F}_{\tau-1}] \} \right\|_{\Lambda_t^{-1}}^2 + 8t^2 \epsilon^2/\lambda. \tag{J.43}$$

Now we fix $V' \in \mathcal{V}_\epsilon$ and define $\varepsilon_t \in \mathbb{R}^t$ by letting $[\varepsilon_t]_\tau = V'(x_\tau) - \mathbb{E}[V'(x_\tau)|\mathcal{F}_{\tau-1}]$ for any $\tau \geq 1$. We define an operator $\Phi \colon \mathcal{H} \to \mathbb{R}^t$ as $\Phi = [\phi_1^\top, \ldots, \phi_t^\top]^\top$ and let $K_t = \Phi_t \Phi_t^\top \in \mathbb{R}^{t \times t}$. Using this notation, we have $\Lambda_t = \lambda \cdot I_\mathcal{H} + \Phi_t^\top \Phi_t$ and

$$\left\| \sum_{\tau=1}^{t} \phi_\tau \{ V'(x_\tau) - \mathbb{E}[V'(x_\tau)|\mathcal{F}_{\tau-1}] \} \right\|_{\Lambda_t^{-1}}^2 = \|\Phi_t^\top \varepsilon_t\|_{\Lambda_t^{-1}}^2 = \varepsilon_t^\top \Phi_t \Lambda_t^{-1} \Phi_t^\top \varepsilon_t$$

$$= \varepsilon_t^\top \Phi_t \Phi_t^\top (K_t + \lambda \cdot I)^{-1} \varepsilon_t = \varepsilon_t^\top K_t (K_t + \lambda \cdot I)^{-1} \varepsilon_t, \tag{J.44}$$

where the third inequality follows from (H.14). Setting $\lambda = 1 + \eta$ for some $\eta > 0$, we have

$$(K_t + \eta \cdot I)\big[ K_t + (1 + \eta) \cdot I \big]^{-1} = (K_t + \eta \cdot I)\big[ I + (K_t + \eta \cdot I) \big]^{-1} = \big[ (K_t + \eta \cdot I)^{-1} + I \big]^{-1},$$

which implies that

$$\varepsilon_t^\top K_t (K_t + \lambda \cdot I)^{-1} \varepsilon_t \leq \varepsilon_t^\top (K_t + \eta \cdot I) \big[ I + (K_t + \eta \cdot I) \big]^{-1} \varepsilon_t$$

$$= \varepsilon_t^\top \big[ (K_t + \eta \cdot I)^{-1} + I \big]^{-1} \varepsilon_t. \tag{J.45}$$

Notice that each entry of $\varepsilon_t$ is bounded by $H$ in absolute value since $V'$ is bounded in $[0, H]$. By combining (J.43), (J.44), (J.45), Lemma J.1, and taking a union bound over $\mathcal{V}_\epsilon$, for any $\delta \in (0, 1)$, we obtain that, with probability at least $1 - \delta$,

$$\sup_{V' \in \mathcal{V}_\epsilon} \left\| \sum_{\tau=1}^{t} \phi_\tau \{ V'(x_\tau) - \mathbb{E}[V'(x_\tau)|\mathcal{F}_{\tau-1}] \} \right\|_{\Lambda_t^{-1}}^2$$

$$\leq H^2 \cdot \mathrm{logdet}[(1 + \eta) \cdot I + K_t] + 2H^2 \cdot \log(\mathcal{N}_\epsilon/\delta) \tag{J.46}$$

holds simultaneously for all $t \geq 1$. Moreover, notice that $(1 + \eta) \cdot I + K_t = [I + (1 + \eta)^{-1} \cdot K_t] \cdot [(1 + \eta) \cdot I]$, which implies that

$$\mathrm{logdet}\big[ (1 + \eta) \cdot I + K_t \big] = \mathrm{logdet}\big[ I + (1 + \eta)^{-1} \cdot K_t \big] + t \ln(1 + \eta)$$

$$\leq \mathrm{logdet}\big[ I + (1 + \eta)^{-1} \cdot K_t \big] + \eta t. \tag{J.47}$$

Finally, combining (J.43), (J.46), and (J.47), we conclude that, simultaneously for all $t \geq 1$, (J.40) holds with probability at least $1 - \delta$, which concludes the proof. $\qquad \square$

**Lemma J.3** ([1]). Let $\{\phi_t\}_{t \geq 1}$ be a sequence in the RKHS $\mathcal{H}$. Let $\Lambda_0 \colon \mathcal{H} \to \mathcal{H}$ be defined as $\lambda \cdot \mathcal{I}_{\mathcal{H}}$ where $\lambda \geq 1$ and $\mathcal{I}_{\mathcal{H}}$ is the identity mapping on $\mathcal{H}$. For any $t \geq 1$, we define a self-adjoint and positive-definite operator $\Lambda_t$ by letting $\Lambda_t = \Lambda_0 + \sum_{j=1}^{t} \phi_j \phi_j^\top$. Then, for any $t \geq 1$, we have

$$\sum_{j=1}^{t} \min\{1, \phi_j^\top \Lambda_{j-1}^{-1} \phi_j\} \leq 2\mathrm{logdet}(I + K_t/\lambda),$$

where $K_t \in \mathbb{R}^{t \times t}$ is the Gram matrix obtained from $\{\phi_j\}_{j \in [t]}$, i.e., for any $j, j' \in [t]$, the $(j, j')$-th entry of $K_t$ is $\langle \phi_j, \phi_{j'} \rangle_{\mathcal{H}}$. Moreover, if we further have $\sup_{t \geq 0}\{\|\phi_t\|_{\mathcal{H}}\} \leq 1$, then it holds that

$$\mathrm{logdet}(I + K_t/\lambda) \leq \sum_{j=1}^{t} \phi_j^\top \Lambda_{j-1}^{-1} \phi_j \leq 2\mathrm{logdet}(I + K_t/\lambda).$$

*Proof.* Note that we have $\log(1+x) \leq x \leq 2\log(1+x)$ for all $x \in [0,1]$. Since $\Lambda_t^{-1}$ is a self-adjoint and positive-definite operator, this implies that

$$\sum_{j=1}^{t} \min\{1, \phi_j^\top \Lambda_{j-1}^{-1} \phi_j\} \leq \sum_{j=1}^{t} 2\log\big(\min\{2, 1 + \phi_j^\top \Lambda_{j-1}^{-1} \phi_j\}\big) \leq 2\sum_{j=1}^{t} \log\big(1 + \phi_j^\top \Lambda_{j-1}^{-1} \phi_j\big).$$

(J.48)

Moreover, when additionally it is the case that $\sup_{j \geq 1} \|\phi_j\|_{\mathcal{H}} \leq 1$ for all $j \geq 0$, we have

$$\phi_j^\top \Lambda_{j-1}^{-1} \phi_j = \langle \phi_j, \Lambda_{j-1}^{-1}\phi_j \rangle_{\mathcal{H}} \leq \|\phi_j\|_{\mathcal{H}} \cdot \big\|\Lambda_{j-1}^{-1}\phi_j\big\|_{\mathcal{H}} \leq [\lambda_{\min}(\Lambda_0)]^{-1} \cdot \|\phi_j\|_{\mathcal{H}}^2 \leq 1. \quad \text{(J.49)}$$

Hence, applying the basic inequality $\log(1 + x) \leq x \leq 2\log(1 + x)$ to (J.49), we have

$$\sum_{j=1}^{t} \log\big(1 + \phi_j^\top \Lambda_{j-1}^{-1} \phi_j\big) \leq \sum_{j=1}^{t} \phi_j^\top \Lambda_{j-1}^{-1} \phi_j \leq 2\sum_{j=1}^{t} \log\big(1 + \phi_j^\top \Lambda_{j-1}^{-1} \phi_j\big). \quad \text{(J.50)}$$

For any $j \geq 1$, let $\Lambda_{j-1}^{1/2} \colon \mathcal{H} \to \mathcal{H}$ be the self-adjoint and positive-definite operator that is the square-root operator of $\Lambda_{j-1}$. Specifically, let $\{\sigma_\ell\}_{\ell \geq 1}$ be the eigenvalues of $\Lambda_{j-1}$ and let $\{v_\ell\}_{\ell \geq 1}$ be the corresponding eigenfunctions. Then $\Lambda_{j-1}^{1/2} = \sum_{\ell \geq 1} \sigma_\ell^{1/2} \cdot v_\ell v_\ell^\top$. Using this notation, for any $j \geq 1$, by the definition of $\Lambda_j$, we have

$$\Lambda_j = \Lambda_{j-1} + \phi_j \phi_j^\top = \Lambda_{j-1}^{1/2}\big(\mathcal{I}_{\mathcal{H}} + \Lambda_{j-1}^{-1/2} \phi_j \phi_j^\top \Lambda_{j-1}^{-1/2}\big)\Lambda_{j-1}^{1/2},$$

which implies that

$$\mathrm{logdet}(\Lambda_j) = \mathrm{logdet}(\Lambda_{j-1}) + \mathrm{logdet}\big(\mathcal{I}_{\mathcal{H}} + \Lambda_{j-1}^{-1/2} \phi_j \phi_j^\top \Lambda_{j-1}^{-1/2}\big)$$
$$= \mathrm{logdet}\big(\Lambda_{j-1}\big) + \mathrm{logdet}\big(1 + \phi_j^\top \Lambda_{j-1}^{-1} \phi_j\big) \quad \text{(J.51)}$$

Moreover, by direct computation, for any $t \geq 1$, we have

$$\det(\Lambda_t \Lambda_0^{-1}) = \det(I + K_t/\lambda). \quad \text{(J.52)}$$

Hence, combining (J.51), and (J.52), we obtain that

$$\sum_{j=1}^{t} \log\big(1 + \phi_j^\top \Lambda_{j-1}^{-1} \phi_j\big) = \mathrm{logdet}(\Lambda_t \Lambda_0^{-1}) = \mathrm{logdet}(I + K_t/\lambda). \quad \text{(J.53)}$$

Finally, combining (J.48), (J.50) and (J.53), we conclude the proof of this lemma. $\qquad \square$

[Supplementary Material 2]

# A    Related Work

Our work belongs to the vast literature on establishing provably efficient RL methods without having access to a generative model or a explorative behavioral policy. The tabular setting is well studied the existing works. See, e.g., [33, 52, 6, 21, 65, 35, 56] and the references therein. It is shown in [6, 35] that any RL algorithm necessarily incurs a $\Omega(\sqrt{SAT})$ regret under the tabular setting, where $S$ and $A$ are the cardinalities of the state and action spaces, respectively. Thus, the algorithms designed for the tabular setting cannot be directly applied to the function approximation setting where the number of states is gigantic. When function approximation is employed, [77, 78, 36, 12, 80, 73, 5, 83, 37] focus on the (generalized) linear setting where the value function (or the transition model) can be represented using a linear transform of a known feature mapping. Among these works, our work is most related to [36]. In particular, in our kernel setting, when kernel function has a finite rank, both our LSVI algorithm and the corresponding regret bound are reduced to the those established in [36]. However, their sample complexity or regret bounds all diverge when the dimension of the feature mapping goes to infinity and thus cannot be directly extended to the kernel setting. Another closely related work is [71], which studies a similar optimistic LSVI algorithm for general function approximation. Their work focuses on value function classes with bounded eluder dimensions [57, 51] and it is unclear whether their construction of the bonus function can be extended to the kernel or neural settings. Besides, [78] also study a kernelized MDP model where the transition model can be directly estimated. Under a slightly more general model, [5] recently propose an optimistic model-based algorithm via value-targeted regression, where the model class is allowed to be general functions with bounded eluder dimension. In another recent work, [37] study a nonlinear control problem where the system dynamics belongs to a known RKHS and can be directly estimated from the data. As opposed to these works, we do not pose an explicit assumption on the transition model and our proposed algorithm is model-free. Furthermore, regret or sample complexity results have also been studied beyond linear function approximation. However, these algorithms are either computational challenging [39, 34, 20, 22] or require additional assumptions on the transition model that might be restrictive [74, 75, 24]

In addition, our work is also related to the literature on contextual bandits with kernel or [62, 38, 63, 67, 18, 28] neural network functions [84], which are special cases of our episodic MDP with the episode length equal to one. The construction of our bonus function are adopted from these works. However, our reinforcement learning problem has temporal dependence caused by state transitions according to the Markov transition kernel, which is absent in bandit models. Specifically, the covering number $N_\infty(\epsilon^*)$ in Table 1 arises due to such an additional structure captures the fundamental challenge of temporally extended exploration in RL. When applying our algorithm to kernel contextual bandits, the regret bound reduces to $d_{\text{eff}} \cdot \sqrt{T}$ where $d_{\text{eff}}$ is the effective dimension of the RKHS. Such a regret bound matches those in [62, 18].

Furthermore, our analysis of the optimistic LSVI algorithm is akin to the recent study of the optimization and generalization of over-parameterized neural networks via the framework of the neural tangent kernel [32]. Most of these works focus on the supervised learning [19, 32, 76, 25, 26, 3, 2, 85, 17, 44, 4, 15, 16, 43]. In contrast, our algorithm incorporates an additional bonus term in the least-squares problem and thus requires novel analysis.

# B    Additional Background

In this section, we present the background of reproducing kernel Hilbert space and overparameterized neural networks.

## B.1    Reproducing Kernel Hilbert Space

In the next section, we aim to estimate the optimal value function $Q_h^\star$ using functions in a reproducing kernel Hilbert space (RKHS) [31]. To this end, hereafter, to simplify the notation, we let $z = (x, a)$ denote a state-action pair and denote $\mathcal{Z} = \mathcal{S} \times \mathcal{A}$. Without loss of generality, we regard $\mathcal{Z}$ as

a compact subset of $\mathbb{R}^d$ where the dimension $d$ is assumed fixed. This can be achieved if there exists a known embedding mapping $\psi_{\text{embed}}\colon \mathcal{Z} \to \mathbb{R}^d$ that pre-processes the input $(x, a)$. Let $\mathcal{H}$ be an RKHS defined on $\mathcal{Z}$ with kernel function $K\colon \mathcal{Z} \times \mathcal{Z} \to \mathbb{R}$, which contains a family of functions defined on $\mathcal{Z}$. Let $\langle \cdot, \cdot \rangle_{\mathcal{H}}\colon \mathcal{H} \times \mathcal{H} \to \mathbb{R}$ and $\| \cdot \|_{\mathcal{H}}\colon \mathcal{H} \to \mathbb{R}$ denote the inner product and RKHS norm on $\mathcal{H}$, respectively. Since $\mathcal{H}$ is an RKHS, there exists a feature mapping $\phi\colon \mathcal{Z} \to \mathcal{H}$ such that $f(z) = \langle f(\cdot), \phi(z) \rangle_{\mathcal{H}}$ for all $f \in \mathcal{H}$ and all $z \in \mathcal{Z}$. Moreover, for any $x, y \in \mathcal{Z}$, we have $K(x, y) = \langle \phi(x), \phi(y) \rangle_{\mathcal{H}}$. In this work, we assume that the kernel function $K$ is uniformly bounded in the sense that $\sup_{z \in \mathcal{Z}} K(z, z) < \infty$. Without loss of generality, we assume that $\sup_{z \in \mathcal{Z}} K(z, z) \le 1$, which implies that $\|\phi(z)\|_{\mathcal{H}} \le 1$ for all $z \in \mathcal{Z}$.

Furthermore, let $\mathcal{L}^2(\mathcal{Z})$ be the space of square-integrable functions on $\mathcal{Z}$ with respect to the Lebesgue measure and let $\langle \cdot, \cdot \rangle_{\mathcal{L}^2}$ be the inner product on $\mathcal{L}^2(\mathcal{Z})$. The kernel function $K$ induces a integral operator $T_K\colon \mathcal{L}^2(\mathcal{Z}) \to \mathcal{L}^2(\mathcal{Z})$ defined as

$$T_K f(z) = \int_{\mathcal{Z}} K(z, z') \cdot f(z') \, \mathrm{d}z', \qquad \forall f \in \mathcal{L}^2(\mathcal{Z}). \tag{B.1}$$

By Mercer's Theorem [64], the integral operator $T_K$ has countable and positive eigenvalues $\{\sigma_i\}_{i \ge 1}$ and the corresponding eigenfunctions $\{\psi_i\}_{i \ge 1}$ form an orthonormal basis of $\mathcal{L}^2(\mathcal{Z})$. Moreover, the kernel function admits a spectral expansion

$$K(z, z') = \sum_{i=1}^{\infty} \sigma_i \cdot \psi_i(z) \cdot \psi_j(z'). \tag{B.2}$$

Then, the RKHS $\mathcal{H}$ can be written as a subset of $\mathcal{L}^2(\mathcal{Z})$ as

$$\mathcal{H} = \left\{ f \in \mathcal{L}^2(\mathcal{Z})\colon \sum_{i=1}^{\infty} \frac{\langle f, \psi_i \rangle_{\mathcal{L}^2}^2}{\sigma_i} < \infty \right\},$$

and the inner product of $\mathcal{H}$ can be written as

$$\langle f, g \rangle_{\mathcal{H}} = \sum_{i=1}^{\infty} 1/\sigma_i \cdot \langle f, \psi_i \rangle_{\mathcal{L}^2} \cdot \langle g, \psi_i \rangle_{\mathcal{L}^2}, \qquad \text{for all} \quad f, g \in \mathcal{H}.$$

By such a construction, the scaled eigenfunctions $\{\sqrt{\sigma_i}\psi_i\}_{i \ge 1}$ form an orthogonal basis of RKHS $\mathcal{H}$ and the feature mapping $\phi(z) \in \mathcal{H}$ can be written as $\phi(z) = \sum_{i=1}^{\infty} \sigma_i \psi_i(z) \cdot \psi_i$ for any $z \in \mathcal{Z}$.

## B.2 Overparameterized Neural Networks

In addition to RKHS, we also study the setting where the value functions are approximated by overparameterized neural networks. In the sequel, we define the class of neural networks that will be used in the algorithm.

Recall that we denote $\mathcal{Z} = \mathcal{S} \times \mathcal{A}$ and view it as a subset of $\mathbb{R}^d$. For neural networks, we further regard $\mathcal{Z}$ as a subset of the unit sphere in $\mathbb{R}^d$. That is, $\|z\|_2 = 1$ for all $z = (x, a) \in \mathcal{Z}$. A two-layer neural network $f(\cdot; b, W)\colon \mathcal{Z} \to \mathbb{R}$ with $2m$ neurons and weights $(b, W)$ is defined as

$$f(z; b, W) = \frac{1}{\sqrt{2m}} \sum_{j=1}^{2m} b_j \cdot \mathrm{act}(W_j^\top z), \qquad \forall z \in \mathcal{Z}. \tag{B.3}$$

Here $\mathrm{act}\colon \mathbb{R} \to \mathbb{R}$ is the activation function, $b_j \in \mathbb{R}$ and $W_j \in \mathbb{R}^d$ for all $j \in [2m]$, and $b = (b_1, \ldots, b_{2m})^\top \in \mathbb{R}^{2m}$ and $W = (W_1, \ldots, W_{2m}) \in \mathbb{R}^{2dm}$. During training, we initialize $(b, W)$ via the symmetric initialization scheme [30, 9] as follows. For any $j \in [m]$, we set $b_j \overset{\text{i.i.d.}}{\sim} \mathrm{Unif}(\{-1, 1\})$ and $W_j \overset{\text{i.i.d.}}{\sim} N(0, I_d/d)$, where $I_d$ is the identity matrix in $\mathbb{R}^d$. For any $j \in \{m+1, \ldots, 2m\}$, we set $b_j = -b_{j-m}$ and $W_j = W_{j-m}$. We remark that such an initialization implies that the initial neural network is a zero function, which is used only to simply the theoretical analysis. Besides, for ease of presentation, during training we fix $b$ at its initial value and only optimize over $W$. Moreover, we denote $f(z; b, W)$ by $f(z; W)$ to simplify the notation.

Furthermore, we assume that the neural network in is overparameterized in the sense that the width $2m$ is much larger than the number of episodes $T$. Overparameterization is shown to be pivotal for

neural training in both theory and practice [49, 2, 4]. Under the such a regime, the dynamics of training neural networks are well captured by the framework of neural tangent kernel (NTK) [32]. Specifically, let $\varphi(\cdot; W) \colon \mathcal{Z} \to \mathbb{R}^{2md}$ be the gradient of $f(; W)$ with respect to $W$, which is given by

$$\varphi(z; W) = \nabla_W f(z; W) = \big(\nabla_{W_1} f(z; W), \ldots, \nabla_{W_{2m}} f(z; W)\big), \qquad \forall z \in \mathcal{Z}. \qquad \text{(B.4)}$$

Let $W^{(0)}$ be the initial value of $W$. Condition on the realization of $W^{(0)}$, we define a kernel matrix $K_m \colon \mathcal{Z} \to \mathcal{Z}$ as

$$K_m(z, z') = \big\langle \varphi(z; W^{(0)}), \varphi(z'; W^{(0)}) \big\rangle, \qquad \forall (z, z') \in \mathcal{Z} \times \mathcal{Z}. \qquad \text{(B.5)}$$

When $m$ is sufficiently large, for all $W$ that is in a neighborhood of $W^{(0)}$, it can be shown that $f(\cdot, W)$ is close to its linearization at $W^{(0)}$,

$$f(\cdot; W) \approx \widehat{f}(\cdot; W) = f(\cdot, W^{(0)}) + \big\langle \phi(\cdot; W^{(0)}), W - W^{(0)} \big\rangle = \big\langle \phi(\cdot; W^{(0)}), W - W^{(0)} \big\rangle. \qquad \text{(B.6)}$$

The linearized function $\widehat{f}(\cdot; W)$ belongs to the RKHS with kernel $K_m$. Moreover, as $m$ goes to infinity, due to random initialization, $K_m$ converges to a kernel $K_{\mathrm{ntk}} \colon \mathcal{Z} \times \mathcal{Z}$, dubbed as neural tangent kernel (NTK), which is given by

$$K_{\mathrm{ntk}}(z, z') = \mathbb{E}\big[\mathrm{act}'(w^\top z) \cdot \mathrm{act}'(w^\top z') \cdot z^\top z'\big], \qquad (z, z') \in \mathcal{Z} \times \mathcal{Z}, \qquad \text{(B.7)}$$

where $\mathrm{act}'$ is the derivative of the activation function, and the expectation in (B.7) is taken with respect to $w \sim N(0, I_d/d)$.

## C  Kernel and Neural Optimistic Least-Squares Value Iteration

In this section, we lay out the details of KOVI and NOVI, which are omitted for brevity. We remark that the loss function $L_h^t$ in Line 7 of Algorithm 4 is given in (C.1) and its global minimizer $\widehat{W}_h^t$ can be efficiently obtained by first-order optimization methods.

---

**Algorithm 2** Kernelized Optimistic Least-Squares Value Iteration (KOVI)

---

1: **Input:** Parameters $\lambda$ and $\beta$.
2: **for** episode $t = 1, \ldots, T$ **do**
3:     Receive the initial state $x_1^t$.
4:     Set $V_{H+1}^t$ as the zero function.
5:     **for** step $h = H, \ldots, 1$ **do**
6:         Compute the response $y_h^t \in \mathbb{R}^{t-1}$, the Gram matrix $K_h^t \in \mathbb{R}^{(t-1) \times (t-1)}$, and function $k_h^t$ as in (3.6) and (3.7), respectively.
7:         Compute
8:             $\alpha_h^t = (K_h^t + \lambda \cdot I)^{-1} y_h^t,$
9:             $b_h^t(\cdot, \cdot) = \lambda^{-1/2} \cdot \big[K(\cdot, \cdot; \cdot, \cdot) - k_h^t(\cdot, \cdot)^\top (K_h^t + \lambda I)^{-1} k_h^t(\cdot, \cdot)\big]^{1/2}.$
10:         Obtain value functions

$$Q_h^t(\cdot, \cdot) \leftarrow \min\{k_h^t(\cdot, \cdot)^\top \alpha_h^t + \beta \cdot b_h^t(\cdot, \cdot), H - h + 1\}^+, \qquad V_h^t(\cdot) = \max_a Q_h^t(\cdot, a).$$

11:     **end for**
12:     **for** step $h = 1, \ldots, H$ **do**
13:         Take action $a_h^t \leftarrow \mathrm{argmax}_{a \in \mathcal{A}} Q_h^t(x_h^t, a)$.
14:         Observe the reward $r_h(x_h^t, a_h^t)$ and the next state $x_{h+1}^t$.
15:     **end for**
16: **end for**

---

### C.1  Neural Optimistic Value Iteration

In this subsection, we estimate the value functions $\{Q_h^\star\}_{h \in [H]}$ using overparameterized neural networks. We aim to estimate each $Q_h^\star$ using a neural network given in (B.3), which is initialized via the symmetric initialization scheme [30, 9] introduced in §B.2. Moreover, for simplicity, we assume

---

**Algorithm 3** Neural Optimistic Least-Squares Value Iteration (NOVI)

---

1: **Input:** Parameters $\lambda$ and $\beta$.
2: Initialize the network weights $(b^{(0)}, W^{(0)})$ via the symmetric initialization scheme.
3: **for** episode $t = 1, \ldots, T$ **do**
4:     Receive the initial state $x_1^t$.
5:     Set $V_{H+1}^t$ as the zero function.
6:     **for** step $h = H, \ldots, 1$ **do**
7:        Solve the neural network optimization problem $\widehat{W}_h^t = \operatorname{argmin}_W L_h^t(W)$.
8:        Update $\Lambda_h^t = \Lambda_h^{t-1} + \varphi(x_h^{t-1}, a_h^{t-1}; \widehat{W}_h^t)\varphi(x_h^{t-1}, a_h^{t-1}; \widehat{W}_h^t)^\top$.
9:        Obtain the bonus function $b_h^t$ defined in (C.4).
10:       Obtain value functions

$$Q_h^t(\cdot, \cdot) \leftarrow \min\big\{ f\big(\cdot, \cdot; \widehat{W}_h^t\big) + \beta \cdot b_h^t(\cdot, \cdot), H - h + 1 \big\}^+, \qquad V_h^t(\cdot) = \max_a Q_h^t(\cdot, a).$$

11:     **end for**
12:     **for** step $h = 1, \ldots, H$ **do**
13:        Take action $a_h^t \leftarrow \operatorname{argmax}_{a \in \mathcal{A}} Q_h^t(x_h^t, a)$.
14:        Observe the reward $r_h(x_h^t, a_h^t)$ and the next state $x_{h+1}^t$.
15:     **end for**
16: **end for**

---

that all the neural networks share the same initial weights, denoted by $(b^{(0)}, W^{(0)})$. Besides, we fix $b = b^{(0)}$ in (B.3) and only update the value of $W \in \mathbb{R}^{2md}$.

Under such a neural setting, we replace the least-squares regression in (3.2) by a nonlinear ridge regression. In particular, for any $(t, h) \in [T] \times [H]$, we define the loss function $L_h^t : \mathbb{R}^{2md} \to \mathbb{R}$ as

$$L_h^t(W) = \sum_{\tau=1}^{t-1} \big[ r_h(x_h^\tau, a_h^\tau) + V_{h+1}^t(x_{h+1}^\tau) - f(x_h^\tau, a_h^\tau; W) \big]^2 + \lambda \cdot \big\| W - W^{(0)} \big\|_2^2, \qquad \text{(C.1)}$$

where $\lambda > 0$ is the regularization parameter. Then we define $\widehat{Q}_h^t$ as

$$\widehat{Q}_h^t(\cdot, \cdot) = f\big(\cdot, \cdot; \widehat{W}_h^t\big), \qquad \text{where} \qquad \widehat{W}_h^t = \operatorname*{argmin}_{W \in \mathbb{R}^{2md}} L_h^t(W). \qquad \text{(C.2)}$$

Here we assume that there is an optimization oracle that returns the global minimizer of the loss function $L_h^t$. It has been shown in a large body of literature that, when $m$ is sufficiently large, with random initialization, simple optimization methods such as gradient descent provably find the global minimizer of the empirical loss function at a linear rate of convergence [26, 25, 4]. Thus, such an optimization oracle can be realized by gradient descent with sufficiently large number of iterations and the computational cost of realizing such a oracle is polynomial in $m$, $T$, and $H$.

It remains to construct the bonus function $b_h^t$. Recall that we define $\varphi(\cdot; W) = \nabla_W f(\cdot; W)$ in (B.4). We define matrix $\Lambda_h^t \in \mathbb{R}^{2md \times 2md}$ as

$$\Lambda_h^t = \lambda \cdot I_{2md} + \sum_{\tau=1}^{t-1} \varphi\big(x_h^\tau, a_h^\tau; \widehat{W}_h^{\tau+1}\big) \varphi\big(x_h^\tau, a_h^\tau; \widehat{W}_h^{\tau+1}\big)^\top, \qquad \text{(C.3)}$$

which can be recursively computed by letting

$$\Lambda_h^1 = \lambda \cdot I_{2md}, \qquad \Lambda_h^t = \Lambda_h^{t-1} + \varphi\big(x_h^{t-1}, a_h^{t-1}; \widehat{W}_h^t\big) \varphi\big(x_h^{t-1}, a_h^{t-1}; \widehat{W}_h^t\big)^\top, \qquad \forall t \geq 2.$$

Then the bonus function $b_h^t$ is defined as

$$b_h^t(x, a) = \big[ \varphi\big(x, a; \widehat{W}_h^t\big)^\top (\Lambda_h^t)^{-1} \varphi\big(x, a; \widehat{W}_h^t\big) \big]^{1/2}, \qquad \forall (x, a) \in \mathcal{S} \times \mathcal{A}. \qquad \text{(C.4)}$$

Finally, we obtain the value functions $Q_h^t$ and $V_h^t$ via (3.5), with $\widehat{Q}_h^t$ and $b_h^t$ defined in (C.2) and (C.4), respectively. By letting $\pi^t$ be the greedy policy with respect to $\{Q_h^t\}_{h \in [H]}$, we obtain the Neural Optimistic Least-Squares Value Iteration (NOVI) algorithm, whose details are stated in Algorithm 4 in §F.

The intuition of the bonus term in (C.4) can be understood via the connection between overparameterized neural networks and NTK. Specifically, when $m$ is sufficiently large, it can be shown that each $\widehat{W}_h^t$ is not far from the initial value $W^{(0)}$. When this is the case, suppose we replace the neural tangent features $\{\varphi(\cdot; \widehat{W}_h^\tau)\}_{\tau \in [T]}$ in (C.3) and (C.4) by $\varphi(\cdot; W^{(0)})$, then $b_h^t$ recovers the UCB bonus in linear contextual bandits and linear MDPs with feature mapping $\varphi(\cdot; W^{(0)})$ [1, 36, 73]. Moreover, when $m$ converges to infinity, it will become the UCB bonus defined in (3.8) for the RKHS setting with the kernel being $K_{\mathrm{ntk}}$. Thus, when the neural networks are overparameterized, value functions $\{Q_h^t\}_{h \in [H]}$ are approximately elementwise upper bounds of the optimal value functions and thus we achieve optimism approximately.

# D    Theory of Neural Optimistic Least-Squares Value Iteration

In this section, we establish the regret of NOVI. Throughout this subsection, we let $\mathcal{H}$ be the RKHS whose kernel function is $K_{\mathrm{ntk}}$ define in (B.7). Also recall that we regard $\mathcal{Z} = \mathcal{S} \times \mathcal{A}$ as a subset of the unit sphere $\mathbb{S}^{d-1} = \{z \in \mathbb{R}^d \colon \|z\|_2 = 1\}$. Moreover, let $(b^{(0)}, W^{(0)})$ be the initial value of the network weights obtained via the symmetric initialization scheme introduced in §B.2. Conditioning on the randomness of the initialization, we define a finite-rank kernel $K_m \colon \mathcal{Z} \times \mathcal{Z} \to \mathbb{R}$ by letting $K_m(z, z') = \langle \nabla_W f(z; b^{(0)}, W^{(0)}), \nabla_W f(z'; b^{(0)}, W^{(0)}) \rangle$. Notice that the rank of $K_m$ is $md$, where $m$ is much larger than $T$ and $H$ and is allowed to increase to infinity. Besides, with a slight abuse of notation, we define

$$\mathcal{Q}^\star = \left\{ f_\alpha(z) = \int_{\mathbb{R}^d} \mathrm{act}'(w^\top z) \cdot z^\top \alpha(w) \, \mathrm{d}p_0(w) \colon \alpha \colon \mathbb{R}^d \to \mathbb{R}^d, \|\alpha\|_{2,\infty} \leq R_Q H/\sqrt{d} \right\}, \quad \text{(D.1)}$$

where $R_Q$ is a positive number, $p_0$ is the density of $N(0, I_d/d)$, and we define $\|\alpha\|_{2,\infty} = \sup_w \|\alpha(w)\|_2$. That is, $\mathcal{Q}^\star$ consists of functions that can be expressed as infinite number of random features. As shown in Lemma C.1 of [30], $\mathcal{Q}^\star$ is a dense subset of the RKHS $\mathcal{H}$. Thus, when $R_Q$ is sufficiently large, $\mathcal{Q}^\star$ in (D.1) is an expressive function class. We impose the following condition on $\mathcal{Q}^\star$.

**Assumption D.1.** We assume that for any $h \in [H]$ and any $Q \colon \mathcal{S} \times \mathcal{A} \to [0, H]$, we have $\mathbb{T}_h^\star Q \in \mathcal{Q}^\star$.

Assumption D.1 is in the same vein as Assumption 4.1. Here we focus on $\mathcal{Q}^*$ instead of an RKHS norm ball of NTK only due to technical considerations. However, since functions of the form in (D.1) are dense in $\mathcal{H}$, Assumptions D.1 and 4.1 are indeed very similar.

To characterize the value function class associated with NOVI, for any discrete set $\mathcal{D} \subseteq \mathcal{Z}$, similar to (C.3), we define

$$\overline{\Lambda}_{\mathcal{D}} = \lambda \cdot I_{2md} + \sum_{z \in \mathcal{D}} \varphi(z; W^{(0)}) \varphi(z; W^{(0)})^\top,$$

where $\varphi(\cdot; W^{(0)})$ is the neural tangent feature defined in (B.4). With a slight abuse of notation, for any $R, B > 0$, we let $\mathcal{Q}_{\mathrm{ucb}}(h, R, B)$ denote that class of functions that take the form of

$$Q(z) = \min\left\{ \langle \varphi(z; W^{(0)}), W \rangle + \beta \cdot [\varphi(z; W^{(0)})^\top (\overline{\Lambda}_{\mathcal{D}})^{-1} \varphi(z; W^{(0)})]^{1/2}, H - h + 1 \right\}^+, \quad \text{(D.2)}$$

where $W \in \mathbb{R}^{2md}$ satisfies $\|W\|_2 \leq R$, $\beta \in [0, B]$, and $\mathcal{D}$ has cardinality no more than $T$. Intuitively, when both $R$ and $B$ are sufficiently large, $\mathcal{Q}_{\mathrm{ucb}}(h, R, B)$ contains the counterpart of neural-based value function $Q_h^t$ that is based on neural tangent features. When $m$ is sufficiently large, it is expected that $Q_h^t$ is well-approximately by functions in $\mathcal{Q}_{\mathrm{ucb}}(h, R, B)$ where the approximation error decays with $m$. It is worth noting the class of linear functions of $\varphi(\cdot; W^{(0)})$ forms an RKHS with kernel $K_m$ in (B.5). Any function $f$ in this class can be written as $f(\cdot) = \langle \varphi(\cdot; W^{(0)}), W_f \rangle$ for some $W_f \in \mathbb{R}^{2md}$. Moreover, the RKHS norm of $f$ is given by $\|W_f\|_2$. Thus, $\mathcal{Q}_{\mathrm{ucb}}(h, R, B)$ defined above coincides with the counterpart defined in (4.4) with the kernel function being $K_m$. We set $R_T = H\sqrt{2T/\lambda}$ and let $N_\infty(\epsilon; h, B)$ denote the $\epsilon$-covering number of $\mathcal{Q}_{\mathrm{ucb}}(h, R_T, B)$ with respect to the $\ell_\infty$-norm on $\mathcal{Z}$.

In the following theorem, we present a general regret bound for NOVI.

**Theorem D.2.** Under Assumptions D.1, We also assume that $m$ is sufficiently large such that $m = \Omega(T^{13}H^{14} \cdot (\log m)^3)$. In Algorithm 4, we let $\lambda$ be a sufficiently large constant and let $\beta = B_T$ which satisfies inequality

$$16\Gamma_{K_m}(T, \lambda) + 16 \cdot \log N_\infty(\epsilon^*, h+1, B_T) + 32 \cdot \log(2TH) + 4R_Q^2 \cdot (1 + \lambda/d) \le (B_T/H)^2 \tag{D.3}$$

for all $h \in [H]$. Here $\epsilon^* = H/T$ and $\Gamma_{K_m}(T, \lambda)$ is the maximal information gain defined for kernel $K_m$. In addition, for the neural network in (B.3), we assume the activation function act is $C_{\text{act}}$-smooth, i.e., its derivative $\text{act}'$ is $C_{\text{act}}$-Lipschitz, and $m$ is sufficiently large such that

$$m = \Omega\big(\beta^{12} \cdot T^{13} \cdot H^{14} \cdot (\log m)^3\big). \tag{D.4}$$

Then with probability at least $1 - (T^2 H^2)^{-1}$, we have

$$\text{Regret}(T) = 5\beta H \cdot \sqrt{T \cdot \Gamma_{K_m}(T, \lambda)} + 10\beta T H \cdot \iota, \tag{D.5}$$

where we define $\iota = T^{7/12} \cdot H^{1/6} \cdot m^{-1/12} \cdot (\log m)^{1/4}$.

This theorem shows that, when $m$ is sufficiently large, NOVI enjoys a similar regret bound as KOVI. Specifically, the choice of $\beta$ in (D.3) is similar to that in (4.5) for kernel $K_m$. Here we set $\lambda$ to be an absolute constant as $\sup_z K_m(z, z) \le 1$ no longer holds. In addition, here we assume that $\text{act}'$ is $C_{\text{act}}$-Lipschitz on $\mathbb{R}$, which can be relaxed to only assuming $\text{act}'$ is Lipschitz continous on a bounded interval of $\mathbb{R}$ that contains $w^\top z$ with high probability, where $w$ is drawn from the initial distribution of $W_j$, $j \in [m]$.

Moreover, comparing (D.6) and (D.5) we observe that, when $m$ is sufficiently large, NOVI can be viewed as a misspecified version of KOVI for the RKHS with kernel $K_m$, where the model misspecification error is $\text{err}_{\text{mis}} = 10\beta \cdot \iota$. Specifically, the first term in (D.5) is the same as that in (D.6), where the choice of $\beta$ and $\Gamma_{K_m}(T, \lambda)$ reflect the intrinsic complexity of $K_m$. Whereas the second term is equal to $\text{err}_{\text{mis}} \cdot TH$, which arises due to approximating neural network value functions by functions in $\mathcal{Q}_{\text{ucb}}(h, R_T, B_T)$, which are constructed using kernel functions with feature mapping $\varphi(\cdot; W^{(0)})$. Moreover, when $\beta$ is bounded by a polynomial of $TH$, to make $\text{err}_{\text{mis}} \cdot TH$ negligible, it suffices to let $m$ be a polynomial of $TH$. That is, when the network width is a polynomial of the total number of steps, NOVI achieves the same performance as KOVI.

Furthermore, when neglecting the constants and logarithmic terms in (D.3), we simplify the regret bound in (D.5) into

$$\text{Regret}(T) = \mathcal{O}\Big(H^2 \cdot \Big[\Gamma_{K_m}(T, \lambda) + \max_{h \in [H]} \sqrt{\Gamma_{K_m}(T, \lambda) \cdot \log N_\infty(\epsilon^*, h, B_T)}\Big] \cdot \sqrt{T} + \text{err}_{\text{mis}} \cdot T\Big).$$

which depends on the intrinsic complexity of $K_m$ through both the effective dimension $\Gamma_{K_m}(T, \lambda)$ and the log-covering number $\log N_\infty(\epsilon^*, h, B_T)$. To obtain a more concrete regret bounds, in the following, we pose an assumption on the spectral structure of $K_m$.

**Assumption D.3** (Eigenvalue Decay of the Empirical NTK). Conditioning on the randomness of $(b^{(0)}, W^{(0)})$, let $K_m$ be the kernel induced by the neural tangent features $\nabla f(\cdot; b^{(0)}, W^{(0)})$. Let $T_{K_m}$ be the integral operator induced by $K_m$ and the Lebesgue measure on $\mathcal{Z}$ and let $\{\sigma_j\}_{j \ge 1}$ and $\{\psi_j\}_{j \ge 1}$ be its eigenvalues and eigenvectors, respectively. We assume that $\{\sigma_j\}_{j \ge 1}$ and $\{\psi_j\}_{j \ge 1}$ satisfy either one of the two decay conditions specified in Assumption 4.3. Here we assume the constants $C_1, C_2, C_\psi, \gamma$, and $\tau$ do not depend on $m$.

Here we assume that $K_m$ satisfies Assumption 4.3. Since $K_m$ depends on the initial network weights, which are random, this assumption should be better understood in the limit sense. Specifically, as $m$ goes to infinity, $K_m$ converges to $K_{\text{ntk}}$, which is determined by both the activation function and the distribution of the initial network weights. Thus, if the RKHS with kernel $K_{\text{ntk}}$ satisfy Assumption 4.3, when $m$ is sufficiently large, it is reasonable to expect that such a condition also holds for $K_m$. Due to the space limit, we present concrete examples of $K_{\text{ntk}}$ satisfying Assumption 4.3 in §G.3 in the appendix.

Now we are ready to characterize the performances of NOVI for each case separately.

**Corollary D.4.** Under Assumptions D.1 and D.3, we assume the activation function is $C_{\mathrm{act}}$-smooth and the number of neurons of the neural network satisfies (D.4). Besides, in Algorithm 4 we let $\lambda$ be a sufficiently large constant and set $\beta = B_T$ as in (4.8). Then exists an absolute constant $C_r$ such that, with probability at least $1 - (T^2 H^2)^{-1}$, we have

$$\mathrm{Regret}(T) \leq \begin{cases} C_r \cdot H^2 \cdot \sqrt{\gamma^3 T} \cdot \log(\gamma T H) + 10\beta T H \cdot \iota & \gamma\text{-finite spectrum,} \\ C_r \cdot H^2 \cdot \sqrt{(\log T)^{3/\gamma} \cdot T} \cdot \log(T H) + 10\beta T H \cdot \iota & \gamma\text{-exponential decay,} \end{cases} \tag{D.6}$$

where we define $\iota = T^{7/12} \cdot H^{1/6} \cdot m^{-1/12} \cdot (\log m)^{1/4}$.

Corollary D.4 is parallel to Corollary 4.4, with an additional misspecification error $10\beta T H \cdot \iota$. It remains to see whether there exist concrete neural networks that induce NTKs satisfying each eigenvalue decay condition. As we will show in §G.3, neural network with quadratic and sine activation functions induce NTKs satisfying the finite-spectrum and exponential eigenvalue decay conditions, respectively. Corollary D.4 can be directly applied to these concrete examples to obtain sublinear regret bounds.

# E   Proofs of the Main Results

In this section, we provide the proofs of Theorems 4.2 and D.2. The proofs of the supporting lemmas and auxiliary results are deferred to the appendix.

## E.1   Proof of Theorem 4.2

*Proof.* For simplicity of presentation, we define the temporal-difference (TD) error as

$$\delta_h^t(x, a) = r_h(x, a) + (\mathbb{P}_h V_{h+1}^t)(x, a) - Q_h^t(x, a), \qquad \forall (x, a) \in \mathcal{S} \times \mathcal{A}. \tag{E.1}$$

Here $\delta_h^t$ is a function on $\mathcal{S} \times \mathcal{A}$ for all $h \in [H]$ and $t \in [T]$. Note that $V_h^t(\cdot) = \max_{a \in \mathcal{A}} Q_h^t(\cdot, a)$. Intuitively, $\{\delta_h^t\}_{h \in [H]}$ quantifies the how far the $\{Q_h^t\}_{h \in [H]}$ are from satisfying the Bellman optimality equation in (2.2). Next, recall that $\pi^t$ is the policy executed in the $t$-th episode, which generates a trajectory $\{(x_h^t, a_h^t)\}_{h \in [H]}$. For any $h \in [H]$ and $t \in [T]$, we further define $\zeta_{t,h}^1, \zeta_{t,h}^2 \in \mathbb{R}$ as

$$\zeta_{t,h}^1 = \left[ V_h^t(x_h^t) - V_h^{\pi^t}(x_h^t) \right] - \left[ Q_h^t(x_h^t, a_h^t) - Q_h^{\pi^t}(x_h^t, a_h^t) \right], \tag{E.2}$$

$$\zeta_{t,h}^2 = \left[ (\mathbb{P}_h V_{h+1}^t)(x_h^t, a_h^t) - (\mathbb{P}_h V_{h+1}^{\pi^t})(x_h^t, a_h^t) \right] - \left[ V_{h+1}^t(x_{h+1}^t) - V_{h+1}^{\pi^t}(x_{h+1}^t) \right]. \tag{E.3}$$

By definition, $\zeta_{t,h}^1$ and $\zeta_{t,h}^2$ capture two sources of randomness—the randomness of choosing an action $a_h^t \sim \pi_h^t(\cdot \,|\, x_h^t)$ and that of drawing the next state $x_{h+1}^t$ from $\mathbb{P}_h(\cdot \,|\, x_h^t, a_h^t)$, respectively. As we will see in Appendix §H.3, $\{\zeta_{t,h}^1, \zeta_{t,h}^2\}$ form a bounded martingale difference sequence with respect to a properly chosen filtration, which enables us to bound their total sum via the Azuma-Hoeffding inequality [7].

To establish an upper bound on the regret, the following lemma first decomposes the regret into three parts using the notation defined above. Similar regret decomposition results also appear in [12, 29].

**Lemma E.1** (Regret Decomposition). The temporal-difference error is the mapping $\delta_h^t : \mathcal{S} \times \mathcal{A} \to$ defined in (E.1) for all $(t, h) \in [T] \times [H]$. We can thus write the regret as

$$\mathrm{Regret}(T) = \underbrace{\sum_{t=1}^{T} \sum_{h=1}^{H} \left[ \mathbb{E}_{\pi^\star}[\delta_h^t(x_h, a_h) \,|\, x_1 = x_1^t] - \delta_h^t(x_h^t, a_h^t) \right]}_{(i)} + \underbrace{\sum_{t=1}^{T} \sum_{h=1}^{H} (\zeta_{t,h}^1 + \zeta_{t,h}^2)}_{(ii)}$$

$$\underbrace{\sum_{t=1}^{T} \sum_{h=1}^{H} \mathbb{E}_{\pi^\star} \left[ \langle Q_h^t(x_h, \cdot), \pi_h^\star(\cdot \,|\, x_h) - \pi_h^t(\cdot \,|\, x_h) \rangle_{\mathcal{A}} \,|\, x_1 = x_1^t \right]}_{(iii)}, \tag{E.4}$$

where $\zeta_{t,h}^1$ and $\zeta_{t,h}^2$ are defined in (E.2) and (E.3), respectively.

*Proof.* See Appendix §H.1 for a detailed proof. □

Returning to the main proof, notice that $\pi_h^t$ is the greedy policy with respect to $Q_h^t$ for all $(t, h) \in [T] \times [H]$. We have

$$\big\langle Q_h^t(x_h, \cdot), \pi_h^\star(\cdot \,|\, x_h) - \pi_h^t(\cdot \,|\, x_h)\big\rangle_{\mathcal{A}} = \big\langle Q_h^t(x_h, \cdot), \pi_h^\star(\cdot \,|\, x_h)\big\rangle_{\mathcal{A}} - \max_{a \in \mathcal{A}} Q_h^t(x_h, a) \leq 0,$$

for all $x_h \in \mathcal{S}$. Thus, Term (iii) in (E.4) is non-positive. Then, by Lemma E.1, we can upper bound the regret by

$$\text{Regret}(T) \leq \underbrace{\left\{ \sum_{t=1}^T \sum_{h=1}^H \big[\mathbb{E}_{\pi^\star}[\delta_h^t(x_h, a_h) \,|\, x_1 = x_1^t] - \delta_h^t(x_h^t, a_h^t)\big] \right\}}_{\text{(i)}} + \underbrace{\left[ \sum_{t=1}^T \sum_{h=1}^H (\zeta_{t,h}^1 + \zeta_{t,h}^2) \right]}_{\text{(ii)}}.$$

(E.5)

For Term (i), since we do not observe trajectories from $\pi^*$, which is unknown, it appears that $\mathbb{E}_{\pi^*}[\delta_h^t(x_h, a_h) \,|\, x_1 = x_1^t]$ cannot be estimated. Fortunately, however, by adding the bonus term in Algorithm 2, we ensure that the temporal-difference error $\delta_h^t$ is a non-positive function, as shown in the following lemma.

**Lemma E.2** (Optimism). Let $\lambda = 1 + 1/T$ and $\beta = B_T$ in Algorithm 2, where $B_T$ satisfies (4.5). Under Assumptions 4.1, with probability at least $1 - (2T^2H^2)^{-1}$, we have that the following holds for all $(t, h) \in [T] \times [H]$ and $(x, a) \in \mathcal{S} \times \mathcal{A}$:

$$-2\beta \cdot b_h^t(x, a) \leq \delta_h^t(x, a) \leq 0.$$

*Proof.* See Appendix §H.2 for a detailed proof. □

Applying Lemma E.2 to Term (i) in (E.5), we obtain that

$$\text{Term (i)} \leq \left[ \sum_{t=1}^T \sum_{h=1}^H -\delta_h^t(x_h^t, a_h^t) \right] \leq 2\beta \cdot \left[ \sum_{t=1}^T \sum_{h=1}^H b_h^t(x_h^t, a_h^t) \right] \qquad \text{(E.6)}$$

holds with probability at least $1 - (2T^2H^2)^{-1}$, where $\beta$ is equal to $B_T$ as specified in (4.5).

Finally, it remains to bound the sum of bonus terms in (E.6). As we show in (H.17), using the feature representation of $\mathcal{H}$, we can write each $b_h^t(x_h^t, a_h^t)$ as

$$b_h^t(x_h^t, a_h^t) = \big[\phi(x_h^t, a_h^t)^\top (\Lambda_h^t)^{-1} \phi(x_h^t, a_h^t)\big]^{1/2},$$

where $\Lambda_h^t = \lambda \cdot I_{\mathcal{H}} + \sum_{\tau=1}^{t-1} \phi(x_h^t, a_h^t)\phi(x_h^t, a_h^t)^\top$ is a self-adjoint and positive-definite operator on $\mathcal{H}$ and $\mathcal{I}_{\mathcal{H}}$ is the identity mapping on $\mathcal{H}$. Thus, combining the Cauchy-Schwarz inequality and Lemma J.3, we have, for any $h \in [H]$, with probability at least $1 - (2T^2H^2)^{-1}$ the following:

$$\text{Term (i)} \leq 2\beta \cdot \sqrt{T} \cdot \sum_{h=1}^H \left[ \sum_{t=1}^T \phi(x_h^t, a_h^t)^\top (\Lambda_h^t)^{-1} \phi(x_h^t, a_h^t) \right]^{1/2}$$

$$\leq 2\beta \cdot \sum_{h=1}^H \big[2T \cdot \text{logdet}(I + K_h^T/\lambda)\big]^{1/2} = 4\beta H \cdot \sqrt{T \cdot \Gamma_K(T, \lambda)}, \qquad \text{(E.7)}$$

where $\Gamma_K(T, \lambda)$ is the maximal information gain defined in (4.2) with parameter $\lambda$.

It remains to bound Term (ii) in (E.5), which is the purpose of the following lemma.

**Lemma E.3.** For $\zeta_{t,h}^1$ and $\zeta_{t,h}^2$ defined respectively in (E.2) and (E.3) and for any $\zeta \in (0, 1)$, with probability at least $1 - \zeta$, we have

$$\sum_{t=1}^T \sum_{h=1}^H (\zeta_{t,h}^1 + \zeta_{t,h}^2) \leq \sqrt{16TH^3 \cdot \log(2/\zeta)}.$$

*Proof.* See Appendix §H.3 for a detailed proof. □

Setting $\zeta = (2T^2H^2)^{-1}$ in Lemma E.3 we obtain that

$$\text{Term (ii)} = \sum_{t=1}^{T}\sum_{h=1}^{H}(\zeta_{t,h}^1 + \zeta_{t,h}^2) \leq \sqrt{16TH^3 \cdot \log(4T^2H^2)} = \sqrt{32TH^3 \cdot \log(2TH)} \quad \text{(E.8)}$$

holds with probability at least $1 - (2TH)^{-1}$.

Therefore, combining (4.5), (E.5), and (E.8), we conclude that, with probability at least $1 - (T^2H^2)^{-1}$, the regret is bounded by

$$\text{Regret}(T) \leq 4\beta H \cdot \sqrt{T \cdot \Gamma_K(T, \lambda)} + \sqrt{32TH^3 \cdot \log(2TH)} \leq 5\beta H \cdot \sqrt{T \cdot \Gamma_K(T, \lambda)},$$

where the last inequality follows from the choice of $\beta = B_T$, which implies that

$$\beta \geq H \cdot \sqrt{16\log(TH)} \geq \sqrt{32H \cdot \log(2TH)}.$$

This concludes the proof of Theorem 4.2. $\qquad\qquad\square$

## E.2 Proof of Theorem D.2

*Proof.* The proof of Theorem D.2 is similar to that of Theorem 4.2. Recall that we let $\mathcal{Z}$ denote $\mathcal{S} \times \mathcal{A}$ for simplicity. Recall also that for all $(t, h) \in [T] \times [H]$, we define the temporal-difference (TD) error $\delta_h^t \colon \mathcal{Z} \to \mathbb{R}$ in (E.1) and define random variables $\zeta_{t,h}^1$ and $\zeta_{t,h}^2$ in (E.2) and (E.3), respectively.

Then, combining Lemma E.1 and the fact that $\pi^t$ is the greedy policy with respect to $\{Q_h^t\}_{h \in [H]}$, we bound the regret by

$$\text{Regret}(T) \leq \underbrace{\left\{\sum_{t=1}^{T}\sum_{h=1}^{H}\big[\mathbb{E}_{\pi^\star}[\delta_h^t(x_h, a_h) \,|\, x_1 = x_1^t] - \delta_h^t(x_h^t, a_h^t)\big]\right\}}_{\text{(i)}} + \underbrace{\left[\sum_{t=1}^{T}\sum_{h=1}^{H}(\zeta_{t,h}^1 + \zeta_{t,h}^2)\right]}_{\text{(ii)}}.$$

$$\text{(E.9)}$$

Here, Term (ii) is a sum of a martingale difference sequence. By setting $\zeta = (4T^2H^2)^{-1}$ in Lemma E.3, with probability at least $1 - (4T^2H^2)^{-1}$, we have

$$\text{Term (ii)} = \sum_{t=1}^{T}\sum_{h=1}^{H}(\zeta_{t,h}^1 + \zeta_{t,h}^2) \leq \sqrt{16TH^3 \cdot \log(8T^2H^2)} \leq H \cdot \sqrt{32TH \log(2TH)}. \quad \text{(E.10)}$$

It remains to bound Term (i) in (E.9). To this end, we aim to establish a counterpart of Lemma E.2 for neural value functions, which shows that, by adding a bonus term $\beta \cdot b_h^t$, the TD error $\delta_h^t$ is always a non-positive function approximately. This implies that bounding Term (i) in (E.9) reduces to controlling $\sum_{t=1}^{T}\sum_{h=1}^{H} b_h^t(x_h^t, a_h^t)$.

Note that the bonus functions $b_h^t$ are constructed based on the neural tangent features $\varphi(\cdot; \widehat{W}_h^t)$ and the matrix $\Lambda_h^t$. In order to relate $\sum_{t=1}^{T}\sum_{h=1}^{H} b_h^t(x_h^t, a_h^t)$ to the maximal information gain of the empirical NTK $K_m$, we define $\overline{\Lambda}_h^t$ and $\overline{b}_h^t$, by analogy with $\Lambda_h^t$ and $b_h^t$, as follows:

$$\overline{\Lambda}_h^t = \lambda \cdot I_{2md} + \sum_{\tau=1}^{t-1} \varphi(x_h^\tau, a_h^\tau; W^{(0)})\varphi(x_h^\tau, a_h^\tau; W^{(0)})^\top, \qquad \overline{b}_h^t(z) = \big[\varphi(z; W^{(0)})^\top (\overline{\Lambda}_h^t)^{-1}\varphi(z; W^{(0)})\big]^{1/2}.$$

In the following lemma, we bound the TD error $\delta_h^t$ using $\overline{b}_h^t$ and show that $b_h^t$ and $\overline{b}_h^t$ are close in the $\ell_\infty$-norm on $\mathcal{Z}$ when $m$ is sufficiently large.

**Lemma E.4** (Optimism). Let $\lambda$ be an absolute constant and let $\beta = B_T$ in Algorithm 4, where $B_T$ satisfies (D.3). Under the assumptions made in Theorem D.2, with probability at least $1 - (2T^2H^2)^{-1} - m^2$, it holds for all $(t, h) \in [T] \times [H]$ and $(x, a) \in \mathcal{S} \times \mathcal{A}$ that

$$-5\beta \cdot \iota - 2\beta \cdot \overline{b}_h^t(x, a) \leq \delta_h^t(x, a) \leq 5\beta \cdot \iota, \qquad \sup_{(x,a)\in\mathcal{Z}} \big|b_h^t(x, a) - \overline{b}_h^t(x, a)\big| \leq 2\iota, \quad \text{(E.11)}$$

where we define $\iota = T^{7/12} \cdot H^{1/12} \cdot m^{-1/12} \cdot (\log m)^{1/4}$.

*Proof.* See Appendix §H.4 for a detailed proof. $\qquad\qquad\square$

Applying Lemma E.2 to Term (i) in (E.5), we obtain that

$$\text{Term (i)} \leq \left[\sum_{t=1}^{T}\sum_{h=1}^{H} -\delta_h^t(x_h^t, a_h^t)\right] + 5TH \cdot \iota \leq 2\beta \cdot \left[\sum_{t=1}^{T}\sum_{h=1}^{H} \bar{b}_h^t(x_h^t, a_h^t)\right] + 10\beta TH \cdot \iota \quad \text{(E.12)}$$

holds with probability at least $1 - (2T^2H^2)^{-1} - m^{-2}$, where $\beta = B_T$. Moreover, combining the Cauchy-Schwarz inequality and Lemma J.3, we have

$$\sum_{t=1}^{T}\sum_{h=1}^{H} \bar{b}_h^t(x_h^t, a_h^t) \leq \sqrt{T} \cdot \sum_{h=1}^{H}\left[\sum_{t=1}^{T} \varphi(x_h^t, a_h^t; W^{(0)})^\top (\overline{\Lambda}_h^t)^{-1}\varphi(x_h^t, a_h^t; W^{(0)})\right]^{1/2}$$

$$\leq 2H \cdot \sqrt{T \cdot \Gamma_{K_m}(T, \lambda)}, \quad \text{(E.13)}$$

where $\Gamma_K(T, \lambda)$ is the maximal information gain defined in (4.2) for kernel $K_m$.

Notice that $(2T^2H^2)^{-1} + m^{-2} + (4T^2H^2)^{-1} \leq (T^2H^2)^{-1}$. Thus, combining (E.9), (E.10), (E.12), and (E.13), we obtain that

$$\text{Regret}(T) \leq 4\beta H \cdot \sqrt{T \cdot \Gamma_{K_m}(T, \lambda)} + 10\beta TH \cdot \iota + H \cdot \sqrt{32TH\log(2TH)}$$

$$\leq 5\beta H \cdot \sqrt{T \cdot \Gamma_{K_m}(T, \lambda)} + 10\beta TH \cdot \iota$$

holds with probability at least $1 - (2T^2H^2)^{-1}$. Here the last inequality follows from the fact that

$$\beta \geq H \cdot \sqrt{32\log(TH)} \geq \sqrt{32H\log(2TH)}.$$

This concludes the proof of Theorem D.2. $\qquad\qquad\qquad\qquad\qquad\qquad\qquad\qquad\qquad\square$

# F   Neural Optimistic Least-Squares Value Iteration

In this section, we provide the pseudocode for NOVI, which was omitted in the main text for brevity. We remark that the loss function $L_h^t$ in Line 7 is given in (C.1) and its global minimizer $\widehat{W}_h^t$ can be efficiently obtained by first-order optimization methods.

---
**Algorithm 4** Neural Optimistic Least-Squares Value Iteration (NOVI)

---
1: **Input:** Parameters $\lambda$ and $\beta$.
2: Initialize the network weights $(b^{(0)}, W^{(0)})$ via the symmetric initialization scheme.
3: **for** episode $t = 1, \ldots, T$ **do**
4:     Receive the initial state $x_1^t$.
5:     Set $V_{H+1}^t$ as the zero function.
6:     **for** step $h = H, \ldots, 1$ **do**
7:         Solve the neural network optimization problem $\widehat{W}_h^t = \text{argmin}_W L_h^t(W)$.
8:         Update $\Lambda_h^t = \Lambda_h^{t-1} + \varphi(x_h^{t-1}, a_h^{t-1}; \widehat{W}_h^t)\varphi(x_h^{t-1}, a_h^{t-1}; \widehat{W}_h^t)^\top$.
9:         Obtain the bonus function $b_h^t$ defined in (C.4).
10:        Obtain value functions

$$Q_h^t(\cdot, \cdot) \leftarrow \min\{f(\cdot, \cdot; \widehat{W}_h^t) + \beta \cdot b_h^t(\cdot, \cdot), H - h + 1\}^+, \qquad V_h^t(\cdot) = \max_a Q_h^t(\cdot, a).$$

11:     **end for**
12:     **for** step $h = 1, \ldots, H$ **do**
13:         Take action $a_h^t \leftarrow \text{argmax}_{a \in \mathcal{A}} Q_h^t(x_h^t, a)$.
14:         Observe the reward $r_h(x_h^t, a_h^t)$ and the next state $x_{h+1}^t$.
15:     **end for**
16: **end for**

---

# G   Proofs of the Corollaries

In this section, we prove Corollaries 4.4 and D.4, which establish the regret for KOVI and NOVI under each specific eigenvalue decay condition. in Appendix §G.3 we provide concrete examples of neural

tangent kernels that satisfy Assumption 4.3 and show how to apply Corollaries 4.4 and D.4 to these examples.

## G.1 Proof of Corollary 4.4

*Proof.* To prove this corollary, it suffices to verify that for each eigenvalue decay condition specified in Assumption 4.3, $B_T$ defined in (4.8) satisfies the condition in (4.5). Recall that we set $\lambda = 1 + 1/T$ in Algorithm 2 and denote $R_T = 2H\sqrt{\Gamma_K(T, \lambda)}$, $\epsilon^* = H/T$. Also recall that we let $N_\infty(\epsilon, h, B)$ denote the $\epsilon$-covering number of $\mathcal{Q}_{\mathrm{ucb}}(h, R_T, B)$ with respect to the $\ell_\infty$-norm. In the sequel, we consider the two cases separately.

**Case (i): $\gamma$-Finite Spectrum.** When $\mathcal{H}$ has at most $\gamma$ nonzero eigenvalues, by Lemma I.5, we have $\Gamma_K(T, \lambda) \leq C_K \cdot \gamma \log T$, where $C_K$ is an absolute constant. Moreover, by Lemma I.1, for any $h \in [H]$, we have

$$\log N_\infty(\epsilon^*, h, B_T) \leq C_N \cdot \gamma \cdot \left\{1 + \log\left[2\sqrt{\Gamma(T, \lambda)} \cdot T\right]\right\} + C_N \cdot \gamma^2 \cdot \left[1 + \log(B_T \cdot T/H)\right]$$
$$\leq 2C_N \cdot \gamma^2 + C' \cdot \gamma \cdot \log(\gamma T) + C_N \cdot \gamma^2 \cdot \log(B_T \cdot T/H), \tag{G.1}$$

where $C_N > 0$ is the absolute constant given in Lemma I.1 and $C'$ is an absolute constant that depends on $C_N$ and $C_K$. Thus, setting $B_T = C_b \cdot \gamma H \cdot \sqrt{\log(dTH)}$ in (G.1), the left-hand side (LHS) of (4.5) is bounded by

$$\text{LHS of (4.5)} \leq 8C_K \cdot \gamma \log T + 16 C_N \cdot \gamma^2 + 8C' \cdot \gamma \cdot \log(\gamma T) +$$
$$8C_N \cdot \gamma^2 \cdot \log(C_b \cdot \gamma T \cdot \sqrt{\log(dTH)}) + 16 \cdot \log(TH) + 22 + 2R_Q^2$$
$$\leq \gamma^2 \cdot \left[\overline{C}_1 \cdot \log(\gamma TH) + 8C_N \cdot \log(C_b)\right], \tag{G.2}$$

where $\overline{C}_1$ is an absolute constant that depends on $C'$, $C_N$, $C_K$, and $R_Q$. Thus, setting $C_b$ as a sufficiently large constant, by (G.2), we have

$$\text{LHS of (4.5)} \leq C_b^2 \cdot \gamma^2 \cdot \log(dTH) = (B_T/H)^2,$$

which establishes (4.5) for the first case. Thus, applying Theorem 4.2 we obtain that

$$\text{Regret}(T) \leq 8B_T \cdot H \cdot \sqrt{T \cdot \Gamma_K(T, \lambda)} \leq C_{r,1} \cdot H^2 \cdot \sqrt{\gamma^3 T} \cdot \log(\gamma TH) = \widetilde{\mathcal{O}}(H^2\sqrt{\gamma^3 T})$$

holds with probability at least $1 - (T^2H^2)^{-1}$, where $C_{r,1}$ is an absolute constant and $\widetilde{\mathcal{O}}(\cdot)$ omits the logarithmic factor. Therefore, we conclude the first case.

**Case (ii): $\gamma$-Exponential Decay.** For the second case, by Lemma I.5 we have

$$\Gamma_K(T, \lambda) \leq C_K \cdot (\log T)^{1+1/\gamma}, \tag{G.3}$$

where $C_K$ is an absolute constant. Thus, by the choice of $B_T$ in (4.8), when $C_b$ is sufficiently large, it holds that $R_T = 2H\sqrt{\Gamma_K(T, \lambda)} \leq B_T$. Then by Lemma I.1 we have

$$\log N_\infty(h, \epsilon^*, B_T) \leq C_N \cdot \left[1 + \log(R_T/\epsilon^*)\right]^{1+1/\gamma} + C_N \cdot \left[1 + \log(B_T/\epsilon^*)\right]^{1+2/\gamma}$$
$$\leq 2C_N \cdot \left[1 + \log(B_T/\epsilon^*)\right]^{1+2/\gamma} = 2C_N \cdot \left\{1 + \log\left[C_b T \cdot \sqrt{\log(TH)} \cdot (\log T)^{1/\gamma}\right]\right\}^{1+2/\gamma},$$

where the absolute constant $C_N$ is given by Lemma I.1. By direct computation, there exists an absolute constant $\overline{C}_2$ such that

$$\log N_\infty(h, \epsilon^*, B_T) \leq 2C_N \cdot \left[1 + \log(C_b) + \overline{C}_2 \cdot \log T + 1/2 \cdot \log\log H\right]^{1+2/\gamma}. \tag{G.4}$$

Thus, combining (G.3) and (G.4), the left-hand side of (4.5) is bounded by

$$\text{LHS of (4.5)} \leq 8C_K \cdot (\log T)^{1+1/\gamma} + 16C \cdot \left[1 + \log(C_b) + \overline{C}_2 \cdot \log T + 1/2 \cdot \log\log H\right]^{1+2/\gamma}$$
$$+ 16 \cdot \log(TH) + 22 + 2R_Q^2$$
$$\leq \overline{C}_3 \cdot \left[(\log T)^{1+2/\gamma} + (\log\log H)^{1+2/\gamma} + \log(C_b)\right], \tag{G.5}$$

where $\overline{C}_3$ is an absolute constant that does not depend on $C_b$. Thus, when $C_b$ is sufficiently large, (G.5) implies that

$$\text{LHS of (4.5)} \leq \overline{C}_3 \cdot \left[(\log T)^{1+2/\gamma} + (\log\log H)^{1+2/\gamma} + \log(C_b)\right] \leq C_b^2 \cdot (\log T)^{2/\gamma} \cdot \log(TH) = (B_T/H)^2.$$

Thus, for the case of $\gamma$-exponential eigenvalue decay, (4.5) holds true for $B_T$ defined in (4.8).

Finally, applying Theorem 4.2 and combining (4.8) and (G.3), we obtain that

$$\text{Regret}(T) \le C_{r,2} \cdot H^2 \cdot \log(TH) \cdot \sqrt{(\log T)^{3/\gamma} \cdot T},$$

where $C_{r,2}$ is an absolute constant. Thus we conclude the second case. Therefore, we conclude the proof of Corollary 4.4. □

## G.2 Proof of Corollary D.4

*Proof.* By Theorem D.2, we have

$$\text{Regret}(T) = 5\beta H \cdot \sqrt{T \cdot \Gamma_{K_m}(T, \lambda)} + 10\beta TH \cdot \iota, \tag{G.6}$$

where $\beta = B_T$ satisfies (D.3) and $\iota = T^{7/12} \cdot H^{1/6} \cdot m^{-1/12} \cdot (\log m)^{1/4}$. When Assumption D.3 holds, thanks to the similarity between (4.5) and (D.3), it can be similarly shown that $B_T$ defined in (4.8) satisfies the inequality in (D.3) when $C_b$ is sufficiently large. Moreover, Lemma I.5 provides upper bounds on $\Gamma_{K_m}(T, \lambda)$ for the two eigenvalue decay conditions. Finally, combining (4.8), (G.6), and Lemma I.5, we conclude the proof of Corollary D.4. □

## G.3 Examples of Kernels Satisfying Assumption 4.3

In the following, we introduce concrete kernels and neural tangent kernels that satisfy Assumption 4.3. We consider each eigenvalue decay condition separately.

**Case (i): $\gamma$-Finite Spectrum.** Consider the polynomial kernel $K(z, z') = (1 + \langle z, z' \rangle)^n$ defined on the unit ball $\{z \in \mathbb{R}^d \colon \|z\|_2 \le 1\}$, where $n$ is a fixed number. By direct computation, the kernel function can be written as

$$K(z, z') = \sum_{\alpha \colon \|\alpha\|_1 \le n} z^\alpha \cdot z'^\alpha,$$

where $\alpha = (\alpha_1, \dots, \alpha_d) \in \mathbb{N}^d$ is a multi-index and $z^\alpha$ is a monomial with degree $\alpha$. It can be shown that all monomials in $\mathbb{R}^d$ with degree no more than $n$ are linearly independent. Thus, the dimension of such an RKHS is $\binom{n+d}{d}$; i.e., it satisfies the $\gamma$-finite spectrum condition with $\gamma = \binom{n+d}{d}$.

Furthermore, for a finite-dimensional NTK, we consider the quadratic activation function $\text{act}(u) = u^2$. Note that we assume $\mathcal{Z} = \mathbb{S}^{d-1}$ for the neural network setting. Moreover, in (B.3), instead of sampling $W_j \sim N(0, I_d/d)$ for all $j \in [d]$, we draw $W_j$ uniformly over the unit sphere $\mathbb{S}^{d-1}$. Then it holds that $|W_j^\top z| \le 1$ for all $j \in [2m]$ and $z \in \mathbb{S}^{d-1}$. Here we let the distribution be $\text{Unif}(\mathbb{S}^{d-1})$ in order to ensure that the $\text{act}'$ is Lipschitz continuous on $\{W_j^\top z \colon z \in \mathbb{S}^{d-1}\} \subseteq [-1, 1]$ for any $W_j$ sampled from the initial distribution, which is required when utilizing Proposition C.1 in [30] in the proof of Lemma E.4. Note that the covariance of $W_j$ is still $I_d/d$. Then by (B.7), the NTK is given by

$$K_{\text{ntk}}(z, z') = \mathbb{E}_{w \sim \text{Unif}(\mathbb{S}^{d-1})}[2(w^\top z) \cdot 2(w^\top z') \cdot (z^\top z')] = 4/d \cdot (z^\top z')^2, \qquad \forall z, z' \in \mathbb{S}^{d-1}. \tag{G.7}$$

Thus, $K_{\text{ntk}}(z, z')$ can be written as a univariate function of the inner product $\langle z, z' \rangle$. To characterize the spectral property $K_{\text{ntk}}$, we first introduce some background on spherical harmonic functions on $\mathbb{S}^{d-1}$, which are closely related to inner product kernels on $\mathbb{S}^{d-1} \times \mathbb{S}^{d-1}$.

Let $\mu$ be the uniform measure on $\mathbb{S}^{d-1}$. For any $j \ge 0$, let $\mathcal{Y}_j(d)$ be the set of all homogeneous harmonics of degree $j$ on $\mathbb{S}^{d-1}$, which is a finite-dimensional subspace of $\mathcal{L}_\mu^2(\mathbb{S}^{d-1})$, the space of square-integrable functions on $\mathbb{S}^{d-1}$ with respect to $\mu$. It can be shown that the dimensionality of $\mathcal{Y}_j(d)$ is given by $N(d, j)$, which is defined as

$$N(d, j) = \frac{(2j + d - 2)(d + j - 3)!}{j!(d-2)!}. \tag{G.8}$$

In addition, let $\{Y_{j,\ell}\}_{\ell \in [N(d,j)]}$ be an orthonormal basis of $\mathcal{Y}_j(d)$, then $\{Y_{j,\ell}\}_{\ell \in [N(d,j)], j \in \mathbb{N}}$ form an orthonormal basis of $\mathcal{L}_\mu^2(\mathbb{S}^{d-1})$. In the next lemma, we present the Funk-Hecke formula [48, page 30], which relates spherical harmonics to inner product kernels.

**Lemma G.1** (Funk-Hecke formula)**.** Let $k\colon [-1,1] \to \mathbb{R}$ be a continuous function, which gives rise to an inner product kernel $K(z,z') = k(\langle z,z' \rangle)$ on $\mathbb{S}^{d-1} \times \mathbb{S}^{d-1}$. For any $\ell \geq 2$, let $|\mathbb{S}^{\ell-1}|$ be the Lebesgue measure of $\mathbb{S}^{\ell-1}$, which is given by $|\mathbb{S}^{\ell-1}| = 2\pi^{\ell/2}/\Gamma(\ell/2)$, where $\Gamma(\cdot)$ is the Gamma function. Moreover, for any $j \geq 0$, let $Y_j\colon \mathbb{S}^{d-1} \to \mathbb{R}$ be any function in $\mathcal{Y}_j(d)$. Then for any $z \in \mathbb{S}^{d-1}$, we have

$$\int_{\mathbb{S}^{d-1}} K(z,z') Y_j(z') \, \mathrm{d}\mu(z') = \left[ \frac{|\mathbb{S}^{d-2}|}{|\mathbb{S}^{d-1}|} \cdot \int_{-1}^{1} k(u) \cdot P_j(u;d) \cdot (1-u^2)^{(d-3)/2} \, \mathrm{d}u \right] \cdot Y_j(z), \tag{G.9}$$

where $P_j(\cdot; d)$ is the $j$-th Legendre polynomial in dimension $d$, which is given by

$$P_j(u;d) = \frac{(-1/2)^j \cdot \Gamma(\frac{d-1}{2})}{\Gamma(\frac{2j+d-1}{2})} \cdot (1-u^2)^{(3-d)/2} \cdot \left( \frac{\mathrm{d}}{\mathrm{d}u} \right)^j \left[ (1-u^2)^{j+(d-3)/2} \right].$$

Thus, by the Funk-Hecke formula, for any inner product kernel $K$, its integral operator $T_K\colon \mathcal{L}_\mu^2(\mathbb{S}^{d-1}) \to \mathcal{L}_\mu^2(\mathbb{S}^{d-1})$ has eigenvalues

$$\varrho_j = \frac{|\mathbb{S}^{d-2}|}{|\mathbb{S}^{d-1}|} \cdot \int_{-1}^{1} k(u) \cdot P_j(u;d) \cdot (1-u^2)^{(d-3)/2} \, \mathrm{d}u, \qquad \forall j \geq 1, \tag{G.10}$$

each with multiplicity $N(d,j)$. Moreover, for each eigenvalue $\varrho_j$, the corresponding eigenfunctions are spherical harmonics $\{Y_{j,\ell}\}_{\ell \in [N(d,j)]}$. Furthermore, to compute the eigenvalues in (G.10), we can use Rodrigues' rule [48, page 23], as follows.

**Lemma G.2** (Rodrigues' Rule)**.** For any $j \geq 0$, let $f\colon [-1,1] \to \mathbb{R}$ be any $j$-th continuously differentiable function. Then we have

$$\int_{-1}^{1} f(t) \cdot P_j(u;d) \cdot (1-u^2)^{(d-3)/2} \, \mathrm{d}u = R_j(d) \cdot \int_{-1}^{1} f^{(j)}(u) \cdot (1-u^2)^{(2j+d-3)/2} \, \mathrm{d}t,$$

where $f^{(j)}$ is the $j$-th order derivative of $f$ and $R_j(d) = 2^{-j} \cdot \Gamma((d-1)/2) \cdot [\Gamma((2j+d-1)/2)]^{-1}$ is the $j$-th Rodrigues constant.

Now we consider the NTK given in (G.7), which is the inner product kernel induced by the univariate function $k_1(u) = 4/d \cdot u^2$. Note that $k_1^{(3)}$ is a zero function. Combining Lemma G.2 and (G.10), we observe that $\varrho_j = 0$ for all $j \geq 3$. In addition, by direct computation, we have that

$$\varrho_1 = R_1(d) \cdot (8/d) \cdot \int_{-1}^{1} u \cdot (1-u^2)^{(d-1)/2} \, \mathrm{d}u = 0,$$

and $\varrho_0, \varrho_2 > 0$. Thus, $K_{\mathrm{ntk}}$ given in (G.7) has $N(d,0) + N(d,2) = d(d+1)/2$ nonzero eigenvalues, each with value $\varrho_2$. This implies that the NTK induced by neural networks with quadratic activation satisfies the $\gamma$-finite spectrum condition with $\gamma = d(d+1)/2$. For such a class of neural networks, Corollary D.4 asserts that the regret of NOVI is $\widetilde{\mathcal{O}}(H^2 d^3 \cdot \sqrt{T} + \beta T H \cdot \iota)$.

**Case (ii): $\gamma$-exponential Decay.** Now we consider the squared exponential kernel

$$K(z,z') = \exp(-\|z-z'\|_2^2 \cdot \sigma^{-2}) = k_2(\langle z,z' \rangle), \qquad \forall z,z' \in \mathbb{S}^{d-1}, \tag{G.11}$$

where $\sigma > 0$ is an absolute constant and we define $k_2(u) = \exp[-2\sigma^{-2} \cdot (1-u)]$. Note that $d$ is regarded as a fixed number. Applying Lemmas G.1 and G.2, we obtain the following lemma that bounds the eigenvalues of $T_K$.

**Lemma G.3** (Theorem 2 in [47])**.** For the squared quadratic kernel in (G.11), the corresponding integral operator has eigenvalues $\{\rho_j\}_{j \geq 0}$, where each $\rho_j$ is defined in (G.10) with $k$ replaced by $k_2$. Moreover, each $\varrho_j$ has multiplicity $N(d,j)$ and the corresponding eigenfunctions are $\{Y_{j,\ell}\}_{\ell \in [N(d,j)]}$. Finally, when $\sigma$ in (G.11) satisfy $\sigma^2 \geq 2/d$, $\{\varrho_j\}_{j \geq 0}$ form a decreasing sequence that satisfy

$$A_1 \cdot (2e/\sigma^2)^j \cdot (2j+d-2)^{-(2j+d-1)/2} < \varrho_j < A_2 \cdot (2e/\sigma^2)^j \cdot (2j+d-2)^{-(2j+d-1)/2} \tag{G.12}$$

for all $j \geq 0$, where $A_1, A_2$ are absolute constants that only depend on $d$ and $\sigma$.

The $\ell_\infty$-norm of each eigenfunction $Y_{j,\ell}$ is given by the following lemma.

**Lemma G.4** (Lemma 3 in [47]). For any $d \geq 2$, $j \geq 0$, and any $\ell \in [N(d,j)]$, we have

$$\|Y_{j,\ell}\|_\infty = \sup_{z \in \mathbb{S}^{d-1}} |Y_{j,\ell}(z)| \leq \sqrt{N(d,j)/|\mathbb{S}^{d-1}|}.$$

Now, let $\tau > 0$ be a sufficiently small constant. Combining Lemmas G.3 and G.4, we have

$$\varrho_j^\tau \cdot \|Y_{j,\ell}\|_\infty \leq C \cdot \left(\frac{2e}{\sigma^2 \cdot (2j+d-2)}\right)^{-j\cdot\tau} \cdot \sqrt{N(d,j) \cdot (2j+d-2)^{-(d-1)\cdot\tau}}, \qquad (G.13)$$

where $C$ is a constant depending on $d$ and $\sigma$. By the definition of $N(d,j)$ in (G.8), when $j$ is sufficiently large, it holds that

$$N(d,j) \asymp \frac{(2j+d-2) \cdot \sqrt{d+j-3} \cdot [(d+j-3)/e]^{d+j-3}}{\sqrt{j} \cdot (j/e)^j} \asymp j^{d-2}, \qquad (G.14)$$

where we utilize the Stirling's formula and neglect constants involving $d$. Then, combining (G.13) and (G.14), we have

$$\sup_{j \geq 0} \sup_{\ell \in [N(d,j)]} \varrho_j^\tau \cdot \|Y_{j,\ell}\|_\infty \leq C_\varrho, \qquad (G.15)$$

for some absolute constant $C_\varrho > 0$. Renaming the eigenvalues and eigenvectors as $\{\sigma_j, \psi_j\}_{j \geq 1}$ in the descending order of the eigenvalues, (G.15) equivalently states that $\sup_{j \geq 1} \sigma_j^\tau \cdot \|\psi_j\|_\infty \leq C_\varrho$.

Furthermore, to show that the squared exponential kernel satisfy the $\gamma$-exponential decay condition, we notice that

$$\sigma_j = \varrho_t \qquad \text{for } \sum_{i=1}^{t-1} N(d,i) \leq j < \sum_{i=1}^{t} N(d,i). \qquad (G.16)$$

Then by (G.14), this implies that $\sigma_j \asymp \rho_t$ for $(t-1)^{d-1} \leq j \leq t^{d-1}$ when $j$ is sufficiently large. Thus, by Lemma G.3 we further obtain that

$$\sigma_j \asymp (2e/\sigma^2)^{j^{\frac{1}{d-1}}} \cdot (2j^{\frac{1}{d-1}} + d - 2)^{-j^{\frac{1}{d-1}} - (d-1)/2}$$
$$\asymp \exp\left(c_1 \cdot j^{\frac{1}{d-1}}\right) \cdot \exp\left(c_2 - j^{\frac{1}{d-1}} \cdot \log j\right) \leq \exp(-c \cdot j^{1/d}),$$

where $c$, $c_1$, and $c_2$ are constants depending on $d$. Therefore, we have shown that the squared exponential kernel satisfies the $\gamma$-exponential decay condition with $\gamma = 1/d$. Combining this with (G.15), we conclude that it satisfies Assumption 4.3.

In the sequel, we construct an NTK that satisfies Assumption 4.3. Specifically, we adopt the sine activation function and slightly modify the neural network in (B.3) by employing an intercept for each neuron. That is,

$$f(z; b, W, \theta) = \frac{1}{\sqrt{m}} \sum_{j=1}^{m} b_j \cdot \sin(W_j^\top z + \theta_j).$$

To initialize the network weights $(b, W, \theta)$, we set $b_j = -b_{j-m}$, $W_j = W_{j-m}$, and $\theta_j = \theta_{j-m}$ for any $j \in \{m+1, \ldots, 2m\}$. For any $j \in [m]$, we independently sample $b_j \sim \text{Unif}(\{-1,1\})$, $W_j \sim N(0, I_d)$, and $\theta_j \sim \text{Unif}([0, 2\pi])$. Only $W$ is updated during training.

For such a neural network, the corresponding NTK is given by

$$K_{\text{ntk}}(z, z') = 2\mathbb{E}\left[(z^\top z') \cdot \cos(w^\top z + \theta) \cdot \cos(w^\top z' + \theta)\right]$$
$$= (z^\top z') \cdot \exp(-\|z - z'\|_2^2/2) = (z^\top z') \cdot \exp[(z^\top z') - 1] = k_3(\langle z, z'\rangle), \qquad (G.17)$$

where we define $k_3(u) = u \cdot \exp(u-1)$. Here the second equality follows from [54]. By construction, such an NTK is closely related to the squared quadratic kernel in (G.11). To see that it satisfy the $\gamma$-exponential decay condition, let $\{\varrho_j\}_{j \geq 0}$ and $\{\widetilde{\varrho}_j\}_{j \geq 0}$ denote the eigenvalues of the NTK in (G.17) and the inner product kernel induced by $k_2(u) = \exp(u-1)$, respectively. By Lemma G.1, we have

$$\rho_j = C_1 \cdot \int_{-1}^{1} k_3(u) \cdot P_j(u; d) \cdot (1-u^2)^{(d-3)/2} \, du = C_1 \cdot \int_{-1}^{1} k_2(u) \cdot u \cdot P_j(u; d) \cdot (1-u^2)^{(d-3)/2} \, du$$
$$= C_2 \cdot j/(2j+d-2) \cdot \widetilde{\varrho}_{j-1} + C_2 \cdot (j+d-2)/(2j+d-2) \cdot \widetilde{\varrho}_{j+1} \leq C_2(\widetilde{\rho}_{j-1} + \widetilde{\rho}_{j+1}), \qquad (G.18)$$

where $C_1$ and $C_2$ are constants and in the second equality, we utilize the following recurrence relation of Legendre polynomials:
$$u \cdot P_j(u;d) = j/(2j+d-2) \cdot P_{j-1}(u;d) + (j+d-2)/(2j+d-2) \cdot P_{j+1}(u;d).$$
Notice that $\{\widetilde{\varrho}_j\}_{j \geq 0}$ satisfy (G.12). Thus, combining (G.12) and (G.18), we obtain (G.15). Moreover, when ordering all the eigenvalues of $K_{\mathrm{ntk}}$ in the descending order and renaming them as $\{\sigma_j\}_{j \geq 1}$, similar to (G.16), we have

$$\sigma_j \leq C_2 \cdot (\widetilde{\rho}_{t-1} + \widetilde{\rho}_{t+1}) \qquad \text{for } \sum_{i=1}^{t-1} N(d,i) \leq j < \sum_{i=1}^{t} N(d,i). \tag{G.19}$$

Using a similar analysis, we can show that $\{\sigma_j\}_{j \geq 1}$ satisfy the $\gamma$-exponential eigenvalue decay condition with $\gamma = 1/d$. Therefore, we have shown that the NTK given in (G.17) satisfy Assumption 4.3.

# H   Proofs of the Supporting Lemmas

## H.1   Proof of Lemma E.1

*Proof.* For ease of presentation, before presenting the proof, we first define two operators $\mathbb{J}_h^\star$ and $\mathbb{J}_{t,h}$ respectively by letting
$$(\mathbb{J}_h^\star f)(x) = \langle f(x,\cdot), \pi_h^\star(\cdot \,|\, x) \rangle_{\mathcal{A}}, \quad (\mathbb{J}_{t,h} f)(x) = \langle f(x,\cdot), \pi_h^t(\cdot \,|\, x) \rangle_{\mathcal{A}}, \tag{H.1}$$
for any $(t,h) \in [T] \times [H]$ and any function $f : \mathcal{S} \times \mathcal{A} \to \mathbb{R}$. Moreover, for any $(t,h) \in [T] \times [H]$ and any state $x \in \mathcal{S}$, we define
$$\xi_h^t(x) = (\mathbb{J}_h Q_h^t)(x) - (\mathbb{J}_{t,h} Q_h^t)(x) = \langle Q_h^t(x,\cdot), \pi_h^\star(\cdot \,|\, x) - \pi_h^t(\cdot \,|\, x) \rangle_{\mathcal{A}}. \tag{H.2}$$
After introducing this notation, to prove (E.4) we decompose the instantaneous regret at the $t$-th episode into the following two terms,
$$V_1^\star(x_1^t) - V_1^{\pi^t}(x_1^t) = \underbrace{V_1^\star(x_1^t) - V_1^t(x_1^t)}_{(i)} + \underbrace{V_1^t(x_1^t) - V_1^{\pi^t}(x_1^t)}_{(ii)}. \tag{H.3}$$
In the sequel, we consider the two terms in (H.3) separately.

**Term (i).** By the definitions of the value function $V_h^\star$ in (2.2) and the operator $\mathbb{J}_h^\star$ in (H.1), we have $V_h^\star = \mathbb{J}_h^\star Q_h^\star$. Similarly, for all the algorithms, we have $V_h^t(x) = \langle Q_h^t(x,\cdot), \pi_h^t(\cdot \,|\, x) \rangle$ for all $x \in \mathcal{S}$. Thus, by the definition of $\mathbb{J}_{t,h}$ in (H.1), we have $V_h^t = \mathbb{J}_{t,h} Q_h^t$. Thus, using $\xi_h^t$ defined in (H.2), for any $(t,h) \in [T] \times [H]$, we have
$$V_h^\star - V_h^t = \mathbb{J}_h^\star Q_h^\star - \mathbb{J}_{t,h} Q_h^t = \left( \mathbb{J}_h^\star Q_h^\star - \mathbb{J}_h^\star Q_h^t \right) + \left( \mathbb{J}_h^\star Q_h^t - \mathbb{J}_{t,h} Q_h^t \right)$$
$$= \mathbb{J}_h^\star(Q_h^\star - Q_h^t) + \xi_h^t, \tag{H.4}$$
where the last equality follows from the definition of $\xi_h^t$ in (H.2) and the fact that $\mathbb{J}_h^\star$ is a linear operator. Moreover, by the definition of the temporal-difference error $\delta_h^t$ in (E.1) and the Bellman optimality condition, we have
$$Q_h^\star - Q_h^t = \left( r_h + \mathbb{P}_h V_{h+1}^\star \right) - \left( r_h + \mathbb{P}_h V_{h+1}^t - \delta_h^t \right) = \mathbb{P}_h(V_{h+1}^\star - V_{h+1}^t) + \delta_h^t. \tag{H.5}$$
Thus, combining (H.4) and (H.5), we obtain that
$$V_h^\star - V_h^t = \mathbb{J}_h^\star \mathbb{P}_h(V_{h+1}^\star - V_{h+1}^t) + \mathbb{J}_h^\star \delta_h^t + \xi_h^t, \qquad \forall (t,h) \in [T] \times [H]. \tag{H.6}$$
Equivalently, for all $x \in \mathcal{S}$, and all $(t,h) \in [T] \times [H]$, we have
$$V_h^\star(x) - V_h^t(x) = \mathbb{E}_{a \sim \pi_h^\star(\cdot \,|\, x)} \left\{ \mathbb{E} \left[ V_{h+1}^\star(x_{h+1}) - V_{h+1}^t(x_{h+1}) \,\middle|\, x_h = x, a_h = a \right] \right\}$$
$$+ \mathbb{E}_{a \sim \pi_h^\star(\cdot \,|\, x)} \left[ \delta_h^t(x,a) \right] + \xi_h^t(x).$$
Then, by recursively applying (H.6) for all $h \in [H]$, we have
$$V_1^\star - V_1^t = \left( \prod_{h=1}^{H} \mathbb{J}_h^\star \mathbb{P}_h \right) (V_{H+1}^\star - V_{H+1}^k) + \sum_{h=1}^{H} \left( \prod_{i=1}^{h-1} \mathbb{J}_i^\star \mathbb{P}_i \right) \mathbb{J}_h^\star \delta_h^t + \sum_{h=1}^{H} \left( \prod_{i=1}^{h-1} \mathbb{J}_i^\star \mathbb{P}_i \right) \xi_h^t. \tag{H.7}$$

Furthermore, notice that we have $V_{H+1}^\star = V_{H+1}^k = \mathbf{0}$. Thus, (H.7) can be equivalently written as

$$V_1^\star(x) - V_1^t(x) = \mathbb{E}_{\pi^\star}\left[\sum_{h=1}^H \langle Q_h^t(x_h, \cdot), \pi_h^\star(\cdot \,|\, x_h) - \pi_h^t(\cdot \,|\, x_h)\rangle_{\mathcal{A}} + \delta_h^t(x_h, a_h) \,\bigg|\, x_1 = x\right],$$

where we utilize the definition of $\xi_h^t$ given in (H.2). Thus, we can write Term (i) on the right-hand side of (H.3) as

$$V_1^\star(x_t^t) - V_1^t(x_t^t) = \sum_{h=1}^H \mathbb{E}_{\pi^\star}\left[\langle Q_h^t(x_h, \cdot), \pi_h^\star(\cdot \,|\, x_h) - \pi_h^t(\cdot \,|\, x_h)\rangle_{\mathcal{A}} \,\big|\, x_1 = x_t^t\right]$$

$$+ \sum_{h=1}^H \mathbb{E}_{\pi^\star}[\delta_h^t(x_h, a_h) \,|\, x_1 = x_t^t], \qquad \forall t \in [T]. \tag{H.8}$$

**Term (ii).** It remains to bound the second term on the right-hand side of (H.3). By the definition of the temporal-difference error $\delta_h^t$ in (E.1), for any $(t, h) \in [T] \times [H]$, we have

$$\delta_h^t(x_h^t, a_h^t) = r_h(x_h^t, a_h^t) + (\mathbb{P}_h V_{h+1}^t)(x_h^t, a_h^t) - Q_h^t(x_h^t, a_h^t)$$

$$= \left[r_h(x_h^t, a_h^t) + (\mathbb{P}_h V_{h+1}^t)(x_h^t, a_h^t) - Q_h^{\pi^t}(x_h^t, a_h^t)\right] + \left[Q_h^{\pi^t}(x_h^t, a_h^t) - Q_h^t(x_h^t, a_h^t)\right]$$

$$= \left(\mathbb{P}_h V_{h+1}^t - \mathbb{P}_h V_{h+1}^{\pi^t}\right)(x_h^t, a_h^t) + (Q_h^{\pi^t} - Q_h^t)(x_h^t, a_h^t), \tag{H.9}$$

where the last equality follows from the Bellman equation (2.1). Moreover, recall that we define $\zeta_{t,h}^1$ and $\zeta_{t,h}^2$ in (E.2) and (E.3), respectively. Thus, from (H.9) we obtain that

$$V_h^t(x_h^t) - V_h^{\pi^t}(x_h^t) \tag{H.10}$$

$$= V_h^t(x_h^t) - V_h^{\pi^t}(x_h^t) + (Q_h^{\pi^t} - Q_h^t)(x_h^t, a_h^t) + \left(\mathbb{P}_h(V_{h+1}^t - V_{h+1}^{\pi^t})\right)(x_h^t, a_h^t) - \delta_h^t(x_h^t, a_h^t),$$

$$= \left(V_h^t - V_h^{\pi^t}\right)(x_h^t) - (Q_h^t - Q_h^{\pi^t})(x_h^t, a_h^t)$$

$$+ \left(\mathbb{P}_h(V_{h+1}^t - V_{h+1}^{\pi^t})\right)(x_h^t, a_h^t) - (V_{h+1}^t - V_{h+1}^{\pi^t})(x_{h+1}^t) + (V_{h+1}^t - V_{h+1}^{\pi^t})(x_{h+1}^t) - \delta_h^t(x_h^t, a_h^t)$$

$$= \left[V_{h+1}^t(x_{h+1}^t) - V_{h+1}^{\pi^t}(x_{h+1}^t)\right] + \zeta_{t,h}^1 + \zeta_{t,h}^2 - \delta_h^t(x_h^t, a_h^t).$$

Thus, recursively applying (H.10) for all $h \in [H]$, we obtain that

$$V_1^t(x_1^t) - V_1^{\pi^t}(x_1^t) = V_{H+1}^t(x_{H+1}^k) - V_{H+1}^{\pi^k,k}(x_{H+1}^t) + \sum_{h=1}^H (\zeta_{t,h}^1 + \zeta_{t,h}^2) - \sum_{h=1}^H \delta_h^t(x_h^t, a_h^t)$$

$$= \sum_{h=1}^H (\zeta_{t,h}^1 + \zeta_{t,h}^2) - \sum_{h=1}^H \delta_h^t(x_h^t, a_h^t), \qquad \forall t \in [T], \tag{H.11}$$

where the last equality follows from the fact that $V_{H+1}^t(x_{H+1}^t) = V_{H+1}^{\pi^t}(x_{H+1}^t) = 0$. Thus, we have simplified Term (ii) defined in (H.3).

Thus, combining (H.3), (H.8), and (H.11), we obtain that

$$\text{Regret}(T) = \sum_{t=1}^T \left[V_1^\star(x_1^t) - V_1^{\pi^t}(x_1^t)\right]$$

$$= \sum_{t=1}^T \sum_{h=1}^H \mathbb{E}_{\pi^\star}[\delta_h^t(x_h, a_h) \,|\, x_1 = x_t^t] + \sum_{t=1}^T \sum_{h=1}^H (\zeta_{t,h}^1 + \zeta_{t,h}^2) - \sum_{t=1}^T \sum_{h=1}^H \delta_h^t(x_h^t, a_h^t)$$

$$+ \sum_{t=1}^T \sum_{h=1}^H \mathbb{E}_{\pi^\star}\left[\langle Q_h^t(x_h, \cdot), \pi_h^\star(\cdot \,|\, x_h) - \pi_h^t(\cdot \,|\, x_h)\rangle_{\mathcal{A}} \,\big|\, x_1 = x_t^t\right].$$

Therefore, we conclude the proof of this lemma. $\qquad\square$

## H.2   Proof of Lemma E.2

*Proof.* For ease of presentation, we utilize the feature representation induced by the kernel $K$. Let $\phi\colon \mathcal{Z} \to \mathcal{H}$ be the feature mapping such that $K(z, z') = \langle \phi(z), \phi(z')\rangle_{\mathcal{H}}$. For simplicity, we formally

view $\phi(z)$ as a vector and write $\langle\phi(z),\phi(z')\rangle_{\mathcal{H}} = \phi(z)^\top\phi(z')$. Then, any function $f\colon \mathcal{Z} \to \mathbb{R}$ in the RKHS satisfies $f(z) = \langle\phi(z), f\rangle_{\mathcal{H}} = f^\top\phi(z)$. Using the feature representation, we can rewrite the kernel ridge regression in (3.4) as

$$\underset{\theta\in\mathcal{H}}{\text{minimize}}\, L(\theta) = \sum_{\tau=1}^{t-1}\big[r_h(x_h^\tau, a_h^\tau) + V_{h+1}^t(x_{h+1}^\tau) - \langle\phi(x_h^\tau, a_h^\tau), \theta\rangle_{\mathcal{H}}\big]^2 + \lambda\cdot\|\theta\|_{\mathcal{H}}^2. \tag{H.12}$$

We define the feature matrix $\Phi_h^t\colon \mathcal{H} \to \mathbb{R}^{t-1}$ and "covariance matrix" $\Lambda_h^t\colon \mathcal{H} \to \mathcal{H}$ respectively as

$$\Phi_h^t = \big[\phi(z_h^1)^\top, \ldots, \phi(z_h^{t-1})^\top\big]^\top, \qquad \Lambda_h^t = \sum_{\tau=1}^{t-1}\phi(z_h^\tau)\phi(z_h^\tau)^\top + \lambda\cdot I_{\mathcal{H}} = \lambda\cdot I_{\mathcal{H}} + (\Phi_h^t)^\top\Phi_h^t,$$

$$\tag{H.13}$$

where $I_{\mathcal{H}}$ is the identity mapping on $\mathcal{H}$. Thus, the Gram matrix $K_h^t$ in (3.7) is equal to $\Phi_h^t(\Phi_h^t)^\top$. More specifically, here $\Lambda_h^t$ is a self-adjoint and positive-definite operator. For any $f_1, f_2 \in \mathcal{H}$, we denote

$$\Lambda_h^t f_1 = \lambda\cdot f_1 + \sum_{\tau=1}^{t-1}\phi(z_h^\tau)\cdot f_1(x_h^\tau) \in \mathcal{H}, \qquad f_1^\top\Lambda_h^t f_2 = \langle f_1, \Lambda_h^t f\rangle_{\mathcal{H}}.$$

It is not hard to see that all the eigenvalues of $\Lambda_h^t$ are positive and at least $\lambda$. Thus, the inverse operator of $\Lambda_h^t$, denoted by $(\Lambda_h^t)^{-1}$, is well-defined, which is also a self-adjoint and positive-definite operator on $\mathcal{H}$. Similarly, for any $f_1, f_2 \in \mathcal{H}$, we let $f_1^\top(\Lambda_h^t)^{-1}f_2$ denote $\langle f_1, (\Lambda_h^t)^{-1}f_2\rangle_{\mathcal{H}}$. The eigenvalues of $(\Lambda_h^t)^{-1}$ are all bounded in interval $[0, 1/\lambda]$.

In addition, using the feature matrix $\Phi_h^t$ defined in (H.13) and $y_h^t$ defined in (3.6), we can write (H.12) as

$$\underset{\theta\in\mathcal{H}}{\text{minimize}}\, L(\theta) = \|y_h^t - \Phi_h^t\theta\|_2^2 + \lambda\cdot\theta^\top\theta,$$

whose solution is given by $\widehat{\theta}_h^t = (\Lambda_h^t)^{-1}(\Phi_h^t)^\top y_h^t$. and $\widehat{Q}_h^t$ in (3.4) satisfies $\widehat{Q}_h^t(z) = \phi(z)^\top\widehat{\theta}_h^t$.

In the sequel, to further simplify the notation, we let $\Phi$ denote $\Phi_h^t$ when its meaning is clear from the context. Since both $(\Phi\Phi^\top + \lambda\cdot I)$ and $(\Phi^\top\Phi + \lambda\cdot I_{\mathcal{H}})$ are strictly positive definite and

$$(\Phi^\top\Phi + \lambda\cdot I_{\mathcal{H}})\Phi^\top = \Phi^\top(\Phi\Phi^\top + \lambda\cdot I),$$

which implies that

$$(\Lambda_h^t)^{-1}\Phi^\top = (\Phi\Phi^\top + \lambda\cdot I_{\mathcal{H}})^{-1}\Phi^\top = \Phi^\top(\Phi\Phi^\top + \lambda\cdot I)^{-1} = \Phi^\top(K_h^t + \lambda\cdot I)^{-1}. \tag{H.14}$$

Here $I$ is the identity matrix in $\mathbb{R}^{(t-1)\times(t-1)}$. Thus, by (H.14) we have

$$\widehat{\theta}_h^t = (\Lambda_h^t)^{-1}\Phi^\top y_h^t = \Phi^\top(K_h^t + \lambda\cdot I)^{-1}y_h^t = \Phi^\top\alpha_h^t. \tag{H.15}$$

Moreover, $k_h^t$ defined in (3.7) can be written as $k_h^t(z) = \Phi\phi(z)$, which, combined with (H.14), implies

$$\begin{aligned}\phi(z) &= (\Lambda_h^t)^{-1}\Lambda_h^t\phi(z) = (\Lambda_h^t)^{-1}(\Phi^\top\Phi + \lambda\cdot I_{\mathcal{H}})\phi(z)\\ &= (\Lambda_h^t)^{-1}(\Phi^\top\Phi)\phi(z) + \lambda\cdot(\Lambda_h^t)^{-1}\phi(z)\\ &= \Phi^\top(K_h^t + \lambda\cdot I)^{-1}k_h^t(z) + \lambda\cdot(\Lambda_h^t)^{-1}\phi(z).\end{aligned} \tag{H.16}$$

Thus, we can write $\|\phi(z)\|_{\mathcal{H}}^2 = \phi(z)^\top\phi(z)$ as

$$\begin{aligned}\|\phi(z)\|_{\mathcal{H}}^2 &= \phi(z)^\top\cdot\big[\Phi^\top(K_h^t + \lambda\cdot I)^{-1}k_h^t(z) + \lambda\cdot(\Lambda_h^t)^{-1}\phi(z)\big]\\ &= k_h^t(z)^\top(K_h^t + \lambda\cdot I)^{-1}k_h^t(z) + \lambda\cdot\phi(z)(\Lambda_h^t)^{-1}\phi(z),\end{aligned}$$

which implies that we can equivalently write the bonus $b_h^t$ defined in (3.8) as

$$b_h^t(x, a) = \big[\phi(x, a)^\top(\Lambda_h^t)^{-1}\phi(x, a)\big]^{1/2} = \|\phi(x, a)\|_{(\Lambda_h^t)^{-1}}. \tag{H.17}$$

Combining (H.15) and (H.17), we equivalently write $Q_h^t$ in (3.5) as

$$\begin{aligned}Q_h^t(x, a) &= \min\big\{\widehat{Q}_h^t(x, a) + \beta\cdot b_h^t(x, a), H - h + 1\big\}^+\\ &= \min\big\{\phi(x, a)^\top\widehat{\theta}_h^t + \beta\cdot\|\phi(x, a)\|_{(\Lambda_h^t)^{-1}}, H - h + 1\big\}^+.\end{aligned} \tag{H.18}$$

Now we are ready to bound the temporal-difference error $\xi_h^t$ defined in (E.1). Noticing that $V_h^t(x) = \max_a Q_h^t(x,a)$ for all $(t,h) \in [T] \times [H]$, we have
$$\delta_h^t = r_h + \mathbb{P}_h V_{h+1}^t - Q_h^t = \mathbb{T}_h^\star Q_{h+1}^t - Q_h^t,$$
where $\mathbb{T}_h^\star$ is the Bellman optimality operator. Under the Assumption 4.1, for all $(t,h) \in [T] \times [H]$, since $Q_{h+1}^t \in [0,H]$, we have $\mathbb{T}_h^\star Q_{h+1}^t \in \mathcal{Q}^\star$. Using the feature representation of RKHS, there exists $\overline{\theta}_h^t \in \mathcal{Q}^\star$ such that $(\mathbb{T}_h^\star Q_{h+1}^t)(z) = \phi(z)^\top \overline{\theta}_h^t$ for all $z \in \mathcal{Z}$.

In the sequel, we consider the difference between $\phi(z)^\top \widehat{\theta}_h^t$ and $\phi(z)^\top \overline{\theta}_h^t$. To begin with, using (H.16), we can write $\phi(z)^\top \overline{\theta}_h^t$ as
$$\phi(z)^\top \overline{\theta}_h^t = k_h^t(z)^\top (K_h^t + \lambda \cdot I)^{-1}\Phi\overline{\theta}_h^t + \lambda \cdot \phi(z)^\top (\Lambda_h^t)^{-1}\overline{\theta}_h^t. \tag{H.19}$$
Hence, combining (H.15) and (H.19), we have
$$\phi(z)^\top \widehat{\theta}_h^t - \phi(z)^\top \overline{\theta}_h^t = \underbrace{k_h^t(z)^\top (K_h^t + \lambda \cdot I)^{-1}\big(y_h^t - \Phi\overline{\theta}_h^t\big)}_{(i)} - \underbrace{\lambda \cdot \phi(z)^\top (\Lambda_h^t)^{-1}\overline{\theta}_h^t}_{(ii)}. \tag{H.20}$$

We bound Term (i) and Term (ii) on the right-hand side of (H.20) separately. For Term (ii), by the Cauchy-Schwarz inequality, we have
$$\big|\lambda \cdot \phi(z)^\top (\Lambda_h^t)^{-1}\overline{\theta}_h^t\big| \le \big\|\lambda \cdot (\Lambda_h^t)^{-1}\phi(x)\big\|_{\mathcal{H}} \cdot \|\overline{\theta}_h^t\|_{\mathcal{H}} \le R_Q H \cdot \big\|\lambda \cdot (\Lambda_h^t)^{-1}\phi(x)\big\|_{\mathcal{H}} \tag{H.21}$$
$$= R_Q H \cdot \sqrt{\lambda \cdot \phi(z)^\top (\Lambda_h^t)^{-1} \cdot \lambda \cdot I_{\mathcal{H}} \cdot (\Lambda_h^t)^{-1}\phi(x)}$$
$$\le R_Q H \cdot \sqrt{\lambda \cdot \phi(z)(\Lambda_h^t)^{-1} \cdot \Lambda_h^t \cdot (\Lambda_h^t)^{-1}\phi(z)} = \sqrt{\lambda} R_Q H \cdot b_h^t(z).$$
Here the first inequality follows from the Cauchy-Schwarz inequality and the second inequality follows from the fact that $\overline{\theta}_h^t \in \mathcal{Q}^\star$, which implies that $\|\overline{\theta}_h^t\|_{\mathcal{H}} \le R_Q H$. Moreover, the last inequality follows from the fact that $\Lambda_h^t - \lambda \cdot I_{\mathcal{H}}$ is a self-adjoint and positive-semidefinite operator, which means that $f^\top (\Lambda_h^t - \lambda \cdot I_{\mathcal{H}})f \ge 0$ for all $f \in \mathcal{H}$, and the last equality follows from (H.17).

Furthermore, for Term (i), by the Bellman equation in (2.2) and the definition of $y_h^t$ in (3.6), for any $\tau \in [t-1]$, the $\tau$-th entry of $(y_h^t - \Phi\overline{\theta}_h^t)$ can be written as
$$[y_h^t]_\tau - [\Phi\overline{\theta}_h^t]_\tau = r_h(x_h^\tau, a_h^\tau) + V_{h+1}^t(x_{h+1}^\tau) - \phi(x_h^\tau, a_h^\tau)^\top \overline{\theta}_h^t$$
$$= r_h(x_h^\tau, a_h^\tau) + V_{h+1}^t(x_{h+1}^\tau) - (\mathbb{T}_h^\star Q_{h+1}^t)(x_h^\tau, a_h^\tau)$$
$$= V_{h+1}^t(x_{h+1}^\tau) - (\mathbb{P}_h V_{h+1}^t)(x_h^\tau, a_h^\tau). \tag{H.22}$$
Thus, combining (H.14), (H.20), and (H.22) we have
$$\big|k_h^t(z)^\top (K_h^t + \lambda \cdot I)^{-1}\big(y_h^t - \Phi\overline{\theta}_h^t\big)\big|$$
$$= \left|\phi(z)^\top (\Lambda_h^t)^{-1}\left\{\sum_{\tau=1}^{t-1}\phi(x_h^\tau, a_h^\tau) \cdot \big[V_{h+1}^k(x_{h+1}^\tau) - (\mathbb{P}_h V_{h+1}^k)(x_h^\tau, a_h^\tau)\big]\right\}\right|$$
$$\le \|\phi(z)\|_{(\Lambda_h^t)^{-1}} \cdot \left\|\sum_{\tau=1}^{t-1}\phi(x_h^\tau, a_h^\tau) \cdot \big[V_{h+1}^t(x_{h+1}^\tau) - (\mathbb{P}_h V_{h+1}^t)(x_h^\tau, a_h^\tau)\big]\right\|_{(\Lambda_h^t)^{-1}}, \tag{H.23}$$
where the last inequality follows from the Cauchy-Schwarz inequality. In the following, we aim to bound (H.23) by the concentration of self-normalized stochastic processes in the RKHS. However, here $V_{h+1}^t$ depends on the historical data in the first $(t-1)$ episodes and is thus not independent of $\{(x_h^\tau, a_h^\tau, x_{h+1}^\tau)\}_{\tau \in [t-1]}$. To bypass this challenge, in the sequel, we combine the concentration of self-normalized processes and uniform convergence over the function classes that contain each $V_{h+1}^t$.

Specifically, recall that we define function classes $\mathcal{Q}_{\text{ucb}}(h, R, B)$ in (4.4) for any $h \in [H]$, and any $R, B > 0$. We define $\mathcal{V}_{\text{ucb}}(h, R, B)$ as
$$\mathcal{V}_{\text{ucb}}(h, R, B) = \big\{V : V(\cdot) = \max_{a \in \mathcal{A}} Q(\cdot, a) \text{ for some } Q \in \mathcal{Q}_{\text{ucb}}(h, R, B)\big\}. \tag{H.24}$$

In the following, we find a parameter $R_T$ such that $V_h^t \in \mathcal{V}_{\text{ucb}}(h, R_T, B_T)$ holds for all $h \in [H]$ and $t \in [T]$, where $B_T$ is specified in (4.5). Here both $R_T$ and $B_T$ depend on $T$. By (4.4) and (H.18), it

suffices to set $R_T$ as an upper bound of $\|\widehat{\theta}_h^t\|_{\mathcal{H}}$ for all $(t, h) \in [T] \times [H]$. In the following lemma, we bound the RKHS norm of each $\widehat{\theta}_h^t$.

**Lemma H.1** (RKHS Norm of $\widehat{\theta}_h^t$). When $\lambda \geq 1$, for any $(t, h) \in [T] \times [H]$, $\widehat{\theta}_h^t$ defined in (H.15) satisfies

$$\left\|\widehat{\theta}_h^t\right\|_{\mathcal{H}} \leq H\sqrt{2/\lambda \cdot \mathrm{logdet}(I + K_h^t/\lambda)} \leq 2H\sqrt{\Gamma_K(T, \lambda)},$$

where $K_h^t$ is defined in (3.7) and $\Gamma_K(T, \lambda)$ is defined in (I.16).

*Proof.* See §J.1 for a detailed proof. $\square$

By this lemma, in the sequel, we set $R_T = 2H\sqrt{\Gamma_K(T, \lambda)}$. To conclude the proof, we show that the sum of the two terms in (H.20) is bounded by $\beta \cdot \|\phi(z)\|_{(\Lambda_h^t)^{-1}}$, where we set $\beta = B_T$. To this end, for any two value functions $V, V' \colon \mathcal{S} \to \mathbb{R}$, we define their distance as $\mathrm{dist}(V, V') = \sup_{x \in \mathcal{S}} |V(x) - V'(x)|$. For any $\epsilon \in (0, 1/e)$, any $B > 0$, and any $h \in [H]$, we let $N_{\mathrm{dist}}(\epsilon; h, B)$ be the $\epsilon$-covering number of $\mathcal{V}_{\mathrm{ucb}}(h, R_T, B)$ with respect to distance $\mathrm{dist}(\cdot, \cdot)$. Recall that we define $N_\infty(\epsilon; h, B)$ as the $\epsilon$-covering number of $\mathcal{Q}_{\mathrm{ucb}}(h, R_T, B)$ with respect to the $\ell_\infty$-norm on $\mathcal{Z}$. Note that for any $Q, Q' \colon \mathcal{Z} \to \mathbb{R}$, we have

$$\sup_{x \in \mathcal{S}}\left|\max_{a \in \mathcal{A}} Q(x, a) - \max_{a \in \mathcal{A}} Q'(x, a)\right| \leq \sup_{(x,a) \in \mathcal{S} \times \mathcal{A}} |Q(x, a) - Q'(x, a)| = \|Q - Q'\|_\infty.$$

By (H.24) we have $N_{\mathrm{dist}}(\epsilon; h, B) \leq N_\infty(\epsilon; h, B)$. Then, by applying Lemma J.2 with $\delta = (2T^2H^3)^{-1}$ and taking a union bound over $h \in [H]$, we obtain that

$$\left\|\sum_{\tau=1}^{t-1} \phi(x_h^\tau, a_h^\tau) \cdot \left[V_{h+1}^t(x_{h+1}^\tau) - (\mathbb{P}_h V_{h+1}^t)(x_h^\tau, a_h^\tau)\right]\right\|_{(\Lambda_h^t)^{-1}}^2$$

$$\leq \sup_{V \in \mathcal{V}_{\mathrm{ucb}}(h+1, R_T, B_T)} \left\|\sum_{\tau=1}^{t-1} \phi(x_h^\tau, a_h^\tau) \cdot \left[V(x_{h+1}^\tau) - (\mathbb{P}_h V)(x_h^\tau, a_h^\tau)\right]\right\|_{(\Lambda_h^t)^{-1}}^2$$

$$\leq 2H^2 \cdot \mathrm{logdet}(I + K_h^t/\lambda) + 2H^2 t \cdot (\lambda - 1) + 8t^2\epsilon^2/\lambda$$

$$\qquad + 4H^2 \cdot \left[\log N_\infty(\epsilon; h+1, B_T) + \log(2T^2H^3)\right] \tag{H.25}$$

holds uniformly for all $(t, h) \in [T] \times [H]$ with probability at least $1 - (2T^2H^2)^{-2}$, where we utilize the fact that $V_{h+1}^t \in \mathcal{V}_{\mathrm{ucb}}(h+1, R_T, B_T)$. Note that we set $\lambda = 1 + 1/T$. Then, setting $\epsilon$ as $\epsilon^* = H/T$, (H.25) is further reduced to

$$\left\|\sum_{\tau=1}^{t-1} \phi(x_h^\tau, a_h^\tau) \cdot \left[V_{h+1}^t(x_{h+1}^\tau) - (\mathbb{P}_h V_{h+1}^t)(x_h^\tau, a_h^\tau)\right]\right\|_{(\Lambda_h^t)^{-1}}^2$$

$$\leq 4H^2 \cdot \Gamma_K(T, \lambda) + 11H^2 + 4H^2 \cdot \log N_\infty(\epsilon^*; h+1, B_T) + 8H^2 \cdot \log(TH). \tag{H.26}$$

Thus, combining (H.17), (H.20), (H.21), (H.23), and (H.26), we obtain that

$$\left|\phi(z)^\top(\widehat{\theta}_h^t - \overline{\theta}_h^t)\right|$$

$$\leq H \cdot \left\{\left[4 \cdot \Gamma_K(T, \lambda) + 4 \cdot \log N_\infty(\epsilon^*; h+1, B_T) + 8 \cdot \log(TH) + 11\right]^{1/2} + \sqrt{\lambda}R_Q\right\} \cdot b_h^t(z)$$

$$\leq H \cdot \left[8 \cdot \Gamma_K(T, \lambda) + 8 \cdot \log N_\infty(\epsilon^*; h+1, B_T) + 16 \cdot \log(TH) + 22 + 2R_Q^2\lambda\right]^{1/2} \cdot b_h^t(z)$$

$$\leq B_T \cdot b_h^t(z) = \beta \cdot b_h^t(z) \tag{H.27}$$

holds uniformly for all $(t, h) \in [T] \times [H]$ with probability at least $1 - (2T^2H^2)^{-1}$, where the second inequality follows from the elementary inequality $\sqrt{a} + \sqrt{b} \leq \sqrt{2(a^2 + b^2)}$, and the last inequality follows from the assumption on $B_T$ given in (4.5).

Finally, by (H.27) and the definition of the temporal-difference error $\delta_h^t$ in (E.1), we have

$$-\delta_h^t(z) = Q_h^t(z) - \phi(z)^\top \overline{\theta}_h^t \leq \phi(z)^\top(\widehat{\theta}_h^t - \overline{\theta}_h^t) + \beta \cdot b_h^t(z) \leq 2\beta \cdot b_h^t(z). \tag{H.28}$$

In addition, since $Q_{h+1}^t(z) \le H - h$ for all $z \in \mathcal{Z}$, we have $(\mathbb{T}_h^\star Q_{h+1}^t) \le H - h + 1$. Hence, we have

$$\delta_h^t(z) = \phi(z)^\top \overline{\theta}_h^t - \min\big\{\phi(z)^\top \widehat{\theta}_h^t + \beta \cdot b_h^t(z), H - h + 1\big\}^+$$

$$\le \max\big\{\phi(z)^\top \overline{\theta}_h^t - \phi(z)^\top \widehat{\theta}_h^t - \beta \cdot b_h^t(z), \phi(z)^\top \overline{\theta}_h^t - (H - h + 1)\big\} \le 0. \qquad \text{(H.29)}$$

Therefore, combining (H.28) and (H.29), we conclude the proof of Lemma E.2. $\qquad\square$

## H.3 Proof of Lemma E.3

*Proof.* Following [12], we prove this lemma by showing that $\{\zeta_{t,h}^1, \zeta_{t,h}^2\}_{(t,h)\in[T]\times[H]}$ can be written as a bounded martingale difference sequence with respect to a filtration. In particular, we construct the filtration explicitly as follows. For any $(t, h) \in [T] \times [H]$, we define $\sigma$-algebras $\mathcal{F}_{t,h,1}$ and $\mathcal{F}_{t,h,2}$ as follows:

$$\mathcal{F}_{t,h,1} = \sigma\big(\{(x_i^\tau, a_i^\tau)\}_{(\tau,i)\in[t-1]\times[H]} \cup \{(x_i^t, a_i^t)\}_{i\in[h]}\big),$$
$$\mathcal{F}_{t,h,2} = \sigma\big(\{(x_i^\tau, a_i^\tau)\}_{(\tau,i)\in[t-1]\times[H]} \cup \{(x_i^t, a_i^t)\}_{i\in[h]} \cup \{x_{h+1}^t\}\big), \qquad \text{(H.30)}$$

where $\sigma(\cdot)$ denotes the $\sigma$-algebra generated by a finite set. Moreover, for any $t \in [T]$, $h \in [H]$ and $m \in [2]$, we define the timestep index $\tau(t, h, m)$ as

$$\tau(t, h, m) = (t - 1) \cdot 2H + (h - 1) \cdot 2 + m, \qquad \text{(H.31)}$$

which offers an partial ordering over the triplets $(t, h, m) \in [T] \times [H] \times [2]$. Moreover, by the definitions in (H.30), for any $(t, h, m)$ and $(t', h', m')$ satisfying $\tau(k, h, m) \le \tau(k', h', m')$, it holds that $\mathcal{F}_{k,h,m} \subseteq \mathcal{F}_{k',h',m'}$. Thus, the sequence of $\sigma$-algebras $\{\mathcal{F}_{t,h,m}\}_{(t,h,m)\in[T]\times[H]\times[2]}$ forms a filtration.

Furthermore, for any $(t, h) \in [T] \times [H]$, since both $Q_h^t$ and $V_h^t$ are obtained based on the trajectories of the first $(t - 1)$ episodes, they are both measurable with respect to $\mathcal{F}_{t,1,1}$, which is a subset of $\mathcal{F}_{t,h,m}$ for all $h \in [H]$ and $m \in [2]$. Thus, by (H.30), $\zeta_{t,h}^1$ defined in (E.2) and $\zeta_{t,h}^2$ defined in (E.3) are measurable with respect to $\mathcal{F}_{t,h,1}$ and $\mathcal{F}_{t,h,2}$, respectively. In addition, note that $a_h^t \sim \pi_h^t(\cdot \,|\, x_h^t)$ and that $x_{h+1}^t \sim \mathbb{P}_h(\cdot \,|\, x_h^t, a_h^t)$. Thus, we have

$$\mathbb{E}\big[\zeta_{t,h}^1 \,\big|\, \mathcal{F}_{t,h-1,2}\big] = 0, \qquad \mathbb{E}\big[\zeta_{t,h}^2 \,\big|\, \mathcal{F}_{t,h,1}\big] = 0, \qquad \text{(H.32)}$$

where we identify $\mathcal{F}_{t,0,2}$ with $\mathcal{F}_{t-1,H,2}$ for all $t \ge 2$ and let $\mathcal{F}_{1,0,2}$ be the empty set. Combining (H.31) and (H.32), we can define a martingale $\{M_{t,h,m}\}_{(t,h,m)\in[T]\times[H]\times[2]}$ indexed by $\tau(t, k, m)$, defined in (H.31), as follows. For any $(t, h, m) \in [T] \times [H] \times [2]$, we define

$$M_{t,h,m} = \bigg\{ \sum_{(s,g,\ell)} \zeta_{s,g}^\ell : \tau(s, g, \ell) \le \tau(t, h, m) \bigg\}; \qquad \text{(H.33)}$$

that is, $M_{t,h,m}$ is the sum of all terms of the form $\zeta_{s,g}^\ell$ defined in (E.2) or (E.3) such that its timestep index $\tau(s, g, \ell)$ is no greater than $\tau(t, h, m)$. By definition, we have

$$M_{K,H,2} = \sum_{t=1}^T \sum_{h=1}^H (\zeta_{t,h}^1 + \zeta_{t,h}^2). \qquad \text{(H.34)}$$

Moreover, since $V_h^t, Q_h^t, V_h^{\pi^t}$, and $Q_h^{\pi^t}$ all takes values in $[0, H]$, we have $|\zeta_{t,h}^1| \le 2H$ and $|\zeta_{t,h}^2| \le 2H$ for all $(t, h) \in [T] \times [H]$. This means that the martingale $M_{t,h,m}$ defined in (H.33) has uniformly bounded differences. Thus, applying the Azuma-Hoeffding inequality [7] to $M_{T,H,2}$ in (H.34), we obtain that

$$\mathbb{P}\bigg(\bigg| \sum_{t=1}^T \sum_{h=1}^H (\zeta_{t,h}^1 + \zeta_{t,h}^2) \bigg| > t\bigg) \le 2\exp\bigg(\frac{-t^2}{16TH^3}\bigg) \qquad \text{(H.35)}$$

holds for all $t > 0$. Finally, we set the right-hand side of (H.35) to $\zeta$ for some $\zeta \in (0, 1)$, which yields $t = \sqrt{16TH^3 \cdot \log(2/\zeta)}$. Thus, we obtain that

$$\bigg| \sum_{t=1}^T \sum_{h=1}^H (\zeta_{t,h}^1 + \zeta_{t,h}^2) \bigg| \le \sqrt{16TH^3 \cdot \log(2/\zeta)},$$

with probability at least $1 - \zeta$, which concludes the proof. $\qquad\square$

## H.4 Proof of Lemma E.4

*Proof.* The proof of this lemma utilizes the connection between overparameterized neural networks and NTKs. Recall that we denote $z = (x, a)$ and $\mathcal{Z} = \mathcal{S} \times \mathcal{A}$. Also recall that $(b^{(0)}, W^{(0)})$ is the initial value of the network parameters obtained by the symmetric initialization scheme introduced in §B.2. Thus, $f(\cdot; W^{(0)})$ is a zero function. For any $(t, h) \in [T] \times [H]$, since $\widehat{W}_h^t$ is the global minimizer of loss function $L_h^t$ defined in (C.1), we have

$$L_h^t(\widehat{W}_h^t) = \sum_{\tau=1}^{t-1} \big[r_h(x_h^\tau, a_h^\tau) + V_{h+1}^t(x_{h+1}^\tau) - f(x_h^\tau, a_h^\tau; \widehat{W}_h^t)\big]^2 + \lambda \cdot \big\|\widehat{W}_h^t - W^{(0)}\big\|_2^2$$

$$\leq L_h^t(W^{(0)}) = \sum_{\tau=1}^{t-1} \big[r_h(x_h^\tau, a_h^\tau) + V_{h+1}^t(x_{h+1}^\tau)\big]^2 \leq (H - h + 1)^2 \cdot (t - 1) \leq TH^2,$$

(H.36)

where the second-to-last inequality follows from the facts that $V_{h+1}^t$ is bounded by $H - h$ and that $r_h \in [0, 1]$. Thus, (H.36) implies that

$$\big\|\widehat{W}_h^t - W^{(0)}\big\|_2^2 \leq TH^2/\lambda, \qquad \forall(t, h) \in [T] \times [H]. \tag{H.37}$$

That is, each $\widehat{W}_h^t$ belongs to the Euclidean ball $\mathcal{B} = \{W \in \mathbb{R}^{2md} \colon \|W - W^{(0)}\|_2 \leq H\sqrt{T/\lambda}\}$. Here the regularization parameter $\lambda$ is does not depend on $m$ and will be determined later. Notice that the radius of $\mathcal{B}$ does not depend on $m$. When $m$ is sufficiently large, it can be shown that $f(\cdot, W)$ is close to its linearization, $\widehat{f}(\cdot; W) = \langle \varphi(\cdot; W^{(0)}), W - W^{(0)} \rangle$, for all $W \in \mathcal{B}$, where $\varphi(\cdot; W) = \nabla_W f(\cdot; W)$.

Furthermore, recall that the temporal-difference error $\delta_h^t$ is defined as

$$\delta_h^t = r_h + \mathbb{P}_h V_{h+1}^t - Q_h^t = \mathbb{T}_h^\star Q_{h+1}^t - Q_h^t.$$

Under Assumption D.1, we have $\mathbb{T}_h^\star Q_{h+1}^t \in \mathcal{Q}^\star$ for all $(t, h) \in [T] \times [H]$, where $\mathcal{Q}^\star$ is defined in (D.1). That is, for all $(t, h) \in [T] \times [H]$, there exists a function $\alpha_h^t \colon \mathbb{R}^d \to \mathbb{R}^d$ such that

$$(\mathbb{T}_h^\star Q_{h+1}^t)(z) = \int_{\mathbb{R}^d} \mathrm{act}'(w^\top z) \cdot z^\top \alpha_h^t(w) \, \mathrm{d}p_0(w), \qquad \forall(t, h) \in [T] \times [H], \forall z \in \mathcal{Z}. \tag{H.38}$$

Moreover, it holds that $\|\alpha_h^t\|_{2,\infty} = \sup_w \|\alpha_h^t(w)\|_2 \leq R_Q H/\sqrt{d}$.

Now we are ready to bound the temporal-difference error $\delta_h^t$ defined in (E.1). Our proof is decomposed into three steps.

**Step I.** In the first step, we show that, with high probability, $\mathbb{T}_h^\star Q_{h+1}^t$ can be well-approximated by the class of linear functions of $\varphi(\cdot; W^{(0)})$ with respect to the $\ell_\infty$-norm.

Specifically, by Proposition C.1 in [30], with probability at least $1 - m^{-2}$ over the randomness of initialization, for any $(t, h) \in [T] \times [H]$, there exists a function $\widetilde{Q}_h^t \colon \mathcal{Z} \to \mathbb{R}$ that can be written as

$$\widetilde{Q}_h^t(z) = \frac{1}{\sqrt{m}} \sum_{j=1}^m \mathrm{act}'\big(\langle W_j^{(0)}, z \rangle\big) \cdot z^\top \alpha_j, \tag{H.39}$$

where $\|\alpha_j\|_2 \leq R_Q/\sqrt{dm}$ for all $j \in [m]$ and $\{W_j^{(0)}\}_{j \in [2m]}$ are the random weights generated in the symmetric initialization scheme. Moreover, $\widetilde{Q}_h^t$ satisfies that

$$\|\widetilde{Q}_h^t - \mathbb{T}_h^\star Q_{h+1}^t\|_\infty \leq 10 C_{\mathrm{act}} R_Q H \cdot \sqrt{\log(mTH)/m}. \tag{H.40}$$

Also, for any $j \in [2m]$, let $W_j^{(0)}$ and $b_j^{(0)}$ be the $j$-th component of $b^{(0)}$ and $W^{(0)}$, respectively.

Now we show that $\widetilde{Q}_h^t$ in (H.39) can be written as $\varphi(\cdot; W^{(0)})^\top (\widetilde{W}_h^t - W^{(0)})$ for some $\widetilde{W}_h^t \in \mathbb{R}^{2md}$. To this end, we define $\widetilde{W}_h^t = (\widetilde{W}_1, \dots \widetilde{W}_{2m}) \in \mathbb{R}^{2md}$ as follows. For any $j \in [m]$, we let $\widetilde{W}_j = W_j^{(0)} + b_j^{(0)} \cdot \alpha_j/\sqrt{2}$, and for any $j \in \{m+1, \dots, 2m\}$, we let $\widetilde{W}_j = W_j^{(0)} + b_j^{(0)} \cdot \alpha_{j-m}/\sqrt{2}$.

Then, by the symmetric initialization scheme, we have

$$
\begin{aligned}
\widetilde{Q}_h^t(z) &= \frac{1}{\sqrt{2m}} \sum_{j=1}^{m} \sqrt{2} \cdot (b_j^{(0)})^2 \cdot \mathrm{act}'\big(\langle W_j^{(0)}, z \rangle\big) \cdot z^\top \alpha_j \\
&= \frac{1}{\sqrt{2m}} \sum_{j=1}^{m} 1/\sqrt{2} \cdot (b_j^{(0)})^2 \cdot \mathrm{act}'\big(\langle W_j^{(0)}, z \rangle\big) \cdot z^\top \alpha_j \\
&\qquad + \frac{1}{\sqrt{2m}} \sum_{j=1}^{m} 1/\sqrt{2} \cdot (b_j^{(0)})^2 \cdot \mathrm{act}'\big(\langle W_j^{(0)}, z \rangle\big) \cdot z^\top \alpha_{j-m} \\
&= \frac{1}{\sqrt{2m}} \sum_{j=1}^{2m} b_j^{(0)} \cdot \mathrm{act}'\big(\langle W_j^{(0)}, z \rangle\big) \cdot z^\top \big(\widetilde{W}_j - W_j^{(0)}\big) = \varphi(z; W^{(0)})^\top \big(\widetilde{W}_h^t - W^{(0)}\big).
\end{aligned}
$$
(H.41)

Moreover, since $\|\alpha_j\|_2 \le R_Q H/\sqrt{dm}$, we have $\|\widetilde{W}_h^t - W^{(0)}\|_2 \le R_Q H/\sqrt{d}$.

Therefore, for all $(t, h) \in [T] \times [H]$, we have constructed $\widetilde{Q}_h^t$ to be linear in $\varphi(\cdot; W^{(0)})$. Moreover, with probability at least $1 - m^{-2}$ over the randomness of initialization, $\widetilde{Q}_h^t$ is close to $\mathbb{T}_h^\star Q_{h+1}^t$ in the sense that (H.40) holds uniformly for all $(t, h) \in [T] \times [H]$. Thus, we conclude the first step.

**Step II.** In the second step, we show that $Q_h^t$ used in Algorithm 4 can be well approximated by functions based on the feature mapping $\varphi(\cdot; W^{(0)})$.

Recall that the bonus in $Q_h^t$ utilizes matrix $\Lambda_h^t$ defined in (C.3), which involves the neural tangent features $\{\varphi(\cdot; \widehat{W}_h^\tau)\}_{\tau \in [T]}$. Similar to $\Lambda_h^t$, we define $\overline{\Lambda}_h^t$ as

$$
\overline{\Lambda}_h^t = \lambda \cdot I_{2md} + \sum_{\tau=1}^{t-1} \varphi(x_h^\tau, a_h^\tau; W^{(0)}) \varphi(x_h^\tau, a_h^\tau; W^{(0)})^\top,
$$
(H.42)

which adopts the same feature mapping $\varphi(\cdot; W^{(0)})$. To simplify the notation, hereafter, we use $\varphi(\cdot)$ to denote $\varphi(\cdot; W^{(0)})$ when its meaning is clear from the text. Moreover, for any $(t, h) \in [T] \times [H]$, we define the response vector $y_h^t \in \mathbb{R}^{t-1}$ by letting its entries be

$$
[y_h^t]_\tau = r_h(x_h^\tau, a_h^\tau) + V_{h+1}^t(x_{h+1}^\tau), \qquad \forall \tau \in [t-1].
$$
(H.43)

We define the feature matrix $\Phi_h^t \in \mathbb{R}^{(t-1) \times 2md}$ by

$$
\Phi_h^t = \big[\varphi(x_h^1, a_h^1)^\top, \ldots, \varphi(x_h^{t-1}, a_h^{t-1})^\top\big]^\top.
$$
(H.44)

Hence, by (H.42) and (H.44), we have $\overline{\Lambda}_h^t = \lambda \cdot I_{2md} + (\Phi_h^t)^\top \Phi_h^t$. Similar to the bonus function $b_h^t$ defined in (C.4), we define

$$
\overline{b}_h^t = \big[\varphi(x, a)^\top (\overline{\Lambda}_h^t)^{-1} \varphi(x, a)\big]^{1/2} = \|\varphi(x, a)\|_{(\overline{\Lambda}_h^t)^{-1}}.
$$
(H.45)

Similar to $L_h^t$ defined in (C.1), we define another least-squares loss function $\overline{L}_h^t : \mathbb{R}^{2md} \to \mathbb{R}$ as

$$
\overline{L}_h^t(W) = \sum_{\tau=1}^{t-1} \big[r_h(x_h^\tau, a_h^\tau) + V_{h+1}^t(x_{h+1}^\tau) - \langle \varphi(x_h^\tau, a_h^\tau), W - W^{(0)} \rangle\big]^2 + \lambda \cdot \|W - W^{(0)}\|_2^2
$$
(H.46)

and let $\overline{W}_h^t$ be its global minimizer. By direct computation, $\overline{W}_h^t$ can be written in closed form as

$$
\overline{W}_h^t = W^{(0)} + (\overline{\Lambda}_h^t)^{-1}(\Phi_h^t)^\top y_h^t,
$$
(H.47)

where $\overline{\Lambda}_h^t$, $\Phi_h^t$, and $y_h^t$ are defined respectively in (H.42), (H.44), and (H.43). Similar to (H.36), utilizing the fact that $\overline{L}_h^t(\overline{W}_h^t) \le \overline{L}_h^t(W^{(0)})$, we also have $\|\overline{W}_h^t - W^{(0)}\|_2 \le H\sqrt{T/\lambda}$. Then, in a manner similar to the construction of $Q_h^t$ in Algorithm 4, we combine $\overline{b}_h^t$ in (H.45) and $\overline{W}_h^t$ in (H.47) to define $\overline{Q}_h^t : \mathcal{Z} \to \mathbb{R}$ as

$$
\overline{Q}_h^t(x, a) = \min\big\{\varphi(x, a)^\top (\overline{W}_h^t - W^{(0)}) + \beta \cdot \overline{b}_h^t(x, a), H - h + 1\big\}^+.
$$
(H.48)

Note that $\overline{Q}_h^t$ share the same form as $Q$ in (D.2). Thus, we have $\overline{Q}_h^t \in \mathcal{Q}_{\mathrm{ucb}}(h, H\sqrt{T/\lambda}, B)$ for any $B \geq \beta$. Moreover, we define $\overline{V}_h^t(\cdot) = \max_{a \in \mathcal{A}} \overline{Q}_h^t(\cdot, a)$.

In the following, we aim to show that $\overline{Q}_h^t$ is close to $Q_h^t$ when $m$ is sufficiently large. When this is true, $\overline{V}_h^t$ is also close to $V_h^t$. To bound $Q_h^t - \overline{Q}_h^t$, since the truncation operator is non-expansive, by the triangle inequality we have

$$\|Q_h^t - \overline{Q}_h^t\|_\infty \leq \underbrace{\left\| f(\cdot; \widehat{W}_h^t) - \varphi(\cdot)^\top (\overline{W}_h^t - W^{(0)}) \right\|_\infty}_{\text{(i)}} + \underbrace{\beta \cdot \|b_h^t - \overline{b}_h^t\|_\infty}_{\text{(ii)}}. \tag{H.49}$$

Recall that we define $\mathcal{B} = \{W \in \mathbb{R}^{2md} : \|W - W^{(0)}\|_2 \leq H\sqrt{T/\lambda}\}$. To bound the two terms on the right-hand side of (H.49), we utilize the following lemma that quantifies the perturbation of $f(\cdot; W)$ and $\varphi(\cdot; W)$ within $W \in \mathcal{B}$.

**Lemma H.2.** When $TH^2 = \mathcal{O}(m \cdot \log^{-6} m)$, with probability at least $1 - m^{-2}$ with respect to the randomness of initialization, for any $W \in \mathcal{B}$ and any $z \in \mathcal{Z}$, we have

$$\left| f(z, W) - \varphi(z, W^{(0)})^\top (W - W^{(0)}) \right| \leq \overline{C} \cdot T^{2/3} \cdot H^{4/3} \cdot m^{-1/6} \cdot \sqrt{\log m},$$

$$\left\| \varphi(z, W) - \varphi(z, W^{(0)}) \right\|_2 \leq \overline{C} \cdot (TH^2/m)^{1/6} \cdot \sqrt{\log m}, \qquad \|\varphi(z, W)\|_2 \leq \overline{C}.$$

*Proof.* See [3, 30, 13] for a detailed proof. More specifically, this lemma is obtained from Lemmas F.1 and F.2 in [13], which are further based on results in [3, 30]. $\qquad \square$

By Lemma H.2 and triangle inequality, Term (i) on the right-hand side of (H.49) is bounded by

$$\text{Term (i)} \leq \left\| f(\cdot; \widehat{W}_h^t) - \varphi(\cdot)^\top (\widehat{W}_h^t - W^{(0)}) \right\|_\infty + \left\| \varphi(\cdot)^\top (\widehat{W}_h^t - \overline{W}_h^t) \right\|_\infty$$

$$\leq \overline{C} \cdot T^{2/3} \cdot H^{4/3} \cdot m^{-1/6} \cdot \sqrt{\log m} + \overline{C} \cdot \left\| \widehat{W}_h^t - \overline{W}_h^t \right\|_2. \tag{H.50}$$

To bound $\left\| \widehat{W}_h^t - \overline{W}_h^t \right\|_2$, notice that $\widehat{W}_h^t$ and $\overline{W}_h^t$ are the global minimizers of $L_h^t$ in (C.1) and $\overline{L}_h^t$ in (H.46), respectively. Thus, by the first-order optimality condition, we have

$$\lambda \cdot \left( \widehat{W}_h^t - W^{(0)} \right) = \sum_{\tau=1}^{t-1} \left\{ [y_h^t]_\tau - f(z_h^\tau; \widehat{W}_h^t) \right\} \cdot \varphi(z_h^\tau; \widehat{W}_h^t), \tag{H.51}$$

$$\lambda \cdot \left( \overline{W}_h^t - W^{(0)} \right) = \sum_{\tau=1}^{t-1} \left\{ [y_h^t]_\tau - \left\langle \varphi(z_h^\tau; W^{(0)}), \overline{W}_h^t - W^{(0)} \right\rangle \right\} \cdot \varphi(z_h^\tau; W^{(0)}), \tag{H.52}$$

where $[y_h^t]_\tau$ is defined in (H.43) and $z_h^\tau = (x_h^\tau, a_h^\tau)$. In addition, by the definition of $\overline{\Lambda}_h^t$ in (H.42), (H.52) can be equivalently written as

$$\overline{\Lambda}_h^t \left( \overline{W}_h^t - W^{(0)} \right) = \sum_{\tau=1}^{t-1} [y_h^t]_\tau \cdot \varphi(z_h^\tau; W^{(0)}). \tag{H.53}$$

Similarly, for (H.51), by direct computation we have

$$\overline{\Lambda}_h^t \left( \widehat{W}_h^t - W^{(0)} \right) = \sum_{\tau=1}^{t-1} [y_h^t]_\tau \cdot \varphi\left( z_h^\tau; \widehat{W}_h^t \right) \tag{H.54}$$

$$+ \sum_{\tau=1}^{t-1} \left[ \left\langle \varphi(z_h^\tau; W^{(0)}), \widehat{W}_h^t - W^{(0)} \right\rangle \cdot \varphi(z_h^\tau; W^{(0)}) - f(z_h^\tau; \widehat{W}_h^t) \cdot \varphi(z_h^\tau; \widehat{W}_h^t) \right].$$

For any $\tau \in [t-1]$, we have

$$\left\langle \varphi(z_h^\tau; W^{(0)}), \widehat{W}_h^t - W^{(0)} \right\rangle \cdot \varphi(z_h^\tau; W^{(0)}) - f(z_h^\tau; \widehat{W}_h^t) \cdot \varphi(z_h^\tau; \widehat{W}_h^t)$$

$$= \left\langle \varphi(z_h^\tau; W^{(0)}), \widehat{W}_h^t - W^{(0)} \right\rangle \cdot \left[ \varphi(z_h^\tau; W^{(0)}) - \varphi(z_h^\tau; \widehat{W}_h^t) \right]$$

$$+ \left[ \left\langle \varphi(z_h^\tau; W^{(0)}), \widehat{W}_h^t - W^{(0)} \right\rangle - f(z_h^\tau; \widehat{W}_h^t) \right] \cdot \varphi(z_h^\tau; \widehat{W}_h^t). \tag{H.55}$$

Thus, applying Lemma H.2 to (H.55), we have

$$\left\| \left\langle \varphi(z_h^\tau; W^{(0)}), \widehat{W}_h^t - W^{(0)} \right\rangle \cdot \varphi(z_h^\tau; W^{(0)}) - f(z_h^\tau; \widehat{W}_h^t) \cdot \varphi(z_h^\tau; \widehat{W}_h^t) \right\|_2$$

$$\leq \left\| \varphi(z_h^\tau; W^{(0)}) \right\|_2 \cdot \left\| \widehat{W}_h^t - W^{(0)} \right\|_2 \cdot \left\| \varphi(z_h^\tau; W^{(0)}) - \varphi(z_h^\tau; \widehat{W}_h^t) \right\|$$

$$+ \left| \left\langle \varphi(z_h^\tau; W^{(0)}), \widehat{W}_h^t - W^{(0)} \right\rangle - f(z_h^\tau; \widehat{W}_h^t) \right| \cdot \left\| \varphi(z_h^\tau; \widehat{W}_h^t) \right\|_2$$

$$\leq 2\overline{C}^2 \cdot T^{2/3} \cdot H^{4/3} \cdot m^{-1/6} \cdot \sqrt{\log m} \cdot \lambda^{-1/2}, \tag{H.56}$$

where we utilize the fact that $\|\widehat{W}_h^t - W^{(0)}\|_2 \leq H\sqrt{T/\lambda} \leq H\sqrt{T}$. Then, combining (H.53), (H.54), and (H.56), we have

$$\left\| \overline{\Lambda}_h^t \big( \widehat{W}_h^t - \overline{W}_h^t \big) \right\|_2$$

$$\leq \left\| \sum_{\tau=1}^{t-1} [y_h^t]_\tau \cdot \big[ \varphi\big(z_h^\tau; \widehat{W}_h^t\big) - \varphi\big(z_h^\tau; W^{(0)}\big) \big] \right\|_2 + 2\overline{C}^2 \cdot T^{5/3} \cdot H^{4/3} \cdot m^{-1/6} \cdot \sqrt{\log m}$$

$$\leq \overline{C} \cdot T^{7/6} \cdot H^{4/3} \cdot m^{-1/6} \cdot \sqrt{\log m} + 2\overline{C}^2 \cdot T^{5/3} \cdot H^{4/3} \cdot m^{-1/6} \cdot \sqrt{\log m}, \tag{H.57}$$

where in the last inequality we utilize the fact that $[y_h^t]_\tau \in [0, H]$. When $T$ is sufficiently large, the second term in (H.57) dominates. Since the eigenvalues of $(\overline{\Lambda}_h^t)^{-1}$ are all bounded by $1/\lambda$, we have

$$\left\| \widehat{W}_h^t - \overline{W}_h^t \right\|_2 \leq \left\| (\overline{\Lambda}_h^t)^{-1} \right\|_{\mathrm{op}} \cdot \left\| \overline{\Lambda}_h^t \big( \widehat{W}_h^t - \overline{W}_h^t \big) \right\|_2 \leq 1/\lambda \cdot \left\| \overline{\Lambda}_h^t \big( \widehat{W}_h^t - \overline{W}_h^t \big) \right\|_2. \tag{H.58}$$

In the sequel, we set $\lambda$ as

$$\lambda = \overline{C}^2 \cdot (1 + 1/T) \in \big[ \overline{C}^2, 2\overline{C}^2 \big]. \tag{H.59}$$

Thus, combining (H.50), (H.57), (H.58), and (H.59), we have

$$\text{Term (i)} \leq 4 \cdot T^{5/3} \cdot H^{4/3} \cdot m^{-1/6} \cdot \sqrt{\log m} \tag{H.60}$$

where we use the fact that $\overline{C}^2/\lambda \leq 1$.

Furthermore, to bound Term (ii), by the definitions of $b_h^t$ and $\overline{b}_h^t$, for any $z \in \mathcal{Z}$, we have

$$\left| b_h^t(z) - \overline{b}_h^t(z) \right| = \left| \sqrt{\varphi(z; \widehat{W}_h^t)^\top (\Lambda_h^t)^{-1} \varphi(z; \widehat{W}_h^t)} - \sqrt{\varphi(z; W^{(0)})^\top (\overline{\Lambda}_h^t)^{-1} \varphi(z; W^{(0)})} \right|$$

$$\leq \sqrt{\left| \varphi(z; \widehat{W}_h^t)^\top (\Lambda_h^t)^{-1} \varphi(z; \widehat{W}_h^t) - \varphi(z; W^{(0)})^\top (\overline{\Lambda}_h^t)^{-1} \varphi(z; W^{(0)}) \right|}, \tag{H.61}$$

where the inequality follows from the elementary inequality $|\sqrt{x} - \sqrt{y}| \leq \sqrt{|x - y|}$. By the triangle inequality

$$\left| \varphi(z; \widehat{W}_h^t)^\top (\Lambda_h^t)^{-1} \varphi(z; \widehat{W}_h^t) - \varphi(z; W^{(0)})^\top (\overline{\Lambda}_h^t)^{-1} \varphi(z; W^{(0)}) \right|$$

$$\leq \left| \big[ \varphi(z; \widehat{W}_h^t) - \varphi(z; W^{(0)}) \big]^\top (\Lambda_h^t)^{-1} \varphi(z; \widehat{W}_h^t) \right| + \left| \varphi(z; W^{(0)})^\top \big[ (\Lambda_h^t)^{-1} - (\overline{\Lambda}_h^t)^{-1} \big] \varphi(z; \widehat{W}_h^t) \right|$$

$$+ \left| \varphi(z; W^{(0)})^\top (\overline{\Lambda}_h^t)^{-1} \big[ \varphi(z; \widehat{W}_h^t) - \varphi(z; W^{(0)}) \big] \right|. \tag{H.62}$$

Combining Hölder's inequality and Lemma H.2, we bound the first term on the right-hand side of (H.62) by

$$\left| \big[ \varphi(z; \widehat{W}_h^t) - \varphi(z; W^{(0)}) \big]^\top (\Lambda_h^t)^{-1} \varphi(z; \widehat{W}_h^t) \right| \tag{H.63}$$

$$\leq \left\| \varphi(z; \widehat{W}_h^t) - \varphi(z; W^{(0)}) \right\|_2 \cdot \left\| (\Lambda_h^t)^{-1} \right\|_{\mathrm{op}} \cdot \left\| \varphi(z; \widehat{W}_h^t) \right\|_2 \leq \overline{C}^2 \cdot T^{1/6} \cdot H^{1/3} \cdot m^{-1/6} \cdot \lambda^{-1} \cdot \sqrt{\log m},$$

where $\|(\Lambda_h^t)^{-1}\|_{\mathrm{op}}$ is the matrix operator norm of $(\Lambda_h^t)^{-1}$, which is bounded by $1/\lambda$. Similarly, for the third term, we also have

$$\left| \varphi(z; W^{(0)})^\top (\overline{\Lambda}_h^t)^{-1} \big[ \varphi(z; \widehat{W}_h^t) - \varphi(z; W^{(0)}) \big] \right| \leq \overline{C}^2 \cdot T^{1/6} \cdot H^{1/3} \cdot m^{-1/6} \cdot \lambda^{-1} \cdot \sqrt{\log m}. \tag{H.64}$$

For the second term, since both $\Lambda_h^t$ and $\overline{\Lambda}_h^t$ are invertible, we have

$$\left\| (\Lambda_h^t)^{-1} - (\overline{\Lambda}_h^t)^{-1} \right\|_{\mathrm{op}} = \left\| (\Lambda_h^t)^{-1} (\Lambda_h^t - \overline{\Lambda}_h^t)(\overline{\Lambda}_h^t)^{-1} \right\|_{\mathrm{op}}$$

$$\leq \left\| (\Lambda_h^t)^{-1} \right\|_{\mathrm{op}} \cdot \left\| (\overline{\Lambda}_h^t)^{-1} \right\|_{\mathrm{op}} \cdot \left\| \Lambda_h^t - \overline{\Lambda}_h^t \right\|_{\mathrm{op}} \leq \lambda^{-2} \cdot \left\| \Lambda_h^t - \overline{\Lambda}_h^t \right\|_{\mathrm{fro}}. \tag{H.65}$$

By direct computation, we have

$$\|\Lambda_h^t - \overline{\Lambda}_h^t\|_{\mathrm{fro}}$$

$$= \left\|\sum_{\tau=1}^{t-1} \left[\varphi(z_h^\tau; \widehat{W}_h^{\tau+1})\varphi(z_h^\tau; \widehat{W}_h^{\tau+1})^\top - \varphi(z_h^\tau; W^{(0)})\varphi(z_h^\tau; W^{(0)})^\top\right]\right\|_{\mathrm{fro}}$$

$$\leq \sum_{\tau=1}^{t-1} \left\|\varphi(z_h^\tau; \widehat{W}_h^{\tau+1})\varphi(z_h^\tau; \widehat{W}_h^{\tau+1})^\top - \varphi(z_h^\tau; W^{(0)})\varphi(z_h^\tau; W^{(0)})^\top\right\|_{\mathrm{fro}}$$

$$\leq \sum_{\tau=1}^{t-1} \left\|\left[\varphi(z_h^\tau; \widehat{W}_h^{\tau+1}) - \varphi(z_h^\tau; W^{(0)})\right]\varphi(z_h^\tau; \widehat{W}_h^{\tau+1})^\top\right.$$

$$\left. + \varphi(z_h^\tau; W^{(0)})\left[\varphi(z_h^\tau; \widehat{W}_h^{\tau+1}) - \varphi(z_h^\tau; W^{(0)})\right]^\top\right\|_{\mathrm{fro}}.$$

Hence, by Lemma H.2 we can bound $\|\Lambda_h^t - \overline{\Lambda}_h^t\|_{\mathrm{fro}}$ by

$$\|\Lambda_h^t - \overline{\Lambda}_h^t\|_{\mathrm{fro}} \leq 2(t-1) \cdot \overline{C}^2 \cdot T^{1/6} \cdot H^{1/3} \cdot m^{-1/6} \cdot \sqrt{\log m}$$

$$\leq 2\overline{C}^2 \cdot T^{7/6} \cdot H^{1/3} \cdot m^{-1/6} \cdot \sqrt{\log m}. \qquad (\mathrm{H.66})$$

Hence, combining (H.65) and (H.66), the second term on the right-hand side of (H.62) can be bounded by

$$\left|\varphi(z; W^{(0)})^\top\left[(\Lambda_h^t)^{-1} - (\overline{\Lambda}_h^t)^{-1}\right]\varphi(z; \widehat{W}_h^t)\right|$$

$$\leq \left\|\varphi(z; W^{(0)})\right\|_2 \cdot \left\|\varphi(z; \widehat{W}_h^t)\right\|_2 \cdot \left\|(\Lambda_h^t)^{-1} - (\overline{\Lambda}_h^t)^{-1}\right\|_{\mathrm{op}}$$

$$\leq 2\overline{C}^4 \cdot T^{7/6} \cdot H^{1/3} \cdot m^{-1/6} \cdot \lambda^{-2} \cdot \sqrt{\log m}. \qquad (\mathrm{H.67})$$

Notice that $\lambda$ defined in (H.59) satisfies that $\lambda \geq \overline{C}^2$. Thus, combining (H.61)-(H.64), and (H.67), we have

$$\left|b_h^t(z) - \overline{b}_h^t(z)\right| \leq 2 \cdot T^{7/12} \cdot H^{1/6} \cdot m^{-1/12} \cdot (\log m)^{1/4}, \qquad \forall(t, h) \in [T] \times [H], \qquad (\mathrm{H.68})$$

which establishes the second inequality in (E.11). Finally, combining (H.49), (H.60), and (H.68), we conclude that

$$\|Q_h^t - \overline{Q}_h^t\|_\infty \leq 4 \cdot T^{5/3} \cdot H^{4/3} \cdot m^{-1/6} \cdot \sqrt{\log m} + 2\beta \cdot T^{7/12} \cdot H^{1/6} \cdot m^{-1/12} \cdot (\log m)^{1/4}.$$

Note that $\beta > 1$. When $m = \Omega(\beta^{12} \cdot T^{13} \cdot H^{14} \cdot (\log m)^3)$, the second term in the above inequality is the dominating term. Thus, we have

$$\sup_{x \in \mathcal{S}}\left|V_h^t(x) - \overline{V}_h^t(x)\right| \leq \|Q_h^t - \overline{Q}_h^t\|_\infty \leq 4\beta \cdot T^{7/12} \cdot H^{1/6} \cdot m^{-1/12} \cdot (\log m)^{1/4}. \qquad (\mathrm{H.69})$$

This concludes the second step.

**Step III.** In the last step, we establish optimism by comparing $\varphi(\cdot)^\top(\overline{W}_h^t - W^{(0)})$ and the function $\widetilde{Q}_h^t$ defined in (H.39), where $\varphi(\cdot)$ denotes $\varphi(\cdot; W^{(0)})$. By the definition of $\overline{\Lambda}_h^t$ in (H.42), we have

$$\widetilde{W}_h^t - W^{(0)} = (\overline{\Lambda}_h^t)^{-1} \cdot \left[\lambda \cdot \left(\widetilde{W}_h^t - W^{(0)}\right) + (\Phi_h^t)^\top \Phi_h^t\left(\widetilde{W}_h^t - W^{(0)}\right)\right],$$

where $\widetilde{W}_h^t$ is given in (H.41). Hence, combining (H.47), we have

$$\overline{W}_h^t - \widetilde{W}_h^t = -\lambda \cdot (\overline{\Lambda}_h^t)^{-1}\left(\widetilde{W}_h^t - W^{(0)}\right) + (\overline{\Lambda}_h^t)^{-1}(\Phi_h^t)^\top\left[y_h^t - \Phi_h^t\left(\widetilde{W}_h^t - W^{(0)}\right)\right]. \qquad (\mathrm{H.70})$$

Thus, for any $z \in \mathcal{Z}$, by (H.70) we have

$$\varphi(z)^\top\left(\overline{W}_h^t - \widetilde{W}_h^t\right)$$

$$= \underbrace{-\lambda \cdot \varphi(z)^\top(\overline{\Lambda}_h^t)^{-1} \cdot \left(\widetilde{W}_h^t - W^{(0)}\right)}_{(\mathrm{iii})} + \underbrace{\varphi(z)^\top(\overline{\Lambda}_h^t)^{-1}(\Phi_h^t)^\top\left[y_h^t - \Phi_h^t\left(\widetilde{W}_h^t - W^{(0)}\right)\right]}_{(\mathrm{iv})}.$$

$$(\mathrm{H.71})$$

For Term (iii) on the right-hand side of (H.71), by the Cauchy-Schwarz inequality, we have

$$\left| \lambda \cdot \varphi(z)^\top (\overline{\Lambda}_h^t)^{-1} \cdot \left( \widetilde{W}_h^t - W^{(0)} \right) \right| \leq \lambda \cdot \left\| \widetilde{W}_h^t - W^{(0)} \right\|_2 \cdot \left\| (\overline{\Lambda}_h^t)^{-1} \varphi(z) \right\|_2$$

$$\leq \lambda \cdot R_Q H / \sqrt{d} \cdot \sqrt{\varphi(z)^\top (\overline{\Lambda}_h^t)^{-1} (\overline{\Lambda}_h^t)^{-1} \varphi(z)} \leq R_Q H \cdot \sqrt{\lambda/d} \cdot \overline{b}_h^t(z). \qquad \text{(H.72)}$$

For Term (iv) in (H.71), recall that $\widetilde{Q}_h^t(z) = \varphi(z)^\top (\widetilde{W}_h^t - W^{(0)})$. To simplify the notation, let $q^\star \in \mathbb{R}^{t-1}$ denote the vector whose $\tau$-th entry is $(\mathbb{T}_h^\star Q_{h+1}^t)(x_h^\tau, a_h^\tau)$ for any $\tau \in [t-1]$. Then, by (H.40), for any $\tau \in [t-1]$, the $\tau$-th entry of $\Phi_h^t(\widetilde{W}_h^t - W^{(0)})$ satisfies

$$\left| [\Phi_h^t(\widetilde{W}_h^t - W^{(0)})]_\tau - [q^\star]_\tau \right| = \left| [\Phi_h^t(\widetilde{W}_h^t - W^{(0)})]_\tau - (\mathbb{T}_h^\star Q_{h+1}^t)(x_h^\tau, a_h^\tau) \right|$$

$$\leq 10 C_{\text{act}} \cdot R_Q H \cdot \sqrt{\log(mTH)/m}.$$

Moreover, for any $\tau \in [t-1]$, the $\tau$-th entry of $(y_h^t - q^\star)$ can be written as

$$[y_h^t]_\tau - [q^\star]_\tau = r_h(x_h^\tau, a_h^\tau) + V_{h+1}^t(x_{h+1}^\tau) - \varphi(x_h^\tau, a_h^\tau)^\top \overline{\theta}_h^t$$

$$= r_h(x_h^\tau, a_h^\tau) + V_{h+1}^t(x_{h+1}^\tau) - (\mathbb{T}_h^\star Q_{h+1}^t)(x_h^\tau, a_h^\tau)$$

$$= V_{h+1}^t(x_{h+1}^\tau) - (\mathbb{P}_h V_{h+1}^t)(x_h^\tau, a_h^\tau). \qquad \text{(H.73)}$$

Then, by the triangle inequality and (H.73), we have

$$\left| \varphi(z)^\top (\overline{\Lambda}_h^t)^{-1} (\Phi_h^t)^\top \left[ y_h^t - \Phi_h^t(\widetilde{W}_h^t - W^{(0)}) \right] \right|$$

$$\leq \left| \varphi(z)^\top (\overline{\Lambda}_h^t)^{-1} (\Phi_h^t)^\top [y_h^t - q^\star] \right| + \left| \varphi(z)^\top (\overline{\Lambda}_h^t)^{-1} (\Phi_h^t)^\top [q^\star - \Phi_h^t(\widetilde{W}_h^t - W^{(0)})] \right|$$

$$\leq \|\varphi(z)\|_{(\overline{\Lambda}_h^t)^{-1}} \cdot \left\| \sum_{\tau=1}^{t-1} \varphi(x_h^\tau, a_h^\tau) \cdot [V_{h+1}^t(x_{h+1}^\tau) - (\mathbb{P}_h V_{h+1}^t)(x_h^\tau, a_h^\tau)] \right\|_{(\overline{\Lambda}_h^t)^{-1}}$$

$$+ 10 C_{\text{act}} \cdot R_Q H \cdot \sqrt{\log(mTH)/m} \cdot \|\varphi(z)\|_{(\overline{\Lambda}_h^t)^{-1}}. \qquad \text{(H.74)}$$

Recall that we have shown in **Step II** that, with probability at least $1 - m^2$ with respect to the randomness of initialization, (H.69) holds for all $(t, h) \in [T] \times [H]$. To simplify the notation, we denote

$$\text{Err} = 4\beta \cdot T^{7/12} \cdot H^{1/6} \cdot m^{-1/12} \cdot (\log m)^{1/4}. \qquad \text{(H.75)}$$

Moreover, we define functions $\Delta V_1 = V_{h+1}^t - \overline{V}_{h+1}^t$ and $\Delta V_2 = \mathbb{P}_h(V_{h+1}^t - \overline{V}_{h+1}^t)$. Then (H.69) implies that $\sup_{x \in S} |\Delta V_1(x)| \leq \text{Err}$ and $\sup_{z \in \mathcal{Z}} |\Delta V_2(z)| \leq \text{Err}$. By the elementary inequality $(a+b)^2 \leq 2a^2 + 2b^2$, we have

$$\left\| \sum_{\tau=1}^{t-1} \varphi(x_h^\tau, a_h^\tau) \cdot [V_{h+1}^t(x_{h+1}^\tau) - (\mathbb{P}_h V_{h+1}^t)(x_h^\tau, a_h^\tau)] \right\|_{(\overline{\Lambda}_h^t)^{-1}}^2$$

$$\leq 2 \underbrace{\left\| \sum_{\tau=1}^{t-1} \varphi(x_h^\tau, a_h^\tau) \cdot [\overline{V}_{h+1}^t(x_{h+1}^\tau) - (\mathbb{P}_h \overline{V}_{h+1}^t)(x_h^\tau, a_h^\tau)] \right\|_{(\overline{\Lambda}_h^t)^{-1}}^2}_{\text{(v)}}$$

$$+ 2 \left\| \sum_{\tau=1}^{t-1} \varphi(x_h^\tau, a_h^\tau) \cdot [\Delta V_1(x_{h+1}^\tau) - \Delta V_2(x_h^\tau, a_h^\tau)] \right\|_{(\overline{\Lambda}_h^t)^{-1}}^2$$

$$\leq 2 \cdot \text{Term (v)} + 8 \cdot \text{Err}^2 \cdot T^2, \qquad \text{(H.76)}$$

where the last inequality follows from the fact that

$$\left\| \sum_{\tau=1}^{t-1} \varphi(x_h^\tau, a_h^\tau) \cdot [\Delta V_1(x_{h+1}^\tau) - \Delta V_2(x_h^\tau, a_h^\tau)] \right\|_{(\overline{\Lambda}_h^t)^{-1}}^2 \leq 4 \text{Err}^2 \cdot \left\| \sum_{\tau=1}^{t-1} \varphi(x_h^\tau, a_h^\tau) \right\|_{(\overline{\Lambda}_h^t)^{-1}}^2$$

$$\leq 4 \cdot \text{Err}^2 \cdot (t-1) \cdot \lambda^{-1} \cdot \sum_{\tau=1}^{t-1} \|\varphi(x_h^\tau, a_h^\tau)\|_2^2 \leq 4 \cdot \text{Err}^2 \cdot (t-1)^2 \cdot \overline{C}^2 \cdot \lambda^{-1} \leq 4 \cdot \text{Err}^2 \cdot T^2.$$

Here the second-to-last inequality follows from Lemma H.2 and the definition of $\lambda$.

Recall that we define $\overline{b}_h^t(z) = \|\varphi(z)\|_{(\overline{\Lambda}_h^t)^{-1}}$. Combining (H.73), (H.74), and (H.77), we have

$$
\begin{aligned}
\big|\varphi(z)^\top (\overline{\Lambda}_h^t)^{-1}(\Phi_h^t)^\top \big[y_h^t - \Phi_h^t\big(\widetilde{W}_h^t - W^{(0)}\big)\big]\big| & \\
\le \overline{b}_h^t(z) \cdot \big[10 C_{\mathrm{act}} \cdot R_Q H \cdot \sqrt{\log(mTH)/m} + \sqrt{2 \cdot \text{Term (v)}} + 2\sqrt{2} \cdot \mathtt{Err} \cdot T\big] & \\
\le \overline{b}_h^t(z) \cdot \big[R_Q H + \sqrt{2 \cdot \text{Term (v)}}\big], & \quad (\text{H.77})
\end{aligned}
$$

where we apply the elementary inequality $\sqrt{a+b} \le \sqrt{a} + \sqrt{b}$. Here in the last inequality we let $m$ be sufficiently large such that

$$
10 C_{\mathrm{act}} \cdot R_Q H \cdot \sqrt{\log(mTH)/m} + 2\sqrt{2} \cdot \mathtt{Err} \cdot T \le R_Q H.
$$

In the following, we aim to bound Term (v) in (H.77) by combining the concentration of the self-normalized stochastic process and uniform concentration. To characterize the function class that contains each $\overline{V}_h^t$, we define $\widetilde{\varphi} \colon \mathcal{Z} \to \mathbb{R}$ by $\widetilde{\varphi}(z) = \varphi(z)/\overline{C}$. Then, conditioning on the event where the statements in Lemma H.2 are true, we have $\|\widetilde{\varphi}(z)\|_2 \le 1$ for all $z \in \mathcal{Z}$. Furthermore, we define a kernel function $\widetilde{K} \colon \mathcal{Z} \times \mathcal{Z} \to \mathbb{R}$ by letting $\widetilde{K}(z,z') = \widetilde{\varphi}(z)^\top \widetilde{\varphi}(z')$ for all $z, z' \in \mathcal{Z}$. That is, $\widetilde{K}$ is the normalized version of the empirical NTK $K_m$. By construction, $\widetilde{K}$ is a bounded kernel such that $\sup_{z \in \mathcal{Z}} \widetilde{K}(z,z) \le 1$. We can also consider the maximal information gain in (4.2) for $\widetilde{K}$ and $K_m$. These two quantities are linked via

$$
\Gamma_{\widetilde{K}}(T,\sigma) = \Gamma_{K_m}\big(T, \overline{C}^2 \sigma\big), \qquad \forall \sigma > 0. \quad (\text{H.78})
$$

Furthermore, we define $\widetilde{\lambda} = \lambda/\overline{C}^2$ and $\widetilde{\Lambda}_h^t = \overline{\Lambda}_h^t//\overline{C}^2$ for all $(t,h) \in [T] \times [H]$. By the definition of $\lambda$ in (H.59), we have $\widetilde{\lambda} = 1 + 1/T \in [1,2]$. Moreover, by (H.42) we have

$$
\widetilde{\Lambda}_h^t = \widetilde{\lambda} \cdot I_{2md} + \sum_{\tau=1}^{t-1} \widetilde{\varphi}(x_h^\tau, a_h^\tau)\widetilde{\varphi}(x_h^\tau, a_h^\tau)^\top.
$$

Since $\widetilde{\lambda} > 1$, $\widetilde{\Lambda}_h^t$ is an invertible matrix and the eigenvalues of $(\widetilde{\Lambda}_h^t)^{-1}$ are all bounded above by one. Using $\widetilde{\varphi}$ and $\widetilde{\Lambda}_h^t$, we rewrite each $\overline{Q}_h^t$ as follows. For $\overline{W}_h^t$ defined in (H.47), we have

$$
\varphi(x,a)^\top\big(\overline{W}_h^t - W^{(0)}\big) = \overline{C} \cdot \widetilde{\varphi}(x,a)^\top\big(\overline{W}_h^t - W^{(0)}\big), \quad (\text{H.79})
$$

where $\overline{C} \cdot \|\overline{W}_h^t - W^{(0)}\|_2 \le \overline{C} \cdot H\sqrt{T/\lambda} \le H\sqrt{T}$ since $\lambda \ge (\overline{C})^2$. Meanwhile, we also have

$$
\overline{b}_h^t(z) = \|\varphi(z)\|_{(\overline{\Lambda}_h^t)^{-1}} = \big[\widetilde{\varphi}(z)^\top\big(\widetilde{\Lambda}_h^t\big)^{-1}\widetilde{\varphi}(z)\big]^{1/2}. \quad (\text{H.80})
$$

Thus, combining (H.79) and (H.80), $\overline{Q}_h^t$ defined in (H.48) can be written equivalently as

$$
\overline{Q}_h^t(z) = \min\big\{\widetilde{\varphi}(z)^\top \overline{\vartheta}_h^t + \beta \cdot \big\|\widetilde{\varphi}(z)\big\|_{(\widetilde{\Lambda}_h^t)^{-1}}, H - h + 1\big\}^+,
$$

where $\overline{\vartheta}_h^t = \overline{C} \cdot \big(\overline{W}_h^t - W^{(0)}\big)$, which satisfies $\|\overline{\vartheta}_h^t\|_2 \le H\sqrt{T}$.

Let $\mathcal{D}$ be a finite subset of $\mathcal{Z}$ with no more than $T$ elements. For any fixed $\mathcal{D}$, we define

$$
\widetilde{\Lambda}_{\mathcal{D}} = \widetilde{\lambda} \cdot I_{2dm} + \sum_{z \in \mathcal{D}} \widetilde{\varphi}(z)\widetilde{\varphi}(z)^\top \in \mathbb{R}^{2md \times 2md}. \quad (\text{H.81})
$$

For any $h \in [H]$, $R, B > 0$, we let $\widetilde{Q}_{\mathrm{ucb}}(h,R,B)$ consists of functions that take the form of

$$
Q(\cdot) = \min\big\{\widetilde{\varphi}(\cdot)^\top \vartheta + \beta \cdot \big\|\widetilde{\varphi}(\cdot)\big\|_{(\widetilde{\Lambda}_{\mathcal{D}})^{-1}}; H - h + 1\big\}^+,
$$

for some $\vartheta \in \mathbb{R}^{2md}$ with $\|\vartheta\|_2 \le R$ and some $\mathcal{D} \subseteq \mathcal{Z}$. Then $\widetilde{Q}_{\mathrm{ucb}}(h,R,B)$ corresponds to the function class in (4.4) with the kernel being $\widetilde{K}$. Moreover, we define $\widetilde{\mathcal{V}}_{\mathrm{ucb}}(h,R,B)$ as

$$
\widetilde{\mathcal{V}}_{\mathrm{ucb}}(h,R,B) = \big\{V : V(\cdot) = \max_a Q(\cdot, a) \ \text{for some} \ Q \in \widetilde{Q}_{\mathrm{ucb}}(h,R,B)\big\}.
$$

By definition, for all $h \in [H]$ and any $R, B > 0$, we have that $\mathcal{Q}_{\mathrm{ucb}}(h,R,B) = \widetilde{Q}_{\mathrm{ucb}}(h,\overline{C}R,B)$. Meanwhile, since $(\overline{C})^2 \le \lambda \le 2(\overline{C})^2$, for all $R > 0$, we have

$$
\mathcal{Q}_{\mathrm{ucb}}(h,R,B) \subseteq \widetilde{Q}_{\mathrm{ucb}}(h,R\sqrt{\lambda},B) \subseteq \mathcal{Q}_{\mathrm{ucb}}(h,\sqrt{2}R,B). \quad (\text{H.82})
$$

Recall that we define $R_T = H\sqrt{2T/\lambda}$ and let $N_\infty(\epsilon; h, B)$ denote the $\epsilon$-covering number of $\mathcal{Q}_{\mathrm{ucb}}(h,R_T,B)$ with respect to the $\ell_\infty$-norm on $\mathcal{Z}$. Moreover, hereafter, we denote $\epsilon^* = H/T$

and set $B = B_T$ which satisfy (D.3). Since we set $\beta = B_T$ in Algorithm 4, it holds for all $(t, h) \in [T] \times [H]$ that
$$\overline{Q}_h^t \in \widetilde{\mathcal{Q}}_{\text{ucb}}(h, H\sqrt{T}, B) \subseteq \mathcal{Q}_{\text{ucb}}(h, R_T, B), \qquad \overline{V}_h^t \in \widetilde{\mathcal{V}}_{\text{ucb}}(h, H\sqrt{T}, B).$$

Now, to bound Term (v) in (H.77), similar to the analysis the proof of Lemma E.2, we apply the concentration of self-normalized stochastic process and uniform concentration over $\widetilde{\mathcal{V}}_{\text{ucb}}(h, H\sqrt{T}, B_T)$. Specifically, similar to (H.25) and (H.26), with probability at least $1 - (2T^2 H^2)^{-1}$, we have

$$\begin{aligned}
\text{Term (v)} &= \left\| \sum_{\tau=1}^{t-1} \varphi(x_h^\tau, a_h^\tau) \cdot \left[ \overline{V}_{h+1}^t(x_{h+1}^\tau) - (\mathbb{P}_h \overline{V}_{h+1}^t)(x_h^\tau, a_h^\tau) \right] \right\|_{(\overline{\Lambda}_h^t)^{-1}}^2 \\
&= \left\| \sum_{\tau=1}^{t-1} \widetilde{\varphi}(x_h^\tau, a_h^\tau) \cdot \left[ \overline{V}_{h+1}^t(x_{h+1}^\tau) - (\mathbb{P}_h \overline{V}_{h+1}^t)(x_h^\tau, a_h^\tau) \right] \right\|_{(\widetilde{\Lambda}_h^t)^{-1}}^2 \\
&\leq 4H^2 \cdot \Gamma_{\widetilde{K}}(T, \widetilde{\lambda}) + 11H^2 + 4H^2 \cdot \log N_\infty(\epsilon^*, h+1, B_T) + 8H^2 \cdot \log(TH).
\end{aligned}$$
(H.83)

Thus, combining (H.71), (H.72), (H.77), and (H.83), we obtain that
$$\left| \varphi(z)^\top (\overline{W}_h^t - \widetilde{W}_h^t) \right|$$
$$\leq \left| \text{Term (iii)} \right| + \left| \text{Term (iv)} \right| \leq \left[ R_Q H + \sqrt{2 \cdot \text{Term (v)}} + R_Q H \cdot \sqrt{\lambda/d} \right] \cdot \overline{b}_h^t(z)$$
$$\leq H \cdot \left\{ \left[ 8\Gamma_{K_m}(T, \lambda) + 22 + 8 \cdot \log N_\infty(\epsilon^*, h+1, B_T) + 16 \cdot \log(TH) \right]^{1/2} + R_Q \cdot (1 + \sqrt{\lambda/d}) \right\} \cdot \overline{b}_h^t(z).$$
Using the elementary inequality $a + b \leq \sqrt{2(a^2 + b^2)}$, we have
$$\left| \varphi(z)^\top (\overline{W}_h^t - \widetilde{W}_h^t) \right|$$
$$\leq H \cdot \left[ 16\Gamma_{K_m}(T, \lambda) + 16 \cdot \log N_\infty(\epsilon^*, h+1, B_T) + 32 \cdot \log(TH) + 2R_Q^2 \cdot (1 + \sqrt{\lambda/d})^2 \right]^{1/2} \cdot \overline{b}_h^t(z)$$
$$\leq H \cdot \left[ 16\Gamma_{K_m}(T, \lambda) + 16 \cdot \log N_\infty(\epsilon^*, h+1, B_T) + 32 \cdot \log(TH) + 4R_Q^2 \cdot (1 + \lambda/d) \right]^{1/2} \cdot \overline{b}_h^t(z).$$
By the choice of $B_T$ in (D.3), we have that
$$\left| \varphi(z)^\top (\overline{W}_h^t - \widetilde{W}_h^t) \right| = \left| \varphi(z)^\top (\overline{W}_h^t - W^{(0)}) - \widetilde{Q}_h^t(z) \right| \leq \beta \cdot \overline{b}_h^t(z)$$
holds simultaneously for all $(t, h) \in [T] \times [H]$ and $z \in \mathcal{Z}$ with probability at least $1 - (2T^2 H^2)^{-1}$.

Thus, combining this with (H.39) and (H.40), we have
$$\left| \varphi(z)^\top (\overline{W}_h^t - W^{(0)}) - \mathbb{T}_h^\star Q_{h+1}^t(z) \right| \leq \beta \cdot \overline{b}_h^t(z) + 10C_{\text{act}} \cdot R_Q H \cdot \sqrt{\log(mTH)/m}. \quad \text{(H.84)}$$
By the definition of $\overline{Q}_h^t$ in (H.48), we have
$$\begin{aligned}
\overline{Q}_h^t(z) - \mathbb{T}_h^\star Q_{h+1}^t(z) &\leq \varphi(z)^\top (\overline{W}_h^t - W^{(0)}) - \mathbb{T}_h^\star Q_{h+1}^t(z) + \beta \cdot \overline{b}_h^t(z) \\
&\leq 2\beta \cdot \overline{b}_h^t(z) + 10C_{\text{act}} \cdot R_Q \cdot \sqrt{\log(mTH)/m}.
\end{aligned}$$
(H.85)
Moreover, since $\mathbb{T}_h^\star Q_{h+1}^t \leq H - h + 1$, by (H.84) we have
$$\begin{aligned}
\mathbb{T}_h^\star Q_{h+1}^t(z) - \overline{Q}_h^t(z) &= \mathbb{T}_h^\star Q_{h+1}^t(z) - \min\left\{ \varphi(x, a)^\top (\overline{W}_h^t - W^{(0)}) + \beta \cdot \overline{b}_h^t(x, a), H - h + 1 \right\}^+ \\
&= \max\left\{ \mathbb{T}_h^\star Q_{h+1}^t(z) - \varphi(z)^\top (\overline{W}_h^t - W^{(0)}) - \beta \cdot \overline{b}_h^t(z), 0 \right\}^+ \\
&\leq 10C_{\text{act}} \cdot R_Q \cdot \sqrt{\log(mTH)/m}.
\end{aligned}$$
(H.86)
Let $\iota$ denote $T^{7/12} \cdot H^{1/12} \cdot m^{-1/12} \cdot (\log m)^{1/4}$. When $m$ is sufficiently large, it holds that
$$10C_{\text{act}} \cdot R_Q \cdot \sqrt{\log(mTH)/m} \leq \iota \leq \beta \cdot \iota.$$
Meanwhile, combining the definition of the TD error $\delta_h^t$ in (E.1) and (H.69), we have
$$\begin{aligned}
\left| \delta_h^t(z) - \left[ \mathbb{T}_h^\star Q_{h+1}^t(z) - \overline{Q}_h^t(z) \right] \right| &= \left| Q_h^t(z) - \overline{Q}_h^t(z) \right| \\
&\leq 4\beta \cdot T^{7/12} \cdot H^{1/12} \cdot m^{-1/12} \cdot (\log m)^{1/4}.
\end{aligned}$$
(H.87)

Finally, combining (H.85), (H.86), and (H.87), we establish that, with probability at least $1 - (2T^2H^2)^{-1}$,

$$\delta_h^t(z) \leq [\mathbb{T}_h^\star Q_{h+1}^t(z) - \overline{Q}_h^t(z)] + 4\beta \cdot \iota \leq 5\beta \cdot \iota$$

$$\delta_h^t(z) \geq [\mathbb{T}_h^\star Q_{h+1}^t(z) - \overline{Q}_h^t(z)] - 4\beta \cdot \iota \geq -2\beta \cdot \overline{b}_h^t(z) - 5\beta \cdot \iota$$

hold for all $(t, h) \in [T] \times [H]$ simultaneously. Finally, combining this with (H.68), we have

$$-2\beta \cdot b_h^t - 9\beta \cdot \iota \leq -2\beta \cdot \overline{b}_h^t - 5\beta \cdot \iota \leq \delta_h^t(z) \leq 5\beta \cdot \iota,$$

which, together with (H.68), concludes the proof of Lemma E.4. $\qquad\square$

# I  Covering Number and Effective Dimension

In this section, we present results on the covering number of the class of value functions that we study and the effective dimension of the corresponding RKHS. Both of these results play a key role in establishing our regret bounds.

## I.1  Covering Number of the Classes of Value Functions

For any $R, B > 0$, any $h \in [H]$, and fixed $\mathcal{D}$, we define $\mathcal{Q}_{\text{ucb}}(h, R, B)$ as the function class that contains functions on $\mathcal{Z}$ that take the following form:

$$Q(z) = \min\big\{\theta(z) + \beta \cdot \lambda^{-1/2} \big[K(z, z) - k_t(z)^\top (K_t + \lambda I)^{-1} k_t(z)\big]^{1/2}, H - h + 1\big\}^+, \quad \text{(I.1)}$$

where $\theta \in \mathcal{H}$ satisfies $\|\theta\|_{\mathcal{H}} \leq R$, $\beta \in [0, B]$, $h \in [H]$, and $\mathcal{D} = \{z^\tau = (x^\tau, a^\tau),\}_{\tau \in [t]}$ is a finite subset of $\mathcal{Z}$ with $t$ elements, where $t \leq T$. Here $T$ is the total number of the episodes. Moreover, $K_t \in \mathbb{R}^{t \times t}$ and $k_t \colon \mathcal{Z} \to \mathbb{R}^t$ are defined similarly as in (3.7) based on state-action pairs in $\mathcal{D}$, that is,

$$K_t = [K(z^\tau, z^{\tau'})]_{\tau, \tau' \in [t]} \in \mathbb{R}^{t \times t}, \qquad k_t(z) = \big[K(z^1, z), \ldots K(z^t, z)\big]^\top \in \mathbb{R}^t.$$

By definition, $Q$ in (I.1) is determined by $Q_0 \in \mathcal{H}$ and a bonus term constructed using $\mathcal{D}$. Thus, the function $Q_h^t$ constructed in Algorithm 2 belongs to $\mathcal{Q}_{\text{ucb}}(h, R, B)$ when $\beta \leq B$ and $\|\alpha_h^t\|_{\mathcal{H}} \leq R$. In the following, for any $\epsilon \in (0, 1)$, we let $\mathcal{C}(\mathcal{Q}_{\text{ucb}}(h, R, B), \epsilon)$ be the minimal $\epsilon$-cover of $\mathcal{Q}_{\text{ucb}}(h, R, B)$ with respect to the $\ell_\infty$-norm on $\mathcal{Z}$. That is, for any $Q \in \mathcal{Q}_{\text{ucb}}(h, R, B)$, there exists $Q' \in \mathcal{C}(\mathcal{Q}_{\text{ucb}}(h, R, B), \epsilon)$ satisfying $\|Q - Q'\|_\infty \leq \epsilon$. Moreover, among all function classes that possess such a property, $\mathcal{C}(\mathcal{Q}_{\text{ucb}}(h, R, B), \epsilon)$ has the smallest cardinality. Thus, by definition, $|\mathcal{C}(\mathcal{Q}_{\text{ucb}}(h, R, B), \epsilon)|$ is the $\epsilon$-covering number of $\mathcal{Q}_{\text{ucb}}(h, R, B)$ with respect to the $\ell_\infty$-norm on $\mathcal{Z}$.

In addition, based on $\mathcal{Q}_{\text{ucb}}(h, R, B)$, we define the function class $\mathcal{V}_{\text{ucb}}(h, R, B)$ as

$$\mathcal{V}_{\text{ucb}}(h, R, B) = \big\{V \colon V(\cdot) = \max_a Q(\cdot, a) \text{ for some } Q \in \mathcal{Q}_{\text{ucb}}(h, R, B)\big\}. \quad \text{(I.2)}$$

For any two value functions $V_1, V_2 \colon \mathcal{S} \to \mathbb{R}$, we denote their supremum norm distance as

$$\text{dist}(V_1, V_2) = \sup_{x \in \mathcal{S}} \big|V_1(x) - V_2(x)\big|. \quad \text{(I.3)}$$

For any $\epsilon \in (0, 1)$, we let $\mathcal{C}(\mathcal{V}_{\text{ucb}}(h, R, B), \epsilon)$ denote the minimal $\epsilon$-cover of $\mathcal{V}_{\text{ucb}}(h, R, B)$ with respect to $\text{dist}(\cdot, \cdot)$ defined in (I.3).

The main result of this section is a set of upper bounds on the size of $\mathcal{C}(\mathcal{V}_{\text{ucb}}(h, R, B), \epsilon)$ under the two eigenvalue decay conditions specified in Assumption 4.3.

**Lemma I.1.** Let Assumption 4.3 hold and $\lambda$ be bounded in $[c_1, c_2]$, where both $c_1$ and $c_2$ are absolute constants. Then, for any $h \in [H]$, any $R, B > 0$, and any $\epsilon \in (0, 1/e)$, there exists a positive constant $C_N$ such that

$$\log\big|\mathcal{C}\big(\mathcal{V}_{\text{ucb}}(h, R, B), \epsilon\big)\big| \leq \log\big|\mathcal{C}\big(\mathcal{Q}_{\text{ucb}}(h, R, B), \epsilon\big)\big| \quad \text{(I.4)}$$

$$\leq \begin{cases} C_N \cdot \gamma \cdot \big[1 + \log(R/\epsilon)\big] + C_N \cdot \gamma^2 \cdot \big[1 + \log(B/\epsilon)\big] & \text{case (i)}, \\ C_N \cdot \big[1 + \log(R/\epsilon)\big]^{1+1/\gamma} + C_N \cdot \big[1 + \log(B/\epsilon)\big]^{1+2/\gamma} & \text{case (ii)}, \end{cases} \quad \text{(I.5)}$$

where cases (i) and (ii) above correspond to the two eigenvalue decay conditions specified in Assumption 4.3, respectively. Moreover, here $C_N$ in (I.4) is independent of $T$, $H$, $R$, and $B$, and only depends on $C_\psi$, $C_1$, $C_2$, $c_1$, $c_2$, $\gamma$, and $\tau$.

*Proof.* For any fixed subset $\mathcal{D} = \{z^\tau\}_{\tau \in [t]}$ of $\mathcal{Z}$ with size $t \in [T]$, we define $\Phi_\mathcal{D} \colon \mathcal{H} \to \mathbb{R}^t$ and $\Lambda_\mathcal{D} \colon \mathcal{H} \to \mathcal{H}$ respectively as

$$\Phi_\mathcal{D} = \left[\phi(z^1)^\top, \phi(z^2)^\top \ldots, \phi(z^t)^\top\right]^\top,$$

$$\Lambda_\mathcal{D} = \sum_{\tau=1}^t \phi(z^\tau)\phi(z^\tau)^\top + \lambda \cdot I_\mathcal{H} = \lambda \cdot I_\mathcal{H} + (\Phi_\mathcal{D})^\top \Phi_\mathcal{D}, \tag{I.6}$$

where $\phi \colon \mathcal{Z} \to \mathcal{H}$ is the feature mapping of $\mathcal{H}$ and $\mathcal{I}_\mathcal{H}$ is the identity mapping on $\mathcal{H}$. Then, we can equivalently write $Q_1 \in \mathcal{Q}_{\mathrm{ucb}}(h, R, B)$ as

$$Q_1(z) = \phi(z)^\top \theta_1 + \beta \cdot \sqrt{\phi(z)^\top \Lambda_{\mathcal{D}_1}^{-1} \phi(z)}, \tag{I.7}$$

where $\theta_1 \in \mathcal{H}$ has an RKHS norm bounded by $R$, $\beta_1 \in [0, B]$, and $\mathcal{D}_1$ is a finite subset of $\mathcal{Z}$ with size $t_1 \leq T$. Let $V_1(\cdot) = \max_{a \in \mathcal{A}} Q_1(\cdot, a)$. Combining (I.2) and (I.7), we can write $V_1 \in \mathcal{V}_{\mathrm{ucb}}(h, R, B)$ as

$$V_1(\cdot) = \min\left\{\max_a\left\{\phi(\cdot, a)^\top \theta_1 + \beta_1 \cdot \sqrt{\phi(\cdot, a)^\top \Lambda_{\mathcal{D}_1}^{-1} \phi(\cdot, a)}\right\}, H - h + 1\right\}^+, \tag{I.8}$$

Similar to $V_1$ in (I.8), consider any $V_2 \colon \mathcal{S} \to \mathbb{R}$ that can be written as

$$V_2(\cdot) = \min\left\{\max_a\left\{f_1(\cdot, a) + \beta_2 \cdot f_2(\cdot, a)\right\}, H - h + 1\right\}^+, \tag{I.9}$$

where $Q_2 = f_1 + \beta_2 \cdot f_2$ for some $f_1, f_2 \colon \mathcal{Z} \to \mathbb{R}$ and $\beta_2 \in [0, B]$. Since both $\min\{\cdot, H - h + 1\}^+$ and $\max_a$ are contractive mappings, by (I.8) and (I.9) we have

$$\mathrm{dist}(V_1, V_2) \leq \sup_{(x,a) \in \mathcal{Z}} \left|\left[\phi(x, a)^\top \theta_1 + \beta_1 \cdot \sqrt{\phi(x, a)^\top \Lambda_{\mathcal{D}_1}^{-1} \phi(x, a)}\right]\right.$$

$$\left. - \left[f(x, a) + \beta_2 \cdot f_2(x, a)\right]\right| = \|Q_1 - Q_2\|_\infty,$$

which implies that

$$\log\left|\mathcal{C}\left(\mathcal{V}_{\mathrm{ucb}}(h, R, B), \epsilon\right)\right| \leq \log\left|\mathcal{C}\left(\mathcal{Q}_{\mathrm{ucb}}(h, R, B), \epsilon\right)\right|.$$

Moreover, by the triangle inequality, we have

$$\|Q_1 - Q_2\|_\infty \leq \sup_{(x,a) \in \mathcal{Z}} \left|\phi(x, a)^\top \theta_1 - f_2(x, a)\right| + |\beta_1 - \beta_2| \cdot \sup_{(x,a) \in \mathcal{Z}} \left\|\phi(x, a)\right\|_{\Lambda_{\mathcal{D}_1}^{-1}}$$

$$+ B \cdot \sup_{(x,a) \in \mathcal{Z}} \left|\left\|\phi(x, a)\right\|_{\Lambda_{\mathcal{D}_1}^{-1}} - f_2(x, a)\right|, \tag{I.10}$$

where we denote $\left\|\phi(x, a)\right\|_{\Lambda_{\mathcal{D}_1}^{-1}}^2 = \phi(x, a)^\top \Lambda_{\mathcal{D}_1}^{-1} \phi(x, a)$. Notice that by the reproducing property we have $\phi(x, a)^\top \theta = \langle \theta, \phi(x, a)\rangle_\mathcal{H} = \theta(x, a)$ for all $\theta \in \mathcal{H}$ and $(x, a) \in \mathcal{Z}$. Also note that

$$\left\|\phi(x, a)\right\|_{\Lambda_{\mathcal{D}_1}^{-1}}^2 \leq 1/\lambda \cdot \left\|\phi(x, a)\right\|^2 \leq 1/\lambda \cdot K(z, z) \leq 1/\lambda.$$

Thus, by (I.10) we have

$$\|Q_1 - Q_2\|_\infty \leq \sup_{(x,a) \in \mathcal{Z}} \left|\theta_1(x, a) - f_1(x, a)\right| + |\beta_1 - \beta_2|/\lambda$$

$$+ B \cdot \sup_{(x,a) \in \mathcal{Z}} \left|\left\|\phi(x, a)\right\|_{\Lambda_{\mathcal{D}_1}^{-1}} - f_2(x, a)\right|. \tag{I.11}$$

Thus, by (I.11), to get the covering number of $\mathcal{Q}_{\mathrm{ucb}}(h, R, B)$ with respect to $\mathrm{dist}(\cdot, \cdot)$, it suffices to bound the covering numbers of the RKHS norm ball $\{f \in \mathcal{H} \colon \|f\|_\mathcal{H} \leq R\}$, the interval $[0, B]$, and the set of functions that are of the form of $\|\phi(\cdot)\|_{\Lambda_\mathcal{D}^{-1}}$, respectively.

Notice that, by the definition in (I.6), $\Lambda_\mathcal{D} \colon \mathcal{H} \to \mathcal{H}$ is a self-adjoint operator on $\mathcal{H}$ with eigenvalues bounded in $[0, 1/\lambda]$. To simplify the notation, we define the function class $\mathcal{F}(\lambda)$ as

$$\mathcal{F}(\lambda) = \left\{\|\phi(\cdot)\|_\Upsilon = \left[\phi(\cdot)^\top \Upsilon \phi(\cdot)\right]^{1/2} \colon \|\Upsilon\|_{\mathrm{op}} \leq 1/\lambda\right\}, \tag{I.12}$$

where $\Upsilon \colon \mathcal{H} \to \mathcal{H}$ in (I.12) is a self-adjoint operator on $\mathcal{H}$ whose eigenvalues are all bounded by $1/\lambda$ in magnitude. Here, the operator norm of $\Upsilon$ is defined as

$$\|\Upsilon\|_{\mathrm{op}} = \sup\{f^\top \Upsilon f \colon f \in \mathcal{H}, \|f\|_\mathcal{H} = 1\} = \sup\{\langle f, \Upsilon f\rangle_\mathcal{H} \colon f \in \mathcal{H}, \|f\|_\mathcal{H} = 1\}.$$

Thus, by definition, for any finite subset $\mathcal{D}$ of $\mathcal{Z}$, $\|\phi(\cdot)\|_{\Lambda_\mathcal{D}^{-1}}$ belongs to $\mathcal{F}(\lambda)$, where $\Lambda_\mathcal{D}$ is defined in (I.6). For any $\epsilon \in (0, 1)$, we let $N_\infty(\epsilon, \mathcal{F}, \lambda)$ denote the $\epsilon$-covering number of $\mathcal{F}(\lambda)$ in (I.12)

with respect to the $\ell_\infty$-norm. Moreover, let $N_\infty(\epsilon, \mathcal{H}, R)$ denote the $\epsilon$-covering number of $\{f \in \mathcal{H} : \|f\|_{\mathcal{H}} \le R\}$ with respect to the $\ell_\infty$-norm and let $N(\epsilon, B)$ denote the $\epsilon$-covering number of the interval $[0, B]$ with respect the Euclidean distance. Then, by (I.11) we obtain that

$$\big|\mathcal{C}\big(\mathcal{Q}_{\mathrm{ucb}}(h, R, B), \epsilon\big)\big| \le N_\infty(\epsilon/3, \mathcal{H}, R) \cdot N(\epsilon \cdot \lambda/3, B) \cdot N_\infty\big(\epsilon/(3B), \mathcal{F}, \lambda\big). \tag{I.13}$$

As shown in [69, Corollary 4.2.13], it holds that

$$N(\epsilon \cdot \lambda/3, B) \le 1 + 6B/(\epsilon \cdot \lambda) \le 1 + 6B/\epsilon, \tag{I.14}$$

where the last inequality follows from the fact that $\lambda \in [1, 2]$.

It remains to bound the first and the third terms on the right-hand side of (I.13) separately. We establish the $\ell_\infty$-covering of the RKHS norm ball and $F(\lambda)$ in the following two lemmas, respectively.

**Lemma I.2** ($\ell_\infty$-norm covering number of RKHS ball). *For any $\epsilon \in (0, 1)$, we let $N_\infty(\epsilon, \mathcal{H}, R)$ denote the $\epsilon$-covering number of the RKHS norm ball $\{f \in \mathcal{H} : \|f\|_{\mathcal{H}} \le R\}$ with respect to the $\ell_\infty$-norm. Consider the two eigenvalue decay conditions given in Assumption 4.3. Then, under Assumption 4.3, there exist absolute constants $C_3$ and $C_4$ such that*

$$\log N_\infty(\epsilon, \mathcal{H}, R) \le \begin{cases} C_3 \cdot \gamma \cdot \big[\log(R/\epsilon) + C_4\big] & \gamma\text{-finite spectrum}, \\ C_3 \cdot \big[\log(R/\epsilon) + C_4\big]^{1+1/\gamma} & \gamma\text{-exponential decay}, \end{cases}$$

*where $C_3$ and $C_4$ are independent of $T$, $H$, $R$, and $\epsilon$, and only depend on absolute constants $C_\psi$, $C_1$, $C_2$, $\gamma$, and $\tau$ specified in Assumption 4.3.*

*Proof.* See §J.2 for a detailed proof. $\qquad\square$

**Lemma I.3.** *For any $\epsilon \in (0, 1/e)$, let $N_\infty(\epsilon, \mathcal{F}, \lambda)$ be the $\epsilon$-covering number of function class $\mathcal{F}(\lambda)$ with respect to the $\ell_\infty$-norm, where $\mathcal{F}(\lambda)$ is defined in (I.12). Here we assume that $\lambda$ is bounded in $[c_1, c_2]$, where both $c_1$ and $c_2$ are absolute constants. Then, under Assumption 4.3, there exist absolute constants $C_5$ and $C_6$ such that*

$$\log N_\infty(\epsilon, \mathcal{F}, \lambda) \le \begin{cases} C_5 \cdot \gamma^2 \cdot \big[\log(1/\epsilon) + C_6\big] & \gamma\text{-finite spectrum}, \\ C_5 \cdot \big[\log(1/\epsilon) + C_6\big]^{1+2/\gamma} & \gamma\text{-exponential decay} \end{cases}$$

*where $C_5$ and $C_6$ only depend on $C_\psi$, $C_1$, $C_2$, $\gamma$, $\tau$, $c_1$, and $c_2$, and do not rely on $T$, $H$, or $\epsilon$.*

*Proof.* See §J.3 for a detailed proof. $\qquad\square$

Finally, we conclude the proof by combining Lemmas I.2 and I.3. Specifically, by (I.13) and (I.14), we have

$$\log\big|\mathcal{C}\big(\mathcal{Q}_{\mathrm{ucb}}(h, R, B), \epsilon\big)\big| \le \log N_\infty(\epsilon/3, \mathcal{H}, R) + \log N(\epsilon \cdot \lambda/3, B) + \log N_\infty\big(\epsilon/(3B), \mathcal{F}, \lambda\big) \tag{I.15}$$

$$\le \log\big[1 + 6B/(\epsilon \cdot \lambda)\big] + \log N_\infty(\epsilon/3, \mathcal{H}, R) + \log N_\infty\big(\epsilon/(3B), \mathcal{F}, \lambda\big).$$

We consider the two eigenvalue decay conditions separately. For the $\gamma$-finite spectrum case, by Lemmas I.2 and I.3 and (I.15) we have

$$\log\big|\mathcal{C}\big(\mathcal{Q}_{\mathrm{ucb}}(h, R, B), \epsilon\big)\big|$$

$$\le \log\big[1 + 6B/(\epsilon \cdot \lambda)\big] + C_3 \cdot \gamma \cdot \big[\log(3R/\epsilon) + C_4\big] + C_5 \cdot \gamma^2 \cdot \big[\log(3B/\epsilon) + C_6\big]$$

$$\le C_N \cdot \gamma \cdot \big[1 + \log(R/\epsilon)\big] + C_N \cdot \gamma^2 \cdot \big[1 + \log(B/\epsilon)\big],$$

where $C_N$ is an absolute constant. Similarly, for the case where the eigenvalues satisfy the $\gamma$-exponential decay condition, by Lemmas I.2 and I.3 we have

$$\log\big|\mathcal{C}\big(\mathcal{Q}_{\mathrm{ucb}}(h, R, B), \epsilon\big)\big|$$

$$\le \log\big[1 + 6B/(\epsilon \cdot \lambda)\big] + C_3 \cdot \big[\log(3R/\epsilon) + C_4\big]^{1+1/\gamma} + C_5 \cdot \big[\log(3B/\epsilon) + C_6\big]^{1+2/\gamma}$$

$$\le C_N \cdot \big[1 + \log(R/\epsilon)\big]^{1+1/\gamma} + C_N \cdot \big[1 + \log(B/\epsilon)\big]^{1+2/\gamma}$$

for some absolute constant $C_N > 0$. Therefore, we conclude the proof. $\qquad\square$

## I.2 Effective Dimension of RKHS

**Definition I.4** (Maximal information gain). For any fixed integer $T$ and any $\sigma > 0$, we define the maximal information gain associated with the RKHS $\mathcal{H}$ as

$$\Gamma_K(T, \sigma^2) = \sup_{\mathcal{D} \subseteq \mathcal{Z}} \left\{ 1/2 \cdot \log\det(I + \sigma^{-2} \cdot K_{\mathcal{D}}) \right\}, \qquad (\text{I.16})$$

where the supremum is taken over all discrete subsets of $\mathcal{Z}$ with cardinality no more than $T$, and $K_D$ is the Gram matrix induced by $\mathcal{D} \subseteq \mathcal{Z}$, which is defined similarly as in (3.7). Here the subscript $K$ in $\Gamma_K(T, \sigma^2)$ denotes the kernel function of $\mathcal{H}$.

The maximal information gain naturally arises in Gaussian process regression. Specifically, let $f \sim \text{GP}(0, K)$ be draw from the Gaussian process with covariance kernel $K$. Let $\mathcal{D} = \{z_1, \ldots, z_{|\mathcal{D}|}\}$ be a subset of $\mathcal{Z}$ with $|\mathcal{D}| \leq T$ elements. Suppose that we observe noisy observations of $f$ at points in $\mathcal{D}$. That is, for any $z_i \in \mathcal{D}$, we have $y_i = f(z_i) + \epsilon_i$, where $\epsilon_i \sim N(0, \sigma^2)$ is a random Gaussian noise. We let $y_{\mathcal{D}}$ denote the vector whose entries are $y_i$. Then, the information gain of $y_{\mathcal{D}}$ is defined as the mutual information between $f$ and the observations $y_{\mathcal{D}}$, denoted by $I(f, y_{\mathcal{D}})$. By direct computation, we have

$$I(f, y_{\mathcal{D}}) = 1/2 \cdot \log\det(I + \sigma^{-2} \cdot K_{\mathcal{D}}).$$

The mutual information $I(f, y_{\mathcal{D}})$ quantifies the reduction of the uncertainty about $f$ when we observe $y_{\mathcal{D}}$. Thus, the maximal mutual information $\Gamma_K(T, \sigma^2)$ characterizes the maximal possible reduction of the uncertainty of $f$ when having no more than $T$ observations.

Moreover, we note that, when $\sigma^2$ is a constant, $\Gamma_K(T, \sigma^2)$ depends on the eigenvalue decay of the RKHS and thus can be viewed as an effective dimension of the RKHS. Specifically, as shown in [62], when the kernel is the $d$-dimensional linear kernel, $\Gamma_K(T, \sigma^2) = \mathcal{O}(d \log T)$. Moreover, for the squared exponential kernel that satisfies the exponential eigenvalue decay condition, the maximal information gain is $\mathcal{O}((\log T)^{d+1})$. In the following lemma, similar to Theorem 5 in [62], we establish upper bounds on the maximal information gain of the RKHS under the eigenvalue decay conditions specified in Assumption 4.3.

**Lemma I.5** (Theorem 5 in [62]). Let $\mathcal{Z}$ be a compact subset of $\mathbb{R}^d$ and $K \colon \mathcal{Z} \times \mathcal{Z} \to \mathbb{R}$ be the RKHS kernel of $\mathcal{H}$. We assume that $K$ is a bounded kernel in the sense that $\sup_{z \in \mathcal{Z}} K(z, z) \leq 1$, and $K$ is continuously differentiable on $\mathcal{Z} \times \mathcal{Z}$. Moreover, let $T_K$ be the integral operator induced by $K$ and the Lebesgue measure on $\mathcal{Z}$, whose definition is given in (B.1). Let $\{\sigma_j\}_{j \geq 1}$ be the eigenvalues of $T_K$ in the descending order. We assume that $\{\sigma_j\}_{j \geq 1}$ satisfy either one of the following three eigenvalue decay conditions:

(i) $\gamma$-finite spectrum: We have $\sigma_j = 0$ for all $j \geq \gamma + 1$, where $\gamma$ is a positive integer.

(ii) $\gamma$-exponential eigenvalue decay: There exist constants $C_1, C_2 > 0$ such that $\sigma_j \leq C_1 \exp(-C_2 \cdot j^{\gamma})$ for all $j \geq 1$, where $\gamma > 0$ is positive constant.

Let $\sigma$ be bounded in interval $[c_1, c_2]$ with $c_1$ and $c_2$ being absolute constants. Then, for conditions (i)–(iii) respectively, we have

$$\Gamma_K(T, \sigma^2) \leq \begin{cases} C_K \cdot \gamma \cdot \log T & \gamma\text{-finite spectrum,} \\ C_K \cdot (\log T)^{1+1/\gamma} & \gamma\text{-exponential decay,} \end{cases}$$

where $C_K$ is an absolute constant that depends on $d$, $\gamma$, $C_1$, $C_2$, $C$, $c_1$, and $c_2$.

We note that Lemma I.5 is a generalization of Theorem 5 in [62], which establishes the maximal information gain for the linear, squared exponential, and Matérn kernels, respectively. Specifically, the squared exponential kernel satisfies the $\gamma$-exponential eigenvalue decay condition with $\gamma = 1/d$. Lemma I.5 implies that the $\Gamma_K(T, \sigma^2) = \mathcal{O}((\log T)^{d+1})$, which matches Theorem 5 in [62].

*Proof.* The proof of this lemma is based on a modification of that of Theorem 5 in [62]. To begin with, for any $j \in \mathbb{N}$, we define $B_K(j) = \sum_{s > j} \sigma_s$, i.e., the sum of eigenvalues with indices larger

than $j$. Then, we use the following lemma obtained from [62] to bound $\Gamma_K(T, \sigma^2)$ using function $B_K$.

**Lemma I.6** (Theorem 8 in [62])**.** Under the same condition as in Lemma I.5, for any fixed $\tau > 0$, we denote $C_\tau = 2\mu(\mathcal{Z}) \cdot (2\tau + 1)$ where $\mu(\mathcal{Z})$ is the Lebesgue measure of $\mathcal{Z}$. Let $n_T$ denote $C_\tau \cdot T^\tau \cdot \log T$. Then, for any $T_\star \in \{1, \ldots, n_T\}$, we have
$$\Gamma_K(T, \sigma^2) \leq T_\star \cdot \log(T \cdot n_T/\sigma^2) + C_\tau \cdot \sigma^{-2} \cdot \log T \cdot \left[T^{\tau+1} \cdot B_K(T_\star) + 1\right] + \mathcal{O}(T^{1-\tau/d}).$$

*Proof.* See [62] for a detailed proof. $\qquad\square$

In the following, we choose proper $\tau$ and $T_\star$ in Lemma I.6 for the two eigenvalue decay conditions separately.

**Case (i): $\gamma$-Finite Spectrum.** When $\sigma_j = 0$ for all $j \geq \gamma + 1$, we set $\tau = d$ and $T_\star = \gamma$ in Lemma I.6. Then we have $B_K(T_\star) = 0$ and $n_T = C_d \cdot T^d \cdot \log T$. When $T$ is sufficiently large, it holds that $T_\star < n_T$. Then Lemma I.6 implies that
$$\Gamma_K(T, \sigma^2) \leq \gamma \cdot \log\left(C_d \cdot T^{d+1} \cdot \log T/\sigma^2\right) + C_d \cdot \sigma^{-2} \cdot \log T + \mathcal{O}(1) \leq C_K \cdot \gamma \cdot \log T,$$
for some absolute constant $C_K > 0$. Thus, we conclude the proof for the first case.

**Case (ii): $\gamma$-Exponential Decay.** When $\{\sigma_j\}_{j \geq 1}$ satisfies the $\gamma$-exponential eigenvalue decay condition, for any $T_\star \in \mathbb{N}$, we have
$$B_K(T_\star) = \sum_{j > T_\star} \sigma_j \leq C_1 \cdot \sum_{j > T_\star} \exp(-C_2 \cdot j^\gamma) \leq C_1 \cdot \int_{T_\star}^\infty \exp(-C_2 \cdot u^\gamma) \, \mathrm{d}u. \qquad (I.17)$$
In a manner similar to the derivation of (J.16), by direct computation we have
$$\int_{T_\star}^\infty \exp(-C_2 \cdot u^\gamma) \, \mathrm{d}u \leq \begin{cases} C_2^{-1} \cdot \exp(-C_2 \cdot T_\star^\gamma), & \text{if } \gamma \geq 1, \\ 2 \cdot (\gamma \cdot C_2)^{-1} \cdot \exp(-C_2 \cdot T_\star^\gamma) \cdot T_\star^{1-\gamma}, & \text{if } \gamma \in (0, 1). \end{cases} \qquad (I.18)$$
In the following, we set $\tau = d$. Then we have $n_T = C_d \cdot T^d \cdot \log T$ where $C_d = 2\mu(\mathcal{Z}) \cdot (2d+1)$. Then we have
$$\log(T \cdot n_T) = \log(C_d) + \log \cdot (T^{d+1} \cdot \log T) \leq \log(C_d) + 2(d+1) \cdot \log T, \qquad (I.19)$$
when $T$ is sufficiently large. Moreover, combining Lemma I.6 and (I.19), when $\sigma$ is sandwiched by absolute constants $c_1$ and $c_2$, we have
$$\Gamma_K(T, \sigma^2) \leq \widetilde{C}_1 \cdot T_\star \cdot \log T + \widetilde{C}_2 \cdot \log T \cdot \left[T^{d+1} \cdot B_K(T_\star) + 1\right] + \widetilde{C}_3, \qquad (I.20)$$
where $\widetilde{C}_1$, $\widetilde{C}_2$, and $\widetilde{C}_3$ are absolute constants that depend on $d, \gamma, c_1, c_2, C_1$, and $C_2$. Now we choose $T_\star$ such that
$$\exp(C_2 \cdot T_\star^\gamma) \asymp T \cdot n_T = C_d \cdot T^{d+1} \cdot \log T, \qquad (I.21)$$
that is, $T_\star = \widetilde{C}_4 \cdot (\log T)^{1/\gamma}$ where $\widetilde{C}_4$ is an absolute constant. Notice that $T_\star < n_T$ when $T$ is sufficiently large.

Thus, combining (I.17), (I.18), and (I.21), for $\gamma \geq 1$, we have
$$\begin{aligned} \log T \cdot &\left[T^{d+1} \cdot B_K(T_\star) + 1\right] \\ &\leq C_1 \cdot C_2^{-1} \log T \cdot T^{d+1} \cdot \exp(-C_2 \cdot T_\star^\gamma) + \log T \leq 2\log T, \end{aligned} \qquad (I.22)$$
where the last inequality follows from (I.21). Similarly, for $\gamma \in (0, 1)$, by (I.17), (I.18), and (I.21), we have
$$\begin{aligned} \log T \cdot &\left[T^{d+1} \cdot B_K(T_\star) + 1\right] \\ &\leq 2C_1 \cdot (\gamma \cdot C_2)^{-1} \cdot \exp(-C_2 \cdot T_\star^\gamma) \cdot \log T \cdot T^{d+1} \cdot T_\star^{1-\gamma} + \log T \asymp (\log T)^{1/\gamma-1} + \log T. \end{aligned}$$
$$(I.23)$$

Thus, combining (I.20), (I.22), (I.23), we conclude that
$$\Gamma_K(T, \sigma^2) \leq C_K \cdot \log(T)^{1+1/\gamma}$$
for any $\gamma \geq 0$, where $C_K$ is an absolute constant that depends on $d, \gamma, c_1, c_2, C_1$, and $C_2$. Thus, we conclude the proof for the second case. Therefore, we conclude the proof of Lemma I.5. $\qquad\square$

# J Proofs of Auxiliary Results

In this section, we provide the proofs of the auxiliary results.

## J.1 Proof of Lemma H.1

*Proof.* For any function $f \in \mathcal{H}$, using the feature representation induced by the kernel $K$, we have

$$\left| \langle f, \widehat{\theta}_h^t \rangle_{\mathcal{H}} \right| = \left| f^\top \widehat{\theta}_h^t \right| \le \left| f^\top (\Lambda_h^t)^{-1} \Phi^\top y_h^t \right|$$

$$= \left| f^\top (\Lambda_h^t)^{-1} \sum_{\tau=1}^{t-1} \phi(x_h^\tau, a_h^\tau) \cdot [r_h(x_h^\tau, a_h^\tau) + V_{h+1}^t(x_{h+1}^\tau)] \right|, \tag{J.1}$$

where we let $\Phi$ denote $\Phi_h^t$ defined in (H.13) for simplicity. Since $|r_h(x_h^\tau, a_h^\tau)| \le 1$ and $|V_{h+1}^t(x_{h+1}^\tau)| \le H - h$, we have $|[r_h(x_h^\tau, a_h^\tau) + V_{h+1}^t(x_{h+1}^\tau)]| \le H$ for all $h \in [H]$ and $\tau \in [t-1]$. Then, by (J.1) and the Cauchy-Schwarz inequality, we have

$$\left| \langle f, \widehat{\theta}_h^t \rangle_{\mathcal{H}} \right| \le H \cdot \sum_{\tau=1}^{t-1} \left| f^\top (\Lambda_h^t)^{-1} \phi(x_h^\tau, a_h^\tau) \right|$$

$$\le H \cdot \left[ \sum_{\tau=1}^{t-1} f^\top (\Lambda_h^t)^{-1} f \right]^{1/2} \cdot \left[ \sum_{\tau=1}^{t-1} \phi(x_h^\tau, a_h^\tau)^\top (\Lambda_h^t)^{-1} \phi(x_h^\tau, a_h^\tau) \right]^{1/2}$$

$$\le H/\sqrt{\lambda} \cdot \|f\|_{\mathcal{H}} \cdot \left[ \sum_{\tau=1}^{t-1} \phi(x_h^\tau, a_h^\tau)^\top (\Lambda_h^t)^{-1} \phi(x_h^\tau, a_h^\tau) \right]^{1/2}, \tag{J.2}$$

where the last inequality follows from the fact that $(\Lambda_h^t)^{-1} \colon \mathcal{H} \to \mathcal{H}$ is a self-adjoint and positive-definite operator whose eigenvalues are bounded by $1/\lambda$. Furthermore, by Lemma J.3, we have

$$\left[ \sum_{\tau=1}^{t-1} \phi(x_h^\tau, a_h^\tau)^\top (\Lambda_h^t)^{-1} \phi(x_h^\tau, a_h^\tau) \right] \le 2\log\det(I + K_h^t/\lambda). \tag{J.3}$$

Thus, combining (J.2), (J.3), and the fact that $\lambda \ge 1$, we obtain that

$$\left| \langle f, \widehat{\theta}_h^t \rangle_{\mathcal{H}} \right| \le H \cdot \|f\|_{\mathcal{H}} \cdot \sqrt{2/\lambda \cdot \log\det(I + K_h^t/\lambda)} \le H \cdot \|f\|_{\mathcal{H}} \cdot \sqrt{2 \cdot \log\det(I + K_h^t/\lambda)}.$$

Finally, utilizing the definition of $\Gamma_K(T, \lambda)$ in (I.16), we conclude the proof of this lemma. $\square$

## J.2 Proof of Lemma I.2

*Proof.* Recall that we have defined the integral operator $T_K \colon \mathcal{L}^2(\mathcal{Z}) \to \mathcal{L}^2(\mathcal{Z})$ defined in (B.1), which has eigenvalues $\{\sigma_j\}_{j \ge 0}$ and eigenvectors $\{\psi_j\}_{j \ge 0}$. Moreover, $\{\psi_j\}$ and $\{\sqrt{\sigma_j} \cdot \psi_j\}_{j \ge 0}$ are orthonormal bases of $\mathcal{L}_2(\mathcal{Z})$ and $\mathcal{H}$, respectively. Then, any $\in \mathcal{H}$ with $\|f\|_{\mathcal{H}} \le R$ can be written as

$$f = \sum_{j=1}^{\infty} w_j \cdot \sqrt{\sigma_j} \cdot \psi_j, \tag{J.4}$$

where $\{w_j\}_{j \ge 0}$ satisfy $\sum_{j=1}^{\infty} w_j^2 = \|f\|_{\mathcal{H}}^2 \le R^2$. Let $m$ be any positive integer and let $\Pi_m \colon \mathcal{H} \to \mathcal{H}$ denote the projection onto the subspace spanned by $\{\psi_j\}_{j \in [m]}$, i.e., $\Pi_m(f) = \sum_{j=1}^{m} w_j \cdot \sqrt{\sigma_j} \cdot \psi_j$ for any $f \in \mathcal{H}$ written as in (J.4). Then we have

$$\|f - \Pi_m(f)\|_{\infty} = \sum_{j=m+1}^{\infty} |w_j| \cdot \sqrt{\sigma_j} \cdot \sup_{z \in \mathcal{Z}} |\psi_j(z)|. \tag{J.5}$$

In the following, we consider the two eigenvalue decay conditions specified in Assumption 4.3 separately.

**Case (i): $\gamma$-Finite Spectrum.** Consider the case where $\sigma_j = 0$ for all $j > \gamma$. Then, by the definition of $\Pi_m$, we have $f = \Pi_\gamma(f)$ for all $f \in \mathcal{H}$. That is, (J.4) is reduced to

$$f = \sum_{j=1}^{\gamma} w_j \cdot \sqrt{\sigma_j} \cdot \psi_j,$$

where $\{w_j\}_{j\in[\gamma]}$ satisfies $\sum_{j=1}^{\gamma} w_j^2 \leq R^2$. Let $\mathcal{C}_\gamma(\epsilon, R)$ be the minimal $\epsilon$-cover of the $\gamma$-dimensional Euclidean ball $\{w \in \mathbb{R}^\gamma : \|w\|_2 \leq R\}$ with respect to the Euclidean norm. Then, by construction, there exists $\widetilde{w} \in \mathbb{R}^\gamma$ such that $\sum_{j=1}^{\gamma}(w_j - \widetilde{w}_j)^2 \leq \epsilon^2$. Then, by the Cauchy-Schwarz inequality, we have

$$\left\| f - \sum_{j=1}^{\gamma} \widetilde{w}_j \cdot \sqrt{\sigma_j} \cdot \psi_j \right\|_\infty = \sup_{z \in \mathcal{Z}} \left| \sum_{j=1}^{\gamma}(w_j - \widetilde{w}_j) \cdot \sqrt{\sigma_j} \cdot \psi_j(z) \right| \tag{J.6}$$

$$= \left[ \sum_{j=1}^{\gamma}(w_j - \widetilde{w}_j)^2 \right]^{1/2} \cdot \sup_{z \in \mathcal{Z}} \left\{ \left[ \sum_{j=1}^{\gamma} \sigma_j \cdot |\psi_j(z)|^2 \right]^{1/2} \right\} \leq \epsilon \cdot \sup_z \sqrt{K(z,z)} \leq \epsilon,$$

where the last equality follows from the fact that $K(z, z) = \sum_{j=1}^{\gamma} \sigma_j \cdot |\psi_j(z)|^2$. Thus, the $\epsilon$-covering of $\{f \in \mathcal{H} : \|f\|_{\mathcal{H}} \leq R\}$ is bounded by the cardinality of $\mathcal{C}_\gamma(\epsilon, R)$. As shown in [69, Corollary 4.2.13], we have

$$\left| \mathcal{C}_\gamma(\epsilon, R) \right| \leq (1 + 2R/\epsilon)^\gamma. \tag{J.7}$$

Thus, combining (J.6) and (J.7), we have

$$\log N_\infty(\epsilon, \mathcal{H}, R) \leq \gamma \cdot \log(1 + 2R/\epsilon) \leq C_3 \cdot \gamma \cdot \left[ \log(R/\epsilon) + C_4 \right],$$

where both $C_3$ and $C_4$ are absolute constants. Thus, we conclude the proof for the first case.

**Case (ii): $\gamma$-Exponential Decay.** In the following, we assume the eigenvalues $\{\sigma_j\}_{j\geq 1}$ satisfy the $\gamma$-exponential decay condition and $\|\psi_j\|_\infty \leq C_\psi \cdot \sigma_j^{-\tau}$ for all $j \geq 1$. Thus, by (J.5) we have

$$\|f - \Pi_m(f)\|_\infty \leq \sum_{j=m+1}^{\infty} C_\psi \cdot |w_j| \cdot \sigma_j^{1/2-\tau}$$

$$\leq \sum_{j=m+1}^{\infty} C_\psi \cdot C_1^{1/2-\tau} \cdot |w_j| \cdot \exp\left[ -C_2 \cdot (1/2-\tau) \cdot j^\gamma \right]. \tag{J.8}$$

To simplify the notation, we define $C_{1,\tau} = C_\psi \cdot C_1^{1/2-\tau}$ and $C_{2,\tau} = C_2 \cdot (1 - 2\tau)$. Then, applying the Cauchy-Schwarz inequality to (J.8), we have

$$\|f - \Pi_m(f)\|_\infty \leq C_{1,\tau} \cdot \left( \sum_{j=m+1}^{\infty} |w_j|^2 \right)^{1/2} \cdot \left[ \sum_{j=m+1}^{\infty} \exp(-C_{2,\tau} \cdot j^\gamma) \right]^{1/2}$$

$$\leq C_{1,\tau} \cdot R \cdot \left[ \sum_{j=m+1}^{\infty} \exp(-C_{2,\tau} \cdot j^\gamma) \right]^{1/2}, \tag{J.9}$$

where the second inequality follows from the fact that $\sum_{j\geq 1} w_j^2 \leq R^2$. Since $\gamma > 0$, $\exp(-u^\gamma)$ is monotonically decreasing in $u$. Thus, we have

$$\sum_{j=m+1}^{\infty} \exp(-C_{2,\tau} \cdot j^\gamma) \leq \int_m^\infty \exp(-C_{2,\tau} \cdot u^\gamma) \, \mathrm{d}u. \tag{J.10}$$

In the following, we bound the integral in (J.10) by considering the cases where $\gamma \geq 1$ and $\gamma \in (0, 1)$ separately. First, when $\gamma \geq 1$, since $d \geq 1$, we have $u^{\gamma-1} \geq 1$ for all $u \geq d$. Hence, we have

$$\int_m^\infty \exp(-C_{2,\tau} \cdot u^\gamma) \, \mathrm{d}u \leq \int_m^\infty u^{\gamma-1} \cdot \exp(-C_{2,\tau} \cdot u^\gamma) \, \mathrm{d}u$$

$$\leq \int_{m^\gamma}^\infty \exp(-C_{2,\tau} \cdot v) \, \mathrm{d}v = C_{2,\tau}^{-1} \cdot \exp(-C_{2,\tau} \cdot m^\gamma), \tag{J.11}$$

where the second inequality follows from the change of variable $v = u^\gamma$ and the fact that $\gamma \geq 1$. Second, when $\gamma < 1$, by letting $v = u^\gamma$, we have

$$\int_m^\infty \exp(-C_{2,\tau} \cdot u^\gamma) \, \mathrm{d}u = \frac{1}{\gamma} \cdot \int_{m^\gamma}^\infty \exp(-C_{2,\tau} \cdot v) \cdot v^{1/\gamma-1} \, \mathrm{d}v = \frac{1}{\gamma \cdot C_{2,\tau}} \int_{m^\gamma}^\infty v^{1/\gamma-1} \, \mathrm{d}[-\exp(-C_{2,\tau} \cdot v)]$$

$$= \frac{1}{\gamma \cdot C_{2,\tau}} \cdot \exp(-C_{2,\tau} \cdot m^\gamma) \cdot m^{1-\gamma} + \frac{(1-\gamma)}{\gamma^2 \cdot C_{2,\tau}} \int_{m^\gamma}^\infty \exp(-C_{2,\tau} \cdot v) \cdot v^{1/\gamma-2} \, \mathrm{d}v, \tag{J.12}$$

where the last equality follows from integration by parts. Moreover, by direct calculation, we have

$$\frac{1}{\gamma} \int_{m^\gamma}^\infty \exp(-C_{2,\tau} \cdot v) \cdot v^{1/\gamma - 2} \, \mathrm{d}v \leq \frac{1}{m^\gamma} \cdot \frac{1}{\gamma} \int_{m^\gamma}^\infty \exp(-C_{2,\tau} \cdot v) \cdot v^{1/\gamma - 1} \, \mathrm{d}v$$

$$= \frac{1}{m^\gamma} \int_m^\infty \exp(-C_{2,\tau} \cdot u^\gamma) \, \mathrm{d}u, \tag{J.13}$$

where the first inequality follows from the fact that $v \geq m^\gamma$ in the integral and the second equality follows from letting $u = v^{1/\gamma}$. Then, combining (J.12) and (J.13), we have

$$\int_m^\infty \exp(-C_{2,\tau} \cdot u^\gamma) \, \mathrm{d}u$$

$$\leq \frac{1}{\gamma \cdot C_{2,\tau}} \cdot \exp(-C_{2,\tau} \cdot m^\gamma) \cdot m^{1-\gamma} + \frac{1/\gamma - 1}{C_{2,\tau} \cdot m^\gamma} \cdot \int_m^\infty \exp(-C_{2,\tau} \cdot u^\gamma) \, \mathrm{d}u. \tag{J.14}$$

Thus, when $m$ is sufficiently large such that $m^\gamma \cdot C_{2,\tau} > 2/\gamma - 2$, by (J.14) we have

$$\int_m^\infty \exp(-C_{2,\tau} \cdot u^\gamma) \, \mathrm{d}u \leq \left(1 - \frac{1/\gamma - 1}{C_{2,\tau} m^\gamma}\right)^{-1} \cdot \frac{1}{\gamma \cdot C_{2,\tau}} \exp(-C_{2,\tau} \cdot m^\gamma) \cdot m^{1-\gamma}$$

$$\leq \frac{2}{\gamma \cdot C_{2,\tau}} \exp(-C_{2,\tau} \cdot m^\gamma) \cdot m^{1-\gamma}. \tag{J.15}$$

Therefore, combining (J.10), (J.11), and (J.15), we obtain that

$$\int_m^\infty \exp(-C_{2,\tau} \cdot u^\gamma) \, \mathrm{d}u \leq \begin{cases} C_{2,\tau}^{-1} \cdot \exp(-C_{2,\tau} \cdot m^\gamma), & \text{if } \gamma \geq 1, \\ 2 \cdot (\gamma \cdot C_{2,\tau})^{-1} \cdot \exp(-C_{2,\tau} \cdot m^\gamma) \cdot m^{1-\gamma}, & \text{if } \gamma \in (0,1). \end{cases} \tag{J.16}$$

In the sequel, we let $m^*$ be the smallest integer such that

$$\int_m^\infty \exp(-C_{2,\tau} \cdot u^\gamma) \, \mathrm{d}u \leq \left(\frac{\epsilon}{2C_{1,\tau} \cdot R}\right)^2, \qquad \forall m \geq m^*. \tag{J.17}$$

Hence, combining (J.9), (J.10), and (J.17), we have $\|f - \Pi_{m^*}(f)\|_\infty \leq \epsilon/2$ for any $f \in \mathcal{H}$ with $\|f\|_\mathcal{H} \leq R$. Note, moreover, that $C_{1,\tau}, C_{2,\tau}$, and $\gamma$ are all absolute constants. By (J.16) and (J.17), there exist absolute constants $C_{1,m}$ and $C_{2,m}$ such that

$$m^* \leq C_{1,m} \cdot \left[\log(R/\epsilon) + C_{2,m}\right]^{1/\gamma}. \tag{J.18}$$

Finally, it remains to approximate $\Pi_{m^*}(f)$ up to error $\epsilon/2$ for $m^*$ specified in (J.17). By the expansion of $f$ in (J.4), we have $\Pi_{m^*}(f) = \sum_{j=1}^{m^*} w_j \cdot \sqrt{\sigma_j} \cdot \psi_j$. For any $m^*$ real numbers $\{\widetilde{w}_j\}_{j \in [m^*]}$, by the Cauchy-Schwarz inequality, we have

$$\left|\left[\Pi_{m^*}(f)\right](z) - \sum_{j=1}^{m^*} \widetilde{w}_j \cdot \sqrt{\sigma_j} \cdot \psi_j(z)\right| = \left|\sum_{j=1}^{m^*} (w_j - \widetilde{w}_j) \cdot \sqrt{\sigma_j} \cdot \psi_j(z)\right|$$

$$\leq \left[\sum_{j=1}^{m^*} (w_j - \widetilde{w}_j)^2\right]^{1/2} \cdot \left\{\sum_{j=1}^{m^*} \sigma_j \cdot [\psi_j(z)]^2\right\}^{1/2} \leq \sqrt{K(z,z)} \cdot \left[\sum_{j=1}^{m^*} (w_j - \widetilde{w}_j)^2\right]^{1/2}, \tag{J.19}$$

where the last inequality follows from the fact that $K(z,z) = \sum_{j=1}^\infty \sigma_j \cdot [\psi_j(z)]^2$. Under Assumption 4.3, we have $\sup_{z \in \mathcal{Z}} K(z,z) \leq 1$. Notice that $\sum_{j=1}^{m^*} \omega_j^2 \leq \|f\|_\mathcal{H}^2 \leq R^2$. Let $\mathcal{C}_{m^*}(\epsilon/2, R)$ be the minimal $\epsilon/2$-cover of $\{w \in \mathbb{R}^{m^*} : \|w\|_2 \leq R\}$ with respect to the Euclidean norm. By definition, for any $f \in \mathcal{H}$ with $\|f\|_\mathcal{H} \leq R$, there exist $\widetilde{w} \in \mathcal{C}_{m^*}(\epsilon/2, R)$ such that $\sum_{j=1}^{m^*} (w_j - \widetilde{w}_j)^2 \leq \epsilon^2/4$. Therefore, by (J.19) we have

$$\left\|f - \sum_{j=1}^{m^*} \widetilde{w}_j \cdot \sqrt{\sigma_j} \cdot \psi_j\right\|_\infty \leq \|f - \Pi_{m^*}(f)\|_\infty + \left\|\Pi_{m^*}(f) - \sum_{j=1}^{m^*} \widetilde{w}_j \cdot \sqrt{\sigma_j} \cdot \psi_j\right\|_\infty \leq \epsilon, \tag{J.20}$$

which implies that the $\epsilon$-covering number of the RKHS norm ball $\{f \in \mathcal{H} : \|f\|_\mathcal{H} \leq R\}$ is bounded by the cardinality of $\mathcal{C}_{m^*}(\epsilon/2, R)$, i.e., $N_\infty(\epsilon, \mathcal{H}, R) \leq |\mathcal{C}_{m^*}(\epsilon/2, R)|$. As shown in [69, Corollary

4.2.13], we have

$$\left|\mathcal{C}_{m^*}(\epsilon/2, R)\right| \leq (1 + 4R/\epsilon)^{m^*}. \tag{J.21}$$

Therefore, combining (J.18) and (J.21), we have

$$\log N_\infty(\epsilon, \mathcal{H}, R) \leq m^* \cdot \log(1 + 4R/\epsilon) \leq C_{1,m} \cdot \left[\log(R/\epsilon) + C_{2,m}\right]^{1/\gamma} \cdot \left[\log(1 + 4R/\epsilon)\right]$$
$$\leq C_3 \cdot \left[\log(R/\epsilon) + C_4\right]^{1+1/\gamma},$$

where $C_3$ and $C_4$ are absolute constants that only depend on $C_\Psi$, $C_1$, $C_2$, $\gamma$, and $\tau$, which are specified in Assumption 4.3. Thus we conclude the proof of this lemma. $\qquad\square$

## J.3   Proof of Lemma I.3

*Proof.* As shown in §B.1, the feature mapping $\phi\colon \mathcal{Z} \to \mathcal{H}$ satisfies

$$\phi(z) = \sum_{j=1}^\infty \sigma_j \cdot \psi_j(z) \cdot \psi_j = \sum_{j=1}^\infty \sqrt{\sigma_j} \cdot \psi_j(z) \cdot (\sqrt{\sigma_j} \cdot \psi_j). \tag{J.22}$$

That is, when expanding $\phi(z) \in \mathcal{H}$ in the basis $\{\sqrt{\sigma_j} \cdot \psi_j\}_{j \geq 0}$ as in (J.4), the $j$-th coefficient is equal to $\sqrt{\sigma_j} \cdot \psi_j(z)$ for all $j \geq 1$. Similar to the proof of Lemma I.2, in the following, we consider the two eigenvalue decay conditions separately.

**Case (i): $\gamma$-Finite Spectrum.** When $\mathcal{H}$ has only $\gamma$ nonzero eigenvalues, for any $z \in \mathcal{Z}$, we define a vector $w_z \in \mathbb{R}^\gamma$ by letting its $j$-th entry be $\sqrt{\sigma_j} \cdot \psi_j(z)$ for all $j \in [\gamma]$. Moreover, for any self-adjoint operator $\Upsilon\colon \mathcal{H} \to \mathcal{H}$ satisfying $\|\Upsilon\|_{\mathrm{op}} \leq 1/\lambda$, we define a matrix $A_\Upsilon \in \mathbb{R}^{\gamma \times \gamma}$ as follows. For any $j, k \in [\gamma]$, we define the $(j, k)$-th entry of $A_\Upsilon$ as

$$[A_\Upsilon]_{j,k} = \left\langle \sqrt{\sigma_j} \cdot \psi_j, \sqrt{\sigma_k} \cdot \Upsilon\psi_k \right\rangle_{\mathcal{H}}.$$

By (J.22) and the definition of $A_\Upsilon$, we have

$$\|\phi(z)\|_\Upsilon^2 = \sum_{j,k=1}^\gamma \sqrt{\sigma_j} \cdot \psi_j(z) \cdot \sqrt{\sigma_k} \cdot \psi_k(z) \cdot [A_\Upsilon]_{j,k} = w_z^\top A_\Upsilon w_z. \tag{J.23}$$

With a slight abuse of notation, we define $\mathcal{C}_\gamma(\epsilon, \lambda)$ denote the minimal $\epsilon^2$-cover of

$$\left\{ A \in \mathbb{R}^{\gamma \times \gamma}\colon \|A\|_{\mathrm{fro}} \leq \sqrt{\gamma}/\lambda \right\}$$

with respect to the Frobenius norm. Then by definition, there exists $\widetilde{A}_\Upsilon \in \mathcal{C}_\gamma(\epsilon, \lambda)$ such that $\|A_\Upsilon - \widetilde{A}_\Upsilon\|_{\mathrm{fro}} \leq \epsilon^2$, which implies that

$$\left| w_z^\top A_\Upsilon w_z - w_z^\top \widetilde{A}_\Upsilon w_z \right| \leq \|w_z\|_2^2 \cdot \|A_\Upsilon - \widetilde{A}_\Upsilon\|_{\mathrm{op}} \leq \|A_\Upsilon - \widetilde{A}_\Upsilon\|_{\mathrm{fro}} \leq \epsilon^2, \tag{J.24}$$

where we use the fact that

$$\|w_z\|_2^2 = \sum_{j=1}^\gamma |w_j|^2 = \sum_{j=1}^\gamma \sigma_j \cdot |\psi_j(z)|^2 = K(z, z) \leq 1.$$

Thus, combining (J.23) and (J.24), and utilizing Corollary 4.2.13 in [69], we have

$$\log N_\infty(\epsilon, \mathcal{F}, \lambda) \leq \log\left|\mathcal{C}_\gamma(\epsilon, \lambda)\right| \leq \gamma^2 \cdot \log\left[1 + 8\sqrt{\gamma}/(\lambda \cdot \epsilon^2)\right] \leq C_5 \cdot \gamma^2 \cdot \left[\log(1/\epsilon) + C_6\right],$$

where $C_5$ and $C_6$ are absolute constants that depend solely on $\lambda$ and $\gamma$. Thus, we conclude the proof for the first case.

**Case (ii): $\gamma$-Exponential Decay.** In the following, we focus on the second case where the eigenvalues satisfy the $\gamma$-exponential decay condition. For any $m \in \mathbb{N}$, we define $\Pi_m\colon \mathcal{H} \to \mathcal{H}$ as the projection operator onto the subspace spanned by $\{\psi_j\}_{j \in [m]}$. Then, by the Cauchy-Schwarz inequality and Assumption 4.3, for any $z \in \mathcal{Z}$, by (J.22) we have

$$\left\|\phi(z) - \Pi_m\big[\phi(z)\big]\right\|_{\mathcal{H}} = \left\| \sum_{j=m+1}^\infty \sqrt{\sigma_j} \cdot \psi_j(z) \cdot \sqrt{\sigma_j} \cdot \psi_j \right\|_{\mathcal{H}} = \left\{ \sum_{j=m+1}^\infty \sigma_j \cdot [\psi_j(z)]^2 \right\}^{1/2}$$
$$\leq \left( \sum_{j=m+1}^\infty \sigma_j \cdot \|\psi_j\|_\infty^2 \right)^{1/2} \leq C_\psi \cdot \left( \sum_{j=m+1}^\infty \sigma_j^{1-2\tau} \right)^{1/2}, \tag{J.25}$$

where the second equality follows from the fact that $\{\sqrt{\sigma_j} \cdot \psi_j\}_{j \geq 0}$ form an orthonormal basis of $\mathcal{H}$, the first inequality follows from taking a supremum over $z \in \mathcal{Z}$, and the last inequality follows from the assumption that $\|\psi_j\|_\infty \leq C_\psi \cdot \sigma_j^{-\tau}$. Then, for any self-adjoint operator $\Upsilon \colon \mathcal{H} \to \mathcal{H}$ satisfying $\|\Upsilon\|_{\mathrm{op}} \leq 1/\lambda$ and any $z \in \mathcal{Z}$, by (J.25) and the triangle inequality we have

$$\left| \|\phi(z)\|_\Upsilon - \|\Pi_m[\phi(z)]\|_\Upsilon \right| \leq \|\phi(z) - \Pi_m[\phi(z)]\|_\Upsilon \leq C_\psi/\sqrt{\lambda} \cdot \left( \sum_{j=m+1}^{\infty} \sigma_j^{1-2\tau} \right)^{1/2}. \quad \text{(J.26)}$$

Note that the eigenvalues $\{\sigma_j\}_{j \geq 0}$ admit $\gamma$-exponential decay under Assumption 4.3. We now upper bound the right-hand side of (J.26) by

$$\sup_{z \in \mathcal{Z}} \left| \|\phi(z)\|_\Upsilon - \|\Pi_m[\phi(z)]\|_\Upsilon \right| \leq C_\psi/\sqrt{\lambda} \cdot \left\{ \sum_{j=m+1}^{\infty} C_1^{1-2\tau} \cdot \exp\left[ -C_2 \cdot (1-2\tau) \cdot j^\gamma \right] \right\}^{1/2}. \quad \text{(J.27)}$$

To simplify the notation, we define $C_{3,\tau} = C_\psi \cdot C_1^{1/2-\tau}/\sqrt{\lambda}$ and $C_{4,\tau} = C_2 \cdot (1-2\tau)$, which are both absolute constants. Then, by (J.27) and the monotonicity of $\exp(-u^\gamma)$, we further obtain

$$\sup_{z \in \mathcal{Z}} \left| \|\phi(z)\|_\Upsilon - \|\Pi_m[\phi(z)]\|_\Upsilon \right| \leq C_{3,\tau} \cdot \left[ \int_m^\infty \exp(-C_{4,\tau} \cdot u^\gamma) \, \mathrm{d}u \right]^{1/2}. \quad \text{(J.28)}$$

Here we can take the supremum over $\mathcal{Z}$ because the right-hand side of (J.27) does not depend on $z$. Note that we have shown in (J.16) that

$$\int_m^\infty \exp(-C_{4,\tau} \cdot u^\gamma) \, \mathrm{d}u \leq \begin{cases} C_{4,\tau}^{-1} \cdot \exp(-C_{4,\tau} \cdot m^\gamma), & \text{if } \gamma \geq 1, \\ 2 \cdot (\gamma \cdot C_{4,\tau})^{-1} \cdot \exp(-C_{4,\tau} \cdot m^\gamma) \cdot m^{1/\gamma - 1}, & \text{if } \gamma \in (0,1), \end{cases} \quad \text{(J.29)}$$

where for the case of $\gamma \in (0,1)$, (J.29) holds for sufficient large $m$ such that $m^\gamma \cdot C_{4,\tau} > 2/\gamma - 2$.

We now define $m^*$ as the smallest integer such that

$$\int_{m^*}^\infty \exp(-C_{4,\tau} \cdot u^\gamma) \, \mathrm{d}u \leq \left[ \epsilon/(2C_{3,\tau}) \right]^2. \quad \text{(J.30)}$$

By (J.29), since both $C_{3,\tau}$, $C_{4,\tau}$ and $\gamma$ are absolute constants, there exist absolute constants $C_{3,m}$ and $C_{4,m}$ such that

$$m^* \leq C_{3,m} \cdot \left[ \log(1/\epsilon) + C_{4,m} \right]^{1/\gamma}. \quad \text{(J.31)}$$

It is worth noting that the choice of $m^*$ in (J.31) is uniform over all $z \in \mathcal{Z}$. Moreover, by (J.28), for such an $m^*$, it holds that

$$\sup_{z \in \mathcal{Z}} \left| \|\phi(z)\|_\Upsilon - \|\Pi_{m^*}[\phi(z)]\|_\Upsilon \right| \leq \epsilon/2. \quad \text{(J.32)}$$

Thus, it remains to approximate $\|\Pi_{m^*}[\phi(z)]\|_\Upsilon$ up to accuracy $\epsilon/2$. Note that the subspace spanned by $\{\psi_j\}_{j \in [m^*]}$ is $m^*$-dimensional. When restricted to such a subspace, $\Upsilon$ can be expressed using a matrix $A_\Upsilon \in \mathbb{R}^{m^* \times m^*}$. Specifically, for any $j, k \in [m^*]$, we define the $(j,k)$-th entry of $A_\Upsilon$ as

$$[A_\Upsilon]_{j,k} = \left\langle \sqrt{\sigma_j} \cdot \psi_j, \sqrt{\sigma_k} \cdot \Upsilon \psi_k \right\rangle_{\mathcal{H}}. \quad \text{(J.33)}$$

Moreover, let $w_z \in \mathbb{R}^{m^*}$ be a vector whose $j$-th entry is given by $\sqrt{\sigma_j} \cdot \psi_j(z)$, $\forall j \in [m^*]$. Then, by (J.33) it holds that

$$\|\Pi_{m^*}[\phi(z)]\|_\Upsilon^2 = \left\langle \Pi_{m^*}[\phi(z)], \Upsilon \Pi_{m^*}[\phi(z)] \right\rangle_{\mathcal{H}} = w_z^\top A_\Upsilon w_z. \quad \text{(J.34)}$$

Also, since $\|\Upsilon\|_{\mathrm{op}} \leq 1/\lambda$, the matrix operator norm of $A_\Upsilon$ is bounded by $1/\lambda$; i.e., $\|A_\Upsilon\|_{\mathrm{op}} \leq 1/\lambda$. This means that the Frobenius norm of $A_\Upsilon$ is bounded by $\sqrt{m^*}/\lambda$. Let $\mathcal{C}_{m^*}(\epsilon/2, \lambda)$ denote the minimal $\epsilon^2/4$-cover of $\{A \in \mathbb{R}^{m^* \times m^*} \colon \|A\|_{\mathrm{fro}} \leq \sqrt{m^*}/\lambda\}$ with respect to the Frobenius norm. By definition, there exists $\widetilde{A}_\Upsilon \in \mathcal{C}_{m^*}(\epsilon/2, \lambda)$ such that $\|A_\Upsilon - \widetilde{A}_\Upsilon\|_{\mathrm{fro}} \leq \epsilon^2/4$. Hence, we have

$$\left| w_z^\top A_\Upsilon w_z - w_z^\top \widetilde{A}_\Upsilon w_z \right| \leq \|w_z\|_2^2 \cdot \|A_\Upsilon - \widetilde{A}_\Upsilon\|_{\mathrm{op}} \leq \|A_\Upsilon - \widetilde{A}_\Upsilon\|_{\mathrm{fro}} \leq \epsilon^2/4. \quad \text{(J.35)}$$

Finally, for any $z \in \mathcal{Z}$, we define

$$f_\Upsilon(z) = w_z^\top \widetilde{A}_\Upsilon w_z = \sum_{j,k=1}^{m^*} \sqrt{\sigma_j \cdot \sigma_k} \cdot \psi_j(z) \cdot \psi_k(z) \cdot \left[ \widetilde{A}_\Upsilon \right]_{jk}, \quad \text{(J.36)}$$

where $[\widetilde{A}_\Upsilon]_{jk}$ is the $(j,k)$-th entry of $\widetilde{A}_\Upsilon$ and $m^*$ is specified in (J.30). We remark that $f_\Upsilon \colon \mathcal{Z} \to \mathbb{R}$ is well defined since $m^*$ does not depend on $z$.

Finally, combining (J.32), (J.34), (J.35), and (J.36), we obtain
$$
\big\| \|\phi(z)\|_\Upsilon - f_\Upsilon \big\|_\infty = \sup_{z \in \mathcal{Z}} \big| \|\phi(z)\|_\Upsilon - f_\Upsilon(z) \big|
$$
$$
\leq \sup_{z \in \mathcal{Z}} \Big| \|\phi(z)\|_\Upsilon - \big\|\Pi_{m^*}[\phi(z)]\big\|_\Upsilon \Big| + \sup_{z \in \mathcal{Z}} \Big| \big\|\Pi_{m^*}[\phi(z)]\big\|_\Upsilon - f_\Upsilon(z) \Big|
$$
$$
\leq \epsilon/2 + \sup_{z \in \mathcal{Z}} \Big| \sqrt{w_z^\top A_\Upsilon w_z} - \sqrt{w_z^\top \widetilde{A}_\Upsilon w_z} \Big| \leq \epsilon/2 + \sup_{z \in \mathcal{Z}} \sqrt{\big| w_z^\top A_\Upsilon w_z - w_z^\top \widetilde{A}_\Upsilon w_z \big|} \leq \epsilon.
$$
This implies that $\{f_\Upsilon \colon \Upsilon \in \mathcal{C}_{m^*}(\epsilon, \lambda)\}$ forms an $\epsilon$-cover of $\mathcal{F}(\lambda)$ in (I.12). Hence, we have that
$$
N_\infty(\epsilon, \mathcal{F}, \lambda) \leq \big| \mathcal{C}_{m^*}(\epsilon/2, \lambda) \big|. \tag{J.37}
$$
Furthermore, using Corollary 4.2.13 in [69], we have
$$
\big| \mathcal{C}_{m^*}(\epsilon/2, \lambda) \big| \leq \big[ 1 + 8\sqrt{m^*}/(\lambda \cdot \epsilon^2) \big]^{m^{*2}}. \tag{J.38}
$$
Combining (J.31), (J.37), and (J.38), we finally have
$$
\log N_\infty(\epsilon, \mathcal{F}, \lambda) \leq m^{*2} \cdot \log \big[ 1 + 8\sqrt{m^*}/(\lambda \cdot \epsilon^2) \big]
$$
$$
\leq C_{3,m}^2 \cdot \big[ \log(1/\epsilon) + C_{4,m} \big]^{2/\gamma} \cdot \log \Big\{ 1 + 8 C_{3,m}^{1/2} \cdot \big[ \log(1/\epsilon) + C_{4,m} \big]^{1/(2\gamma)} / (\lambda \cdot \epsilon^2) \Big\}
$$
$$
\leq C_5 \cdot \big[ \log(1/\epsilon) + C_6 \big]^{1+2/\gamma},
$$
where $C_5$ and $C_6$ are absolute constants that depend on $C_\psi$, $C_1$, $C_2$, $\tau$, $\gamma$, and $\lambda$, but are independent of $T$, $H$, and $\epsilon$. Here in the last inequality we use the fact that $\log(1/\epsilon) \leq 1/\epsilon$, which holds when $\epsilon \leq 1/e$. Therefore, we conclude the proof for the second case and thus conclude the proof of the lemma. $\qquad \square$

## J.4  Technical Lemmas

Next, we present a few concentration inequalities. The first one provides concentration for standard self-normalized processes.

**Lemma J.1** (Concentration of Self-Normalized Processes in RKHS [18]). *Let $\mathcal{H}$ be an RKHS defined over $\mathcal{X} \subseteq \mathbb{R}^d$ with kernel function $K(\cdot, \cdot) \colon \mathcal{X} \times \mathcal{X} \to \mathbb{R}$. Let $\{x_\tau\}_{\tau=1}^\infty \subseteq \mathcal{X}$ be a discrete time stochastic process that is adapted to the filtration $\{\mathcal{F}_t\}_{t=0}^\infty$. That is, $x_\tau$ is $\mathcal{F}_{\tau-1}$ measurable for all $\tau \geq 1$. Let $\{\epsilon_t\}_{\tau=1}^\infty$ be a real-valued stochastic process such that (i) $\epsilon_\tau \in \mathcal{F}_\tau$ and (ii) $\epsilon_\tau$ is zero-mean and $\sigma$-sub-Gaussian conditioning on $\mathcal{F}_{\tau-1}$:*
$$
\mathbb{E}[\epsilon_\tau | \mathcal{F}_{\tau-1}] = 0, \qquad \mathbb{E}[e^{\lambda \epsilon_\tau} | \mathcal{F}_{\tau-1}] \leq e^{\lambda^2 \sigma^2 / 2}, \qquad \forall \lambda \in \mathbb{R}.
$$
*Moreover, for any $t \geq 2$, let $E_t = (\epsilon_1, \ldots, \epsilon_{t-1})^\top \in \mathbb{R}^{t-1}$ and $K_t \in \mathbb{R}^{(t-1) \times (t-1)}$ be the Gram matrix of $\{x_\tau\}_{\tau \in [t-1]}$. Then, for any $\eta > 0$ and any $\delta \in (0,1)$, with probability at least $1 - \delta$, simultaneously for all $t \geq 1$, we have*
$$
E_t^\top \big[ (K_t + \eta \cdot I)^{-1} + I \big]^{-1} E_t \leq \sigma^2 \cdot \operatorname{logdet} \big[ (1 + \eta) \cdot I + K_t \big] + 2\sigma^2 \cdot \log(1/\delta). \tag{J.39}
$$
*Moreover, if $K_t$ is positive definite for all $t \geq 2$ with probability one, then the inequality in (J.39) also holds with $\eta = 0$.*

*Proof.* See Theorem 1 in [18] for a detailed proof. $\qquad \square$

**Lemma J.2** (Lemma D.4 of [35]). *Let $\{x_\tau\}_{\tau=1}^\infty$ and $\{\phi_\tau\}_{\tau=1}^\infty$ be $\mathcal{S}$-valued and $\mathcal{H}$-valued stochastic processes adapted to filtration $\{\mathcal{F}_\tau\}_{\tau=0}^\infty$, respectively, where we assume that $\|\phi_\tau\|_\mathcal{H} \leq 1$ for all $\tau \geq 1$. Moreover, for any $t \geq 1$, we let $K_t \in \mathbb{R}^{t \times t}$ be the Gram matrix of $\{\phi_\tau\}_{\tau \in [t]}$ and define an operator $\Lambda_t \colon \mathcal{H} \to \mathcal{H}$ as $\Lambda_t = \lambda \cdot I_\mathcal{H} + \sum_{\tau=1}^t \phi_\tau \phi_\tau^\top$ with $\lambda > 1$. Let $\mathcal{V} \subseteq \{V \colon \mathcal{S} \to [0, H]\}$ be a class of bounded functions on $\mathcal{S}$. Then for any $\delta \in (0,1)$, with probability at least $1 - \delta$, we have*

simultaneously for all $t \geq 1$ that

$$\sup_{V \in \mathcal{V}} \left\| \sum_{\tau=1}^{t} \phi_\tau \{ V(x_\tau) - \mathbb{E}[V(x_\tau)|\mathcal{F}_{\tau-1}] \} \right\|_{\Lambda_t^{-1}}^2 \tag{J.40}$$
$$\leq 2H^2 \cdot \mathrm{logdet}(I + K_t/\lambda) + 2H^2 t(\lambda - 1) + 4H^2 \log(\mathcal{N}_\epsilon/\delta) + 8t^2\epsilon^2/\lambda,$$

where $\mathcal{N}_\epsilon$ is the $\epsilon$-covering number of $\mathcal{V}$ with respect to the distance $\mathrm{dist}(\cdot, \cdot)$.

*Proof.* Let $\mathcal{V}_\epsilon \subseteq \{V : \mathcal{S} \to [0, H]\}$ be the minimal $\epsilon$-cover of $\mathcal{V}$ such that $N_\epsilon = |\mathcal{V}_\epsilon|$. Then for any $V \in \mathcal{V}$, there exists a value function $V' : \mathcal{S} \to \mathbb{R}$ in $\mathcal{N}_\epsilon$ such that $\mathrm{dist}(V, V') \leq \epsilon$. Let $\Delta_V = V - V'$. By the inequality $(a + b)^2 \leq 2a^2 + 2b^2$, we have

$$\left\| \sum_{\tau=1}^{t} \phi_\tau \{ V(x_\tau) - \mathbb{E}[V(x_\tau)|\mathcal{F}_{\tau-1}] \} \right\|_{\Lambda_t^{-1}}^2 \tag{J.41}$$
$$\leq 2 \cdot \left\| \sum_{\tau=1}^{t} \phi_\tau \{ V'(x_\tau) - \mathbb{E}[V'(x_\tau)|\mathcal{F}_{\tau-1}] \} \right\|_{\Lambda_t^{-1}}^2 + 2 \cdot \left\| \sum_{\tau=1}^{t} \phi_\tau \{ \Delta_V(x_\tau) - \mathbb{E}[\Delta_V(x_\tau)|\mathcal{F}_{\tau-1}] \} \right\|_{\Lambda_t^{-1}}^2.$$

To bound the first term on the right-hand side of (J.41), we apply Lemma J.1 to $V'$ and take a union bound over $V' \in \mathcal{V}_\epsilon$. While for the second term, since $\sup_{x \in \mathcal{S}} |\Delta_V(x)| \leq \epsilon$, we have

$$\left\| \sum_{\tau=1}^{t} \phi_\tau \{ \Delta_V(x_\tau) - \mathbb{E}[\Delta_V(x_\tau)|\mathcal{F}_{\tau-1}] \} \right\|_{\Lambda_t^{-1}}^2 \leq t^2 \cdot (2\epsilon)^2/\lambda = 4t^2\epsilon^2/\lambda. \tag{J.42}$$

Thus, combining (J.41) and (J.42), we have

$$\sup_{V \in \mathcal{V}} \left\| \sum_{\tau=1}^{t} \phi_\tau \{ V(x_\tau) - \mathbb{E}[V(x_\tau)|\mathcal{F}_{\tau-1}] \} \right\|_{\Lambda_t^{-1}}^2$$
$$\leq \sup_{V' \in \mathcal{V}_\epsilon} 2 \cdot \left\| \sum_{\tau=1}^{t} \phi_\tau \{ V'(x_\tau) - \mathbb{E}[V'(x_\tau)|\mathcal{F}_{\tau-1}] \} \right\|_{\Lambda_t^{-1}}^2 + 8t^2\epsilon^2/\lambda. \tag{J.43}$$

Now we fix $V' \in \mathcal{V}_\epsilon$ and define $\varepsilon_t \in \mathbb{R}^t$ by letting $[\varepsilon_t]_\tau = V'(x_\tau) - \mathbb{E}[V'(x_\tau)|\mathcal{F}_{\tau-1}]$ for any $\tau \geq 1$. We define an operator $\Phi : \mathcal{H} \to \mathbb{R}^t$ as $\Phi = [\phi_1^\top, \ldots, \phi_t^\top]^\top$ and let $K_t = \Phi_t \Phi_t^\top \in \mathbb{R}^{t \times t}$. Using this notation, we have $\Lambda_t = \lambda \cdot I_{\mathcal{H}} + \Phi_t^\top \Phi_t$ and

$$\left\| \sum_{\tau=1}^{t} \phi_\tau \{ V'(x_\tau) - \mathbb{E}[V'(x_\tau)|\mathcal{F}_{\tau-1}] \} \right\|_{\Lambda_t^{-1}}^2 = \|\Phi_t^\top \varepsilon_t\|_{\Lambda_t^{-1}}^2 = \varepsilon_t^\top \Phi_t \Lambda_t^{-1} \Phi_t^\top \varepsilon_t$$
$$= \varepsilon_t^\top \Phi_t \Phi_t^\top (K_t + \lambda \cdot I)^{-1} \varepsilon_t = \varepsilon_t^\top K_t (K_t + \lambda \cdot I)^{-1} \varepsilon_t, \tag{J.44}$$

where the third inequality follows from (H.14). Setting $\lambda = 1 + \eta$ for some $\eta > 0$, we have

$$(K_t + \eta \cdot I)[K_t + (1 + \eta) \cdot I]^{-1} = (K_t + \eta \cdot I)[I + (K_t + \eta \cdot I)]^{-1} = [(K_t + \eta \cdot I)^{-1} + I]^{-1},$$

which implies that

$$\varepsilon_t^\top K_t (K_t + \lambda \cdot I)^{-1} \varepsilon_t \leq \varepsilon_t^\top (K_t + \eta \cdot I)[I + (K_t + \eta \cdot I)]^{-1} \varepsilon_t$$
$$= \varepsilon_t^\top [(K_t + \eta \cdot I)^{-1} + I]^{-1} \varepsilon_t. \tag{J.45}$$

Notice that each entry of $\varepsilon_t$ is bounded by $H$ in absolute value since $V'$ is bounded in $[0, H]$. By combining (J.43), (J.44), (J.45), Lemma J.1, and taking a union bound over $\mathcal{V}_\epsilon$, for any $\delta \in (0, 1)$, we obtain that, with probability at least $1 - \delta$,

$$\sup_{V' \in \mathcal{V}_\epsilon} \left\| \sum_{\tau=1}^{t} \phi_\tau \{ V'(x_\tau) - \mathbb{E}[V'(x_\tau)|\mathcal{F}_{\tau-1}] \} \right\|_{\Lambda_t^{-1}}^2$$
$$\leq H^2 \cdot \mathrm{logdet}[(1 + \eta) \cdot I + K_t] + 2H^2 \cdot \log(\mathcal{N}_\epsilon/\delta) \tag{J.46}$$

holds simultaneously for all $t \geq 1$. Moreover, notice that $(1 + \eta) \cdot I + K_t = [I + (1 + \eta)^{-1} \cdot K_t] \cdot [(1 + \eta) \cdot I]$, which implies that

$$\mathrm{logdet}[(1 + \eta) \cdot I + K_t] = \mathrm{logdet}[I + (1 + \eta)^{-1} \cdot K_t] + t \ln(1 + \eta)$$
$$\leq \mathrm{logdet}[I + (1 + \eta)^{-1} \cdot K_t] + \eta t. \tag{J.47}$$

Finally, combining (J.43), (J.46), and (J.47), we conclude that, simultaneously for all $t \geq 1$, (J.40) holds with probability at least $1 - \delta$, which concludes the proof. $\qquad\square$

**Lemma J.3** ([1])**.** Let $\{\phi_t\}_{t \geq 1}$ be a sequence in the RKHS $\mathcal{H}$. Let $\Lambda_0 \colon \mathcal{H} \to \mathcal{H}$ be defined as $\lambda \cdot \mathcal{I}_{\mathcal{H}}$ where $\lambda \geq 1$ and $\mathcal{I}_{\mathcal{H}}$ is the identity mapping on $\mathcal{H}$. For any $t \geq 1$, we define a self-adjoint and positive-definite operator $\Lambda_t$ by letting $\Lambda_t = \Lambda_0 + \sum_{j=1}^{t} \phi_j \phi_j^\top$. Then, for any $t \geq 1$, we have

$$\sum_{j=1}^{t} \min\{1, \phi_j^\top \Lambda_{j-1}^{-1} \phi_j\} \leq 2\mathrm{logdet}(I + K_t/\lambda),$$

where $K_t \in \mathbb{R}^{t \times t}$ is the Gram matrix obtained from $\{\phi_j\}_{j \in [t]}$, i.e., for any $j, j' \in [t]$, the $(j, j')$-th entry of $K_t$ is $\langle \phi_j, \phi_{j'} \rangle_{\mathcal{H}}$. Moreover, if we further have $\sup_{t \geq 0}\{\|\phi_t\|_{\mathcal{H}}\} \leq 1$, then it holds that

$$\mathrm{logdet}(I + K_t/\lambda) \leq \sum_{j=1}^{t} \phi_j^\top \Lambda_{j-1}^{-1} \phi_j \leq 2\mathrm{logdet}(I + K_t/\lambda).$$

*Proof.* Note that we have $\log(1+x) \leq x \leq 2\log(1+x)$ for all $x \in [0, 1]$. Since $\Lambda_t^{-1}$ is a self-adjoint and positive-definite operator, this implies that

$$\sum_{j=1}^{t} \min\{1, \phi_j^\top \Lambda_{j-1}^{-1} \phi_j\} \leq \sum_{j=1}^{t} 2\log\bigl(\min\{2, 1 + \phi_j^\top \Lambda_{j-1}^{-1} \phi_j\}\bigr) \leq 2\sum_{j=1}^{t} \log\bigl(1 + \phi_j^\top \Lambda_{j-1}^{-1} \phi_j\bigr).$$
(J.48)

Moreover, when additionally it is the case that $\sup_{j \geq 1} \|\phi_j\|_{\mathcal{H}} \leq 1$ for all $j \geq 0$, we have

$$\phi_j^\top \Lambda_{j-1}^{-1} \phi_j = \langle \phi_j, \Lambda_{j-1}^{-1} \phi_j \rangle_{\mathcal{H}} \leq \|\phi_j\|_{\mathcal{H}} \cdot \|\Lambda_{j-1}^{-1} \phi_j\|_{\mathcal{H}} \leq [\lambda_{\min}(\Lambda_0)]^{-1} \cdot \|\phi_j\|_{\mathcal{H}}^2 \leq 1. \quad (J.49)$$

Hence, applying the basic inequality $\log(1 + x) \leq x \leq 2\log(1 + x)$ to (J.49), we have

$$\sum_{j=1}^{t} \log\bigl(1 + \phi_j^\top \Lambda_{j-1}^{-1} \phi_j\bigr) \leq \sum_{j=1}^{t} \phi_j^\top \Lambda_{j-1}^{-1} \phi_j \leq 2\sum_{j=1}^{t} \log\bigl(1 + \phi_j^\top \Lambda_{j-1}^{-1} \phi_j\bigr). \quad (J.50)$$

For any $j \geq 1$, let $\Lambda_{j-1}^{1/2} \colon \mathcal{H} \to \mathcal{H}$ be the self-adjoint and positive-definite operator that is the square-root operator of $\Lambda_{j-1}$. Specifically, let $\{\sigma_\ell\}_{\ell \geq 1}$ be the eigenvalues of $\Lambda_{j-1}$ and let $\{v_\ell\}_{\ell \geq 1}$ be the corresponding eigenfunctions. Then $\Lambda_{j-1}^{1/2} = \sum_{\ell \geq 1} \sigma_\ell^{1/2} \cdot v_\ell v_\ell^\top$. Using this notation, for any $j \geq 1$, by the definition of $\Lambda_j$, we have

$$\Lambda_j = \Lambda_{j-1} + \phi_j \phi_j^\top = \Lambda_{j-1}^{1/2}\bigl(\mathcal{I}_{\mathcal{H}} + \Lambda_{j-1}^{-1/2} \phi_j \phi_j^\top \Lambda_{j-1}^{-1/2}\bigr)\Lambda_{j-1}^{1/2},$$

which implies that

$$\mathrm{logdet}(\Lambda_j) = \mathrm{logdet}(\Lambda_{j-1}) + \mathrm{logdet}\bigl(\mathcal{I}_{\mathcal{H}} + \Lambda_{j-1}^{-1/2} \phi_j \phi_j^\top \Lambda_{j-1}^{-1/2}\bigr)$$
$$= \mathrm{logdet}\bigl(\Lambda_{j-1}\bigr) + \mathrm{logdet}\bigl(1 + \phi_j^\top \Lambda_{j-1}^{-1} \phi_j\bigr) \quad (J.51)$$

Moreover, by direct computation, for any $t \geq 1$, we have

$$\det(\Lambda_t \Lambda_0^{-1}) = \det(I + K_t/\lambda). \quad (J.52)$$

Hence, combining (J.51), and (J.52), we obtain that

$$\sum_{j=1}^{t} \log\bigl(1 + \phi_j^\top \Lambda_{j-1}^{-1} \phi_j\bigr) = \mathrm{logdet}(\Lambda_t \Lambda_0^{-1}) = \mathrm{logdet}(I + K_t/\lambda). \quad (J.53)$$

Finally, combining (J.48), (J.50) and (J.53), we conclude the proof of this lemma. $\qquad\square$