[Reviews · NeurIPS 2020]

Review 1

Summary and Contributions: This paper studies the exploration problem in episodic reinforcement learning with kernel and neural network function approximations. The proposed algorithm is an optimistic version of least-squares value iteration, where the solution to the standard LSVI is further added by a bonus function for exploration. Under assumptions on the underlying RKHS or NTK function classes, the proposed algorithms are shown to achieve a H^2 \sqrt{T} \delta_F regret, where \delta_F depends on the effective dimension of the RKHS or NTK. —————— Updated after author response and reviewer discussion: After reading the authors’ response and discussing it with another reviewer, I would suggest the authors revise the paper in the following aspects regarding presentation and discussion. First, state clearly in the introduction (maybe also abstract) that this paper makes the assumption that the transition model is characterized by the RKHS class — I think you already did but it doesn’t hurt to emphasize it. Also, revise the sentence “propose the first provable efficient RL algorithm [...] without any additional assumptions on the sampling model” (lines 44-46), e.g., by changing the term “sampling model” to be “generative model” or “simulator”, as such a term is ambiguous. Second, discuss clearly the assumption that the transition model is characterized by the RKHS class, especially how it covers linear MDPs and Lipschitz MDPs as special cases by choosing different kernels. The other reviewer suggests adding the T regret under model misspecification, but I don’t think it is necessary — it is up for you to decide. Also, give concrete examples of such kernels, as promised in the authors’ response. In summary, the other reviewer and I don’t find any technical issues. The main debate is about the presentation and discussion, which I believe is easily fixable. Given the significance and depth of the results (especially the theory of DRL), I would keep my score and advocate for acceptance (as the rest two reviewers).

Strengths: 1. The theoretical results seem solid. 2. RL nonlinear function approximation is known to be challenging in both statistical and computation aspects. Statistically, in practice, DRL methods can be sample inefficient. Computationally, these methods may fail to converge. More importantly, RL with nonlinear function approximation has divergent counterexamples. This paper addresses the statistical aspect of this fundamental problem by proposing novel LSVI-type algorithms for both kernel and nonlinear neural network settings with provably sample efficiency. Moreover, these algorithms are also computationally efficient because (1) the kernel version involves kernel ridge regression and (2) the neural setting utilizes the NTK optimization theory for overparameterized NN. Thus, the proposed algorithm might be applicable to real-world deep RL problems where currently heuristic exploration schemes are widely used.

Weaknesses: The limitation of this work is that it only considers RKHS with exponential eigenvalue decay. Although the theoretical framework is general and the extension to polynomial decaying kernels seems involving just direct computations, it would be nice to have a general result that holds for all kernels.

Correctness: I didn't check the proof line by line but the theory seems reasonable and the proof seems correct.

Clarity: The paper is overall very well-written. However, as I mentioned above, the presentation can be further improved by adding: (1) a general result that holds for general RKHS that does not depend on the eigenvalue decay. This would strengthen the argument of this paper. (2) consider other eigenvalue decay conditions such as finite rank kernels and polynomial decay kernels. (3) Give a concrete example for both the RKHS and NTK kernels that satisfy the assumption.

Relation to Prior Work: Yes. This paper discusses and distinguishes from previous work thoroughly.

Reproducibility: Yes

Additional Feedback: My main concerns are stated in the Clarity section. I will elaborate on them as follows. [1] It would be nice to have a general result which depends on the intrinsic complexity of the RKSH, rather than only focusing on the exponentially decaying kernels, which is only a small class of kernels. [2] In the assumption, you also require the magnitude of eigenfunctions do not grow exponentially? Does this hold for the Gaussian RBF kernel, which is the example you gave in the paper for $\gamma = 1/d$. Moreover, are there any NTK that satisfy this assumption? It would be nice to provide an example of such an NTK. For example, does ReLU NTK satisfy the assumption? [3] When the kernel is finite rank, say having only $D$ nonzero eigenvalues, does the algorithm reduces to the LSVI algorithm in Jin et al? If so, how does the regret bound compare? [4] It would be nice to give a concrete example, say the Gaussian RBF kernel. In this case, it seems that the regret is H^2 \sqrt{T} (\log T)^{2+1.5d}. Why is this regret still “sample efficient” when it is exponential in d? It seems that the authors assume that $d$ is fixed. Then it would be nice to clearly define what “sample efficient algorithm” means.


Review 2

Summary and Contributions: This paper considers the problem of low-regret policy optimization in H-stage episodic MDPs with large state and action spaces. The main assumption is the availability of a kernel that exactly captures the dynamics of the MDP. Under this assumption, the paper shows that "kernelized" versions of existing LSVI-based algorithms using optimistic exploration yield algorithms with O(H²sqrt(T)) regret, also depending on various characteristics of the kernel space. An extension of this idea shows similar results for sufficiently large neural networks, using the "neural tangent kernel".

Strengths: This paper shows that the essential ideas of fitted Q-learning with linear function approximation continue to work in the kernel setting, and that the UCB-style reward bonus for exploration can also be expressed using kernel functions. Moreover, it identifies the characteristics of the kernel space (i.e. a particular l∞ covering number and an eigenvalue decay constant) that replace the dimensionality of a linear feature representation in the regret bounds. In the case that the kernel is actually constructed from a finite-dimensional linear function representation, the regret bounds match known results.

Weaknesses: While the paper does show regret bounds, they involve hard-to-reason about quantities like the covering number mentioned above. Furthermore, although the introduction claims the presented algorithm is computationally efficient, the presence of these unknown quantities in the regret bound means that the actual computational complexity is unknown. In particular, I don't see any results on how much computation is needed as a function of the suboptimality achieved. Furthermore, the paper makes very strong assumptions, namely the for /every/ Q-function, the Bellman optimality update lies in the RKHS. The recent work of Zanette et al. (ICML 2020, "Learning Near Optimal Policies with Low Inherent Bellman Error") shows that this assumption is equivalent to having linear dynamics (low Bellman rank), roughly meaning that the kernel /exactly/ captures the transition dynamics of the MDP. Furthermore, as pointed out in that work, departing from this assumption means that reward bonuses can no longer be used to induce exploration. Incidentally, this means that the statement on line 493 about the lack of explicit assumptions on the transition kernel is only marginally true; it may not be explicit, but it is there nonetheless. This means that one cannot simply reduce the complexity of the kernel space, because that risks violating this assumption; the approximation/estimation trade-off is not quantified. The equivalent assumption in the linear function approximation (LFA) setting is somewhat more acceptable, because the computation time needed there is well-specified (not depending on unknown quantities). In light of this, it seems misleading to claim that the computational complexity does not depend on the number of states, because it is not discussed how complex the RKHS needs to be to ensure the assumptions are satisfied for large MDPs. Similar issues exist for the variant of the proof based on neural tangent kernels. That makes the very strong assumption of an optimization oracle for a neural network whose size is polynomial in the number of transitions. It seems misleading to hide this assumption deep in the middle of the paper (page 5) while claiming to be computationally tractable. I would expect the claims of computational tractability to be removed, or some analysis of the cost of actually performing this optimization to the required accuracy. On the whole, the kernelization of the fitted Q-iteration algorithm with UCB-style reward bonuses seems mainly like a technical exercise, leading to little new insight beyond what was already known about MDPs with linear dynamics; the addition of kernels doesn't seem to change very much. Indeed, the extremely strong assumptions along the way make it doubtful how useful these results are, either in theory or in practice. Correctness: I did not have time to thoroughly check all the proofs, but skimmed through some of them and did not find any errors. It is quite possible that I might have missed them. The claims made in the introduction about computational complexity are not really borne out later in the paper, as noted above. To set more realistic expectations for the reader, the nature of the assumptions (perfect realizability, small covering number, optimization oracle for potentially large neural networks) should be mentioned along with or instead of those claims.

Correctness: I did not have time to thoroughly check all the proofs, but skimmed through some of them and did not find any errors. It is quite possible that I might have missed them. The claims made in the introduction about computational complexity are not really borne out later in the paper, as noted above. To set more realistic expectations for the reader, the nature of the assumptions (perfect realizability, small covering number, optimization oracle for potentially large neural networks) should be mentioned along with or instead of those claims.

Clarity: On the whole, the paper is fairly clear, apart from the misleading claims I have already described above.

Relation to Prior Work: The prior work section in the appendix is brief but serviceable. However, the comparison of the strengths and weaknesses of this work compared to existing literature is unfortunately one-sided. In particular, the claims of prior algorithms being computationally challenging (l. 496) apply to this work as well, and the following claim about not requiring additional assumptions on the transition model are wrong; assumption 4.1 is most certainly a (strong) additional assumption.

Reproducibility: Yes

Additional Feedback: My main objections to this paper are not the technical work, but the foundation, the assumptions needed and how they conflict with the claims made. I believe that the technical work could be salvaged if the discussion was more realistic about the benefits and costs. However, I'm not able to judge whether the resulting paper would be sufficiently novel or interesting. *** Post-rebuttal update I've read the author rebuttal, and while they partly address some of my concerns, they don't address the major ones. They credibly claim that the computational complexity of their algorithm is polynomial (although it is arguable if polynomial time algorithms are actually "efficient" if the degree of the polynomial is not understood). They have not addressed two significant concerns: the assumption that the dynamics of the MDP are perfectly realizable by the kernel is *extremely* strong. The previous work [10] along with other work of Jin et al., in the finite-dimensional setting, explicitly considers the effect of non-realizability. Just because kernels can be potentially infinite-dimensional does not mean that it is reasonable to assume they perfectly capture the dynamics of the MDP, which after all can be much more complex. Moreover, even if we somehow had a kernel that exactly captured the MDP, we usually have to make a tradeoff between approximation error and sample complexity by reducing the size of the function class; this is impossible without losing realizability. (The authors miss the point when they say the quantities are fixed when the RKHS is chosen; the algorithm designer has to make this tradeoff precisely when choosing the RKHS, which is one of the main purposes of having performance bounds,) More broadly, the performance bounds make use of hard-to-understand quantities like the covering number and the eigenvalue decay rate. In response to R1, the authors have promised to add concrete examples of RKHS that satisfy the assumptions. I would also like to see conditions on MDPs that would ensure that these examples (gaussian RBFs) exactly capture the transition dynamics. Without this last ingredient, it's hard to say whether this work is "theory for its own sake". The precursor work [10] explores the consequences of using optimistic value functions for exploration with finite-dimensional linear function approximation. This work transfers that to the kernel setting in a very similar way to how Kernel UCB is derived from Linear UCB; this kernel transformation is usually always possible for anything to do with linear approximations. Crucially, however, it does not really address the extra challenges that come with the kernel setting, making no attempt to resolve the strong assumptions. It is unclear whether simply moving to the kernel setting without further insights actually makes the results meaningfully different from the simpler finite-dimensional setting in [10]. I would like to see the authors more realistically discuss the limitations of their work up front, for example the strength of the realizability assumption, the optimization oracle for the neural network, etc. Some of these concerns of mine were alleviated after a discussion with R1, who has suggested some changes to the writing; these should be followed up on.


Review 3

Summary and Contributions: The authors propose a least-square value iteration algorithm with exploration bonus and derive regret bounds under both kernel and neural approximation schemes.

Strengths: The results are mainly theoretical. Such results provide insights for further understanding of the conditions needed for RL algorithms to converge to near-optimal solutions.

Weaknesses: It is not clear how much impact the proposed algorithm has to RL in practice.

Correctness: I did not check the proofs but the results seem plausible.

Clarity: The paper is easy to read, assuming that the reader is already familiar with most of the ideas in the literature this work is building on.

Relation to Prior Work: The authors did a decent job connecting the present work to prior works.

Reproducibility: Yes

Additional Feedback: It would be great if the authors could provide more insights in terms of how the proposed algorithm (such as the use of proper exploration bonus) can be applicable in practice. For example, if the regret bound for NOVI is always worse than that of KOVI, why would one use NOVI? ===== Post-Rebuttal ===== I have read the authors' feedback, I maintain my score. Thank you.


Review 4

Summary and Contributions: The authors proposed two provably efficient reinforcement learning algorithms called Kernel Optimistic least-squares Value Iteration (KOVI) and Neural Optimistic least-squares Value Iteration (NOVI) in episodic Markov Decision Processes with the horizon H, where both achieve O(H^2\sqrt(T))-regret bounds. Compared to the previous works with either tabular learning or linear function approximations, the submission has its novelty on the use of the powerful approximators such as Reproducing Kernel Hilbert Spaces (RKHSs) and overparameterized neural networks.

Strengths: The contribution of the submission is enough in the sense that it is the first provably efficient reinforcement learning methods with kernel and neural network approximations.

Weaknesses: Although the algorithm is shown to be computationally and statistically efficient, it seems difficult to be used in practice for the long-horizon setting (due to H^2 order). However, the submission is highly meaningful since this can give theoretical motivations for later works.

Correctness: Derivations and claims seem to be correct. This is a theoretical work without empirical studies.

Clarity: The submission is clearly written with clear mathematical definitions and notations. Sufficient explanations are given for all assumptions and theorems with the intuitive meaning of each term.

Relation to Prior Work: The relation to prior works is clearly specified in Appendix.

Reproducibility: Yes

Additional Feedback: The submission is extremely well-written, so there are not many comments I could make. The reason I score “accept” is due to the low confidence in my evaluation. One simple question is on Assumption 4.2 and Assumption 4.5. (which deal with assumptions on eigenvalues) and how it affects the proof. --- After I read other reviewers' comments and author response, I decided to keep my score.

[Author Response · NeurIPS 2020]

We would like to thank the reviewers for their helpful comments. We will revise accordingly in the revision.

**Reviewer 1**: (General RKHS) Our KOVI algorithm can be applied for any RKHS in generalized. As shown in the
discussion below Theorem 4.3, we can set $\beta = O(H\sqrt{\log N_\infty})$ and obtain a $H^2\sqrt{\log N_\infty \gamma_T T}$ regret, where $N_\infty$ is
the $\epsilon^*$-covering number of the value function class in Eq. (4.2) in the $\ell_\infty$-norm with $\epsilon^* = H/T$ and $\gamma_T$ is the effective
dimension of the RKHS. Here $\beta$ is set in this way to ensure optimism and $N_\infty$ appears due to a uniform concentration
argument. See Lemma C.2. We will revise Lemma C.2 for handling general RKHS in the revision.
(Eigen-decay conditions) As discussed above, our analysis can be applied to general RKHS. For the general case,
we only need to bound $\log N_\infty$ and $\gamma_T$. When the kernel has rank $r$, $\log N_\infty = \tilde{O}(r^2)$ and $\gamma_T = \tilde{O}(r)$, where $\tilde{O}(\cdot)$
omits $\log T$ terms. Then we obtain a $H^2\sqrt{r^3 T}$ regret, which recovers the linear case in Ref [33]. When the kernel has
polynomial eigen-decay ($\sigma_j \lesssim j^{-\nu}, \nu > 1$), our Lemma D.4 gives an upper bound of $\gamma_T$ and our Lemmas D.2 and D.3
can be modified to bound $\log N_\infty$. It can be shown that NOVI also achieves sublinear regret when $\nu$ is sufficiently large.
(Concrete examples of kernels) The Gaussian RBF kernel on the sphere $\mathbb{S}^{d-1}$ satisfies Assumption 4.2 for any fixed
$\tau \in (0, 1)$. (See {1}). Moreover the NTK induced by sinusoidal activations recovers the Gaussian RBF (See {2}). The
ReLU NTK satisfy the polynomial eigen-decay condition with $\nu = 1/d$. We will add concrete examples in the revision.

**Reviewer 2**: (Computational complexity) The computation complexity of KOVI is dominated by solving $HT$ kernel
ridge regression (KRR) problems, each with no more than $T$ data points. Thus, the total computation needed is
$H\text{poly}(T)$. Moreover, with sublinear regret, to achieve any fixed accuracy level $\varepsilon$, it suffices to set $T = \text{poly}(H, 1/\varepsilon)$.
Thus, the total computation needed to achieve $\varepsilon$ accuracy is polynomial in $H$ and $1/\varepsilon$ and is thus **efficient**. Moreover,
in the low-rank and exponential eigen-decay cases, the regret is $\tilde{O}(\sqrt{T})$, thus $T$ depends on $1/\varepsilon$ only through $\epsilon^{-2}$.
For polynomial decay we can also obtain a sublinear regret, which gives a $\text{poly}(H, 1/\varepsilon)$ computation complexity. For
NOVI, it is well-known that gradient descent converges linearly in training overparameterized NN (Refs [2,3,4,23]).
Since width $m$ is polynomial in $T$ and $H$, the computation in the neural setting is also $\text{poly}(H, 1/\varepsilon)$ and thus **efficient**.
(Assumption 4.1) **(i)** We would like to emphasize that the Bellman rank of the MDP model we consider is infinity as
we consider a infinite-dimensional function class. Our model only fall in the low Bellman-rank framework when the
RKHS kernel is low-rank. In this case, we recover the result in linear setting (Ref [33]). **(ii)** Even when restricted to
the linear case, it seems that [Zanette 2020] did not show that Assumption 4.1 is equivalent to having linear transition.
Instead, their Proposition 2 prove that when the inherent Bellman error is zero for all linear functions with parameters
in the unbounded set $\mathbb{R}^p$, the transition is linear. However, we require the Bellman operator maps any bounded function
with values in $[0, H]$ to a linear function with bounded parameters. [Zanette 2020]'s result does not apply. **(iii)** Our
assumption is implicit as it does not assume the transition to have a particular form and only assume Bellman operator
maps bounded functions to a bounded RKHS ball. **(iv)** Such an assumption is required because without any structural
assumption, the regret lower bound is $\sqrt{|S||A|H^3 T}$, which is infinity when $S \times A$ is an uncountable set. To have
meaningful result, we need to assume the target function $\mathbb{T}_h^\star f$ belongs to a function class with bounded capacity (in
terms of $\ell_\infty$-norm), which is standard in supervised learning. Here we consider the class of infinite-dimensional
RKHS-norm ball. **(v)** The complexity of RKHS is determined by its eigenvalues, and is fixed once the RKHS is
specified. Thus, it seems impossible to "reduce the complexity of kernel space". We show that the regret bound depends
on such intrinsic complexity through the covering number $N_\infty$ and effective dimension $\gamma_T$, both can be computed using
the eigenvalues. Moreover, $\gamma_T$ is previously used in analyzing the regret of kernel bandit (Refs [16,34,49]) and $N_\infty$
captures the temporal structure of MDP, which also appears in linear MDP (Ref [33]). Our work extends previous work
on linear setting to the infinite-dimensional kernel and neural settings with a general framework of regret analysis.

**Reviewer 3**: (Impact on RL practice) Most of the existing deep RL approaches adopt heuristic exploration strategies.
NOVI can be readily incorporated into the framework of neural fitted Q-learning (NFQ) in practice, which is the batch
version of DQN. NOVI proposes to add a bonus term to each NFQ-iteration. When using overparameterized neural
networks, we have proved that such an exploration scheme solves the deep RL problem with sample efficiency.
(KOVI v.s. NOVI) The regret of NOVI is worse than that of KOVI by $\beta T H \cdot \iota$, which is negligible when $m$ is a
polynomial of $T$ and $H$. Thus, KOVI and NOVI essentially have the same regret when $m$ is large. Besides, KOVI solves
kernel ridge regressions, which requires the closed form of the solution. In contrast, NOVI solves the least-squares
problems using gradient descent and can be applied to the case where we do not know the form of kernel function $\tilde{K}$.

**Reviewer 4**: (Long horizon setting) We consider the episodic setting where $H$ is fixed. In this case, the regret lower
bound is $\sqrt{H^3 T}$ (Ref [32]) and our upper bound is $\sqrt{H}$-larger in terms of $H$. The $H^2\sqrt{T}$-upper bound also appear in
various previous works with (generalized) linear function approximation (Refs [33,58,64,65,66], their $T$ is equal to
our $TH$). Moreover, our algorithms can be modified for the infinite-horizon discounted or ergodic settings. In these
settings, we only need to slightly modify Eq.(3.2) due to having different forms of Bellman equation. With the added
bonus term, we can similarly establish the optimism principle and sample complexity upper bounds.
(Assumptions 4.2 and 4.5) The eigen-decay condition captures the intrinsic complexity of RKHS. As discussed in the
first two points for **Reviewer 1**, our results can be extended to general RKHS by bounding the the covering number
$N_\infty$ and the effective dimension $\gamma_T$ under different eigen-decay conditions. Our assumption of exponential decay is
common in nonparametric statistics, which leads to an infinite-dimensional RKHS. We will add results for general
RKHS in revision.

{1}    Mercer's Theorem, Feature Maps, and Smoothing, Minh et al. ICML, 2006.
{2}    Random Features for Large-Scale Kernel Machines, Rahimi and Recht. NeurIPS, 2007.

[Meta-Review · NeurIPS 2020]

This paper studies the exploration problem in episodic reinforcement learning with kernel and neural network function approximations. The authors propose a novel algorithm which is an optimistic version of least-squares value iteration, where the solution to the standard LSVI is further added by a bonus function for exploration. They derive regret bounds for this algorithm for two different function classes: RKHS and NTK. Overall, the technical contribution in this paper seems solid. Some reviewers had some concerns about the assumptions made for the analysis, especially regarding the one assuming that the Bellman optimality update lies in the RKHS. There was a rather long discussion between two reviewers, one of them being more negative about this assumption and what it implies. Although the AC and senior AC agree with some of the points raised by this reviewer, this work still had some novelty to it and has the potential of encouraging further research in this direction. Overall, we recommend acceptance as a poster.